# Methods and Conversations in (Post)Modern Thermodynamics

Francesco Avanzini[1,2,$], Massimo Bilancioni[1,¶], Vasco Cavina[1,©], Sara Dal Cengio[3,•], Massimiliano Esposito[1,†], Gianmaria Falasco[4,a], Danilo Forastiere[4,5,§], Nahuel Freitas[6,£], Alberto Garilli[1,‡], Pedro E. Harunari[1,春], Vivien Lecomte[3,✳], Alexandre Lazarescu[7,↗], Shesha G. Marehalli Srinivas[1,№], Charles Moslonka[8,™], Izaak Neri[9,∥], Emanuele Penocchio[10,¤], William D. Piñeros[1,☏], Matteo Polettini[1,♪], Adarsh Raghu[9,*] Paul Raux[11,12,€], Ken Sekimoto[8,13,✍], and Ariane Soret[1,√]

**1** Department of Physics and Materials Science, University of Luxembourg, Campus Limpertsberg, 162a avenue de la Faïencerie, L-1511 Luxembourg (G. D. Luxembourg)
**2** Department of Chemical Sciences, University of Padova, Via F. Marzolo, 1, I-35131 Padova, Italy
**3** Université Grenoble Alpes, CNRS, LIPhy, FR-38000 Grenoble, France
**4** Department of Physics and Astronomy, University of Padova, Via Marzolo 8, I-35131 Padova, Italy
**5** INFN, Sezione di Padova, via Marzolo 8, I-35131 Padova, Italy
**6** Universidad de Buenos Aires, Facultad de Ciencias Exactas y Naturales, Departamento de Física. Buenos Aires, Argentina
**7** Institut de Recherche en Mathématique et Physique, UCLouvain, Belgium
**8** Laboratoire Gulliver, UMR CNRS 7083, ESPCI Paris, Université PSL
**9** Department of Mathematics, King's College London, Strand, London, WC2R 2LS, UK
**10** Department of Chemistry, Northwestern University, 2145 Sheridan Road, Evanston, IL 60208, USA
**11** Université Paris Cité, CNRS, UMR 8236-LIED, 75013 Paris, France
**12** Université Paris-Saclay, CNRS/IN2P3, IJCLab, 91405 Orsay, France
**13** Laboratoire Matière et Systèmes Complexes, UMR CNRS 7057, Université Paris Cité

$francesco.avanzini@unipd.it  ¶massimo.bilancioni@uni.lu  ©vasco.cavina@uni.lu
•sara.dal-cengio@univ-grenoble-alpes.fr  †massimiliano.esposito@uni.lu   agianmaria.falasco@unipd.it
§danilo.forastiere@unipd.it  £nfreitas@df.uba.ar  ‡alberto.garilli@uni.lu  春pedro.harunari@uni.lu
✳vivien.lecomte@univ-grenoble-alpes.fr  ↗alexandre.lazarescu@gmail.com  №shesha.marehalli@uni.lu
™charles.moslonka@espci.psl.eu  ∥izaak.neri@kcl.ac.uk  ¤emanuele.penocchio@northwestern.edu
☏william.pineros@uni.lu  ♪matteo.polettini@uni.lu  *raghu.adarsh@kcl.ac.uk
€paul.raux@etu.u-paris.fr  ✍ken.sekimoto@espci.psl.eu  √ariane.soret@uni.lu

December 4, 2023

**Lecture notes after the doctoral school (Post)Modern Thermodynamics held at the University of Luxembourg, December 2022, 5-7, covering and advancing continuous-time Markov chains, network theory, stochastic thermodynamics, large deviations, deterministic and stochastic chemical reaction networks, metastability, martingales, quantum thermodynamics, and foundational issues.**

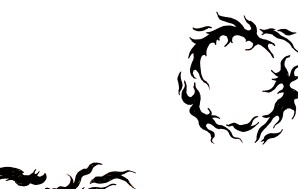

## Foreword

**Massimiliano Esposito**

Progress in nonequilibrium physics has been quite spectacular over the last 25 years. The trigger was without doubt the discovery of fluctuation theorems in many seemingly different contexts (thermostatted dynamics, Hamiltonian dynamics, stochastic dynamics) and the need to come up with a coherent understanding of the relation amongst these various fluctuation theorems. This effort converged into what is often called nowadays, broadly speaking, stochastic thermodynamics, encompassing the description of both classical and quantum systems. But at the same time that stochastic thermodynamics was being consolidated, research in the field also diversified and started interfacing with many new research areas.

The doctoral school (Post)Modern Thermodynamics that was held at the University of Luxembourg in December 2022, 5-7, represents that evolution. The first three courses covered classical topics in stochastic thermodynamics: defining Markov jump processes, analyzing them with elements of network theory, constructing the thermodynamic quantities (e.g. entropy production, local detailed balance, affinities), and analysing the role of coarse-graining. The emphasis on first-passage times and martingales in courses 1 and 9 is already a move toward more recent developments in stochastic thermodynamics, which showed that various first-passage problems can be constrained by thermodynamics. Courses 6 and 4 reflect the importance that the study of chemical reaction networks using stochastic thermodynamics has taken over the last decade. These systems are ideal to study the fate of stochastic thermodynamics in the macroscopic limit. Indeed their deterministic description can be viewed as emerging from a large volume limit. Course 5 introduces large deviation theory, which provides a powerful tool to characterize how nonequilibrium fluctuations scale in such a macroscopic limit. Course 8 analyses instead metastability, another phenomenon arising in the macroscopic limit (or equivalently in the low noise limits), and uses it to provide key insight into the central concept of local detailed balance. Course 10 focuses instead on quantum thermodynamics, another field that has attracted significant attention over the last decade and that aims at extending the concepts from classical stochastic thermodynamics to quantum systems and making contact with concepts from quantum information theory.

Matteo Polettini took the lead in organizing this doctoral school and did it in a truly collective manner, involving many young researchers. He also added a philosophical flavor to it, with a session dedicated to discussing foundational issues in thermodynamics and maybe decoding the enigmatic title of the school. A report on this session is the object of the last course 11.

We received numerous very enthusiastic feedback from the participants showing that the event was a real success. The hope is that the present lecture notes will help many young researchers enter the rapidly evolving field of nonequilibrium thermodynamics.

## Introduction

**Pedro Harunari, Vasco Cavina, William Piñeros and Matteo Polettini**

In early December 2022, amidst the first snowflakes and city lights illuminating Christmas markets, we gathered in the small city of Luxembourg for an immersive week dedicated to thermodynamics. The school+workshop provocatively named (post)Modern Thermodynamics brought together approximately 130 physicists to delve into ongoing research, established principles and interpretations of thermodynamics, and their recent unfoldings.

The first part of the event was a school aimed at, but not limited to, graduate students. Experts in their fields gave 2-hour-long lectures on topics such as stochastic thermodynamics, quantum thermodynamics, mathematical methods for statistical physics, and chemical reaction networks. Our goal was to provide introductory courses on relevant areas of research and improve the toolboxes of young researchers. More than textbook-style lectures, we asked lecturers to include a twist of their own research, making the process a bit more personal and communicating state-of-the-art developments to the students. This resulted in ten different lectures about the frontiers of thermodynamics by teachers with different original takes on a theory that puzzled physicists for a good part of the last three centuries.

The disparity of background, experiences and points of view of our teachers profoundly enriched the scientific discussion during the school but also compelled us to look for a way to increase its cohesion. In a manner befitting good physicists, we decided to run an experiment and to assign to each of our lecturers an "angel", whose role was to help in preparing the lectures and avoid overlaps within the program. Another important task assigned to the angels was to help in standardizing the notation among the various contributions: Much confusion arises from assigning distinct names/symbols to the same ideas such as e.g. entropy production versus entropy production rate, order of indices, negative versus positive signs conventions for directed heat flows, etc. During the lectures and in this version of them, we and the angels did our best to dissipate this confusion and strived to present a treatment that is notationally and conceptually homogeneous.

Given the pedagogic character of the lectures, we prepared the following notes with the help of some young participants of the (post)Modern Thermodynamics school. The contribution of the students to the event was invaluable, their curiosity and excitement being the main driving forces behind all the scientific discussions. Having them collaborate on these lecture notes was therefore a natural way to try and capture their energy and enthusiasm on the exposition of these subjects.

Of course, a team composed primarily of students preparing the notes was not without its challenges despite the healthy number of volunteers. We identified room for improvement in distributing workload and coordinating the various perspectives from members of different seniorities. Nonetheless we found we could partially integrate these differences via an internal "peer-review" process where team members could discuss and receive feedback on each other's content. Altogether, we believe the effort was an instructive exercise in collaborative academic writing for both students and organizers.

Our final communal experiment, and indeed the content of the last and namesake lecture 11 of the school, was an open-ended discussion to reflect on fundamental issues in and about thermodynamics. This took the form of a set of suggested questions, both by organizers and participants alike, in which everyone could also contribute an answer anonymously online for later in-person discussions. One of the most surprising outcomes of the survey was the lack of a common interpretation on the foundations of thermodynamics, showing how even in the presence of universally accepted experimental results and a well-established formalism there is still room for disparate conceptual beliefs. These differences sparkled an interesting philosophical debate that was highly appreciated by participants of the school+workshop and lead us to believe that there could be more space for this kind of discussion at scientific conferences.

The event itself was held free of charge following funds from already awarded grants. We were additionally able to offer financial aid, by means of free accommodation, to all students who requested it. This allowed participants from less fortunate backgrounds to join the school and attend in person. In total, 20 students benefitted from this opportunity, increasing the overall pool of students.

Post-(post)Modern Thermodynamics, we ran a survey that revealed that most partic-

ipants were satisfied with the chosen themes and overall organization. The criticism that stood out was the lack of time for relaxation and discussion in view of the dense program. We believe this problem could be circumvented by relaxing the number of lectures/talks or by making clear that participants understand they are not obliged to attend all the sessions as science is social, and conferences are made for exchanging.

We tried to make this school an opportunity for young researchers to grow, a "rich and challenging environment for the individual to explore", using the word of Noam Chomsky (to whom our social event, a widely participated billiard tournament, was dedicated).

We hope these lecture notes and the reported experience will serve the community of thermodynamicists for many more such future events.

## Outline

The first three lectures, Chaps. 1, 2 and 3, deal with the dynamics of Markov processes defined over networks, and how thermodynamics is tied to them. Lectures 4 and 5 address chemical reaction networks from deterministic and stochastic standpoints, presenting both analytical and numerical approaches, with particular attention to their nonequilibrium thermodynamics. Lectures 6 and 7 address the large deviation principle for describing fluctuations of observables in the limits of long times and large systems, and Lecture 8 discusses in more detail the metastable states that arise between these limits. Lecture 9 introduces martingale theory and its recently discovered connections to thermodynamics. Lecture 10 overviews the mathematical framework of quantum statistics and presents a consistent approach to defining thermodynamic quantities at the quantum level. Finally, lecture 11 wraps up with a discussion on thermodynamics fundamentals considering the input of school attendees, as we elaborated above.

We prepared a tentative unified notation (see Appendix A) and invited lecturers/angels to discuss and employ it at will. By no means did we intend to impose a standard on the field, but rather implement a time-saving tool when reading across the different sets of lectures and their related concepts. Some lecturers agreed to it, some others preferred maintaining diversity as a pedagogical tool. Interestingly, the parts that were more agreed upon were the more standard mathematical ones (linear algebra, graph theory) while little agreement was found in the use of symbols for thermodynamic quantities, apart from $S$ for entropy. In the meantime, the issue became a topic of informal discussion during the event, and even more so the disambiguation of concepts such as "reversibility", "microscopic reversibility", "detailed balance", etc.

# 1 Markov Chains and First-Passages

**Ken Sekimoto, Pedro Harunari and Charles Moslonka.**[1] *The time evolution of numerous systems across scientific domains is well described by the mathematics of Markov chains. We will introduce the formalism, its predictions, and how to use it both analytically and numerically. The problem of first-passage times in continuous-time Markov chains (CTMCs) will be motivated and solved. Finally, we will connect the discussed points to thermodynamics and the discrete-time version.*

## 1.1 Introduction

In this lecture we will introduce the main concepts and methods used in the study of Markov Chains.

In particular, the first part, which was conducted by Ken Sekimoto, will be centered around Markov chains in continuous time. Here our main goal is to set up different tools, such as the transition network (TN), master equations, and modified networks to adapt to the first-passage time problems.

In the second half-lecture (Subsections 1.7 and 1.8), conducted by Pedro Harunari, we are going to connect the different tools developed for continuous-time Markov chains to the discrete-time framework. We will first introduce those elements in the first part, and then we will present more recent developments about the first-passage time of *transitions*.

## 1.2 Continuous-time Markov Chains: basic notions

### 1.2.1 Notations and definitions

For basic notations, we will denote by $\{a, b, c, \dots\}$ or $\{a_1, a_2, \dots\}$ the discrete set of states, and $\hat{X}_t$ is the random variable representing the state $X_t$ of the system at time $t$, where $t \in [0; +\infty)$. The time evolution of $X$ is a stochastic process, and its history $\{X_t, t \in \mathbb{R}^+\}$ is also a random variable. The sample space $\Omega$ is the set of all possible histories. We will usually denote $X_0$ by $a_0$.

We recall the Markov property for such a stochastic process:

**Definition 1** $X_t$ is a continuous-time **Markovian** process with respect to $t$ if the conditional probability $p\left(\hat{X}_{t+dt} = a | \hat{X}_{[0,t]}\right)$ is independent of $X_s$ for all $s < t$.

The statistics of $\hat{X}_{t+dt}$ only depends on the realization of $\hat{X}_t$. It is usually said that the system *forgets* the past after every time step of length $dt$.

### 1.2.2 Transition rates

For two different states $a \neq b$, the conditional probability $p(\hat{X}_{t+dt} = b | \hat{X}_t = a)$ is of the order $O(dt)$ for a Markovian process. We denote by $R_{ba} > 0$ the proportionality coefficient, such that:

$$p(\hat{X}_{t+dt} = b | \hat{X}_t = a) = R_{ba}dt + O(dt^2). \tag{1.1}$$

For several destinations $\{a_1, a_2, a_3\}$ we have in a similar way:

$$p(\hat{X}_{t+dt} = a_i | \hat{X}_t = a) = R_{a_i a}dt + O(dt^2). \tag{1.2}$$

A Markovian process is characterized by the set of states and the transition rates among

---

[1]KS was the main lecturer; PH was the angel and lectured the second half; CM wrote this chapter.

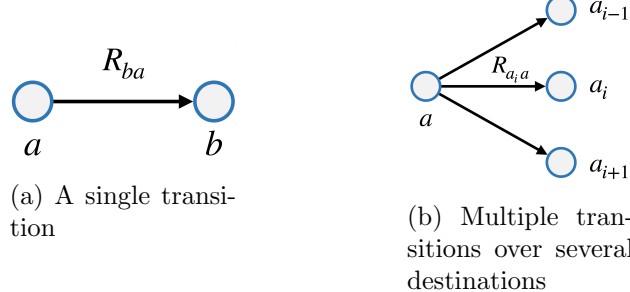

(a) A single transition

(b) Multiple transitions over several destinations

Figure 1.1: Possible transition configurations illustrating Eqs.(1.1) and (1.2)

them. Note that, to the linear order $O(dt)$, the transition rates do not interfere with each other.

In physics, one may consider discrete problems derived from an underlying continuous process thanks to coarse-graining procedures. As an example, let us consider a random walker traveling in Europe, as in Fig.1.2. We only measure the country code with respect to time. Right after crossing a border, the walker has - for a short time period - a large probability of re-crossing the same border. Thus, if the temporal coarse-graining was not introduced, one might see multiple erratic transitions between two country codes before the walker finally moves far enough from the border.

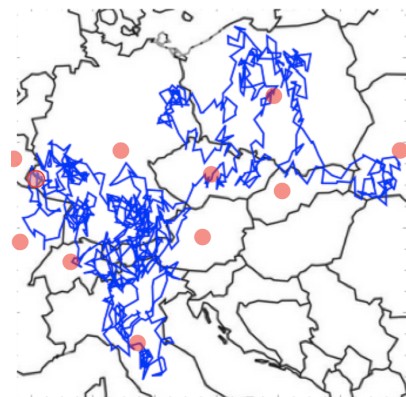

Figure 1.2: Continuous trajectory of a random traveler in Europe, with the different countries shown in red.

This discretization is typically non-Markovian. To obtain a Markovian trajectory in the new discrete-state continuous-time model, we need to weaken the time resolution of the trajectory, that is, introducing a time step $\Delta t$ such that faster phenomena are integrated over. More precisely, transitions $a \rightarrow b$ such that $R_{ba} \gtrsim (\Delta t)^{-1}$ should not appear in the discrete-state model. Thermodynamically, this state coarse-graining is equivalent to adding a heat bath to mask details. Descriptions with different resolutions can thus have different thermodynamics.

### 1.2.3 Transition networks

We use a network -or graph- representation for each Markov chain, in which the nodes are the states of the system, and the directed edges represent the non-zero transition rates. For the following part of the lecture, we consider ergodic ergodic transition networks i.e from any node, all the other nodes are reachable through directed edges. See Fig.1.4.

A prototype model: $\hat{X}_{t=0} = a_0$

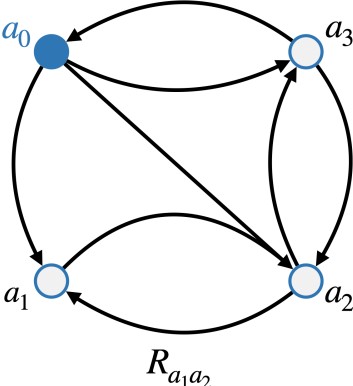

Figure 1.3: An example of ergodic transition network with four different states

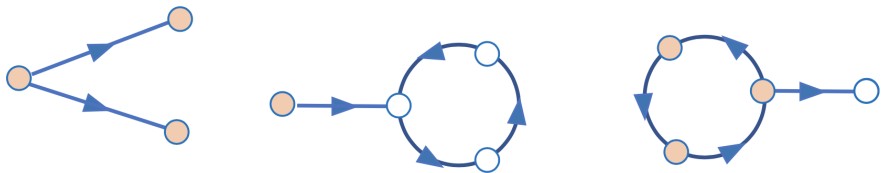

Figure 1.4: Three examples of non-ergodic transition networks. The nodes colored in red represent potentially unreachable states.

**Remark 1** In some other lectures of this School and Workshop, we encounter chemical reaction networks (CRN), where each node represents the state of particular constituent molecules, instead of the state of the *whole* system. This description will not be discussed in this chapter.

## 1.3 Simulating a trajectory: Gillespie's algorithm

Now that the basic notions of continuous-time Markov chains have been introduced, we may ask ourselves how to generate sample histories so that the statistical properties are verified. The main idea of such an algorithm is to generate a list of jumps at specific times, e.g $X_t = a \to a_i$ with $i \in \{1, \ldots, n\}$.

A first but naive idea is to try a jump at every small time segment $\delta t$. This method may work but is not practical, as it is quite inefficient and could be approximate if $\delta t$ is too big.

A better -and exact - approach is the Gillespie algorithm. The idea is to generate a waiting time $\hat{T}$ between consecutive jumps. The probability of having $\hat{T} > \tau$ where $\tau > 0$ is:

$$p(\hat{T} > \tau) = \exp\left(-\sum_{i=1}^{n} R_{a_i a} \tau\right). \tag{1.3}$$

PROOF The event $\hat{T} > \tau$ is equivalent to having no transitions during time intervals $\left(\frac{\tau}{M}\right) k \le t < \left(\frac{\tau}{M}\right)(k+1)$ for every $k \in \{0, 1, \ldots M - 1\}$.

Thus, we have:

$$p(\hat{T} > \tau) = \left(1 - \sum_{i=1}^{n} R_{a_i a} \frac{\tau}{M}\right)^M \xrightarrow{M \to \infty} \exp\left(-\sum_{i=1}^{n} R_{a_i a} \tau\right). \quad \blacksquare$$

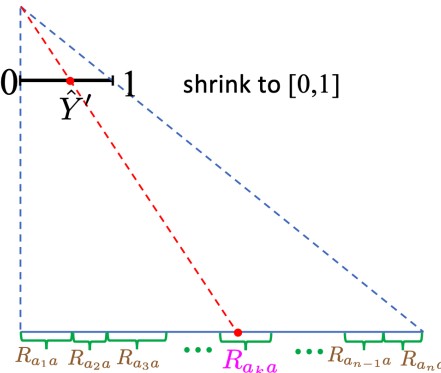

Figure 1.5: Choice protocol for the arrival state: the transition rates are "shrinked" to map the interval $[0, 1]$.

We then generate such a waiting time $\hat{T}$ through a uniform random variable $\hat{Y}$. We have:

$$p(\hat{T} > \tau) = p(e^{-\sum_{i=1}^n R_{a_i a} \hat{T}} < e^{-\sum_{i=1}^n R_{a_i a} \tau}) \tag{1.4}$$

$$= e^{-\sum_{i=1}^n R_{a_i a} \tau}. \tag{1.5}$$

Introducing $\hat{Y} := e^{-\sum_{i=1}^n R_{a_i a} \hat{T}}$ and $y := e^{-\sum_{i=1}^n R_{a_i a} \tau}$, we have:

$$p(\hat{Y} < y) = y \Rightarrow \hat{Y} \text{ is a uniform random variable on } [0, 1]. \tag{1.6}$$

After generating $\hat{Y}$ with a built-in function, we can find $\hat{T}$ such that $\hat{Y} = e^{-\sum_{i=1}^n R_{a_i a} \hat{T}}$.

To determine the arrival state, we notice that, in a Markovian process, the destination is determined at the last infinitesimal interval $dt$. We thus have:

$$p(\text{destination is } a_k) = \frac{R_{a_k a}}{\sum_{i=1}^n R_{a_i a}} \tag{1.7}$$

and the state can be decided with another uniform random variable on $[0, 1]$, see Fig.1.5.

**Remark 2** We can generalize this idea: given a 1D probability density $\rho(x)$, we can construct a random variable $\hat{X}$ that obeys $\rho(x)$. The cumulative probability up to $x \in \mathbb{R}$ is:

$$p(\hat{X} < x) = \int_{-\infty}^x \rho(x') dx'.$$

Since this is equivalent to $p(\int_{-\infty}^{\hat{X}} \rho(x') dx' < \int_{-\infty}^x \rho(x') dx') = \int_{-\infty}^x \rho(x') dx'$, we can define $\hat{Y} := \int_{-\infty}^{\hat{X}} \rho(x') dx'$, which is a uniform random variable on $[0, 1]$, and find $\hat{X}$ by this relation.

**_Exercise_ 1** [2] _Let $\rho(x, y) \in \mathbb{R}^2$ a probability density of a two-component random variable $\hat{Z} = (\hat{X}, \hat{Y})$. Describe a way of generating $\hat{Z}$ from two, mutually independent, uniform random variables on $[0, 1]$ denoted $(\hat{\xi}, \hat{\eta})$._

**_Exercise_ 2** _Consider the individual scores of a class of 134 students, with their rank sorting by increasing order (Fig.1.6). We denote by $r$ the rank and by $\psi(r) \in [0, 20]$ the score of the $r$-ranked student. We consider the score as a random variable._

---

[2]The solution to the exercises can be found at the end of this section.

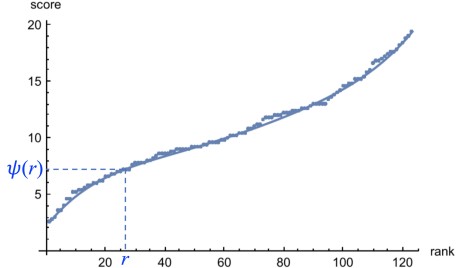

Figure 1.6: Score vs Rank of 134 students. The dotted curve is the actual data, and the smooth curve represents the continuous fitting score $= \psi(\text{rank} = r)$

- *Show that $p(\text{score} < \psi(r)) = r/134$*

- *Find the score probability density - over the interval $[0, 20]$ - using $\psi(r)$ without inverting $\psi(r)$.*

In Sec. 5, a *python* implementation of the algorithm is given for a simple chemical reaction network.

## 1.4 First-passage time problems

Consider a Markovian transition network such as the one depicted in Fig.1.3. Given an initial condition $\hat{X}_{t=0} = a_0$ and the ergodic hypothesis, the probability of the process $\hat{X}_t$ never visiting a state $a_i$ is zero. We can, therefore, define a time $\hat{T}_{FP}$ at which $\hat{X}_t$ visits $a_i$ *for the first time*. The random variable $\hat{T}_{FP}$ is called the first-passage time (usually abridged FPT [1]), and is a special case of stopping-time. We have in particular $p(\hat{T}_{FP} < +\infty) = 1$. Numerically, the sampling of $\hat{T}_{FP}$ can be done with a Gillespie algorithm. Moreover, we can obtain analytical results for its statistics, such as $p(\hat{T}_{FP} > \tau)$.

### 1.4.1 Master equation

Let us consider $N (\gg 1)$ copies of the transition network, starting at $\hat{X}_{t=0} = a_0$. For $t > 0$, each copy evolves independently. At a time $t$ we find $\simeq N p_t(a_i)$ copies in the state $\hat{X}_t = a_i$, with $0 \leq p_t(a_i) \leq 1$ and $\sum_{i=0}^{n} p_t(a_i) = 1$. Between $t$ and $t+dt$, the *population* i.e the probability of finding the system in a certain state, changes by $N p_{t+dt}(a_i) - N p_t(a_i)$. In parallel, counting all the possible transitions coming to the state $a_i$ from other states allow us to write the population influx as: $+ \sum_{k(\neq i)} (R_{a_i a_k} dt) (N p_t(a_k))$. Likewise, the probability out-flux coming from the state $a_i$ to all of the other states is: $- \sum_{k(\neq i)} (R_{a_k a_i} dt) (N p_t(a_i))$. Therefore, we have the following equality between time-variation and flux:

$$N p_{t+dt}(a_i) - N p_t(a_i) = \sum_{k(\neq i)} (R_{a_i a_k} dt) (N p_t(a_k)) - \sum_{k(\neq i)} (R_{a_k a_i} dt) (N p_t(a_i)). \tag{1.8}$$

Dividing by $N dt$ and taking the $N \to \infty$ limit, we obtain the so-called master equation:

$$\frac{dp_t(a_i)}{dt} = \sum_{k(\neq i)} R_{a_i a_k} p_t(a_k) - \sum_{k(\neq i)} R_{a_k a_i} p_t(a_i). \tag{1.9}$$

We can define the net probability flow from $a_i$ to $a_k$: $J_{a_k a_i} := -R_{a_i a_k} p_t(a_k) + R_{a_k a_i} p_t(a_i)$, also known as current, so that:

$$\frac{dp_t(a_i)}{dt} = - \sum_{k(\neq i)} J_{a_k a_i}. \tag{1.10}$$

The probability flow $J_{a_k a_i}$ can be seen as the difference of two *semi*-flows: the out-going flow

$$\mathcal{J}_{a_k a_i} := R_{a_k a_i}\, p_t(a_i) \tag{1.11}$$

and the in-coming flow

$$\mathcal{J}_{a_i a_k} := R_{a_i a_k}\, p_t(a_k). \tag{1.12}$$

Thus

$$J_{a_k a_i} = \mathcal{J}_{a_k a_i} - \mathcal{J}_{a_i a_k}. \tag{1.13}$$

The semi-flows characterize the effect of each individual possible transition, and are of particular importance when considering the transition network modifications that will be introduced in Section 1.5. For a pair of states $a_i$ and $a_k$, we say that the $a_i \leftrightarrow a_k$ transition is reciprocal if $\mathcal{J}_{a_k a_i} = \mathcal{J}_{a_i a_k}$. In this case, there is no net flow between them.

We have now switched from an individual history framework to a flow of population framework. We can now obtain a formal solution to the set of master equations. We regroup the state probabilities in a column vector: $\vec{p}_t := (p_t(a_0), \ldots, p_t(a_n))^{\dagger}$. We also introduce the diagonal elements, such that: $R_{a_i a_i} := -\sum_{k(\neq i)} R_{a_k a_i}$. We can now write all of the master equations as a vector-matrix equation:

$$\frac{d\vec{p}_t}{dt} = \mathbf{R}\,\vec{p}_t. \tag{1.14}$$

The matrix $\mathbf{R}$ is called the rate matrix, and its off-diagonal elements are the transition rates: $(\mathbf{R})_{ki} = R_{a_k a_i}$. A formal solution for every $t$ is thus:

$$\vec{p}_t = e^{\mathbf{R}t}\,\vec{p}_0. \tag{1.15}$$

**Remark 3** We recall the definition of the exponential of a matrix $\mathbf{M}$:

$$e^{\mathbf{M}} := \sum_{n=0}^{\infty} \frac{\mathbf{M}^n}{n!}. \tag{1.16}$$

We can write the propagator–that is the path integral from an initial state to a particular later state (in this case $p(\hat{X}_t = a_k | X_0 = a_i)$)–as:

$$p(\hat{X}_t = a_k | X_0 = a_i) = \left(e^{\mathbf{R}t}\right)_{a_k a_i}. \tag{1.17}$$

### 1.4.2   First-Passage time from master equation

We can use the vector master equation to study the statistics of $\hat{T}_{FP}$, through the usage of *absorbing boundary conditions* (for more details, the reader may refer to Chapter XII of [2]). Figure 1.7 represents qualitatively the procedure. The master equation allows to generate individual trajectories up to a time $t$. In a way, we know the intersection (and the subsequent statistics) of the trajectories $\{a(t)\}$ with a vertical line of coordinate $t$ (Fig.1.7(a)). The first-passage time problem is, in this framework, a sort of reverse problem. We want to know the (first) intersection-time of the trajectories with a horizontal line representing a particular state ($a^*$ in Fig.1.7(b)). The main idea is to modify the transition network, by introducing particular absorbing states, meaning that all out-going transitions from them are removed. In the trajectory-space of Figure 1.7, that means that once a trajectory has reached the state of interest $a^*$, it becomes stationary (Fig.1.7(c)). We then solve the master equation (Eq.1.14) for the now-modified rate matrix. From this we can deduce the statistics of interest, such as $p(\hat{T}_{FP} < t)$ or the mean first-passage time. Below we detail this procedure applied to the network depicted in Fig.1.3.

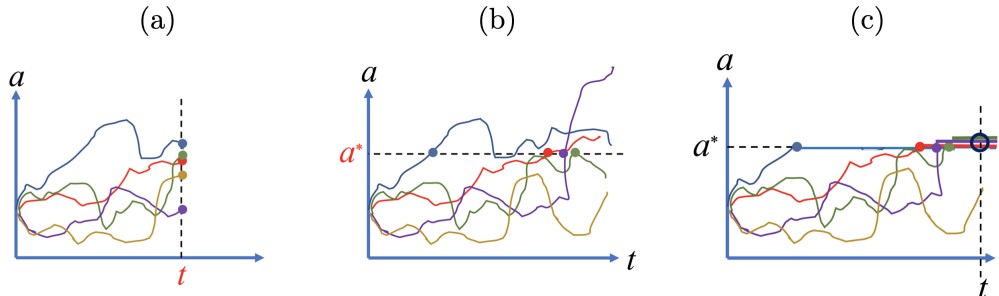

Figure 1.7: **(a)** Trajectories generated by the master equation $\frac{d\vec{p}_t}{dt} = \mathbf{R}\,\vec{p}_t$ representing a state variable $a$ with respect to time $t$. **(b)** Illustration of the first-passage time of each trajectory at the state $a^*$. **(c)** The same trajectories on the modified network, with an absorbing state at $a^*$.

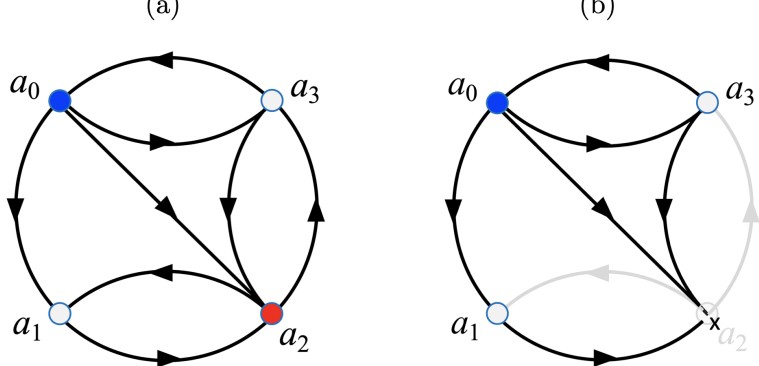

Figure 1.8: **(a)**: Example of a first-passage time problem at the state $a_2$ (in red) where $a_0$ (in blue) is the starting state. **(b)**: Modified transition network, where $a_2$ is now an absorbing state (cross), with out-going transitions having been removed (light-gray lines).

Now, we provide a step-by-step example procedure of network modification to obtain first-passage times. We consider the 4-states transition network depicted in Figs. 1.3 and 1.8(a), with a $(4 \times 4)$ rate matrix $\mathbf{R}$. Our goal is to compute the statistics of the FPT reaching the state $a_2$ starting from $a_0$.

- We first remove the destination node (or nodes if we consider more than one state of arrival) of the FPT problem, in this case $a_2$, and replace it with an absorbing state, that is a state from which no transitions are possible. This transformation of the transition network is depicted in Fig.(1.8)(b). We denote the modified rate matrix by $\mathbf{R}^*$, Eq.(1.19).

- We now consider the **reduced** state space, where all the absorbing states have been removed. In our example, the reduced state space is $(a_0, a_1, a_3)$, and the corresponding probability vector is $\vec{p_t^*} = (p_t^*(a_0), p_t^*(a_1), p_t^*(a_3))^\dagger$. The reduced master equation reads:

$$\frac{d\vec{p_t^*}}{dt} = \mathbf{R}^* \vec{p_t^*} \tag{1.18}$$

  with

$$\mathbf{R}^* = \begin{pmatrix} -R_{a_1 a_0} - R_{a_2 a_0} - R_{a_3 a_0} & 0 & R_{a_0 a_3} \\ R_{a_1 a_0} & -R_{a_2 a_1} & 0 \\ R_{a_3 a_0} & 0 & -R_{a_0 a_3} - R_{a_2 a_3} \end{pmatrix}. \tag{1.19}$$

- From Eq.(1.18), we obtain the solution, given the initial condition $\vec{p_0^*}$: $\vec{p_t^*} = \exp(\mathbf{R}^* t)\, \vec{p_0^*}$

- We can now compute the cumulative probability of the first-passage as an integral of the probability semi-flow towards the absorbing state over time:

$$p(\hat{T}_{FP} < t) = \int_0^t [\mathcal{J}_{a_2 a_0}^* + \mathcal{J}_{a_2 a_1}^* + \mathcal{J}_{a_2 a_3}^*] ds \tag{1.20}$$

$$= \int_0^t [R_{a_2 a_0} p_s^*(a_0) + R_{a_2 a_1} p_s^*(a_1) + R_{a_2 a_3} p_s^*(a_3)] ds \tag{1.21}$$

$$= 1 - (p_t^*(a_0) + p_t^*(a_1) + p_t^*(a_3)) = 1 - \vec{1^*} \cdot \vec{p_t^*} \tag{1.22}$$

  with $\vec{1^*} = (1, 1, \ldots 1)$ the unit row vector over the reduced state space. While Eq.(1.23) is understood by the complementary event, $\hat{T}_{FP} \geq t$, it can also be derived from the first line using a kind of Gauss-Stokes theorem.

- The FPT probability density $\rho_{FP}(t)$ is then obtained from Eqs.(1.18) and (1.22):

$$\rho_{FP}(t) = \frac{d}{dt} p(\hat{T}_{FP} < t) = -\vec{1^*} \cdot \mathbf{R}^* \cdot \vec{p_t^*}. \tag{1.23}$$

From Eq.(1.23), we can compute the quantities of interest–for example, the mean first-passage time conditioned to the initial state $\vec{p_0^*} = (1, 0, 0)^\mathsf{T}$.:

$$\langle \hat{T}_{FP} | \hat{X}_0 = a_0 \rangle = \int_0^\infty t \rho_{FP}(t) dt$$

$$= \int_0^\infty t \frac{d}{dt} p(\hat{T}_{FP} < t) dt = \int_0^\infty t \frac{d}{dt} \left[ p(\hat{T}_{FP} < t) - 1 \right] dt$$

$$= \left[ t(p(\hat{T}_{FP} < t) - 1) \right]_0^{+\infty} - \int_0^\infty (p(\hat{T}_{FP} < t) - 1) dt$$

$$= 0 + \int_0^\infty \vec{1^*} \cdot \vec{p_t^*} dt$$

$$= \int_0^\infty \vec{1^*} \cdot \exp(\mathbf{R}^* t) \cdot \vec{p_0^*} dt = -\vec{1^*} \cdot \mathbf{R}^{*-1} \cdot \vec{p_0^*} \tag{1.24}$$

where $\mathbf{R}^{*-1}$ denotes the inverse of the modified rate matrix, knowing that $\exp(\mathbf{R}^*t)\vec{p_0^*} \to \vec{0^*}$. It is possible to prove that $\mathbf{R}^*$ is invertible, but it is not a straightforward task; the reader may refer to Ref. [3], in particular its SM2, where it is shown that a survival matrix does not have any vanishing eigenvalue and therefore its determinant is nonzero. Note that the original matrix $\mathbf{R}$ is not invertible since 0 is an eigenvalue. Note also that $\vec{1^*} \cdot \mathbf{R}^* \neq 0$.

**Remark 4** We can derive this last result with a different approach attributed to Kramers [4]. We consider the reduced master equation, Eq.(1.18), complemented by a source term $J$ on the initial state $\vec{p_0^*}$. The equation reads:

$$\frac{d\vec{p_t^*}}{dt} = \mathbf{R}^* \vec{p_t^*} + J\vec{p_0^*}. \tag{1.25}$$

The steady-state, denoted $\vec{p_\infty^*}$, is:

$$\vec{p_\infty^*} = -J\mathbf{R}^{*-1}\vec{p_0^*}. \tag{1.26}$$

By multiplying $\frac{1}{J}\vec{1^*}$ from the left,

$$\frac{1}{J}\vec{1^*} \cdot \vec{p_\infty^*} = -\vec{1^*} \cdot \mathbf{R}^{*-1} \cdot \vec{p_0^*} = \langle \hat{T}_{FP}|\hat{X}_0 = a_0\rangle. \tag{1.27}$$

*Reminder* : **Linear algebra of master equation** We recall the spectral decomposition of the diagonalizable matrix $\mathbf{R}$: $\mathbf{R} = \mathbf{Q}\mathbf{\Lambda}\mathbf{Q}^{-1}$ where $\mathbf{\Lambda} = \mathrm{diag}(\lambda_1, \lambda_2, \ldots, \lambda_n)$ is the diagonal matrix of eigenvalues, and $\mathbf{Q}$ is the eigenbasis representation matrix.

Figure 1.9: Columns and rows of the matrices $\mathbf{Q}$ and $\mathbf{Q}^{-1}$.

We have $\mathbf{R}\mathbf{Q} = \mathbf{Q}\mathbf{\Lambda}$, and the columns of $\mathbf{Q}$ denoted by $\vec{v_\mu}$ are the *right*-eigenvectors of $\mathbf{R}$. Similarly, we have $\mathbf{Q}^{-1}\mathbf{R} = \mathbf{\Lambda}\mathbf{Q}^{-1}$, so the rows of $\mathbf{Q}^{-1}$, denoted by $\vec{u_\nu}$ are the (row) *left*-eigenvectors of $\mathbf{R}$. We have: $\mathbf{Q}^{-1}\mathbf{Q} = I \Leftrightarrow \vec{u_\nu} \cdot \vec{v_\mu} = \delta_{\nu\mu}$ (orthonormality of the dual bases) and $\mathbf{Q}\mathbf{Q}^{-1} = \sum_\mu \vec{v_\mu} \cdot \vec{u_\mu} = I$ (completeness). The latter follows from the former.

Since

$$\begin{aligned}
\exp(\mathbf{R}t) &= \exp\left(\mathbf{Q}\mathbf{\Lambda}\mathbf{Q}^{-1}t\right) \\
&= \sum_{n=0}^{+\infty} \frac{(\mathbf{Q}\mathbf{\Lambda}\mathbf{Q}^{-1})^n t^n}{n!} \\
&= \mathbf{Q}\left(\sum_{n=0}^{+\infty} \frac{\mathbf{\Lambda}^n t^n}{n!}\right)\mathbf{Q}^{-1} = \mathbf{Q}\exp(\mathbf{\Lambda}t)\mathbf{Q}^{-1},
\end{aligned} \tag{1.28}$$

we have

$$\exp(\mathbf{R}t) = \sum_\mu \vec{v_\mu} e^{\lambda_\mu t} \vec{u_\mu}. \tag{1.29}$$

**Remark 5** Numerically the exponential $e^{\mathbf{R}^*t}$ is computed using spectral decomposition, $\mathbf{R}^* = \sum_\mu \vec{v_\mu^*}\lambda_\mu \vec{u_\mu^*}$, that is, $e^{\mathbf{R}^*t} = \sum_\mu \vec{v_\mu^*} e^{\lambda_\mu t} \vec{u_\mu^*}$. Once $e^{\mathbf{R}^*t}$ is obtained, $\vec{p_t^*}$ is given by the matrix-vector product, $\vec{p_t^*} = e^{\mathbf{R}^*t}\vec{p_0^*}$

- The matrix $\mathbf{R}$ or $\mathbf{R}^*$ can have complex eigenvalues.

- All eigenvalues of $\mathbf{R}^*$ must have strictly negative real part because $\vec{p}_t^*$ with any initial state $a_0$ should decay to the reduced zero-vector, $\vec{0}^*$, for $t \to \infty$.

- The (non-reduced) rate matrix $\mathbf{R}$ must have at least one null eigenvalue: the steady-state distribution, $\vec{\nu}_0$ satisfies $\mathbf{R}\vec{\nu}_0 = 0$. The corresponding left null eigenvector $\vec{\nu}_0$ has all components 1, a consequence of the conservation of the total probability: for all $\vec{p}_t$, we have $d/dt\vec{\nu}_0 \cdot \vec{p}_t = \vec{\nu}_0 \cdot \mathbf{R} \cdot \vec{p}_t = 0$. For further spectral properties of the rate matrix, the reader may refer to Perron-Frobenius theorems and their consequences [5].

The FPT problem shows how network modifications can be used to derive quantities of interest of the original network. We can sometimes add nodes or even replicate the original network so as to adapt to more advanced first-passage problems. Below, we study different event statistics on the same model of network, through more convoluted semi-flow re-directions.

## 1.5 Other first event problems

### 1.5.1 First and second transition time

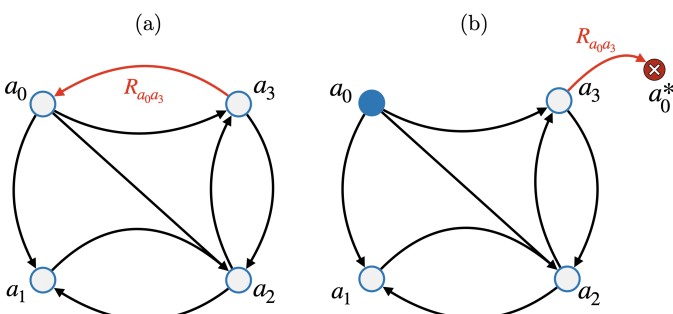

Figure 1.10: **(a)**: Original transition network, with the transition of interest $a_3 \to a_0$. **(b)**: Modified TN to study the first-transition time from $a_3$ to $a_0$, with an additional absorbing state (crossed state $a_0^*$ on the right)

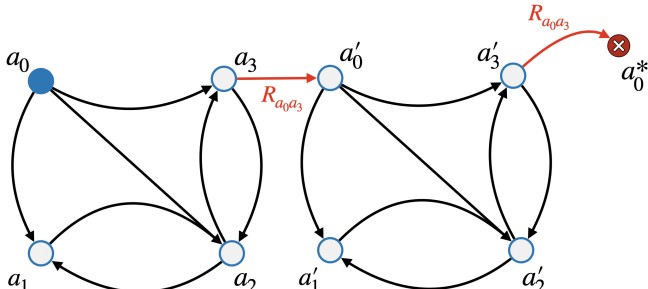

Figure 1.11: Modified TN to study the *second* transition time from $a_3$ to $a_0$, with a replica ($a_i'$ states) and an absorbing state (crossed state $a_0^*$ on the right).

We are now interested in the first *transition* time, which is the first time a specific transition happens in the network. In this case, we can cut the transition of inter-

est, and replace it with a transition to an absorbing state. In this case, the semi-flow, $\mathcal{J}^*_{a_0^* a_3'} = R_{a_0 a_3} p_t^*(a_3')$, gives the transition time statistics.

We can extend this strategy to other transition orders, such as *second* transition times, as shown in Fig.1.10. In this case, we make use of replicas of the system without the transition of interest, linking the two states to account for the first transitions. In the example of Fig.1.10, the second transition statistics are computed with one replica, linking the states $a_3$ and the state $a_0$ of the replica, and then an absorbing state.

### 1.5.2   FPT with competitions

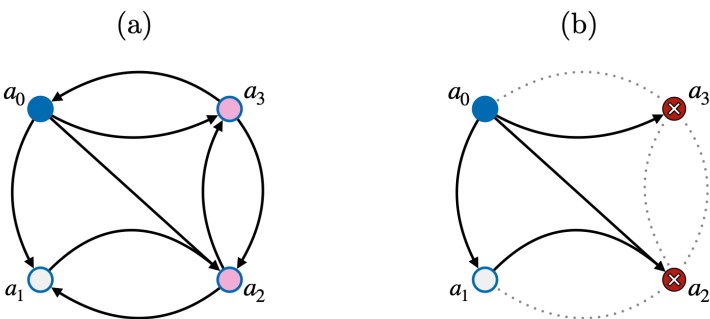

Figure 1.12: **(a)**: Original transition network, starting from $a_0$. **(b)**: Modified TN to study the first-passage time to $a_3$ *before* reaching $a_2$.

We can also put competitions on first-passage times. For example, in Fig.1.12 we study the first-passage time to $a_3$ before the state $a_2$ is reached. In this case we replace *both* states by absorbing states. Then, the modified semi-flow $\mathcal{J}^*_{a_3 a_0} = R_{a_3 a_0} p_t^*(a_0)$, gives the statistics in question. We can evolve this idea to study the first-transition time for $a_3 \to a_2$ before the reversed transition $a_2 \to a_3$. By placing two absorbing states at the end of opposite transitions (Fig. 1.13), one can assess the occurrence of one or the other (see Section 1.8).

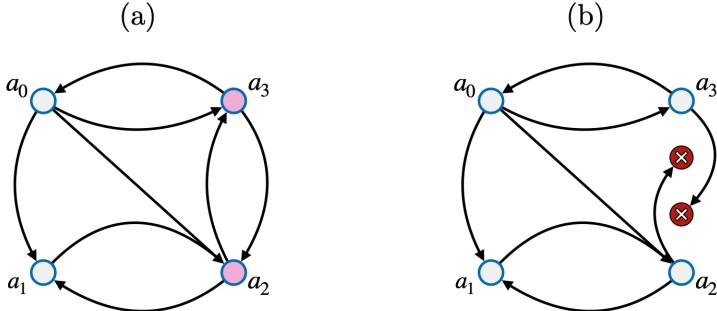

Figure 1.13: **(a)** Original TN in which we study the relative first occurrence of transition between $a_2$ and $a_3$ (in light pink). **(b)** The modified TN where two absorbing states have been added to measure the individual semi-flows.

### 1.5.3   FPT of consecutive transitions

As a last example, we can study FPT of *consecutive transitions*. For example (Fig.1.14) we may want to study the sequence of transitions $a_2 \to a_3 \to a_0$. In this case, we keep the original states $a_3$ and $a_2$, but we add a replica of the intermediate state $a_3$ and an

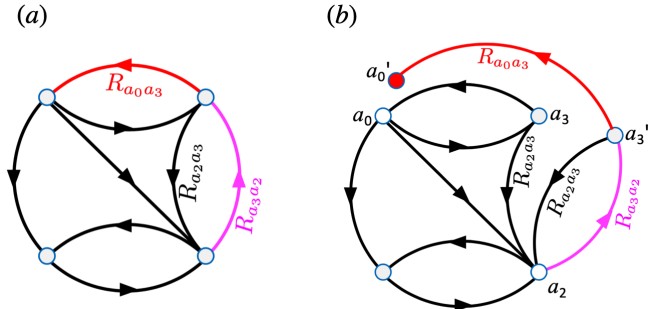

Figure 1.14: **(a)**: Original transition network. **(b)**: Modified TN to study the *consecutive* transitions $a_2 \to a_3 \to a_0$. In this case we compute the probability flow from $a'_3$ to $a'_0$.

absorbing state $a'_0$. Then, the probability flow from $a'_3$ to $a'_0$ gives the FPT density of the consecutive transitions.

## 1.6 "Markovianization" of non-Markov problems

Those network modification and state replication techniques can also be used to perform exact "Markovianization" of non-Markov problems [6]. For example, we consider a 1D

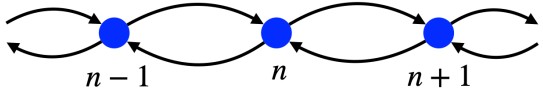

Figure 1.15: Scheme of a 1D chain network on which a non-Markov jump process occurs.

chain of states with jump probabilities depending on the actual state and the previous jump realization. We define the stochastic jump $\hat{X}_k$ that takes the value $+1$ when the walker goes from a state $n$ to the state $n+1$, or the value $-1$ if the walker goes from $n$ to $n-1$ at stage $k$. We suppose that the probability of having $\left( \hat{X}_k = \pm 1 \right)$ depends on the value of the previous jump. The jump probabilities $p(\hat{X}_k = \pm 1 | X_{k-1} = \pm 1)$ are defined as follows:

| | Jump probability | |
|---|---|---|
| Condition | $k$-th jump $= +1$ | $k$-th jump $= -1$ |
| $(k-1)$-th jump $= +1$ | $(1-\theta)w$ | $\theta w$ |
| $(k-1)$-th jump $= -1$ | $\theta w$ | $(1-\theta)w$ |

We can map this problem to a Markovian transition network, *via* a replica of the whole chain, to allow the conditions on $\hat{X}_{k+1}$ to be taken into account.

In the example depicted in Fig.1.16, the states $\{n'\}$ are only reached **from the left** to ensure that the previous transition was $+1$. Similarly, the states $\{n''\}$ are reached **from the right**, ensuring that the previous transition was $-1$. We then link the states with transition rates in accordance with the table to fully describe the Markovian network. We can then apply the previous techniques to obtain, for example, FPT statistics. Note that $n'$ and $n''$ both describe the same original state $n$, and their statistics must be added to obtain the original one.

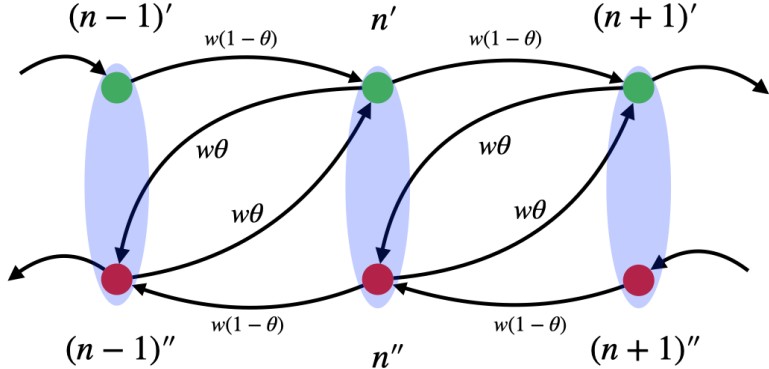

Figure 1.16: Modified transition network derived from Fig.1.15. The non-Markovian process is rendered Markovian thanks to the double chains.

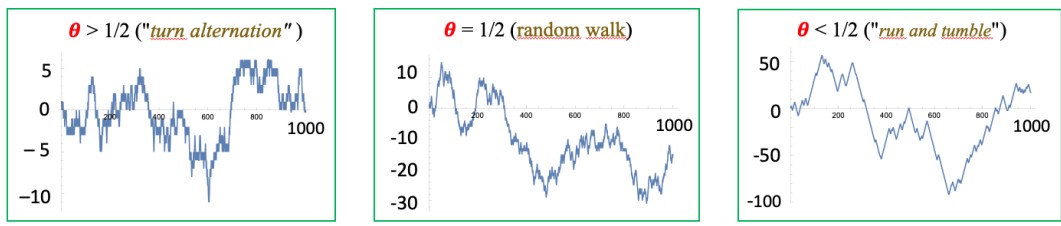

Figure 1.17: Typical trajectories for the three different cases. Note that the smaller $\theta$ is, the more spread trajectories are. This effect is explained qualitatively by the dependency of the diffusion coefficient $D$ with $\theta$ (Eq.1.34).

The random walk exhibits a directional persistence when the parameter $\theta$ differs from $1/2$. The $0 < \theta < 1/2$ case typically represents "run and tumble" processes, whereas $1/2 < \theta < 1$ represent turn alterations (typical trajectories are depicted in Fig.1.17). Otherwise, if $\theta = 1/2$, the process is a Markovian random walk. This "fine-graining" method of Markovianization preserves the time resolution of the underlying process - as opposed to coarse-graining methods (discussed above in Sec.1.2.2).

**Remark 6** The topology of the TN in Fig.1.16 is isomorphic with that of the run-and-tumble model of an active swimmer [6]. In the latter model the the nodes $n'$ and $(n-1'')$ in Fig.1.16 is associated to the position $n$.

**Calculation of the diffusion coefficient** We take, as a time-unit, the period between **consecutive tumbling**. (The step displacement of the tumbling will be taken into account later as the first displacement of this period.) Since the **tumbling** transition rate is $\theta w$, the probability density of this period is $P(t)dt = (\theta w)e^{-\theta wt}dt$. During this period, a number $n$ of forward jumps occurs with a Poissonian distribution $P(n|t) = \frac{e^{-(1-\theta)wt}}{n!}[(1-\theta)wt]^n$. The joint probability is then,

$$P(t,n)dt = \frac{e^{-(1-\theta)wt}}{n!}[(1-\theta)wt]^n(\theta w)e^{-\theta wt}dt. \tag{1.30}$$

Note that, using the gamma function identity $\Gamma(n) = \int_0^\infty e^{-x} x^{n-1} dx = (n-1)!$ with $n \in \mathbb{N}^*$, we have :

$$
\begin{aligned}
P(n) &= \int_0^\infty P(t, n) dt \\
&= \int_0^\infty \frac{e^{-(1-\theta)wt}}{n!} [(1-\theta)wt]^n (\theta w) e^{-\theta wt} dt \\
&= (1-\theta)^n \theta \int_0^\infty \frac{e^{-wt}}{n!} (wt)^n w \, dt \\
&= (1-\theta)^n \theta \frac{1}{n!} \int_0^\infty e^x x^n dx = (1-\theta)^n \theta \qquad \text{with } x = wt.
\end{aligned}
\tag{1.31}
$$

The run length $\ell_n$ is given by $\ell_n = n+1$, in taking account of the initial unit step associated with the tumbling. For the $N$ pairs of periods of forward and backward runs, the total time is $T = (\sum_{k=1}^N t_k^+) + (\sum_{k=1}^N t_k^-)$ and the total displacement is $X = (\sum_{k=1}^N \ell_k^+) - (\sum_{k=1}^N \ell_k^-)$. For each period we apply $P(t_k^\pm, n_k^\pm)$, where $\ell_k^\pm = n_k^\pm + 1$. The quantity of interest is the ratio $\langle X^2 \rangle / (2 \langle T \rangle)$, where $\langle \bullet \rangle$ denotes the expected value, or average. Since different $\ell_k^\pm$'s are mutually independent,

$$
\langle T \rangle = 2N \langle t \rangle \tag{1.32}
$$
$$
\langle X^2 \rangle = 2N(E \langle \ell^2 \rangle - \langle \ell \rangle^2) = 2N(\langle n^2 \rangle - \langle n \rangle^2). \tag{1.33}
$$

From $P(n)$ we find $\langle n^2 \rangle - \langle n \rangle^2 = \frac{(1-\theta)}{\theta^2}$, while from $P(t)$ we find $\langle t \rangle = \frac{1}{\theta \omega}$. Altogether, the diffusion constant is

$$
D = \frac{\langle X^2 \rangle}{2 \langle T \rangle} = \frac{\langle n^2 \rangle - \langle n \rangle^2}{2 \langle t \rangle} = \frac{w}{2} \frac{1-\theta}{\theta}. \tag{1.34}
$$

Here $w$ is the total frequency of jump, either rightward or leftward.

## 1.7 Discrete-time Markov Chains

### 1.7.1 A link between discrete and continuous Markov chains

Continuous-time Transition network           Discrete-time equivalent

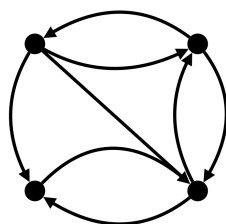 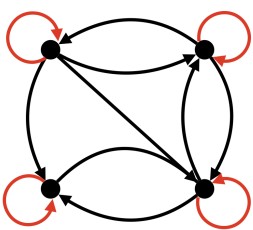

Figure 1.18: Discretization of time on the transition network. Stationary transitions are shown in red.

Let us consider the previously studied transition network, see Fig.1.18. The dots represent the different states of the systems and the arrows are the possible transitions. In discrete-time, the network has a similar shape, but the iteration of steps creates the possibility of sojourns when the system remains in the same state[3]. Those transitions

---

[3]Some transitions may happen at a finer resolution, but are not visible at the present level of description (see Section 1.2.2).

are added to the graph, symbolized by the circular arrows. We now consider time as a discrete variable: $t = n\Delta t$ with $n \in \mathbb{N}$ and $\Delta t$ is the unitary step duration. We define the stochastic (or transfer) [4] matrix $\mathbf{P}$ by

$$\mathbf{P} := \mathbf{I} + \Delta t \mathbf{R}, \tag{1.35}$$

where $\mathbf{R}$ is the rate matrix for the continuous-time master equation. $\mathbf{P}$ defines how the population probability is dynamically flowing in the network. The off-diagonal elements are given by:

$$(P)_{\substack{ab \\ a\neq b}} = \Delta t (R)_{ab} = p(\hat{X}_{n+1} = a | \hat{X}_n = b) \tag{1.36}$$

for $a$ and $b$ two states of the system. Since $(R)_{ab}$ is the probability per unit time that the system jumps from $b$ to $a$, $(P)_{ab}$ is the probability of jumping to state $a$ at time $(n+1)\Delta t$ given that state are in $b$ at time $n\Delta t$. Here $\hat{X}_n$ represents the random variable associated to the state occupied at time $n$, and $p$ is the probability measure.

The diagonal elements are:

$$(P)_{aa} = 1 + \Delta t (R)_{aa} = 1 - \Delta t \sum_{b(\neq a)} (R)_{ba}. \tag{1.37}$$

The diagonal element $(R)_{aa}$ is, according to the continuous-time framework, the negative sum of all transition rates from state $a$ (this results comes from the conservation of the probability norm during processes). $P_{aa}$ is thus the probability of not jumping at all:

$$(P)_{aa} = p(\hat{X}_{n+1} = a | \hat{X}_n = a). \tag{1.38}$$

Note that those transitions do not appear explicitly in the continuous-time framework.

The evolution equation of the probability vector $\vec{p}$ over the state space is:

$$\vec{p}_{n+1} = \mathbf{P}\vec{p}_n \tag{1.39}$$

which gives element-wise:

$$p_{n+1}(b) = \sum_a (P)_{ba} p_n(a) \tag{1.40}$$

$$= \sum_a p(\hat{X}_{n+1} = b | \hat{X}_n = a) p_n(a) \tag{1.41}$$

$$= \sum_{a(\neq b)} (P)_{ba} p_n(a) + (P)_{bb} p_n(b). \tag{1.42}$$

From Eqs.(1.36) and (1.37), we obtain:

$$p_{n+1}(b) = \sum_{a(\neq b)} \Delta t (R)_{ba} p_n(a) + [1 + \Delta t (R)_{bb}] p_n(b) \tag{1.43}$$

or, equivalently:

$$\frac{p_{n+1}(b) - p_n(b)}{\Delta t} = \sum_{a(\neq b)} (R)_{ba} p_n(a) - \sum_{a(\neq b)} (R)_{ab} p_n(b). \tag{1.44}$$

Taking the limit $\Delta t \to 0^+$ in Eq.(1.44) with $n = t$ allow us to verify the continuous-time master equation $\mathrm{d}\vec{p}/\mathrm{d}t = \mathbf{R}\vec{p}$. For this reason, everything said in the discrete-time

---

[4]The value of $\Delta t$ has to be small-enough to ensure the nonnegativity of all its entries, which is a technical detail that will not affect the connection to CTMCs.

framework holds in the continuous-time limit, given that we take a small enough time step $\Delta t$. This equivalence is sometimes used in the other direction, switching from continuous to discrete time to prevent dealing with exponentially-distributed time intervals. This shift of description can make proofs easier with the results holding in both cases as their are a time limit away.

Now we discuss the solution of the evolution equation. In order to write a propagator for the evolution equation, we apply Eq.(1.39) $n$ times:

$$
\begin{aligned}
\vec{p}_n &= \mathbf{P}^n \vec{p}_0 \\
&= (\mathbf{I} + \Delta t \mathbf{R})^n \vec{p}_0.
\end{aligned} \tag{1.45}
$$

$\mathbf{P}^n$ in the above is, therefore, the propagator. If we let $\Delta t \to 0^+$ with $n\Delta t$ fixed, we again recover the exponential propagator of the continuous-time formulation:

$$
\vec{p}_n = (\mathbf{I} + \Delta t \mathbf{R})^{t/\Delta t} \vec{p}_0 \xrightarrow[\Delta t \to 0^+]{} e^{\mathbf{R}t} \vec{p}_0. \tag{1.46}
$$

In the following table we sum up the differences between the continuous and discrete formulations.

| Element | Continuous framework | Discrete framework |
|---|---|---|
| Stochastic Matrix | Rate Matrix $\mathbf{R}$ | Transition probability matrix $\mathbf{P}$ |
| Dynamics | Master Equation $\frac{\mathrm{d}\vec{p}}{\mathrm{d}t} = \mathbf{R}\vec{p}$ | Evolution Equation $\vec{p}_{n+1} = \mathbf{P}\vec{p}_n$ |
| $p(\hat{X}_{t+\Delta t} = b \mid \hat{X}_t = a)$ | $(R)_{ba}\Delta t$ | $(P)_{ba}$ |
| Diagonal elements | $(R)_{aa} = -\sum_{b(\neq a)}(R)_{ba} \leq 0$ | $0 \leq (P)_{aa} \leq 1$ |
| Propagator | $(e^{\mathbf{R}t})_{ba}$ | $(\mathbf{P}^n)_{ba}$ |
| Conservation of prob. | $\sum_b (R)_{ba} = 0$ | $\sum_b (P)_{ba} = 1$ |

Note that the elements of $\mathbf{R}$ have the dimension of the inverse of time whereas the elements of $\mathbf{P}$ are dimensionless. It is also worth mentioning that discrete-time processes are simpler to simulate in a computer program as we do not have to draw random time steps with a Gillespie algorithm (as described in Section 1.3). In the discrete case, $\Delta t$ is fixed and the operator only has to draw the probabilities $(P)_{ba}$ from a uniform distribution between 0 and 1.

### 1.7.2 First-passage times

In a similar way as the continuous-time framework, we can compute the first-passage times by introducing *absorbing* states in the network - as shown in Fig.1.19. The only difference is that the stationary transition (dashed red line) cannot be removed like the other escape probabilities (Fig. 1.19b). Except for this remark, we refer to Section 1.4 for the computation of first-passage times. Note that for the problem to be properly defined, we need to assume the ergodicity of the considered network.

## 1.8 First-transition times and hidden networks

### 1.8.1 Introduction and experimental setups

In this subsection, we are no longer interested in the time when a state is reached for the first time. Instead, we want to measure the first time that a specific transition is realized in the network. This is still a very active topic of research and the reader may refer to the recent articles [7] and [8] for more details. As an illustration, consider the popular model of

(a)                                        (b)

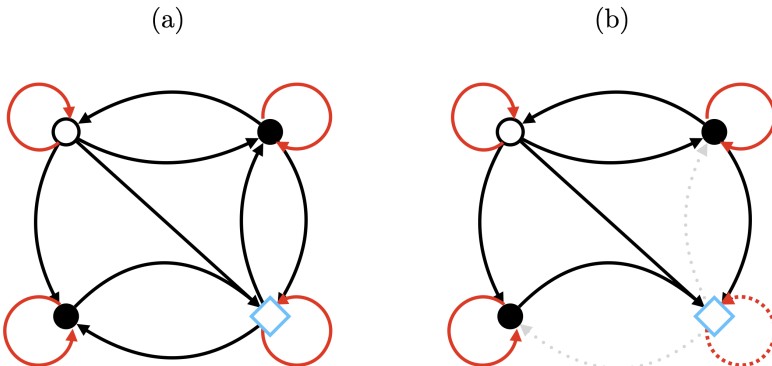

Figure 1.19: In order to compute the first-passage time to the square state from the open circle state the network **(a)** is modified. The arrival state is replaced by an absorbing state where the escape probabilities to other states are 0 **(b)**.

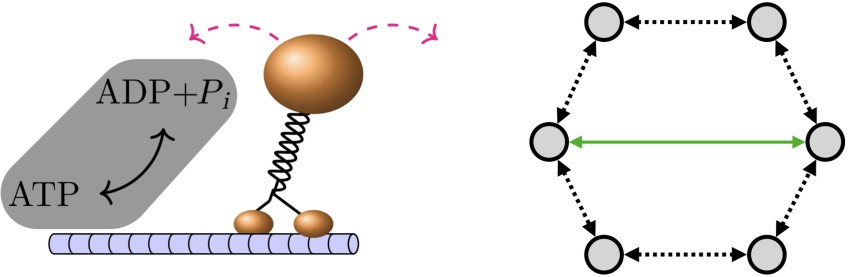

Figure 1.20: **(left)** Schematic of a molecular motor walking along a microtubule, consuming ATP as an energy source and releasing ADP as a byproduct. **(right)** Bicyclic Kinesin model, where the states and the dashed transitions are hidden. Only the transition along the solid line is visible.

a molecular motor moving along a microtubule and the monitoring of its activity (Fig1.20 (Left)). The only visible change in this system is the forward or backward movement of the motor. Indeed the whole metabolism behind is quite difficult experimentally to monitor precisely, and it is impossible to observe each molecule of ATP and ADP going in and out of the system. By contrast, the movement of the molecular motor itself can be measured accurately in experiments. The different states of the molecular motor are modeled by the Kinesin model. In this model, the different metabolic steps are represented by transitions in a network. Yet, the only *visible* transition is the movement of the motor, symbolized by the solid green line in Fig.1.20(right). All the other transitions and states are hidden from the observer.

Another experimental setup is a system of two quantum dots between two electrodes (Fig.1.21(a)). Each quantum dot can be occupied by one electron coming either from the electrodes or the other dot. The states of the system consist of vacant or occupied quantum dots denoted, respectively, by 0 or 1. For the two dots, the possible states are $00, 01, 10$ and $11$. The electrons are free to move in the system and to enter and leave through the electrodes. If we place only one detector between the two quantum dots, we do not have access to all the possible transitions (Fig.1.21(b)). In this case only the $01 \leftrightarrow 10$ transition is visible.

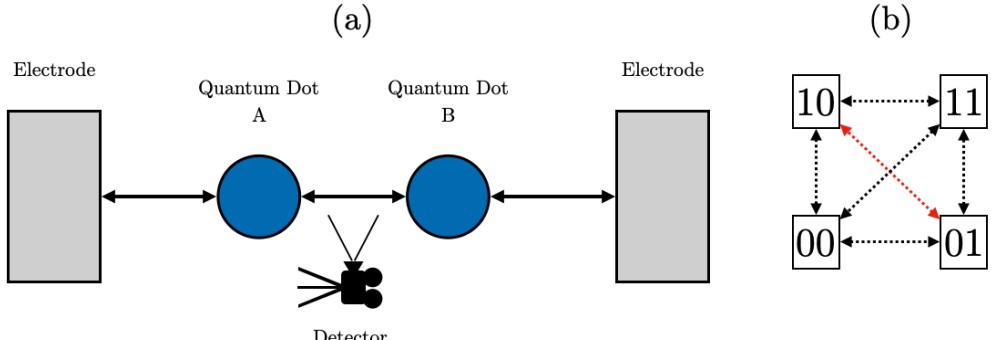

Figure 1.21: Quantum dot experiments. Only one detector is placed **(a)**, so that only a part of the possible transitions are observable. The equivalent network in shown on the right **(b)**. The plain red transition is visible whereas the dotted ones are not.

### 1.8.2   Inference from transition statistics

We now consider only the visible transition and its occurrences in a more formal manner. The observed trajectory (history) $\Gamma$ is therefore a list of waiting times $t_i$ (with $i \in \mathbb{N}$) between occurrences of visible transition $l_i$, starting at $t = 0$. See Fig.1.22 for an illustration.

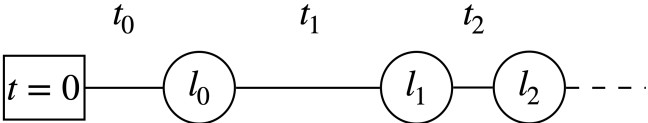

Figure 1.22: Example of a possible trajectory $\Gamma$.

In a similar manner as in Section 1.5, we can map the question of the first occurrence of a specified transition to the question of the first occupation of a specific state, by again modifying the transition network in a particular way with absorbing states, as shown in Fig.1.23. Note that the visible transition rates are kept the same in the modified network. In this case, the first-transition time will be the first-passage time to one of the absorbing states, whatever happens in the hidden part of the system.

Considering a trajectory $\Gamma$ with observed transitions $\{l_i\}_{i\in\mathbb{N}}$, we compute the first-transition time of $l_{i+1}$ given that we start (at $t = 0$) at the end of the preceding transition, $l_i$. With the modification trick in mind, we can now write the density of first-passage time according to Eq.(1.23) of Section 1.4.2:

$$\rho_{\text{PT}}(t) = R_{l_{i+1}} p_t^*(l_{i+1}|l_i(t=0)), \tag{1.47}$$

where $R_l$ is the transition rate of $l$. Here, $p_t^*(l_{i+1}|l_i(t = 0))$ denotes the probability that - in the modified transition network - the system is in the state located at the *tail* of the transition $l_{i+1}$ at time $t$, given that the system is in the state at the *tip (head)* of transition $l_i$ at time $t = 0$. The system can, therefore, perform any succession of hidden transition during 0 and $t$ with no visible transitions occurring, because the absorbing states prevent those events. The time $t$ is hence called the "intertransition time".

From the statistics of the intertransition time (which are experimentally measurable quantities), one can probe characteristics of the whole system, such as the amount of

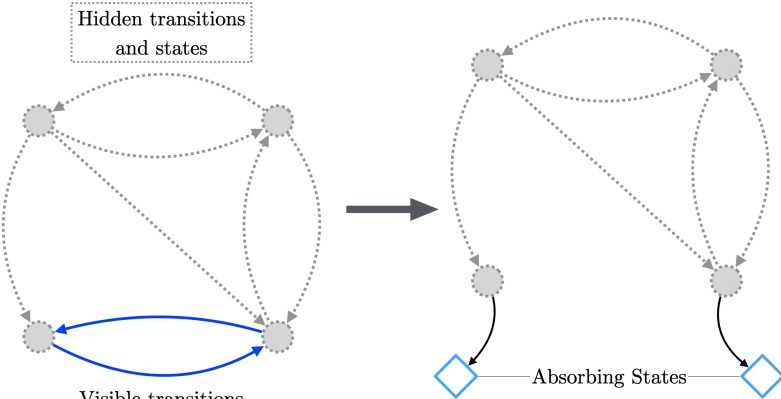

Figure 1.23: Example of a transition network with hidden states (dotted filled circles) and transitions (dotted curves), and one visible transition (solid curves). The network can be modified to switch from a first-transition time problem to a first-passage time problem, thanks to the addition of absorbing states. Note that, in this case, these absorbing states cannot be physically-meaningful initial positions, as they are only computational artifacts.

disorder of the molecular system, bounds for the entropy production rate and the thermodynamic efficiency, insights of the hidden network topology and possibly more quantities that are yet to be found [7,8].

**Solution to the exercises**

**Exercise 1**   Let us first take care of the $x$ component. Similar to the $1D$ case, we define $\hat{X}$ such that:

$$\hat{\xi} = \int_{-\infty}^{\hat{X}} \rho_x(x)\mathrm{d}x \tag{1.48}$$

where $\rho_x(x)$ is the marginal $\rho_x(x) = \int_{\mathbb{R}} \rho(x,y)\mathrm{d}y$. In practice, one needs the inverse function of the cumulative probability distribution which can be computed analytically or numerically.

For a given $\hat{X}$, we define $\hat{Y}$ such that:

$$\hat{\eta} = \int_{-\infty}^{\hat{Y}} \rho(y|\hat{X})\mathrm{d}y \tag{1.49}$$

where $\rho(y|x)$ denotes the conditional probability density: $\rho(y|x) = \rho(x,y)/\rho_x(x)$. Then, for given $(x,y)$, the probability attributed to the surface element $(dx \wedge dy)$ by the random variables $\hat{\xi}$ and $\hat{\eta}$ is:

$$d\xi \wedge d\eta = \rho_x(x)dx \wedge \left( \rho(y|x)dy + \left[ \int_{-\infty}^{y} \frac{\partial}{\partial x}\rho(y'|x)\mathrm{d}y \right] dx \right) \tag{1.50}$$

$$= \rho_x(x)\rho(y|x)(dx \wedge dy) + 0 = \rho(x,y)(dx \wedge dy) \tag{1.51}$$

because $(dx \wedge dx) = 0$. Therefore, $\hat{X}$ and $\hat{Y}$ corresponds to the density $\rho(x,y)$.

**Exercise 2**

- Question 1 : This is the definition of a ranked series. Since the interval of ranks is $[0, 134]$, the fraction $r/134$ is the number of scores *below* the score of the $r$-th student, which is $\psi(r)$. Hence $r/134 = p(\hat{S} < \psi(r))$, where $\hat{S}$ is the random variable corresponding to the score of a randomly chosen student.

- Question 2 : By taking the derivative of the relation, $p(\text{score} < \psi(r)) = \frac{r}{134}$, with respect to $r$, we have $\rho(\psi(r))\psi'(r) = (134)^{-1}$. Then the parametric presentation, $(134\psi'(r))^{-1}$ vs $\psi(r)$ gives the relation $\rho(s)$ vs $s$.

# 2 Network Thermodynamics

**Sara Dal Cengio, Nahuel Freitas and Paul Raux.**[5] *We introduce the basic concepts and computational tools of network theory. We discuss their use in the thermodynamics of Markov jump processes, including the Markov chain tree theorem and the Schnakenberg decomposition of forces and fluxes. We draw connections with the deterministic description of electrical circuits and chemical reaction networks, highlighting similarities and differences.*

## 2.1 Introduction

This chapter is dedicated to motivate the use of graph theory in the field of nonequilibrium thermodynamics. Graph theory found its first application in electricity. Therein, Kirchhoff gave a major contribution to the field by stating the current and the voltage laws, opening up to the understanding of electrical circuits as nonequilibrium systems involving one (type of) potential and current. Since then, graph theory has expanded its scope. Generalised Kirchhoff Current Law and Kirchhoff Voltage Law have been developed to accommodate systems with multiple currents and potentials, such as chemical networks. A vast recent literature on this subject can be found [9–14], but the review by Schnakenberg in 1976 [15] is considered as seminal in the field. In this lecture, we focus on two representatives of this literature: the Markov chain tree theorem and the decomposition of observables on a graph. These notes are organised as follows: in the first subsection, we introduce the elements of graph theory needed to describe a Markov chain on a graph; in the second subsection we recall some elements of linear algebra and use them to prove the Markov chain tree theorem; the last subsection is dedicated to the decomposition of physical observables on the graph.

## 2.2 Elements of graph theory

In this first subsection, we introduce the few basic elements of graph theory that we need in order to illustrate the relevance of graph theory in nonequilibrium thermodynamics. Across this subsection, we will illustrate the definitions on the four vertex graph already used as an example in [15] (See Figure 2.1).

### 2.2.1 Graph, incidence matrix and spanning tree

Let $\mathcal{G}(N, E)$ be a connected graph with $N$ vertices and $E$ edges. We denote respectively $\mathcal{V}$ and $\mathcal{E}$, the vertex space and the edge space of $\mathcal{G}$, so that $\dim(\mathcal{V}) = N$ and $\dim(\mathcal{E}) = E$. We assign to each edge $e$ an arbitrary orientation by defining a source vertex $s(e)$ and a target vertex $t(e)$, so that an edge can be more precisely denoted as $e \equiv (s(e) \rightarrow t(e))$. We emphasize that this orientation is a convention used to describe the network and does not convey any physical meaning concerning the directionality of transitions. Indeed, in the physical cases, transitions along an edge $e$ can be performed in both directions according to micro-reversibility. We denote $-e$ the reverse direction: $-e \equiv (t(e) \rightarrow s(e))$.

**Incidence matrix**    The incidence matrix $\boldsymbol{D}$, characterizing $\mathcal{G}$, is of size $N \times E$, and is defined as follows:

$$\boldsymbol{D}_{i,e} = \delta_{i,t(e)} - \delta_{i,s(e)} \,, \tag{2.1}$$

---

[5]SDC was the lecturer; NF was the angel; PR wrote this chapter.

where $\delta_{i,j}$ denotes the Kronecker delta, and the indices $i$ and $e$ span respectively the set of vertices and edges of $\mathcal{G}$. Eq. (2.1) encodes the fact that each edge has exactly one source and one target. As a consequence, the row vector $\boldsymbol{\ell}^\top = (1,\dots,1)$ is a left null vector of $\boldsymbol{D}$, so that:

$$\sum_i \ell_i \boldsymbol{D}_{ie} = \sum_i \boldsymbol{D}_{ie} = 0\,, \forall e\,. \tag{2.2}$$

It follows that the lines of $\boldsymbol{D}$ are not linearly independent and the rank of the incidence matrix, $\mathrm{rank}\boldsymbol{D}$, is strictly less than $N$. Specifically, since graph $\mathcal{G}$ is connected,

$$\mathrm{rank}\boldsymbol{D} = N - 1 \tag{2.3}$$

**Proof:**

- First, the rank nullity theorem of linear algebra relates $\mathrm{rank}\boldsymbol{D}$ with the dimension of the kernel, $\dim(\ker\boldsymbol{D})$, so that:

$$\mathrm{rank}\boldsymbol{D} = \mathrm{rank}\boldsymbol{D}^\top = N - \dim(\ker\boldsymbol{D}^\top) \tag{2.4}$$

  where $\boldsymbol{D}^\top$ is the transpose of matrix $\boldsymbol{D}$. Thus, to prove that $\mathrm{rank}\boldsymbol{D} = N-1$ is fully equivalent to show that $\dim(\ker\boldsymbol{D}^\top) = 1$.

- We have seen already that the vector $\boldsymbol{\ell}^\top$ belongs to $\ker\boldsymbol{D}^\top$. *Ad absurdum*, let assume that there exists another vector, independent from $\boldsymbol{\ell}$, which belongs to $\ker\boldsymbol{D}^\top$. If such a vector exists, one could build, by linear combination with $\boldsymbol{\ell}$, a non-zero vector $\boldsymbol{\ell}' \in \ker\boldsymbol{D}^\top$ containing a 0 component $\ell'_i = 0$; this is impossible since, using Eq. (2.2) by recursion along the connected graph $\mathcal{G}$, starting from vertex $i$, we find $\ell'_j = 0\ \forall j$. This is against the original assumption $\boldsymbol{\ell}' \neq 0$; thus the kernel of $\boldsymbol{D}$ is uniquely spanned by $\boldsymbol{\ell}$ and $\mathrm{rank}\boldsymbol{D} = N - 1$.

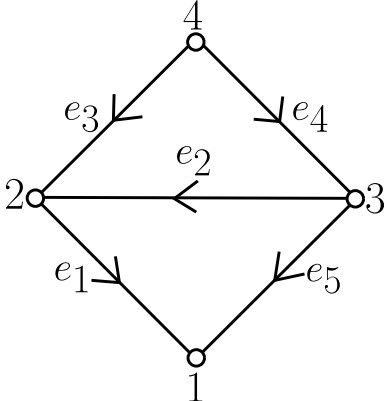

Figure 2.1: This graph has four vertices and five edges. The vertex space is $\mathcal{V} = (1, 2, 3, 4)$. The edge space is $\mathcal{E} = (e_1, e_2, e_3, e_4, e_5)$. Consider for example edge $e_1$, then the arbitrary orientation chosen here gives: $s(e_1) = 2$ and $t(e_1) = 1$ so that $e_1$ is the edge connecting $2 \to 1$.

**Example:** Figure 2.1 shows an oriented graph with four vertices and five edges. From now on, we will denote it $\mathcal{G}_{\mathrm{ex}}$. The vertex space and the edge space are respectively given by $\mathcal{V} = (1, 2, 3, 4)$ and $\mathcal{E} = (e_1, e_2, e_3, e_4, e_5)$. With this ordering of the vertices and the

edges, the incidence matrix of $\mathcal{G}_{\text{ex}}$ reads:

$$\boldsymbol{D} = \begin{pmatrix} 1 & 0 & 0 & 0 & 1 \\ -1 & 1 & 1 & 0 & 0 \\ 0 & -1 & 0 & 1 & -1 \\ 0 & 0 & -1 & -1 & 0 \end{pmatrix} \tag{2.5}$$

The first column in $\boldsymbol{D}$ corresponds to $e_1$. It has only two non-zero entries:

- a 1 for the target vertex of $e_1$ : $t(e_1) = 1$

- a $-1$ for the source vertex of $e_1$ : $s(e_1) = 2$

**Spanning tree**    Let define a spanning tree $\mathcal{T}_\mathcal{G}$ as a sub-graph of $\mathcal{G}$ such that:

- $\mathcal{T}_\mathcal{G}$ contains all the vertices of $\mathcal{G}$.

- All the edges in $\mathcal{T}_\mathcal{G}$ are edges of $\mathcal{G}$.

- $\mathcal{T}_\mathcal{G}$ is connected, i.e. there exists a succession of edges from any vertex to any other.

- $\mathcal{T}_\mathcal{G}$ contains no loop, i.e. no cyclic sequences of edges.

Several remarks follow:

- $\mathcal{T}_\mathcal{G}$ is not unique.

- $\mathcal{T}_\mathcal{G}$ contains exactly $N - 1 = \text{rank}\boldsymbol{D}$ edges, and its corresponding incidence matrix is full column rank. This is due to the absence of loops.

- The definition of spanning tree does not specify its edge orientation. If the edges of a spanning tree are oriented all towards vertex $i$, the spanning tree is said to be rooted in $i$. We denote the set of $i-$rooted spanning trees as $\mathcal{T}^i$. In the following, we denote with $\mathcal{T}$ the set of un-directed spanning tree, i.e. spanning trees with no edge orientation (see Figure 2.2).

**Example:**    The 8 un-directed spanning trees of $\mathcal{G}_{\text{ex}}$ are given in Figure 2.2. The set of 1-rooted spanning trees is given in Figure 2.3. Consider in particular the spanning tree circled in grey in the upper left corner of Figure 2.3, that we denote $\mathcal{T}^1_{\mathcal{G}_{\text{ex}}}$. $\mathcal{T}^1_{\mathcal{G}_{\text{ex}}}$ is one element of the set $\mathcal{T}^1$ and it consists of the three first edges of $\mathcal{G}$: $(e_1, e_2, e_3)$. Accordingly, the incidence matrix of this spanning tree, denoted $\mathcal{D}$, corresponds to the three first columns of $\boldsymbol{D}$ in Eq. (2.5) so that:

$$\mathcal{D} = \begin{pmatrix} 1 & 0 & 0 \\ -1 & 1 & 1 \\ 0 & -1 & 0 \\ 0 & 0 & -1 \end{pmatrix} \tag{2.6}$$

### 2.2.2   Cycles and cocyles

From a given choice of spanning tree $\mathcal{T}_\mathcal{G}$, two subsets of $\mathcal{E}$ can be identified:

- The set of *chords* is the set of edges that do not belong to $\mathcal{T}_\mathcal{G}$. Since removing the chords is equivalent to opening all the loops of $\mathcal{G}$, the cardinality of chords is equal to the number of loops of $\mathcal{G}$, i.e. $c = E - N + 1$ . From the viewpoint of the incidence matrix, the chords correspond to $c$ dependent columns of $\boldsymbol{D}$. We denote by $\alpha$ the index spanning the set of chords. In the example in Figure 2.3, the set of chords whose removal generates $\mathcal{T}^1_{\mathcal{G}_{\text{ex}}}$ is $(e_4, e_5)$.

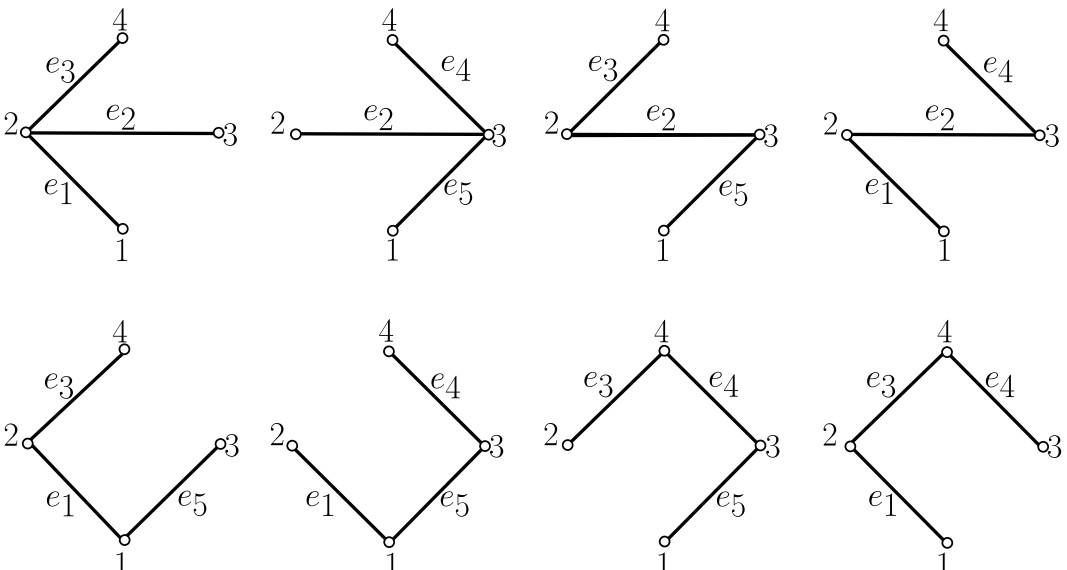

Figure 2.2: $\mathcal{T}$: Set of un-directed spanning trees of $\mathcal{G}_{\text{ex}}$.

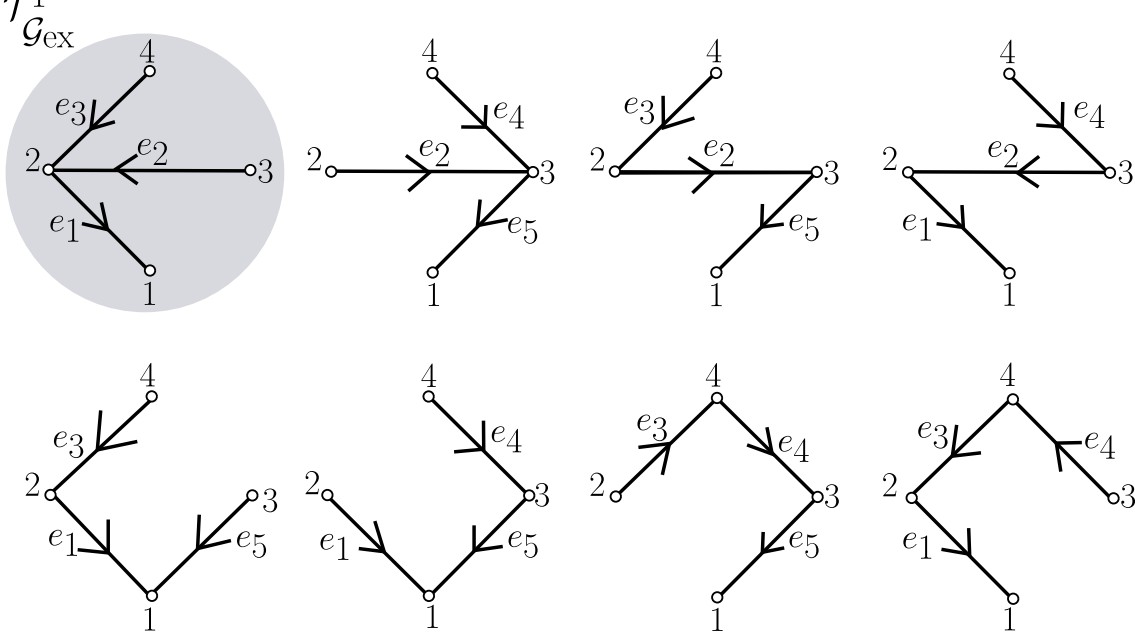

Figure 2.3: $\mathcal{T}^1$: Set of 1-rooted spanning trees of $\mathcal{G}_{\text{ex}}$.

- The set of *cochords* is the set of edges that belong to $\mathcal{T}_{\mathcal{G}}$. From the viewpoint of the incidence matrix, the cochords correspond to linearly independent columns of $\boldsymbol{D}$. This encodes the absence of loops in $\mathcal{T}_{\mathcal{G}}$. We denote by $\gamma$ the index that spans the set of cochords. In the example of Figure. 2.3, the set of cochords of $\mathcal{T}^1_{\mathcal{G}_{\text{ex}}}$ is $(e_1, e_2, e_3)$.

In what follows, we always order the edges of $\mathcal{G}$ such that $1 \leq \gamma \leq N-1$ and $N \leq \alpha \leq E$.

**Cycle**    A cycle is the loop obtained by placing back a chord in $\mathcal{T}_{\mathcal{G}}$ and removing the cochords that do not belong to the loop just created. By convention, we assign to the cycle the orientation of its generating chord. Then, formally we define cycle vectors $\boldsymbol{c}^\alpha \in \mathcal{E}$

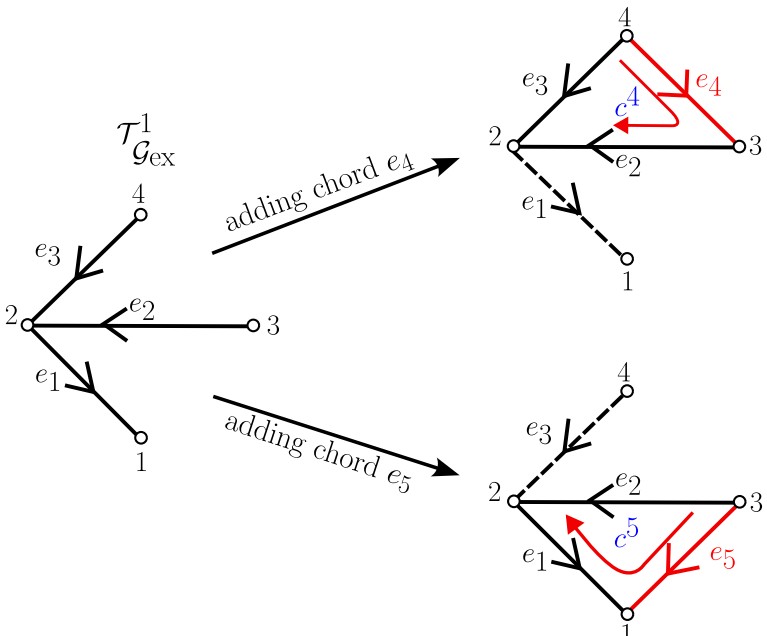

Figure 2.4: Cycles of $\mathcal{G}_{\mathrm{ex}}$ built out of $\mathcal{T}^1_{\mathcal{G}_{\mathrm{ex}}}$.

as vector with components:

$$
c^\alpha_e = \begin{cases} 1 & \text{if } e \text{ belongs to the cycle generated by the chord } \alpha \\ -1 & \text{if } -e \text{ belongs to the cycle generated by the chord } \alpha \\ 0 & \text{otherwise} \end{cases} \tag{2.7}
$$

Given the one-to-one correspondence between chords and cycles, we index them with the same index $\alpha$.

**Example:** Figure 2.4 shows how to obtain the cycles of $\mathcal{G}_{\mathrm{ex}}$ out of $\mathcal{T}^1_{\mathcal{G}_{\mathrm{ex}}}$. For instance, placing back the chord $e_4$, we obtain the cycle $\boldsymbol{c}^{\alpha=4}$. The cochord $e_1$ (dashed line in Figure 2.4) does not belong to $\boldsymbol{c}^{\alpha=4}$, it is thus removed. The orientation of $\boldsymbol{c}^{\alpha=4}$ is given by the orientation of $e_4$ in $\mathcal{G}_{\mathrm{ex}}$. The edges that belong to $\boldsymbol{c}^{\alpha=4}$ are $(e_2, -e_3, e_4)$. Thus the cycle vector of $\boldsymbol{c}^{\alpha=4}$ reads:

$$
\boldsymbol{c}^{\alpha=4} = \begin{pmatrix} 0 \\ 1 \\ -1 \\ 1 \\ 0 \end{pmatrix} \tag{2.8}
$$

**Cocycle**  Removing a cochord from $\mathcal{T}_\mathcal{G}$ creates a cut, i.e. it separates $\mathcal{T}_\mathcal{G}$ into two disconnected components called islands. We name respectively source (target) island the island that contains the source (target) vertex of the generating cochord. Then, a cocycle is defined as the set of edges in $\mathcal{G}$ which reconnects the two islands, from source to target. Accordingly the cocycle vectors $\boldsymbol{c}^\gamma \in \mathcal{E}$ are formally defined as vector with components:

$$
c^\gamma_e = \begin{cases} 1 & \text{if } e \text{ belongs to the cocycle generated by the cochord } \gamma \\ -1 & \text{if } -e \text{ belongs to the cocycle generated by the cochord } \gamma \\ 0 & \text{otherwise} \end{cases} \tag{2.9}
$$

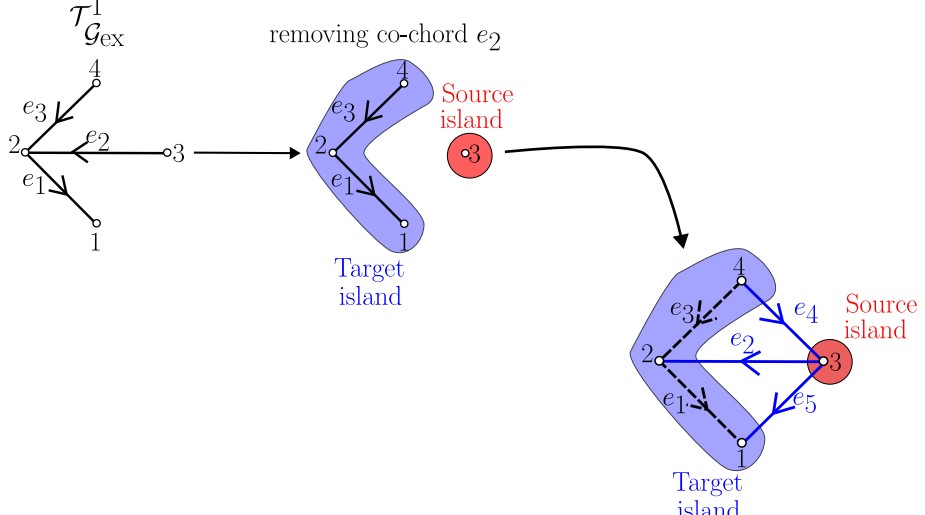

Figure 2.5: Blue edges: cocycle built out of the removal of one cochord from $\mathcal{T}^1_{\mathcal{G}_{\text{ex}}}$.

Given the one-to-one correspondence between cochords and cocycles, we index them with the same index $\gamma$.

**Example:** Figure 2.5 shows how to obtain a cocycle of $\mathcal{G}_{\text{ex}}$. We start by removing the cochord $e_2$ from $\mathcal{T}^1_{\mathcal{G}_{\text{ex}}}$. This creates a cut in the graph, with the two vertices of $e_2$, here $s(e_2) = 3$ and $t(e_2) = 2$, belonging to two disconnected components. Accordingly, we name source island the disconnected component containing vertex 3 and target island the component containing vertex 2. Then, the cocycle generated by $e_2$, denoted as $\boldsymbol{c}^{\gamma=2}$, is the set of edges that reconnects $\mathcal{G}_{\text{ex}}$ from source to target. In this case, these edges are $(e_2, -e_4, e_5)$ and the cocycle vector of $\boldsymbol{c}^{\gamma=2}$ reads:

$$c^{\gamma=2} = \begin{pmatrix} 0 \\ 1 \\ 0 \\ -1 \\ 1 \end{pmatrix} \tag{2.10}$$

Several remarks go as follows:

- $\forall\, \alpha,\ \boldsymbol{c}^{\alpha} \in \ker \boldsymbol{D}$. This means that for all $\alpha$, the following property holds:

$$\sum_e \boldsymbol{D}_{ie} c^{\alpha}_e = 0 \quad \forall i \tag{2.11}$$

  which encodes the fact that, along a cycle, every vertex has exactly one incoming and one outgoing edge.

- All cocycle vectors are linearly independent since, by construction, every cochord belongs to exactly one cocycle. Hence:

$$\boldsymbol{c}^{\gamma} \cdot \mathbf{e}^{\gamma'} = \delta_{\gamma,\gamma'} \quad \forall\ 1 \le \gamma, \gamma' \le N - 1 \tag{2.12}$$

  where $\mathbf{e}^{\gamma}$ is the canonical vector of $\mathcal{E}$ associated with the cochord $\gamma$ so that $\mathrm{e}^{\gamma}_e = \sum_e \delta_{\gamma,e}$.

- All cycle vectors are linearly independent since, by construction, every chord belongs to exactly one cycle. Reformulating this in mathematical terms, we have:

$$\boldsymbol{c}^{\alpha} \cdot \mathbf{e}^{\alpha'} = \delta_{\alpha,\alpha'} \quad \forall \ N \leq \alpha, \alpha' \leq E \tag{2.13}$$

where $\mathbf{e}^{\alpha}$ is the canonical vector of $\mathcal{E}$ associated with the chord $\alpha$ so that $\mathrm{e}_e^{\alpha} \equiv \sum_e \delta_{\alpha,e}$.

- Cycles and cocycles span orthogonal sets, namely:

$$\boldsymbol{c}^{\alpha} \cdot \boldsymbol{c}^{\gamma} = 0 \ \forall \ \alpha, \gamma \tag{2.14}$$

This follows from the definition of cycles and cocycles which ensures that $(\pm e_{\gamma}, e_{\alpha}) \in \boldsymbol{c}^{\alpha}$ iff $(e_{\gamma}, \mp e_{\alpha}) \in \boldsymbol{c}^{\gamma}$.

It follows that cycles and cocycles $(\boldsymbol{c}^{\gamma}, \boldsymbol{c}^{\alpha})$ form a basis for $\mathcal{E}$. Specifically cycles span $\ker \boldsymbol{D}$ and cocycles span its orthogonal complement space, the coimage of $\boldsymbol{D}$, $\mathrm{im} \boldsymbol{D}^{\top}$. We hereafter summarize the major algebraic properties encountered so-far:

$$\ker \boldsymbol{D} \perp \mathrm{im} \boldsymbol{D}^{\top} \tag{2.15}$$

$$\ker \boldsymbol{D} = \mathrm{span} \, \boldsymbol{c}^{\alpha} \tag{2.16}$$

$$\mathrm{im} \boldsymbol{D}^{\top} = \mathrm{span} \, \boldsymbol{c}^{\gamma} \tag{2.17}$$

## 2.3   Network theory applied to CTMC: the Markov chain tree theorem

In this subsection, we are interested in the case in which the graph $\mathcal{G}$ describes a CTMC. Within this framework, it is often of interest to look for the steady-state distribution of the CTMC as it gives a first insight into the dynamics of the system. The Markov chain tree theorem [16] gives a nice interpretation of the stationary state of a Markov chain in terms of algebraic graph theory. The outline of this subsection is the following: first we recall properties of the Markovian dynamics. Then, we recollect the necessary linear algebra basics needed to prove the theorem. Lastly, we provide a proof.

### 2.3.1   Markovian dynamics in a nutshell

For a more complete introduction to Markovian dynamics and its application to stochastic thermodynamics, the reader can refer to [17].
A Markovian dynamics on a connected graph $\mathcal{G}$ satisfies the master equation:

$$\begin{aligned} \partial_t p_i(t) &= \sum_j \left[ r(i|j) p_j(t) - r(j|i) p_i(t) \right] \\ &= \sum_e \boldsymbol{D}_{ie} j_e(t) \end{aligned} \tag{2.18}$$

where $\boldsymbol{D}$ is the incidence matrix of $\mathcal{G}$, $r(i|j)$ is the transition probability rate to jump from $j$ to $i$, and we have introduced the net probability flux along the oriented edge $e$ as:

$$j_e \equiv r(e) p_{s(e)} - r(-e) p_{t(e)} \tag{2.19}$$

where $r(e) \equiv r(t(e)|s(e))$ and $r(-e) \equiv r(s(e)|t(e))$. The second equality in Eq. (2.18) shows that, apart from a minus sign, the master equation can be interpreted as a continuity

equation for the probabilities $p_i$'s, with $\boldsymbol{D}$ a discrete divergence operator. The master equation can also be written as follows:

$$\partial_t p_i(t) = \sum_j \boldsymbol{R}_{ij} p_j(t) \tag{2.20}$$

where we have introduced the $N \times N$ rate matrix defined as:

$$\boldsymbol{R}_{ij} = r(i|j) - \left( \sum_k r(k|i) \right) \delta_{ij} \tag{2.21}$$

Then, probability conservation, namely $\sum_i p_i(t) = 1 \ \forall t$, is ensured by the fact that the elements of the columns of $\boldsymbol{R}$ sum to 0:

$$\sum_i \boldsymbol{R}_{ij} = 0 \ \forall j \tag{2.22}$$

Eq. (2.22) is a null linear combination of the lines of $\boldsymbol{R}$. It implies that the row vector $\boldsymbol{\ell}^\top$ [that we already met] is a left null eigenvector of $\boldsymbol{R}$, and that $\boldsymbol{R}$ has determinant 0. Since the graph is connected, Perron-Frobenius theorem ensures the uniqueness of the left null eigenvector, as well as the existence of a unique stationary state [18], since $\dim(\ker \boldsymbol{R}) = \dim(\ker \boldsymbol{R}^\top) = 1$. Accordingly, this stationary solution $\boldsymbol{p}_\infty$ is obtained as a right null eigenvector of $\boldsymbol{R}$:

$$\sum_j \boldsymbol{R}_{ij} p_j(\infty) = 0 \tag{2.23}$$

Then one way to obtain the stationary probability distribution of our CTMC is to directly compute the right null eigenvector of $\boldsymbol{R}$. However, we will see that a graph theoretical expression of $\boldsymbol{p}(\infty)$ can also be found.

### 2.3.2 Markov chain tree theorem

As introduced in 2.2, let $\mathcal{T}$ be the set of un-rooted spanning tree of $\mathcal{G}$ and $\mathcal{T}^i$ be the corresponding set of $i$-rooted spanning trees. Then, we define the weight vector $\boldsymbol{w} \in \mathcal{V}$ with components:

$$w_i = \sum_{\mathcal{T}} w(\mathcal{T}^i) = \sum_{\mathcal{T}} \prod_{e \in \mathcal{T}^i} r(e) \ \forall i \tag{2.24}$$

given by the sum over all spanning trees of the product of their transition rates with the edges oriented toward $i$. The Markov chain tree theorem states that the stationary distribution is given by the following formula:

$$p_i(\infty) = \frac{w_i}{\sum_j w_j} \tag{2.25}$$

**Example:** Let us illustrate the theorem by computing the stationary probability of a CTMC on a graph $\mathcal{G}$ with vertex space $\mathcal{V} = (1, 2, 3)$ and edge space $\mathcal{E} = (e_1, e_2, e_3)$ (See Figure 2.6). According to the Markov tree chain theorem, to compute the stationary probability to be in state $i$, we need to compute the weight $w_i$. To do so, we need to find all the spanning trees of $\mathcal{G}$ rooted in state $i$. Figure 2.6 shows for instance the three spanning trees rooted in state 2. Then the weight $w_2$ is obtained by summing, among the possible rooted spanning trees, the products of the transition rates. Here, for state 2, we thus have:

$$w_2 = r(1|3)r(2|1) + r(2|1)r(2|3) + r(3|1)r(2|3) \tag{2.26}$$

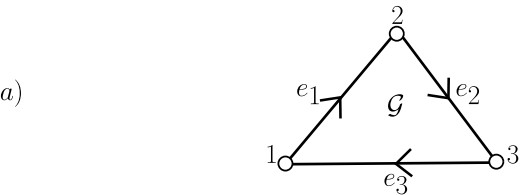

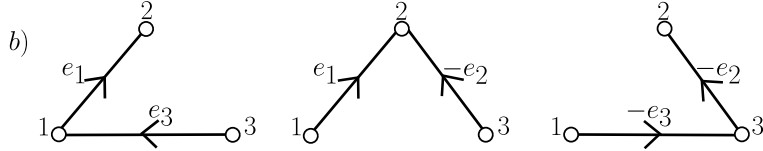

Figure 2.6: a) Oriented graph $G$ with 3 vertices. b) Set of 2-rooted spanning trees of $\mathcal{G}$.

The process can be repeated for states 1 and 3 to obtain $w_1, w_3$. By normalizing Eq. (2.26), we then obtain the stationary probability to be in state 2:

$$p_2(\infty) = \frac{r(1|3)r(2|1) + r(2|1)r(2|3) + r(3|1)r(2|1)}{Z} \tag{2.27}$$

where $Z = \sum_i w_i$.

### 2.3.3 Proof of the theorem

There exists no trivial proof of the above theorem. In this subsection, we propose a compact proof for CTMC following the spirit of [19].

**Linear algebra** We start this proof subsection with a short reminder of linear algebra. Given a square matrix $\boldsymbol{A}$, the cofactor matrix of $\boldsymbol{A}$ denoted $\boldsymbol{C_A}$ is defined as follows:

$$(\boldsymbol{C_A})_{ij} = (-1)^{i+j} (\boldsymbol{M_A})_{ij} \tag{2.28}$$

where $(\boldsymbol{M_A})_{ij}$ is the determinant of the matrix obtained by removing line $i$ and column $j$ from matrix $\boldsymbol{A}$, also called minor of $\boldsymbol{A}$. It can thus be written as $(\boldsymbol{M_A})_{ij} \equiv \det \boldsymbol{A}_{\setminus(i,j)}$. The adjugate matrix is defined as the transpose of $\boldsymbol{C_A}$:

$$\mathbf{adj}\boldsymbol{A} = (\boldsymbol{C_A})^\top \tag{2.29}$$

We will use the following property of the adjugate matrix:

$$\boldsymbol{A}\,\mathbf{adj}\boldsymbol{A} = \mathbf{adj}\boldsymbol{A}\,\boldsymbol{A} = \det \boldsymbol{A}\,\mathbb{1} \tag{2.30}$$

where $\mathbb{1}$ is the identity matrix. Finally, we introduce the Cauchy-Binet formula that will be used later on. Let $\boldsymbol{A}$ and $\boldsymbol{B}$ be two conformable matrices of respective size $m \times n$ and $n \times m$. The Cauchy-Binet formula reads:

$$\det \boldsymbol{A}\boldsymbol{B} = \sum_{\mathcal{S}_m^n} \det \boldsymbol{A}^{(\mathcal{S}_m^n)} \det \boldsymbol{B}^{\top(\mathcal{S}_m^n)} \tag{2.31}$$

where $\mathcal{S}_m^n$ denotes the sub-ensembles of $m$ elements out of $(1, ..., n)$. The cardinality of such sub-ensembles is the binomial coefficient: $\binom{n}{m}$. Then, $\boldsymbol{A}^{(\mathcal{S}_m^n)}$ corresponds to the $m \times m$ matrix whose columns are the columns of $\boldsymbol{A}$ indexed by $\mathcal{S}_m^n$, and the same holds for $\boldsymbol{B}^\top$.

**Usefulness of the adjugate matrix:** As a first lemma, we show that the diagonal of the adjugate matrix of $\boldsymbol{R}$ is related to $\boldsymbol{p}(\infty)$. Since $\det \boldsymbol{R} = 0$, Eq. (2.30) applied to the rate matrix gives:

$$\boldsymbol{R}\,\mathbf{adj}\boldsymbol{R} = \mathbb{0}_N \tag{2.32}$$

where we have denoted $\mathbb{0}_N$ the null matrix of size $N \times N$. Therefore, every column of the adjugate matrix is a right null eigenvector of $\boldsymbol{R}$. Given that $\dim(\ker \boldsymbol{R}) = 1$, it means that all columns of $\mathbf{adj}\boldsymbol{R}$ are proportional to each other. Moreover, using the second equality of Eq. (2.30), we have:

$$\boldsymbol{R}^\top \left(\mathbf{adj}\boldsymbol{R}\right)^\top = \mathbb{0}_N \tag{2.33}$$

meaning that the column vectors of $\left(\mathbf{adj}\boldsymbol{R}\right)^\top$ are in $\ker \boldsymbol{R}^\top$. This in turn implies that every vector of $\ker \boldsymbol{R}^\top$ is proportional to $\boldsymbol{\ell}$. Therefore, denoting $\boldsymbol{v}^i$ the $i$-th column vector of $\left(\mathbf{adj}\boldsymbol{R}\right)^\top$, we have $\forall\, i\ \boldsymbol{v}^i = \alpha_i \boldsymbol{\ell}$ , with $\alpha_i \in \mathbb{R}$. It follows that every columns of $\mathbf{adj}\boldsymbol{R}$ is equal to $(\alpha_1, ..., \alpha_N)^T$. In particular, they are all equal to the vector given by the diagonal entries of $\mathbf{adj}\boldsymbol{R}$. Since this vector lives in $\ker \boldsymbol{R}$, we obtain from it the stationary probability by imposing normalization:

$$p_i(\infty) = \frac{(\mathbf{adj}\boldsymbol{R})_{ii}}{\sum_i (\mathbf{adj}\boldsymbol{R})_{ii}} \tag{2.34}$$

We have shown how the adjugate matrix relates to the stationary distribution of our CTMC. We now turn to the task of drawing a connection to graph theory.

**Connection with graph theory:** We have shown in the last paragraph that the columns of $\mathbf{adj}\boldsymbol{R}$ are all equal. We may thus write:

$$(\mathbf{adj}\boldsymbol{R})_{ij} = (\mathbf{adj}\boldsymbol{R})_{ii} = (\boldsymbol{M}_{\boldsymbol{R}})_{ii} \tag{2.35}$$

where the second equality originates from the definition of the adjugate matrix Eq. (2.29). The connection to graph theory goes as follows. First, we have seen in subsection 2.3.1 that the dynamics of the CTMC can be described either in the space of probabilities $p_i$ with the rate matrix $\boldsymbol{R}$ or in the space of probability fluxes $j_e$ with the matrix $\boldsymbol{D}$. We now show how to relate the two above matrices. First we introduce the matrix $\mathcal{R}$ defined as:

$$\mathcal{R}_{i,e} = r(-e)\delta_{i,t(e)} - r(e)\delta_{i,s(e)} \tag{2.36}$$

Then, we realize that $\boldsymbol{R}$ can be written in terms of $\mathcal{R}$ and $\boldsymbol{D}$ as follows:

$$\boldsymbol{R} = \boldsymbol{D}\mathcal{R}^\top \tag{2.37}$$

Eq. (2.37) makes explicit the relation between the rate matrix of the CTMC and the incidence matrix $\boldsymbol{D}$ containing information on the topology of the underlying network. Taking back equation (2.35), we have:

$$\begin{aligned}
(\mathbf{adj}\boldsymbol{R})_{ii} = (\mathbf{adj}\boldsymbol{R})_{ij} &= (\boldsymbol{M}_{\boldsymbol{R}})_{ii} \\
&= \det \boldsymbol{D}_{\setminus(i,.)} \mathcal{R}^\top_{\setminus(.,i)} \\
&= \sum_{\mathcal{S}^E_{N-1}} \det \boldsymbol{D}^{(\mathcal{S}^E_{N-1})}_{\setminus(i,.)} \det \mathcal{R}^{(\mathcal{S}^E_{N-1})}_{\setminus(.,i)} \\
&= \sum_{\mathcal{T}} \det \boldsymbol{D}^{(\mathcal{T})}_{\setminus(i,.)} \det \mathcal{R}^{(\mathcal{T})}_{\setminus(.,i)}
\end{aligned} \tag{2.38}$$

In the second equality we have replaced $\boldsymbol{R}$ by its definition in terms of $\boldsymbol{D}$ and $\mathcal{R}$. To compute $(\boldsymbol{M}_{\boldsymbol{R}})_{ii}$, we have to remove line $i$ and column $i$ of $\boldsymbol{R}$, which is equivalent to

removing line $i$ of $\boldsymbol{D}$ and column $i$ of $\mathcal{R}^\top$, indicated respectively as $\boldsymbol{D}_{\setminus(i,.)}$ and $\mathcal{R}^\top_{\setminus(.,i)}$. In the third line, we have used the Cauchy-Binet formula introduced earlier. Here the sub-ensembles are the choices of $N-1$ columns of $\boldsymbol{D}_{\setminus(i,.)}$ and $\mathcal{R}_{\setminus(.,i)}$. One then has to compute the determinant of the resulting $(N-1) \times (N-1)$ square matrices. Crucially, the only choices that contribute to the determinant are the ones in which the chosen $N-1$ elements are linearly independent. In graph theory, the set of choices of independent edges is the set of spanning trees. Therefore, the sum on $\mathcal{S}^E_{N-1}$ can be changed on a sum over $\mathcal{T}$. For all $\mathcal{T}$, matrices $\boldsymbol{D}^{(\mathcal{T})}_{\setminus(i,.)}$ and $\mathcal{R}^{(\mathcal{T})}_{\setminus(.,i)}$ are full-rank and can be made upper triangular using elementary row and column operations. Notably, given the common structure of $\boldsymbol{D}$ and $\mathcal{R}$ (see Eq. (2.1) and Eq. (2.36)), the required elementary operations are the same for the two matrices, leaving the sign of the product of determinants unchanged. Once the two matrices are reduced in upper triangular form, their determinant is given by the product of the diagonal entries. By an adequate labeling of the lines and columns, $\boldsymbol{D}^{(\mathcal{T})}_{\setminus(i,.)}$ and $\mathcal{R}^{(\mathcal{T})}_{\setminus(.,i)}$ can be made upper triangular with respectively $-1$ and $-r(e)$ diagonal entries for every edge directed inward vertex $i$, and $+1$ and $+r(-e)$ diagonal entries for every edge directed outward vertex $i$ (see Appendix A in [20] for details). It follows that the resulting product of rates corresponds to a sum over spanning trees rooted in $i$ and we may write:

$$(\mathbf{adj}\boldsymbol{R})_{ii} = \sum_{\mathcal{T}} \prod_{e \in \mathcal{T}^i} r(e) = w_i \tag{2.39}$$

Finally, ensuring the normalization of the probabilities $p_i$'s completes the proof.

## 2.4  Network theory applied to observable decomposition

In this subsection, we apply the notions of cycles and cocycles introduced in subsubsection 2.2.2 to the decomposition of physical observables on $\mathcal{G}$. Thermodynamics imposes strong conditions on the decomposition. We show in particular that we are able to locate thermodynamic processes on subsets of $\mathcal{G}$.

### 2.4.1  Observables decomposition

**Fluxes**   We first recall that $(\boldsymbol{c}^\gamma, \boldsymbol{c}^\alpha)$, as well as $(\boldsymbol{e}^\gamma, \boldsymbol{c}^\alpha)$ and $(\boldsymbol{c}^\gamma, \boldsymbol{e}^\alpha)$ form bases of the edge space $\mathcal{E}$, due to the orthogonality conditions in Eqs. (2.13-2.12). Therefore, since the vector of probability fluxes $\boldsymbol{j}(t)$, whose components are given by Eq. (2.19), belongs to $\mathcal{E}$, we may decompose it as follows:

$$\boldsymbol{j}(t) = \sum_{\gamma=1}^{N-1} \mathcal{J}_\gamma(t) \boldsymbol{e}^\gamma + \sum_{\alpha=N}^{E} \mathcal{J}_\alpha(t) \boldsymbol{c}^\alpha \tag{2.40}$$

where coefficients $\mathcal{J}_\gamma(t) \equiv \boldsymbol{j}(t) \cdot \boldsymbol{c}^\gamma$ and $\mathcal{J}^c_\alpha(t) \equiv \boldsymbol{j}(t) \cdot \boldsymbol{e}^\alpha$, and we used the orthogonality properties Eqs. (2.13-2.12). Therefore, coefficients $\mathcal{J}_\gamma(t)$ for $1 \leq \gamma \leq N-1$ in Eq. (2.40) correspond to the $N-1$ probability fluxes flowing across the cocycles. More can be said about these coefficients. Indeed, at steady-state, the master equation (2.18) reduces to:

$$\boldsymbol{D}\boldsymbol{j}(\infty) = 0 \tag{2.41}$$

Thus, at steady state $\boldsymbol{j}(\infty)$ lives in the kernel of $\boldsymbol{D}$. Since cycles form a basis of $\ker\boldsymbol{D}$, the only term that survives at steady state is the cycle term of the decomposition, so that:

$$\boldsymbol{j}(\infty) = \sum_{\alpha=N}^{E} \mathcal{J}_\alpha(\infty) \boldsymbol{c}^\alpha \tag{2.42}$$

This last equation is precisely the Kirchoff Current Law (KCL) which implies that $N-1$ cocycle fluxes are transient and vanish in the steady state, $\mathcal{J}_\gamma(t \to \infty) \to 0 \, \forall \gamma$. Therefore, the analysis of the steady-state fluxes alone only requires the knowledge of a reduced number of degrees of freedom on the graph, whose number is given by the cardinality of cycles $c$.

**Forces** We now do the same for the affinity vector $\boldsymbol{A}(t) \in \mathcal{E}$ whose components are defined as:

$$A_e(t) = \log\left(\frac{r(e)p_{s(e)}(t)}{r(-e)p_{t(e)}(t)}\right) \tag{2.43}$$

We put forward the following decomposition for the affinity vector:

$$\boldsymbol{A}(t) = \sum_{\gamma=1}^{N-1} \mathcal{A}_\gamma(t)\boldsymbol{c}^\gamma + \sum_{\alpha=N}^{E} \mathcal{A}_\alpha(t)\boldsymbol{e}^\alpha \tag{2.44}$$

where coefficients $\mathcal{A}_\gamma(t) \equiv \boldsymbol{A}(t) \cdot \boldsymbol{e}^\gamma$ and $\mathcal{A}_\alpha(t) \equiv \boldsymbol{A}(t) \cdot \boldsymbol{c}^\alpha$, and we used the orthogonality properties Eqs. (2.13-2.12). Therefore, the coefficients $\mathcal{A}_\alpha$ for $N \le \alpha \le E$ corresponds to the cycle affinities, i.e. the affinities summed along cycles. Let us compute them explicitly by making use of Eq. (2.43):

$$
\begin{aligned}
\forall \alpha, \; \mathcal{A}_\alpha(t) &= \boldsymbol{A}(t) \cdot \boldsymbol{c}^\alpha \\
&= \sum_e A_e(t) c_e^\alpha \\
&= \sum_e c_e^\alpha \log\left(\frac{r(e)p_{s(e)}(t)}{r(-e)p_{t(e)}(t)}\right) \\
&= \log\left(\frac{\prod_e \left(r(e)p_{s(e)}(t)\right)^{c_e^\alpha}}{\prod_e \left(r(-e)p_{t(e)}(t)\right)^{c_e^\alpha}}\right) \\
&= \log\left(\frac{\prod_e (r(e))^{c^\alpha}}{\prod_e (r(-e))^{c_e^\alpha}}\right)
\end{aligned}
\tag{2.45}
$$

where the first equality was obtained using Eqs. (2.13) and (2.14). To go from the fourth to the fifth equality, we used the fact that the probabilities are vertex quantities and therefore their product along a cycle does not depend on the orientation. Several remarks:

- The cycle affinities only depend on the transition rates.

- If $\mathcal{A}_\alpha = 0 \; \forall \alpha$, then Eq. (2.45) ensures that the Kolmogorov criterion is fulfilled, namely that the product of transition rates across all cycles is the same in the forward and backward directions. This is a sufficient and necessary condition for the Markov chain to be reversible. It is also equivalent to the detailed balance property.

- The condition $\mathcal{A}_\alpha = 0 \; \forall \, \alpha$ is also called Kirchoff voltage law (KVL).

- For a reversible dynamics, i.e. when KVL is satisfied, all the affinities are conservative, which implies the existence of a potential vector $\boldsymbol{V}(t) \in \mathcal{V}$ such that:

$$\boldsymbol{A}(t) = \sum_\gamma \mathcal{A}_\gamma(t)\boldsymbol{c}^\gamma = -\boldsymbol{D}^\top \boldsymbol{V}(t) \tag{2.46}$$

where we made use of the decomposition Eq. (2.44) and in the last equality we used the fact that cocycles span the $\mathrm{im}\boldsymbol{D}^\top$. Given that $\boldsymbol{D}^\top$ can be seen as a discrete gradient, Eq. (2.46) is a potential condition for conservative affinities.

- As a last remark, if fluxes fulfill KCL, namely $\boldsymbol{j}(t) \in \ker \boldsymbol{D}$, and affinities fulfill KVL, namely $\boldsymbol{A}(t) \in \operatorname{im} \boldsymbol{D}^\top$, then Tellegen's theorem is satisfied and we have:

$$\boldsymbol{j}(t) \cdot \boldsymbol{A}(t) = \sum_{\alpha,\gamma} \mathcal{J}_\alpha \mathcal{A}_\gamma \boldsymbol{c}^\alpha \cdot \boldsymbol{c}^\gamma = 0 \qquad (2.47)$$

**Entropy production**   Using the decomposition for the fluxes and for the affinities introduced above and the orthogonality relations expressed in subsection 2.2.2, we obtain a decomposition for the entropy production rate (EPR) as follows:

$$
\begin{aligned}
\sigma(t) &\equiv \boldsymbol{A}(t) \cdot \boldsymbol{j}(t) \\
&= \left( \sum_\gamma \mathcal{A}_\gamma(t) \boldsymbol{c}^\gamma + \sum_\alpha \mathcal{A}_\alpha(t) \boldsymbol{e}^\alpha \right) \cdot \left( \sum_{\gamma'} \mathcal{J}_{\gamma'}(t) \boldsymbol{e}^{\gamma'} + \sum_{\alpha'} \mathcal{J}_{\alpha'}(t) \boldsymbol{c}^{\alpha'} \right) \\
&= \sum_\gamma \sum_{\gamma'} \mathcal{A}_\gamma(t) \mathcal{J}_{\gamma'}(t) \underbrace{\boldsymbol{c}^\gamma \cdot \boldsymbol{e}^{\gamma'}}_{=\delta_{\gamma,\gamma'}} + \sum_\gamma \sum_{\alpha'} \mathcal{A}_\gamma(t) \mathcal{J}_{\alpha'}(t) \underbrace{\boldsymbol{c}^\gamma \cdot \boldsymbol{c}^{\alpha'}}_{=0} \\
&\quad + \sum_\alpha \sum_{\gamma'} \mathcal{A}_\alpha(t) \mathcal{J}_{\gamma'}(t) \underbrace{\boldsymbol{e}^\alpha \cdot \boldsymbol{e}^{\gamma'}}_{=0} + \sum_\alpha \sum_{\alpha'} \mathcal{A}_\alpha(t) \mathcal{J}_{\alpha'}(t) \underbrace{\boldsymbol{e}^\alpha \cdot \boldsymbol{c}^{\alpha'}}_{\delta_{\alpha,\alpha'}} \\
&= \sum_\gamma \mathcal{J}_\gamma(t) \mathcal{A}_\gamma(t) + \sum_\alpha \mathcal{J}_\alpha(t) \mathcal{A}_\alpha(t)
\end{aligned}
\qquad (2.48)
$$

Thus there are again two cases:

- At steady-state, we know that the only remaining fluxes are the fluxes across the cycles $\mathcal{J}_\alpha(t)$. Therefore, in the long-time limit, the EPR simplifies to:

$$\sigma(\infty) = \sum_\alpha \mathcal{J}_\alpha(\infty) \mathcal{A}_\alpha(\infty) \qquad (2.49)$$

- If the chain is reversible we have seen that $\mathcal{A}_\alpha = 0 \, \forall \alpha$, therefore the EPR reduces to:

$$\sigma(t) = \sum_\gamma \mathcal{J}_\gamma(t) \mathcal{A}_\gamma(t) \qquad (2.50)$$

and it vanishes in the long-time limit since $\mathcal{J}_\gamma(t \to \infty) \to 0 \, \forall \gamma$.

As a last remark, if the chain is both at steady state and reversible, i.e. if both KCL and KVL are fulfilled, then both terms in Eq. (2.48) vanish. The system is in an equilibrium steady-state with zero entropy production:

$$\sigma(t) = 0 \, \forall t \qquad (2.51)$$

## 2.5   An example of chemical reaction network

As a summary, we apply the Markov chain tree formula and the observable decomposition to the example of a chemical reaction network (CRN) inspired from biochemistry and shown in Figure 2.7. The reaction network describes the activation of a substrate $S \to S^*$ via the enzyme $E$ that binds to the substrate to form the complex $ES$. The latter complex also decomposes in the free enzyme and the new substrate $S^*$. A first observation is that the chemical network is non-linear and cannot be represented as a simple graph. Specifically, reactions 1 and 3 in panel $a)$ of Figure 2.7 involve three species and cannot be represented as simple edges connecting a single vertex (species) to another one. This is

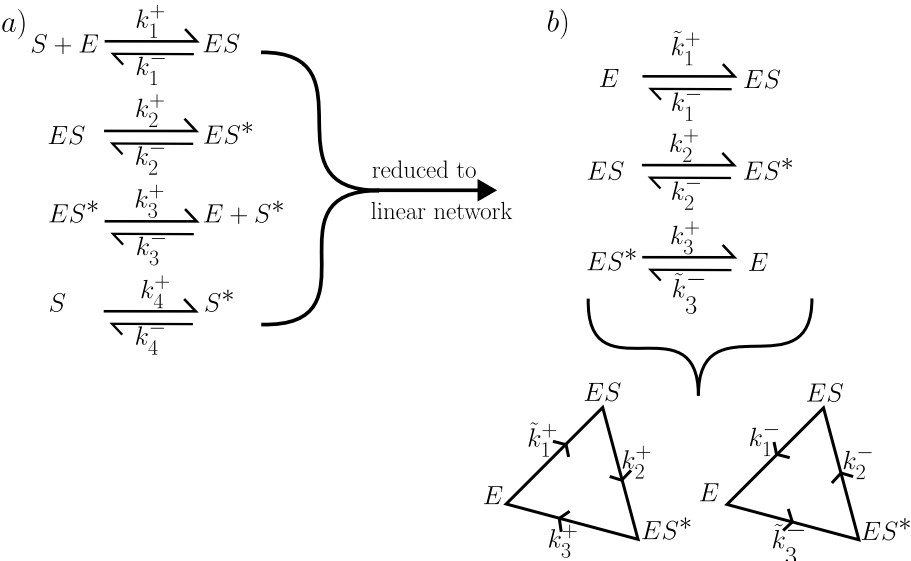

Figure 2.7: An example of CRN. a) Reaction 1 requires the enzyme $E$ to bind to the substrate $S$ to generate the complex $ES$. Similarly, reaction 3 requires the enzyme $E$ to bind to the substrate $S^*$ to form the complex $ES^*$. Consequently, the reaction network is not a graph and its dynamics is non linear. We need a hypergraph to represent it. b) By chemostating $S$ and $S^*$, the network becomes effectively linear. The concentration of $S$ and $S^*$ are no longer dynamical variables and they enter in the dynamics via the rescaled transition rates $\tilde{k}_1^+ \equiv k_1^+ x_S^{\text{chem}}$ and $\tilde{k}_3^- x_{S^*}^{\text{chem}}$.

typically the case in chemical reaction networks which involve interactions many-to-many and thus map to hypergraphs (we refer to [20] for a generalization of the graph-theoretical notions introduced here to hypergraphs). We assume in the following that the CRN follows the mass action kinetics and denote $x_i$ the concentration of species $i$. Using the mass action law, we can for instance write the kinetic equation for the concentration of $ES$ through reaction 1 and 2. It reads:

$$\frac{dx_{ES}}{dt} = k_1^+ x_S x_E - k_1^- x_{ES} - k_2^+ x_{ES} + k_2^- x_{ES^*} \tag{2.52}$$

Clearly, this kinetic equation is non linear due to the first term on the right-hand side. Here to illustrate the results developed in the previous subsections on graph theory, we choose to fix the concentration of $S$ and $S^*$ via external chemostatting. The chemostating is performed by connecting the system with two reservoirs of respectively $S$ and $S^*$, held at concentrations $x_S^{\text{chem}}$ and $x_{S^*}^{\text{chem}}$. Then, the concentrations of $S$ and $S^*$ no longer vary, and the network can be reduced effectively to a linear network on the states $(E, ES, ES^*)$, for which a graphical representation can be exploited (see Figure 2.7). Then, the kinetic equation (2.52) also reduces to a linear equation in the dynamical variables $x_E$ and $x_{ES}$:

$$\frac{dx_{ES}}{dt} = \underbrace{k_1^+ x_S^{\text{chem}}}_{\tilde{k}_1^+} x_E - k_1^- x_{ES} - k_2^+ x_{ES} + k_2^- x_{ES^*} \tag{2.53}$$

We are thus left with a linear continuity equation formally analogous to the master equation (2.18) where the $x_i(t)$'s are the analogue of the probabilities $p_i(t)$ and the $k_i$'s are the analogue of the transition rates $r(i|j)$. Figure 2.7 panel b) represents the graph that can be obtained out of our linearized set of kinetic equations. For CRNs, the analogous of the

incidence matrix is called stoichiometric matrix. Each column of the stoichiometric matrix corresponds to a reaction of the CRN. We denote $S$ the stoichiometric of our example which reads:

$$S = \begin{pmatrix} -1 & 0 & 1 \\ 1 & -1 & 0 \\ 0 & 1 & -1 \end{pmatrix} \tag{2.54}$$

Since the graph of the CRN is unicyclic, the unique cycle vector reads:

$$\boldsymbol{c} = \begin{pmatrix} 1 \\ 1 \\ 1 \end{pmatrix} \tag{2.55}$$

where we dropped the index $\alpha$. Since there is only one cycle, the force driving the CRN out of equilibrium is going to be the cycle affinity $\mathcal{A} = \boldsymbol{c} \cdot \boldsymbol{A}$. We can compute it as follows:

$$\begin{aligned}
\mathcal{A} = \boldsymbol{A} \cdot \boldsymbol{c} &= \log \left( \frac{\tilde{k}_1^+ x_E k_2^+ x_{ES} k_3^+ x_{ES^*}}{k_1^- x_{ES} k_2^- x_{ES^*} \tilde{k}_3^- x_E} \right) \\
&= \underbrace{\log \left( \frac{k_1^+ k_2^+ k_3^+}{k_1^- k_2^- k_3^-} \right)}_{1} + \underbrace{\log \left( \frac{x_S^{\text{chem}}}{x_{S^*}^{\text{chem}}} \right)}_{2}
\end{aligned} \tag{2.56}$$

We find two terms:

1. the first term depends only on the transition rates.

2. the second term depends only on the concentration of the reservoirs that we used to chemostat $S$ and $S^*$.

The first term of equation (2.56) vanishes whenever the transition rates fulfill the Kolmogorov criterion which states that the following ratio must be one:

$$\frac{k_1^+ k_2^+ k_3^+}{k_1^- k_2^- k_3^-} = 1 \tag{2.57}$$

The, $\mathcal{A}$ reduces to:

$$\mathcal{A} = \log \left( \frac{x_S^{\text{chem}}}{x_{S^*}^{\text{chem}}} \right) \tag{2.58}$$

and the out of equilibrium driving of our CRN relies only on the difference between the two chemostats connected to the CRN.

Finally, we can make the link with the Markov chain tree formula. Assuming Eq. (2.57) ensures that the chain is reversible and, in the absence of chemostatting, it relaxes to an equilibrium steady state. We then use a parameterization of the transition rates in terms of the standard chemical potential of thermodynamics $\mu_i^0$ of species $i$, namely:

$$\frac{k_e^+}{k_e^-} = e^{\mu_{s(e)}^0 - \mu_{t(e)}^0} \tag{2.59}$$

This also ensures the following property at the level of spanning trees:

$$\frac{w(\mathcal{T}^i)}{w(\mathcal{T}^j)} = e^{\mu_j^0 - \mu_i^0} \tag{2.60}$$

where $w(\mathcal{T}^i)$ is the product of the transition rates along the rooted spanning tree $\mathcal{T}^i$. Then, the Markov chain tree formula simplifies considerably leading to:

$$x_i(\infty) = \frac{w_i}{\sum_j w_j} = \frac{\sum_\mathcal{T} w(\mathcal{T}^i)}{\sum_j \sum_\mathcal{T} w(\mathcal{T}^j)} \tag{2.61}$$

$$= \frac{\sum_\mathcal{T} w(\mathcal{T}^i)}{\sum_\mathcal{T} w(\mathcal{T}^i) \sum_j e^{\mu_i^0 - \mu_j^0}} = \frac{e^{-\mu_i^0}}{\sum_j e^{-\mu_j^0}} \tag{2.62}$$

which is the equilibrium solution of thermodynamics.

# 3 Stochastic Thermodynamics

**Massimiliano Esposito and Danilo Forastiere.**[6] *These lecture notes contain an informal introduction to stochastic thermodynamics, which is a dynamical theory of mesoscopic systems in contact with external reservoirs which is fully compatible with statistical mechanics, and reproduces its main results when the full system is allowed to reach thermodynamic equilibrium. The stochastic dynamics of discrete systems is introduced in Section 3.1. For simplicity, in Section 3.2 we develop the stochastic dynamics and thermodynamics of a small system in contact with a single ideal reservoir in equilibrium, and at the average level. Then, in Section 3.3 we explain how to treat genuine nonequilibrium situations, in which the external reservoirs are characterized by different temperatures and therefore preclude the system from reaching equilibrium. Section 3.4 explains how to extend the formalism to cases in which the mesoscopic states of the system have internal entropy and how to deal with transfer of matter between reservoirs. Finally, Section 3.5 introduces the fundamental tools needed to develop stochastic thermodynamics at the level of the single stochastic trajectories, namely the fluctuation theorems.*

## 3.1 Stochastic dynamics

We start from a system described by the Master Equation (with the graphical representation given in Fig. 3.1) with a possibly time-dependent generator $W$

$$\mathrm{d}_t p_i = \sum_j W_{ij} p_j = \sum_{j \neq i} \underbrace{(W_{ij}(t) p_j - W_{ji}(t) p_i)}_{J_{ij}} \tag{3.1}$$

where conservation of normalization implies $\sum_j W_{ji} = 0$ or equivalently $W_{ii} = -\sum_{j \neq i} W_{ji}$, and where $J_{ij}$ denotes the probability current from $j$ to $i$.

The steady state is obtained as right-null eigenvector of $W$ and thus satisfies the linear equation $W p^{ss} = 0$. It is unique for an ergodic Markov process[7] by the Perron-Frobenius theorem [2,21]. If a time-dependent driving protocol is applied from the environment, the generator becomes time-dependent; in this case, we can define the *instantaneous steady-state* $p^{ss}(t)$ which satisfies $\sum_i W_{ji}(t) p_i^{ss}(t) = 0$ at every time and for each $j$, and which becomes time-independent when the time-dependent driving is stopped. In absence of time-dependent driving every initial condition relaxes to $p^{ss}$, as it can be seen introducing the *relative entropy* (or *Kullback-Leibler divergence*, [13,22]), which is the positive function defined as

$$D(p|p') \equiv \sum_i p_i \ln \frac{p_i}{p'_i} \tag{3.2}$$

$$= -\sum_i p_i \ln \frac{p'_i}{p_i} \geq -\sum_i p_i \left(1 - \frac{p'_i}{p_i}\right) = 0. \tag{3.3}$$

The inequality follows from the normalization $\sum_i p_i = \sum_i p'_i = 1$ and from the fact that $-\ln x \geq 1 - x$ (which is easily obtained, for example, considering $\ln x = \int_1^x \mathrm{d}s \frac{1}{s} \leq \int_1^x \mathrm{d}s$ for $x > 1$ and analogously for $x < 1$, and which has the geometrical interpretation depicted in Fig. 3.2).

---

[6]ME was the lecturer; DF was the angel and wrote this chapter.
[7]Defined by an irreducible generator $W$.

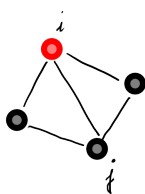

Figure 3.1: Graphical representation of the ME (3.1). In a small time interval $\mathrm{d}t$, a system in the state $i$ can jump to $j$ with a probability given by $W_{ji}\mathrm{d}t$.

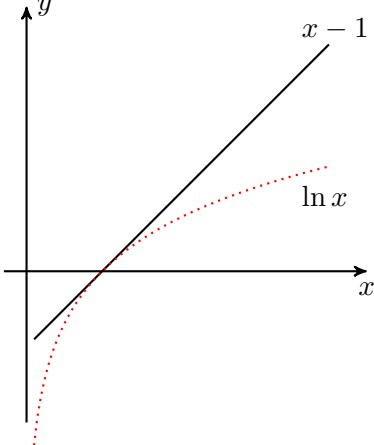

Figure 3.2: Graphical representation of the inequality $\ln x \leq x - 1$.

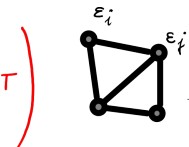

Figure 3.3: Closed system (no flux of matter) evolving at the fixed temperature $T$ of the environment. The transition rates obey local detail balance in the form (3.7) since the system needs relax to equilibrium in absence of driving.

The relative entropy $D(p|p^{ss})$ is zero only at the steady-state and its time-derivative is always negative, meaning that it is a Lyapunov function [23]. In fact

$$-\mathrm{d}_t D(p|p^{ss}(t)) = -\sum_i \mathrm{d}_t p_i \ln \frac{p_i}{p_i^{ss}} - \underbrace{\sum_i p_i \mathrm{d}_t \ln p_i}_{=0} + \sum_i p_i \mathrm{d}_t \ln p_i^{ss} . \tag{3.4}$$

The second term vanishes by normalization using the ME (3.1). The third term only occurs in presence of time-dependent driving protocols. On the other hand, the first term obeys

$$-\sum_i \mathrm{d}_t p_i \ln \frac{p_i}{p_i^{ss}} = -\sum_{ij} W_{ij} p_j \ln \frac{p_i}{p_i^{ss}} = -\sum_{ij} W_{ij} p_j \ln \frac{p_i p_j^{ss}}{p_i^{ss} p_j} \tag{3.5}$$

$$\geq \sum_{ij} W_{ij} p_j \left( 1 - \frac{p_i p_j^{ss}}{p_i^{ss} p_j} \right) = -\sum_i \frac{p_i}{p_i^{ss}} \sum_j W_{ij} p_j^{ss} = 0 \tag{3.6}$$

where we used the ME (3.1) in the first equality, the normalization property of the generator $\sum_i W_{ij} = 0$ in the second and fourth equalities, the inequality $-\ln \geq 1 - x$ and finally the definition of the steady state. Therefore, when no driving is present and $\mathrm{d}_t p^{ss} = 0$, we have the Lyapunov property $\mathrm{d}_t D(p|p^{ss}) \leq 0$.

As a side remark, notice that the matrix $\tilde{W}_{ij} \equiv W_{ij} \frac{p_j^{ss}}{p_i^{ss}}$ is a rate matrix ($\sum_i \tilde{W}_{ij} = 0$) which defines a different Master Equation rate matrix with the same steady state $p^{ss}$ as the original matrix $W$.

## 3.2   Stochastic thermodynamics for a single reservoir of energy

**Basic theory**   We assign the energies of the states $\epsilon_i(t)$, the system is closed and in contact with a reservoir at temperature T (therefore $\beta = \frac{1}{k_{\mathrm{B}} T}$), and we assume *local detailed balance* [24–26]:

$$\frac{W_{ij}}{W_{ji}} = e^{-\beta(\epsilon_i - \epsilon_j)} . \tag{3.7}$$

Examples of rates satisfying eq. (3.7) are given later in this section, and are graphically depicted in Fig. 3.4.

Consider the average energy $\langle E \rangle = \sum_i \epsilon_i(t) p_i(t)$. Its rate of change in times is given by

$$\mathrm{d}_t \langle E \rangle = \underbrace{\sum_i p_i \mathrm{d}_t \epsilon_i}_{\dot{W}} + \underbrace{\sum_i \epsilon_i \mathrm{d}_t p_i}_{\dot{Q}} \tag{3.8}$$

This corresponds to the First Law of thermodynamics. The quantities $\dot{W}$ and $\dot{Q}$ have the dimensions of energy over time but they are not derivatives with respect to time for a generic protocol. A remark on notation: in these notes we use $\mathrm{d}_t, \partial_t$ for total and partial derivatives and the overdot only to mean that a quantity has dimension of a rate of change, but without implying that it is a derivative.

The identification of the heat is motivated by the following identity

$$\dot{Q} = \sum_{ij} W_{ij} p_j \epsilon_i = \sum_{ij} W_{ij} p_j (\epsilon_i - \epsilon_j) = \frac{1}{2} \sum_{ij} J_{ij}(\epsilon_i - \epsilon_j), \tag{3.9}$$

obtained using the conservation of normalization $\sum_i W_{ij} = 0$ in the first equality and a symmetrization over nodes in the second. Inserting local detailed balance (3.7) in eq. (10.31) we get

$$-\frac{\dot{Q}}{T} = \frac{k_B}{2} \sum_{ij} J_{ij} \ln \frac{W_{ij}}{W_{ji}}, \tag{3.10}$$

from which we see that in a transition governed by LDB the entropy change in the reservoir is given by the heat divided by the temperature. The entropy of the system is

$$S = -k_{\mathrm{B}} \sum_i p_i \ln p_i. \tag{3.11}$$

Its time derivative is the entropy change in the system and reads

$$\mathrm{d}_t S = -k_B \sum_i \mathrm{d}_t p_i \ln p_i = -k_B \sum_{ij} W_{ij} p_j (\ln p_i - \ln p_j)$$
$$= -\frac{k_B}{2} \sum_{ij} J_{ij} \ln \frac{p_i}{p_j}. \tag{3.12}$$

Therefore the rate of entropy change in the universe (*i.e.* the entropy production rate) is obtained summing (3.10) and (3.12),

$$\dot{\sigma} \equiv \mathrm{d}_t S - \frac{\dot{Q}}{T} = \frac{k_B}{2} \sum_{ij} (W_{ij} p_j - W_{ji} p_i) \ln \frac{W_{ij} p_j}{W_{ji} p_i} \geq 0, \tag{3.13}$$

where the inequality follows from $(a - b) \ln \frac{a}{b} \geq 0$ for positive $a, b$. The entropy production is zero only when $p$ satisfies detailed balance $W_{ij} p_j^{\mathrm{eq}} = W_{ji} p_i^{\mathrm{eq}}$. The equilibrium state is a steady state as $\sum_j W_{ij} p_j^{\mathrm{eq}} = 0$ for which the canonical distribution holds. This can be obtained from the local detailed balance condition[8] (3.7) by summing over $j$ and imposing normalization

$$1 = \sum_j p_j^{\mathrm{eq}} = \sum_j \frac{W_{ji}}{W_{ij}} p_i^{\mathrm{eq}} = p_i^{\mathrm{eq}} e^{\beta \epsilon_i} \sum_j e^{-\beta \epsilon_j}, \tag{3.14}$$

which implies

$$p_i^{\mathrm{eq}} = \frac{e^{-\beta \epsilon_i}}{\sum_j e^{-\beta \epsilon_j}} \equiv e^{-\beta(\epsilon_i - F^{\mathrm{eq}})}, \tag{3.15}$$

where we introduced the equilibrium free energy $F^{\mathrm{eq}} \equiv -k_B T \ln \sum_j p_j^{\mathrm{eq}}$. Since the system is in contact with a single reservoir, in absence of driving over energies it will relax to equilibrium.

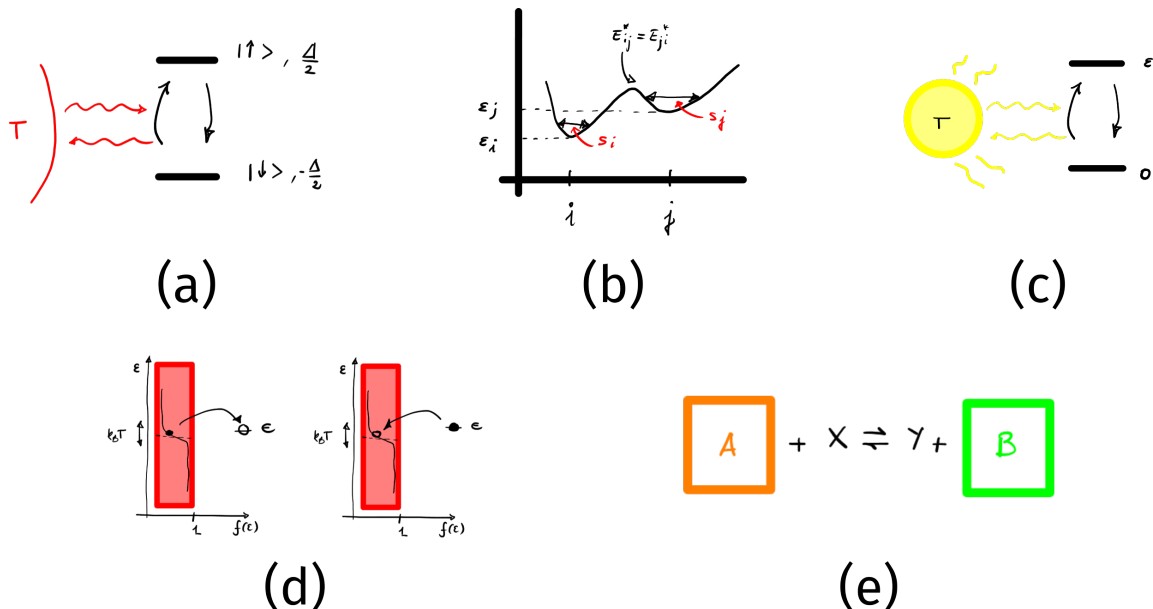

Figure 3.4: Illustration of systems characterized by different types of rates. (a) Spin-boson (b) Arrhenius (c) Bose (d) Fermi (e) Mass-action chemistry

**Examples of rates.** We list here some important examples of rates satisfying LDB. They characterize the systems depicted in Fig. 3.4.

**Spin boson.** The rates at which a spin in contact with a bosonic bath switches between up and down are given by

$$W_+ = \frac{\hbar}{4|\Delta|}\gamma(|\Delta|)(\coth\frac{\beta\Delta}{2} - 1)\,, \tag{3.16a}$$

$$W_- = \frac{\hbar}{4|\Delta|}\gamma(|\Delta|)(\coth\frac{\beta\Delta}{2} + 1)\,. \tag{3.16b}$$

The LDB condition

$$\ln\frac{W^+}{W^-} = -\beta\Delta \tag{3.17}$$

follows from the property $\coth\frac{x}{2} - 1 = \frac{2}{e^x - 1} = e^x(\coth\frac{x}{2} + 1)$. Drude-Ullersma model [2,27] is obtained setting $\gamma(\Delta) = \frac{2}{\pi}\frac{\alpha^2}{\alpha^2 + \Delta^2}$.

**Arrhenius rates.** These rates can be obtained as the coarse-graining of a diffusion process in a multi-well potential. See *e.g.* [2] (Ch. XIII.6) and [26].

$$W_{ij} = \Gamma\exp\left\{-\beta(E^*_{ij} - \phi_j)\right\}\,, \tag{3.18a}$$

$$W_{ji} = \Gamma\exp\left\{-\beta(E^*_{ji} - \phi_i)\right\}\,. \tag{3.18b}$$

The rates involve the free energy $\phi_i = \epsilon_i - Ts_i$, where the entropic contribution arises from the curvature near the minima of the energy [see Fig. 3.4 (b)]. The barrier between

---

[8]For the present case of a system in equilibrium with a single reservoir, local detailed balance and detailed balance are equivalent (see *e.g.* [2] Ch.4).

the two states is symmetric, $E_{ij}^* = E_{ji}^* = E^*$. The LDB (3.7) therefore takes the form

$$\ln \frac{W_{ij}}{W_{ji}} = -\beta(\phi_i - \phi_j)\,. \tag{3.19}$$

**Fermi rates.**   These rates describe the interaction of a two-level system with a bath obeying Fermi statistics. See [28].

Define $x = \epsilon - \mu$. The Fermi statistics is $f(x) = \frac{e^{-\beta x}}{e^{-\beta x}+1}$, therefore $1 - f(x) = \frac{1}{1+e^{-\beta x}}$. The Fermi rates are given by

$$W_+ = \Gamma f(\epsilon - \mu)\,, \tag{3.20a}$$
$$W_- = \Gamma\left(1 - f(\epsilon - \mu)\right)\,, \tag{3.20b}$$

satisfying (3.7).

**Bose rates.**   The Bose statistics is given by

$$n(x) = \frac{e^{-x}}{1 - e^{-x}}\,. \tag{3.21}$$

Correspondingly $1 + n(x) = \frac{1}{1-e^{-x}}$. The Bose rates are given by

$$W_+ = \Gamma n(\epsilon - \mu)\,, \tag{3.22a}$$
$$W_- = \Gamma\left(1 + n(\epsilon - \mu)\right)\,, \tag{3.22b}$$

satisfying (3.7).

**Mass-action chemistry.**   Consider the chemical reaction $A + X \rightleftharpoons Y + B$ occurring in a container of volume $V$, where the concentrations of A and B are externally maintained constant by a chemostatting mechanism [29]. The mass-action transition rates $W_{(N_X,N_Y),(N_X',N_Y')}$ for going from the state with $(N_X', N_Y')$ particles of species $X$ and $Y$ respectively to the one with $(N_X, N_Y)$ particles are given by

$$W_{(N_X-1,N_Y+1),(N_X,N_Y)} = k_+[A]N_X/V\,, \tag{3.23a}$$
$$W_{(N_X,N_Y),(N_X-1,N_Y+1)} = k_-[B](N_Y + 1)/V\,. \tag{3.23b}$$

The kinetic constants obey the constraint [29]

$$\ln \frac{k_+}{k_-} = -\beta(\mu_A^0 + \mu_X^0 - \mu_B^0 - \mu_Y^0) \tag{3.24}$$

where $\mu_i^0$ is the standard chemical potential of the species $i$ and for ideal solutions the chemical potential is obtained from the Gibbs free energy as $\mu_i = \frac{\partial}{\partial N_i} g_i(N_i) = \mu_i^0 + k_B T \ln N_i!$. It is then possible to show that (3.23) satisfy

$$\ln \frac{W_{(N_X-1,N_Y+1),(N_X,N_Y)}}{W_{(N_X,N_Y),(N_X-1,N_Y+1)}} = -\beta\{g_X(N_X - 1) + g(N_Y + 1) - g(N_X) - g(N_y)\}\,. \tag{3.25}$$

Similar considerations can also be applied to non-ideal solutions [30].

**Nonequilibrium free energy**   We can introduce the nonequilibrium free energy:

$$F = E - TS \tag{3.26}$$

When the temperature of the environment is not allowed to vary, the nonequilibrium free energy changes in time according to

$$d_t F = d_t E - T d_t S\,, \tag{3.27}$$

$$= \dot{W} - T\dot{\sigma}\,, \tag{3.28}$$

using first (eq.(3.8)) and second law (eq. (3.13)) in the second equality.

Therefore we can express the entropy production rate in the form

$$\dot{\sigma} = \frac{\dot{W} - d_t F}{T}\,, \tag{3.29}$$

which is the analogous of Kelvin's classic formulation of the second law of thermodynamics.

The nonequilibrium free energy can be related to the relative entropy between the actual state and the equilibrium state at a given temperature. In fact, using the definitions (3.2) and (3.26) we find

$$k_B T D(p|p^{\text{eq}}) = k_B T \sum_i p_i \ln p_i - k_B T \sum_i p_i \ln e^{-\beta(\epsilon_i - F^{\text{eq}})}\,, \tag{3.30}$$

$$= -TS + E - F^{\text{eq}} = F - F^{\text{eq}}\,. \tag{3.31}$$

It is important to note that in absence of driving (*i.e.* when no external power is provided, $\dot{W} = 0$) $F$ is a Lyapunov function for the system, meaning that dynamics will bring it to its minimum value (the equilibrium one). This can be seen using (3.29) combined with (3.31) gives

$$T\dot{\sigma} = \dot{W} - d_t F = -k_B T d_t D(p|p^{\text{eq}}) \geq 0\,. \tag{3.32}$$

In the following, we will also make use of the identity

$$\dot{W} - d_t F^{\text{eq}} = -k_B T \sum_i p_i d_t \ln e^{-\beta(\epsilon_i - F^{\text{eq}})} = -k_B T \sum_i p_i d_t \ln p^{\text{eq}}\,. \tag{3.33}$$

After summing and subtracting $d_t F^{\text{eq}}$ to the entropy production in eq. (3.29), eq. (3.33) gives

$$\dot{\sigma} = -k_B \left( \sum_i p_i d_t p_i^{\text{eq}} + d_t D(p|p^{\text{eq}}) \right) \tag{3.34}$$

$$= -k_B \sum_i d_t p_i \ln \frac{p_i}{p_i^{\text{eq}}} = -k_B \sum_{ij} W_{ij} p_j \ln \frac{p_i}{p_i^{\text{eq}}}\,. \tag{3.35}$$

**Special transformations**

**1.  Reversible transformations.**   If the transformation is extremely slow, every state of the system is at every time only slightly distant from its equilibrium value (see [31] for more details), $p_i = p_i^{\text{eq}} + \delta p_i$. Therefore

$$\ln \frac{p_i}{p_i^{\text{eq}}} = \ln\left(1 + \frac{\delta p_i}{p_i^{\text{eq}}}\right) \simeq \frac{\delta p_i}{p_i^{\text{eq}}} - \frac{1}{2}\left(\frac{\delta p_i}{p_i^{\text{eq}}}\right)^2 + O\left(\left(\frac{\delta p_i}{p_i^{\text{eq}}}\right)^3\right)\,. \tag{3.36}$$

The resulting entropy production is a second order quantity in the relative variation of the states,

$$\dot{\sigma} \approx -k_B \sum_{ij} W_{ij} p_j^{\text{eq}} \left(1 + \frac{\delta p_j}{p_j^{\text{eq}}}\right) \left(\frac{\delta p_i}{p_i^{\text{eq}}} + \frac{1}{2}\left(\frac{\delta p_i}{p_i^{\text{eq}}}\right)^2\right) \tag{3.37}$$

$$= -k_B \sum_{ij} W_{ij} p_j^{\text{eq}} \frac{\delta p_i}{p_i^{\text{eq}}} + k_B \sum_{ij} W_{ij} p_j^{\text{eq}} \frac{\delta p_i}{p_i^{\text{eq}}} \frac{\delta p_j}{p_j^{\text{eq}}} - \frac{k_B}{2} \sum_{ij} W_{ij} p_j^{\text{eq}} \left(\frac{\delta p_i}{p_i^{\text{eq}}}\right)^2, \tag{3.38}$$

since the first sum vanishes by normalization.

**2. Sudden switch.** Driving faster than the typical relaxation rate of the dynamics does not give time to $p$ to change in time. Therefore, like in an isolated system

$$\dot{\sigma} \approx 0 \,, \dot{Q} \approx 0 \,, \mathrm{d}_t S \approx 0 \,. \tag{3.39}$$

Crucially, however, $\dot{W} = \mathrm{d}_t F \neq 0$.

**Nonequilibrium state as a resource** We consider again the dissipated free energy expressed via eq. (3.29)

$$T\dot{\sigma} = \dot{W} - \mathrm{d}_t F^{\text{eq}} - k_B T \mathrm{d}_t D(p|p^{\text{eq}}) \geq 0 \tag{3.40}$$

Integrating it with respect to time and defining the total entropy production $\sigma \equiv \int_0^t dt' \dot{\sigma}(t')$ we obtain the integrated form of the Kelvin formulation of the second law

$$T\sigma = \underbrace{W - \Delta F^{\text{eq}}}_{W_{\text{irr}}} - k_B T D(p(t)|p^{\text{eq}}(t)) + k_B T D(p(0)|p^{\text{eq}}(0)) \geq 0 \,, \tag{3.41}$$

where the equilibrium distribution at different time can change because the energy levels $\{\epsilon_i\}$ are varied by the externally supplied work. Note, however, that the above calculation is restricted to isothermal conditions.

In a transformation between equilibrium states ($p(0) = p^{\text{eq}}(0)$ and $p(t) = p^{\text{eq}}(t)$), the contribution coming from the relative entropy cancels and eq. (3.41) implies that irreversible work is positive ($W_{\text{irr}} \geq 0$), or equivalently

$$W \geq \Delta F^{\text{eq}} \,. \tag{3.42}$$

This is exactly Kelvin's formulation of the second law for transformations between equilibrium states.

However, preparing the system in a nonequilibrium state $p(0) \neq p^{\text{eq}}(0)$ allows the irreversible work

$$W_{\text{irr}} = \underbrace{T\sigma}_{\geq 0} + \underbrace{k_B T D(p(t)|p^{\text{eq}}(t))}_{\geq 0} \underbrace{- k_B T D(p(0)|p^{\text{eq}}(0))}_{\leq 0} \tag{3.43}$$

to become negative due to the nonequilibrium free energy stored in the initial condition, thus achieving extraction from the nonequilibrium initial state, which acts as a resource for the environment [32] (see Fig. 3.5).

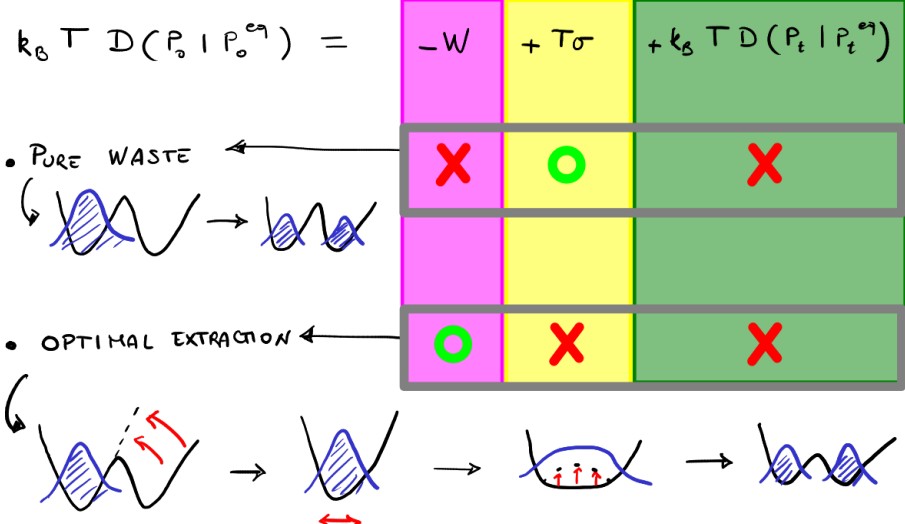

Figure 3.5: Extreme cases for free energy transduction, with red crosses corresponding to vanishing contributions. The resource term $k_B T D(P_0|P_0^{\text{eq}})$ due to the nonequilibrium initial state can be either completely lost ($W = 0$ and $\sigma = k_B D(P_0|P_0^{\text{eq}})$) if no extraction protocol is applied, or optimally extracted via a reversible, quasi-static protocol (for which $\sigma = 0$ and $W = -k_B T D(P_0|P_0^{\text{eq}})$) following a sudden change of the potential from two to single well, followed by an adiabatic deformation back to two wells.

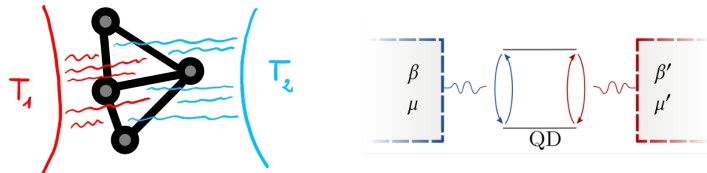

Figure 3.6: Systems in contact with multiple reservoirs. (a) Schematic depiction of a generic system (b) scheme of a 2-levels system in contact with two baths at different temperatures.

## 3.3 ST for multiple reservoirs

We can generalize the above treatment to the important case of multiple reservoirs.

The local detailed balance takes the form

$$\ln \frac{W_{ij}^{(\nu)}}{W_{ji}^{(\nu)}} = -\beta_\nu (\epsilon_i - \epsilon_j), \tag{3.44}$$

where the index $\nu$ now spans the different reservoirs in contact with the system. Each of the reservoirs, in isolation, would impose a well defined equilibrium state as the steady state of the dynamics of the system. However, the simultaneous interaction of the system with different reservoirs frustrates the relaxation to any of the equilibria identified by each of them, and consequently heat and particle flows ensue. The dynamics follows the master equation (3.1), where we assume that the reservoirs affect the system independently of each other, leading to an additive generator $W = \sum_\nu W^{(\nu)}$ [24].

We identify the heat in each reservoir as being

$$\dot{Q}^{(\nu)} = \frac{1}{2} \sum_{ij} \frac{1}{2} (W_{ij}^{(\nu)} p_j - W_{ji}^{(\nu)} p_i) \frac{W_{ij}^{(\nu)}}{W_{ij}^{(\nu)}} \tag{3.45}$$

This gives for the entropy production rate appearing in the second law (3.13)

$$\dot{\sigma} = d_t S - \sum_\nu \frac{\dot{Q}^{(\nu)}}{T} \tag{3.46}$$

$$= k_B \sum_{\nu,i,j} \frac{1}{2} (W_{ij}^{(\nu)} p_j - W_{ji}^{(\nu)} p_i) \ln \frac{W_{ij}^{(\nu)} p_j}{W_{ji}^{(\nu)} p_i} \geq 0, \tag{3.47}$$

where the inequality follows from $(a - b) \ln \frac{a}{b} \geq 0$ for positive $a, b$.

Applying the log-sum inequality [22] $\sum_n a_n \ln \frac{a_n}{b_n} \geq (\sum a_n) \ln \frac{\sum_n a_n}{\sum_n b_n}$ (for nonnegative $a_n, b_n$), to the sum over the index $\nu$ of the reservoirs in eq. (3.47), after defining the probability flow $j_{ij}^\nu = W_{ij}^{(\nu)} p_j$ due to each of the reservoirs, we have

$$\dot{\sigma} = \frac{1}{2} \sum_{\nu,i,j} (j_{ij}^\nu - j_{ji}^\nu) \ln \frac{j_{ij}^\nu}{j_{ji}^\nu} = \sum_{\nu,i,j} j_{ij}^\nu \ln \frac{j_{ij}^\nu}{j_{ji}^\nu} \geq \sum_{i,j} \left( \sum_\nu W_{ij}^\nu \right) p_j \ln \frac{\left( \sum_\nu W_{ij}^\nu \right) p_i}{\left( \sum_\nu W_{ji}^\nu \right) p_j}. \tag{3.48}$$

This means that lumping together rates coming from the interaction with different reservoirs underestimates the entropy production. In particular, there could be detailed balance effective models that do not resolve between different reservoirs and that therefore estimate a vanishing entropy production even for a system which is out of equilibrium.

**Example:** The above fact can be seen easily by considering the 2-levels system $(p_0 = 1 - p_1)$ in contact with two reservoirs of Fig. 3.6 (b) which is governed by the master equation $d_t p_0 = W_{01} p_1 - W_{10} p_0$. It reaches a stationary state when the detailed balance condition $W_{01} p_1 = W_{10} p_0$ is satisfied, thus leading to an estimate of zero entropy production when the latter is computed with the lumped rates $W$. However, since $W = \sum_{\nu=l,r} W^\nu$, the steady state entropy production rate computed using the correct formula (3.47) is nonzero whenever there is a difference in the temperatures of the two reservoirs. Therefore, when in contact with multiple reservoirs the steady state toward which the state of the systems relaxes by virtue of the Perron-Frobenius theorem will not coincide with the equilibrium state given by the Gibbs distribution (3.15). Therefore, in

Figure 3.7: System coupled to energy and matter reservoirs.

absence of external time-dependent driving, after a sufficiently long time the system will be in a nonequilibrium steady state (NESS) characterized by a positive entropy production rate $\dot{\sigma}^{ss} \geq 0$.

The first law now takes the form — obtained following the same steps as for eq. (10.31)

$$\mathrm{d}_t E = \sum_i \mathrm{d}_t \epsilon \, p_i + \sum_i \epsilon \mathrm{d}_t p_i = \dot{W} + \sum_{\nu,i,j} j_{ij}^\nu (\epsilon_i - \epsilon_j) \tag{3.49}$$

$$= \dot{W} + \sum_{\nu,i,j} \frac{1}{2} (j_{ij}^\nu - j_{ji}^\nu)(\epsilon_i - \epsilon_j) = \dot{W} + \sum_\nu \dot{Q}^\nu \,, \tag{3.50}$$

where $\dot{Q}^\nu$ is the heat flowing in the $\nu$-th reservoir.

If we are in presence of reservoirs indexed by $\nu = 0, \ldots, N$, and we choose $T_0$ as the reference temperature which appears in the nonequilibrium free energy definition (3.26), we can rewrite the entropy production rate (3.46)

$$T_0 \dot{\sigma} = T_0 \mathrm{d}_t S - \dot{Q}^0 - \sum_{\nu \geq 1} \frac{T_0}{T_\nu} Q^\nu \tag{3.51}$$

$$= \dot{W} - \mathrm{d}_t F^0 + \sum_{\nu \geq 1} \left(1 - \frac{T_0}{T_\nu}\right) Q^\nu \tag{3.52}$$

$$= \dot{W} - \mathrm{d}_t F^{\mathrm{eq},0} + \sum_{\nu \geq 1} \left(1 - \frac{T_0}{T_\nu}\right) Q^\nu - k_B T_0 \mathrm{d}_t D(p(t)|p^{\mathrm{eq}}(t)) \,. \tag{3.53}$$

Two remarks are needed here. First, the second equality we used the first law (3.50) to solve for $\dot{Q}^0$, the definition of nonequilibrium free energy (3.26) and we have identified the Carnot efficiency $\eta_C^\nu = 1 - \frac{T_\nu}{T_0}$ of the heat exchanged with the $\nu$-th reservoir, which is zero when evaluated for $T_0$ (*i.e.*, no power output can be extracted from a single heat reservoir, that is from an equilibrium environment). Second, in the third equality we made use of the identity for the nonequilibrium free energy in terms of the equilibrium one plus a relative entropy, eq. (3.31).

In a nonequilibrium steady state it reduces to

$$\dot{\sigma}^{ss} = \sum_\nu \left(\frac{1}{T_0} - \frac{1}{T_\nu}\right) \dot{Q}^\nu \,, \tag{3.54}$$

which is analogous to the expression obtained in macroscopic nonequilibrium thermodynamics [33].

## 3.4   Internal entropy and matter transfer

We can generalize the description to include states with internal entropies and to allow for the exchange of particles with the reservoirs.

Each node is now characterized by a number of particles $n_i$, a mesoscopic energy $\epsilon_i = e_i n_i$ ($e_i$ being the average energy per particle in state $i$), and an internal entropy $s_i$. The average values of these observables in a state characterized by the distribution $p$ are respectively $\langle E \rangle = \sum_i \epsilon_i p_i$, $\langle N \rangle = \sum_i n_i p_i$ and $S = \sum_i (s_i - \ln p_i) p_i$. Notice that the Shannon entropy has been slightly extended to take into account the mesoscopic entropy. Energy and particles are conserved quantities for the (isolated) total system obtained considering the mesoscopic subsystem together with all the reservoirs. Given a reservoir with temperature $T^\nu$ and chemical potential $\mu^\nu$, we can define the nonequilibrium free energy associated with the mesoscopic states as $\phi_i = \epsilon_i - \mu^\nu n_i - k_B T^\nu s_i$. The local detailed balance becomes

$$\ln \frac{W_{ij}^\nu}{W_{ji}^\nu} = -\beta^\nu \left( \phi_i^\nu - \phi_j^\nu \right) . \tag{3.55}$$

The fluxes of the conserved quantities can be derived considering their balance equations. For the number of particles we have (repeating the same steps used for eq. (10.31))

$$\mathrm{d}_t \langle N \rangle = \sum_{\nu ij} W_{ij}^\nu p_j n_i = \sum_{\nu ij} \frac{1}{2} (W_{ij}^\nu p_j - W_{ji}^\nu p_i)(n_i - n_j) = \sum_\nu I_N^\nu , \tag{3.56}$$

introducing the particle currents for each reservoir, $I_N^\nu \equiv \sum_{ij} \frac{1}{2} (W_{ij}^\nu p_j - W_{ji}^\nu p_i)(n_i - n_j)$. Analogously, we introduce the energy flows $I_E^\nu \equiv \frac{1}{2} \sum_{ij} (W_{ij}^\nu p_j - W_{ji}^\nu p_i)(\epsilon_i - \epsilon_j)$ to write the first law in a form in which the chemical work contribution $\dot{W}_\mathrm{chem}$ appears explicitly,

$$\mathrm{d}_t \langle E \rangle = \underbrace{\sum_i \mathrm{d}_t \epsilon_i p_i}_{\dot{W}_\mathrm{d}} + \sum_\nu \underbrace{(I_E^\nu - \mu^\nu I_N^\nu)}_{\dot{Q}^\nu} + \underbrace{\sum_\nu \mu^\nu I_N^\nu}_{\dot{W}_\mathrm{chem}} . \tag{3.57}$$

The identification of heat flow with

$$\dot{Q}^\nu = \frac{1}{2} \sum_{ij} (W_{ij}^\nu p_j - W_{ji}^\nu p_i) \left( (\epsilon_i - \mu^\nu n_i) - (\epsilon_j - \mu^\nu n_j) \right) \tag{3.58}$$

$$= \frac{1}{2} \sum_{ij} (W_{ij}^\nu p_j - W_{ji}^\nu p_i) \left( (\phi_i + T^\nu s_i) - (\phi_j + T^\nu s_j) \right) \tag{3.59}$$

takes into account the fact that part of the energy flow is associated to the transfer of particles between different reservoirs, and therefore it should not be counted as heat (as it does not "heat" the reservoir). Indeed, one can easily verify that at equilibrium with a single reservoir eq. (3.59) gives the Clausius formula for the entropy change for a given heat flux once the equilibrium distribution $p_i^\mathrm{eq} = Z^{-1} e^{-\beta \phi_i}$ is introduced.

The second law can obtained computing

$$\mathrm{d}_t S = \sum_i \mathrm{d}_t p_i (s_i - \ln p_i) = \sum_{\nu, i, j} \frac{1}{2} \left( W_{ij}^\nu p_i - W_{ji}^\nu p_i \right) \left( s_i - s_j - \ln \frac{p_i}{p_j} \right) . \tag{3.60}$$

The expression for the entropy production rate, encoding the second law, is consequently

$$\dot{\sigma} = \mathrm{d}_t S - \sum_\nu \frac{\dot{Q}^\nu}{T^\nu} = \sum_{\nu, i, j} \frac{1}{2} \left( W_{ij}^\nu p_j - W_{ji}^\nu p_i \right) \ln \frac{W_{ij}^\nu p_i}{W_{ji}^\nu p_i} \geq 0 . \tag{3.61}$$

Alternatively, using $\dot{Q}^\nu = I_E^\nu - \mu^\nu I_N^\nu$ and using the first law (3.57) to isolate the reservoir with $\nu = 0$, used as a reference, we can write

$$T_0\dot{\sigma} = T_0\mathrm{d}_t S - I_E^0 + \mu^0 I_N^0 - \sum_{\nu \geq 1}(I_E^\nu - \mu^\nu I_N^\nu) \tag{3.62}$$

$$= \dot{W}_{\mathrm{d}} - (\mathrm{d}_t\langle E\rangle - \mu_0\mathrm{d}_t\langle N\rangle - T_0\mathrm{d}_t S) + \sum_\nu\left(\frac{1}{T_0} - \frac{1}{T_\nu}\right)I_E^\nu - \sum_\nu\left(\frac{\mu_\nu}{T_\nu} - \frac{\mu_0}{T_0}\right)I_N^\nu \tag{3.63}$$

and therefore

$$\dot{\sigma} = \frac{\dot{W}_{\mathrm{d}} - \mathrm{d}_t\Phi^0}{T_0} + \sum_{\nu \geq 1}(F_E^\nu I_E^\nu + F_N^\nu I_N^\nu) \tag{3.64}$$

with the nonconservative forces defined as $F_E^\nu = \frac{1}{T_0} - \frac{1}{T_\nu}$ and $F_N^\nu = \frac{\mu_\nu}{T_\nu} - \frac{\mu_0}{T_0}$. The decomposition of the entropy production (3.64) obtained here can be refined using the conceptually similar but more general framework of Refs. [12,13]. There, it is also proved that the number of fundamental force $N_F$ can be expressed as the difference between the number of intensive variables associated to the reservoirs $N_I$ and the number of conserved quantities that characterize the total system $N_C$, namely

$$N_F = N_I - N_C. \tag{3.65}$$

The important case of tight-coupling between currents, namely when the transfer process for different conserved quantities (*e.g.* the one of matter and energy) occurs simultaneously, is discussed in Ref. [34].

## 3.5 Fluctuation theorems and thermodynamic uncertainty relations

Until now, we have considered the average value of thermodynamic observables, where the averaging process sums over all possible stochastic trajectories compatible with the dynamics. Stochastic thermodynamics, however, can be formulated also at the level of the single trajectories [35,36]. Let's indicate with $\Gamma$ a trajectory in the state space. We define an operation of time reversal $\Gamma \to \tilde{\Gamma}$, which changes the order of the visited states and the sign of all the observable which are odd under time-reversal. Furthermore, it reverses the time-dependence contained in the time-dependent driving protocols, if present.

We have the following trajectory level entropy production

$$\sigma(\Gamma) = k_B \ln\frac{P(\Gamma)}{\tilde{P}(\Gamma)} \tag{3.66}$$

which is a trajectory-wise version of the local detailed balance condition (3.7) and is related to the so-called *detailed fluctuation theorem* [37–41].

An important result is the *integral fluctuation theorem* [42], that in this context is an immediate consequence of the rearrangement of eq. (3.66)

$$\left\langle e^{-\frac{\sigma}{k_B}}\right\rangle = \sum_\Gamma P(\Gamma)\frac{\tilde{P}(\tilde{\Gamma})}{P(\Gamma)} = \sum_\Gamma \tilde{P}(\tilde{\Gamma}) = \sum_{\tilde{\Gamma}}\tilde{P}(\tilde{\Gamma}) = 1. \tag{3.67}$$

The third equality is obtained recognizing that the sum over all paths can be enumerated equivalently using the forward or the reversed trajectories.

The average of (3.66) gives the entropy production at the average level

$$\langle \sigma \rangle = k_B \sum_\Gamma P(\Gamma) \ln \frac{P(\Gamma)}{\tilde{P}(\tilde{\Gamma})} \geq 0 \,. \tag{3.68}$$

The inequality is a consequence of Jensen's inequality, as $1 = \langle e^{-\sigma/k_B} \rangle \geq e^{\langle \sigma \rangle/k_B} \geq 1 - \frac{\langle \sigma \rangle}{k_B}$.

Another important class of results concerning fluctuations is the one of thermodynamic uncertainty relations (TUR). TURs are relations which bound the signal-to-noise ratio of a current $J$ (or another measure of the precision of a signal) in terms of the entropy production needed to achieve it. The original TUR [43] states that

$$\mathrm{pr}(J) \equiv \frac{\langle J \rangle^2}{\mathrm{Var}(J)} \leq \frac{\langle \sigma \rangle}{2k_B} \,, \tag{3.69}$$

and therefore provides a bound on a dynamical characterization of a stochastic system (the precision) in terms of a thermodynamic observable (the EPR), which can be exploited either to optimize the former or to infer the latter. For a general perspective unifying different types of uncertainty relations, see [44].

# 4    Deterministic Chemical Reaction Networks

**Francesco Avanzini, Shesha Gopal Marehalli Srinivas, Emanuele Penocchio and Massimiliano Esposito.**[9] *We formulate a nonequilibrium thermodynamic theory for open chemical reaction networks (CRNs) described by deterministic rate equations following the law of mass-action. The conservation laws of CRNs are used to decompose the entropy production into a potential change and two work contributions. One work contribution accounts for the time-dependent manipulation of the chemostatted species. The other accounts for the flows of matter maintained through the CRN by nonconservative forces breaking the detailed balance condition.*

## 4.1   Introduction

Out-of-equilibrium chemical processes are ubiquitous in nature. They constitute for instance the underlying scaffold of information processing [45], oscillations [46], self-replication [47–49], metabolism, and photosynthesis in biosystems [50]. They also play a central role in synthetic chemistry [51], where artificial chemical processes are designed to perform sophisticated tasks. Prototypical examples include self-assembly [52, 53] and molecular machines [54, 55]. These processes operate out of equilibrium: nonzero reaction currents are sustained by continuously harvesting free energy from environment (represented in terms of reservoirs of chemical species called chemostats) [56]. Their energetics can be characterized on rigorous grounds using a nonequilibrium thermodynamic theory for CRNs. Such a theory has been developed in recent years for CRNs undergoing a stochastic [29, 57, 58] or a deterministic dynamics [10, 30, 59–61]. This theory has been applied to study, for instance, the energetic costs of sustaining coherent oscillations [62] and sustaining growth of copolymers [63–65] and biomolecules [66]; the efficiency of central metabolism [67] in prokaryotes; the internal information transfer in a model of chemically-driven self-assembly and an experimental light-driven bimolecular motor [68]. It was also formulated for reaction diffusion systems, where chemical reactions can be used to create patterns [69, 70] and waves [71].

     In these lecture notes, we focus on deterministic CRNs whose dynamics follows the law of mass-action. In particular, we re-derive, in a self-contained way, the formulation of nonequilibrium thermodynamics for CRNs developed in Refs. [29, 30, 60]. The lecture notes are organized as follows. We introduce the basic setup for the description of CRNs in Sec. 4.2 and discuss their dynamics in Sec. 4.3. We then build nonequilibrium thermodynamics on top of the CRN dynamics. We start in Sec. 4.4 where we introduce the notion of thermodynamic consistency: closed CRNs must be detailed balanced, namely, they must relax towards an equilibrium state. This allows us to derive a condition known in stochastic thermodynamics as local detailed balance establishing a correspondence between dynamic (i.e., reaction fluxes) and thermodynamic (i.e., chemical potentials) quantities. In Sec. 4.5, we use the local detailed balance to obtain the nonequilibrium formulation of the first and second law of thermodynamics for closed CRNs. In Sec. 4.6, we derive the first and second law of thermodynamics for open CRNs. Crucially, we show how to use the conservation laws of CRNs (defined in Subs. 4.3.3) to rewrite the second law in such a way as to split the free energy exchanged with the chemostats into two work contributions. One contribution, named driving work, accounts for the free energy exchanged with the chemostats via a time dependent manipulation of CRNs. The other contribution, named nonconservative work, accounts for the nonconservative forces created via the exchanges

---

[9]FA was the lecturer and wrote this chapter; SGMS was the angel; EP and ME contributed to the selection and organization of the chapter content.

with the chemostats that break the detailed balance condition and can maintain CRNs out of equilibrium. This decomposition of the second law is a major result as it identifies the specific mechanism, in terms of nonconservative forces, that can maintain CRNs out of equilibrium.

**Disclaimer.** These lecture notes represent a first (partial) summary of the recent developments in nonequilibrium thermodynamics of CRNs that Francesco Avanzini, Massimiliano Esposito and Emanuele Penocchio (authors are listed in alphabetic order) are planning to review more extensively in another contribution. The course "Thermodynamics of Deterministic Chemical Reaction Networks" of the school (Post)Modern Thermodynamics (5-9 December 2022, Luxembourg) was given by Francesco Avanzini and prepared together with Shesha Gopal Marehalli Srinivas.

## 4.2 Basic Setup

A CRN is defined as a set of chemical species $\{Z_\alpha\}$, identified by $\alpha \in \mathcal{S} = \{1, 2, \ldots, n_s\}$, that are interconverted via chemical reactions, identified by the index $\rho \in \mathcal{R} = \{1, 2, \ldots, n_r\}$. Each reaction $\rho$ is assumed here to be reversible and represented by an equation like

$$\sum_{\alpha \in \mathcal{S}} \nu_{\alpha,\rho}^+ \, Z_\alpha \xrightleftharpoons{\rho} \sum_{\alpha \in \mathcal{S}} \nu_{\alpha,\rho}^- \, Z_\alpha \, , \tag{4.1}$$

where $\nu_{\alpha,\rho}^+$ (resp. $\nu_{\alpha,\rho}^-$) is the stoichiometric coefficients of $Z_\alpha$ in the forward (resp. backward) transformation. The net stoichiometry is encoded in the $(n_s \times n_r)$ stoichiometric matrix $\mathbb{S}$ whose columns are defined as

$$\boldsymbol{S}_\rho := \boldsymbol{\nu}_\rho^- - \boldsymbol{\nu}_\rho^+ \, , \tag{4.2}$$

with $\boldsymbol{\nu}_\rho^\pm := (\ldots, \nu_{\alpha,\rho}^\pm, \ldots)_{\alpha \in \mathcal{S}}^{\mathrm{T}}$.

    *Physical Remark.* We consider here systems, known as ideal dilute solutions, where the $n_s$ species $\{Z_\alpha\}$ are mixed together with a non-reacting, very abundant species $Z_0$ called solvent. The solvent maintains the temperature $T$ and the volume $V$ of the system constant.

    *Example.* We now consider the following CRN,

$$
\begin{aligned}
\mathrm{F} + \mathrm{E} &\xrightleftharpoons{\phantom{xx}1\phantom{xx}} \mathrm{EF} \\
\mathrm{EF} &\xrightleftharpoons{\phantom{xx}2\phantom{xx}} \mathrm{EW} \\
\mathrm{EW} &\xrightleftharpoons{\phantom{xx}3\phantom{xx}} \mathrm{E} + \mathrm{W} \\
\mathrm{S} + \mathrm{EF} &\xrightleftharpoons{\phantom{xx}4\phantom{xx}} \mathrm{EFS} \\
\mathrm{EFS} &\xrightleftharpoons{\phantom{xx}5\phantom{xx}} \mathrm{EW} + \mathrm{P}
\end{aligned} \tag{4.3}
$$

representing a minimal metabolic process. Indeed, the interconversion of the substrate S (representing for instance ADP) into the product P (representing for instance ATP) is catalyzed by the enzyme E and powered by the interconversion of the fuel F (representing for instance nutrients) into the waste W (representing for instance $CO_2$). The species EF and EW are the complexes enzyme-fuel and enzyme-waste, respectively. The species EFS is the complex obtained by binding both the fuel and the substrate to the enzyme.

The stoichiometric matrix of the CRN (4.3) reads

$$
\mathbb{S} = \begin{array}{c} \\ \text{E} \\ \text{EF} \\ \text{EW} \\ \text{EFS} \\ \text{P} \\ \text{W} \\ \text{S} \\ \text{F} \end{array}
\begin{array}{ccccc}
1 & 2 & 3 & 4 & 5 \\
\left( \begin{array}{ccccc}
-1 & 0 & 1 & 0 & 0 \\
1 & -1 & 0 & -1 & 0 \\
0 & 1 & -1 & 0 & 1 \\
0 & 0 & 0 & 1 & -1 \\
0 & 0 & 0 & 0 & 1 \\
0 & 0 & 1 & 0 & 0 \\
0 & 0 & 0 & -1 & 0 \\
-1 & 0 & 0 & 0 & 0
\end{array} \right)
\end{array},
\tag{4.4}
$$

where, for instance, the first column specifies that 1 molecule of E and 1 molecule of F are consumed every time the first reaction occurs (in the forward direction), while 1 molecule of EF is produced.

*Mathematical Remark.* The same stoichiometric matrix might correspond to different CRNs. Consider for instance two CRNs composed of the three species S, P, and E. In one CRN, the species undergo the chemical reaction

$$
\text{S} \rightleftharpoons \text{P} \,.
\tag{4.5}
$$

In the other CRN, the species undergo the chemical reaction

$$
\text{S} + \text{E} \rightleftharpoons \text{P} + \text{E} \,.
\tag{4.6}
$$

According to Eq. (4.2), both CRNs have the same stoichiometric matrix.

**Disclaimer.** The CRN (4.3) has been chosen to illustrate the theory discussed in these lecture notes and does not necessarily represent any realistic chemical process.

## 4.3 Dynamics

The evolution of deterministic CRNs with constant volume is specified by the (vector of the) concentrations of chemical species: $[\boldsymbol{Z}] = (\dots, [Z_\alpha], \dots)_{\alpha \in \mathcal{S}}$.

*Notation Remark.* Throughout these notes, we will omit for compactness of notation the time dependence of any quantity (e.g., concentrations and thermodynamic fluxes, like heat, entropy flow, and entropy production rate).

### 4.3.1 Rate Equation

In open CRNs, the concentration vector follows the rate equation

$$
\frac{\mathrm{d}}{\mathrm{d}t}[\boldsymbol{Z}] = \mathbb{S}\boldsymbol{J}([\boldsymbol{Z}]) + \boldsymbol{I} \,.
\tag{4.7}
$$

where $\boldsymbol{J}([\boldsymbol{Z}])$ is the reaction current vector and $\boldsymbol{I}$ is the exchange current vector.

Each entry $J_\rho$ of $\boldsymbol{J}$ quantifies the net current of reaction $\rho$. If $J_\rho > 0$ (resp. $J_\rho < 0$), reaction $\rho$ occurs in the forward (resp. backward) direction, namely, from left to right (resp. from right to left) in Eq. (4.1). Each net current is given by the difference between the forward and backward fluxes,

$$
J_\rho([\boldsymbol{Z}]) = r_\rho^+([\boldsymbol{Z}]) - r_\rho^-([\boldsymbol{Z}]) \,,
\tag{4.8}
$$

which satisfy the law of mass-action, i.e., the fluxes are proportional to the concentrations of the reactants to the power of their stoichiometric coefficients:

$$r_\rho^\pm([\boldsymbol{Z}]) = k_\rho^\pm [\boldsymbol{Z}]^{\boldsymbol{\nu}_\rho^\pm}, \tag{4.9}$$

with $[\boldsymbol{Z}]^{\boldsymbol{\nu}_\rho^\pm} = \prod_{\alpha \in \mathcal{S}} [Z_\alpha]^{\nu_{\alpha,\rho}^\pm}$ and $k_\rho^\pm$ being the so-called kinetic constants. This is physically justified for ideal dilute solutions: interactions between chemical species are negligible and the solvent is much more abundant, i.e., $[Z_0] \gg \sum_{\alpha \in \mathcal{S}} [Z_\alpha]$. Therefore, reacting species behave as an ideal gas mixture reacting at rates that are proportional to their concentrations.

Each entry $I_\alpha$ of $\boldsymbol{I}$ quantifies the net current at which the species $Z_\alpha$ is exchanged with the environment. This can represent for instance i) exchanges with external particle reservoirs known as chemostats, or ii) the net effect of other non-specified chemical reactions.

We can now split the set of chemical species $\{Z_\alpha\}$, and of the corresponding set of indexes $\mathcal{S}$, into two disjoint sets by using the rate equation (4.7). The $n_X$ species $\{X_\alpha\}$ (with $\alpha \in \mathcal{S}_X$) are not exchanged with the environment, i.e., $I_\alpha = 0$ for the whole dynamics, and are thus called *internal*. The $n_Y$ species $\{Y_\alpha\}$ (with $\alpha \in \mathcal{S}_Y$) are exchanged with the environment, i.e., $I_\alpha \neq 0$ for some moments of the dynamics, and are thus called *exchanged*. By applying the same splitting to the concentration vector

$$[\boldsymbol{Z}] = ([\boldsymbol{X}], [\boldsymbol{Y}])^{\mathrm{T}}, \tag{4.10}$$

and the stoichiometric matrix

$$\mathbb{S} = \begin{pmatrix} \mathbb{S}^X \\ \mathbb{S}^Y \end{pmatrix}, \tag{4.11}$$

the rate equation (4.7) can be rewritten as

$$\frac{\mathrm{d}}{\mathrm{d}t}[\boldsymbol{X}] = \mathbb{S}^X \boldsymbol{J}([\boldsymbol{X}], [\boldsymbol{Y}]), \tag{4.12}$$

$$\frac{\mathrm{d}}{\mathrm{d}t}[\boldsymbol{Y}] = \mathbb{S}^Y \boldsymbol{J}([\boldsymbol{X}], [\boldsymbol{Y}]) + \boldsymbol{I}^Y, \tag{4.13}$$

where $\boldsymbol{I}^Y$ collects the non-null entries of $\boldsymbol{I}$.

*Remark.* In these lecture notes, we consider CRNs coupled with chemostats (the $\mathcal{S}_Y$ species are thus said to be chemostatted). This means that the concentrations $[\boldsymbol{Y}]$ are not dynamical variables, but are controlled by the external chemostats. Equation (4.13) becomes a mere definition of the currents $\boldsymbol{I}^Y$ ensuring that $[\boldsymbol{Y}]$ follow the chemostat-imposed protocol.

*Example.* The reaction rates of the CRN (4.3) (represented in Fig. 4.1) are given by

$$r_1^+([\boldsymbol{Z}]) = k_1^+ [\mathrm{E}][\mathrm{F}], \tag{4.14}$$
$$r_1^-([\boldsymbol{Z}]) = k_1^- [\mathrm{EF}], \tag{4.15}$$
$$r_2^+([\boldsymbol{Z}]) = k_2^+ [\mathrm{EF}], \tag{4.16}$$
$$r_2^-([\boldsymbol{Z}]) = k_2^- [\mathrm{EW}], \tag{4.17}$$
$$r_3^+([\boldsymbol{Z}]) = k_3^+ [\mathrm{EW}], \tag{4.18}$$
$$r_3^-([\boldsymbol{Z}]) = k_3^- [\mathrm{E}][\mathrm{W}], \tag{4.19}$$
$$r_4^+([\boldsymbol{Z}]) = k_4^+ [\mathrm{EF}][\mathrm{S}], \tag{4.20}$$
$$r_4^-([\boldsymbol{Z}]) = k_4^- [\mathrm{EFS}], \tag{4.21}$$
$$r_5^+([\boldsymbol{Z}]) = k_5^+ [\mathrm{EFS}], \tag{4.22}$$
$$r_5^-([\boldsymbol{Z}]) = k_5^- [\mathrm{EW}][\mathrm{P}]. \tag{4.23}$$

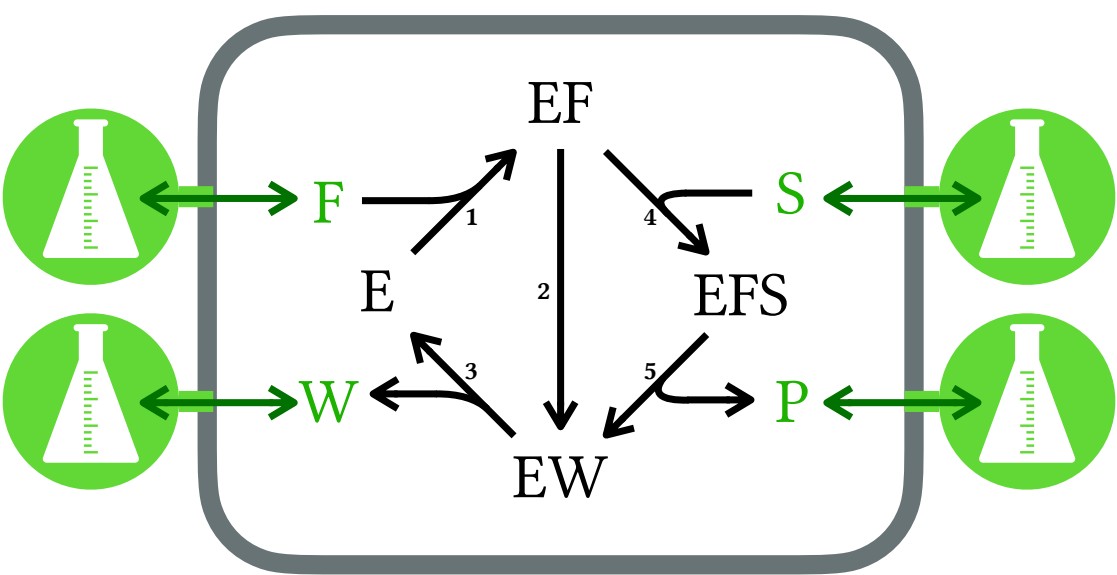

Figure 4.1: Pictorial illustration of the CRN (4.3) when the species F, S, W, and P are exchanged with chemostats. Black (numbered) arrows represent chemical reactions, while green arrows crossing the gray boundary represent the exchange processes with the chemostats (represented by flasks). Here, all reactions are assumed to be reversible even if they are represented with a hypergraph notation by single arrows.

If we now assume that the species P, W, S, and F are exchanged (like in Fig. 4.1), the exchange current vector reads

$$\boldsymbol{I} = \begin{array}{cccccccc} \text{\scriptsize E} & \text{\scriptsize EF} & \text{\scriptsize EW} & \text{\scriptsize EFS} & \text{\scriptsize P} & \text{\scriptsize W} & \text{\scriptsize S} & \text{\scriptsize F} \\ \left( 0 & 0 & 0 & 0 & I_\text{P} & I_\text{W} & I_\text{S} & I_\text{F} \right) \end{array} \tag{4.24}$$

### 4.3.2   Steady State

The steady state $[\boldsymbol{Z}]_\text{ss}$ of the rate equation (4.7), defined as

$$\mathbb{S}\boldsymbol{J}([\boldsymbol{Z}]_\text{ss}) + \boldsymbol{I} = 0\,, \tag{4.25}$$

is said to be an equilibrium steady state if

$$\boldsymbol{J}([\boldsymbol{Z}]_\text{eq}) = 0\,, \tag{4.26}$$

which implies that $\boldsymbol{I} = 0$.

   *Remark.* When CRNs have a well defined equilibrium state, they are said to be *detailed balanced* (not to be confused with *local detailed balance* which will be discussed in Sec. 4.4).

### 4.3.3   Conservation Laws

The $n_\ell \leq n_s$ (linearly independent) left-null eigenvectors of the stoichiometric matrix, identified by the index $\lambda$,

$$\boldsymbol{\ell}^\lambda \cdot \mathbb{S} = 0\,, \tag{4.27}$$

are named conservation laws. Indeed, each scalar

$$L^\lambda = \boldsymbol{\ell}^\lambda \cdot [\boldsymbol{Z}] \tag{4.28}$$

is a conserved quantity of the rate equation (4.7) if CRNs were closed, i.e., $\boldsymbol{I} = 0$ :

$$\frac{\mathrm{d}}{\mathrm{d}t}L^\lambda = \boldsymbol{\ell}^\lambda \cdot \frac{\mathrm{d}}{\mathrm{d}t}[\boldsymbol{Z}] = \boldsymbol{\ell}^\lambda \cdot \mathbb{S}\boldsymbol{J}([\boldsymbol{Z}]) = 0 \,. \tag{4.29}$$

*Physical Remark.* The conservation laws identify fragments of (or entire) molecules, named, *moieties*, that remain intact in all reactions. The corresponding conserved quantities quantify the concentration of these moieties.

*Mathematical Remark.* The set of conservation laws $\{\boldsymbol{\ell}^\lambda\}$ is not unique: a linear combination of conservation laws is still a left-null eigenvector of the stoichiometric matrix. Different sets identify different moieties whose physical interpretation might not be obvious.

*Example.* The stoichiometric matrix (4.4) of the CRN (4.3) admits the following three conservation laws

$$\boldsymbol{\ell}^{\mathrm{E}} = \begin{matrix} \mathrm{E} \\ \mathrm{EF} \\ \mathrm{EW} \\ \mathrm{EFS} \\ \mathrm{P} \\ \mathrm{W} \\ \mathrm{S} \\ \mathrm{F} \end{matrix} \begin{pmatrix} 1 \\ 1 \\ 1 \\ 1 \\ 0 \\ 0 \\ 0 \\ 0 \end{pmatrix}, \quad \boldsymbol{\ell}^{\mathrm{S}} = \begin{matrix} \mathrm{E} \\ \mathrm{EF} \\ \mathrm{EW} \\ \mathrm{EFS} \\ \mathrm{P} \\ \mathrm{W} \\ \mathrm{S} \\ \mathrm{F} \end{matrix} \begin{pmatrix} 0 \\ 0 \\ 0 \\ 1 \\ 1 \\ 0 \\ 1 \\ 0 \end{pmatrix}, \quad \boldsymbol{\ell}^{\mathrm{F}} = \begin{matrix} \mathrm{E} \\ \mathrm{EF} \\ \mathrm{EW} \\ \mathrm{EFS} \\ \mathrm{P} \\ \mathrm{W} \\ \mathrm{S} \\ \mathrm{F} \end{matrix} \begin{pmatrix} 0 \\ 1 \\ 1 \\ 1 \\ 0 \\ 1 \\ 0 \\ 1 \end{pmatrix}. \tag{4.30}$$

The concentrations of the corresponding moieties, i.e., the enzyme, the substrate and the fuel moieties, are given by $L^{\mathrm{E}} = [\mathrm{E}] + [\mathrm{EF}] + [\mathrm{EW}] + [\mathrm{EFS}]$, $L^{\mathrm{S}} = [\mathrm{EFS}] + [\mathrm{P}] + [\mathrm{S}]$, $L^{\mathrm{F}} = [\mathrm{EF}] + [\mathrm{EW}] + [\mathrm{EFS}] + [\mathrm{W}] + [\mathrm{F}]$. The fragments corresponding to these moieties are highlighted in Fig. 4.2 with different colors.

## 4.4 Local Detailed Balance

The local detailed balance condition establishes a correspondence between the dynamics and the thermodynamics, and is crucial to build a nonequilibrium thermodynamic theory on top of a dynamical system. We introduce it here by comparing the dynamic and thermodynamic equilibrium conditions. We then generalize it to nonequilibrium conditions.

*Remark.* The general idea of our derivation of the local detailed balance is the following: i) Dynamic equilibrium relates $[\boldsymbol{Z}]_{\mathrm{eq}}$ to $\{k_\rho^\pm\}$; ii) Thermodynamic equilibrium relates $[\boldsymbol{Z}]_{\mathrm{eq}}$ to the standard chemical potentials $\{\mu_\alpha^\circ(T)\}$; iii) Local detailed balance relates $\{k_\rho^\pm\}$ (dynamics) to $\{\mu_\alpha^\circ(T)\}$ (thermodynamics); iv) Generalization to nonequilibrium.

### 4.4.1 Dynamic equilibrium

The steady state of closed CRNs must be an equilibrium steady state for thermodynamic consistency: if we do not provide (free) energy to dissipative systems such as CRNs, they must reach an equilibrium steady state.

By combining Eqs. (4.8), (4.9) and (4.26), we obtain

$$\frac{k_\rho^+}{k_\rho^-} = [\boldsymbol{Z}]_{\mathrm{eq}}^{\boldsymbol{S}_\rho} \,, \tag{4.31}$$

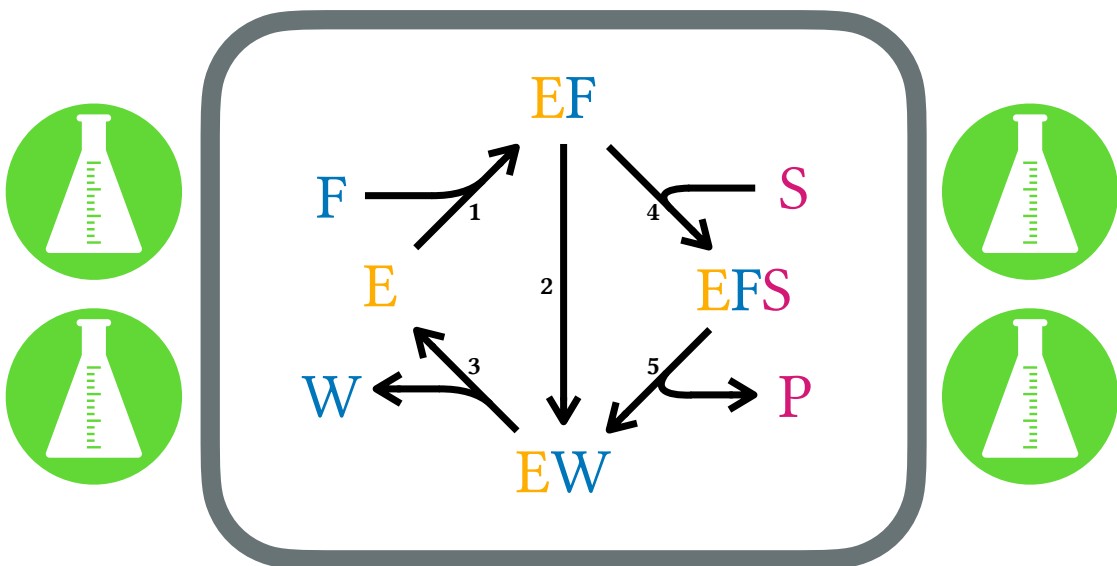

Figure 4.2: Pictorial illustration of the CRN (4.3) where the different moieties identified by the conservation laws in Eq. (4.30) are highlighted with different colors: orange for $\boldsymbol{\ell}^{\mathrm{E}}$, purple for $\boldsymbol{\ell}^{\mathrm{S}}$, and blue for $\boldsymbol{\ell}^{\mathrm{F}}$.

for every $\rho \in \mathcal{R}$ or, equivalently,

$$RT \log \frac{k_\rho^+}{k_\rho^-} = RT \log([\boldsymbol{Z}]_{\mathrm{eq}}) \cdot \boldsymbol{S}_\rho \,. \tag{4.32}$$

Note that hereafter $\log \boldsymbol{v} = (\ldots, \log v_i, \ldots)^{\mathrm{T}}$ for very vector $\boldsymbol{v} = (\ldots, v_i, \ldots)^{\mathrm{T}}$.

### 4.4.2 Thermodynamic equilibrium

We consider here systems with temperature and pressure fixed by the environment they are exposed to. Equilibrium thermodynamics then dictates that the equilibrium state is a global minimum of the Gibbs free energy (density) that, for ideal mixture of chemical species, reads [72]

$$G([\boldsymbol{Z}]_{\mathrm{eq}}, [Z_0]) = \sum_{\alpha \in \mathcal{S}} \mu_\alpha([\boldsymbol{Z}]_{\mathrm{eq}}, [Z_0])[Z_\alpha]_{\mathrm{eq}} + \mu_0([\boldsymbol{Z}]_{\mathrm{eq}}, [Z_0])[Z_0] \tag{4.33}$$

with

$$\mu_\alpha([\boldsymbol{Z}]_{\mathrm{eq}}, [Z_0]) = \hat{\mu}_\alpha^\circ(T) + RT \log \frac{[Z_\alpha]_{\mathrm{eq}}}{\sum_{\beta \in \mathcal{S}}[Z_\beta]_{\mathrm{eq}} + [Z_0]} \,, \tag{4.34}$$

$$\mu_0([\boldsymbol{Z}]_{\mathrm{eq}}, [Z_0]) = \hat{\mu}_0^\circ(T) + RT \log \frac{[Z_0]}{\sum_{\beta \in \mathcal{S}}[Z_\beta]_{\mathrm{eq}} + [Z_0]} \,, \tag{4.35}$$

being the chemical potential of $Z_\alpha$ and of the solvent, respectively; $\hat{\mu}_\alpha^\circ(T)$ and $\hat{\mu}_0^\circ(T)$ being the corresponding temperature-dependent standard chemical potentials; and $R$ and $T$ being the gas constant and the temperature, respectively.

For consistency with mass-action kinetics, we consider the dilute solution limit, i.e., $[Z_0] \gg \sum_{\alpha \in \mathcal{S}}[Z_\alpha]$, which allows us to approximate

$$\log\left(\frac{[Z_\alpha]_{\mathrm{eq}}}{\sum_{\beta \in \mathcal{S}}[Z_\beta]_{\mathrm{eq}} + [Z_0]}\right)[Z_\alpha]_{\mathrm{eq}} \approx \log\left(\frac{[Z_\alpha]_{\mathrm{eq}}}{[Z_0]}\right)[Z_\alpha]_{\mathrm{eq}}, \tag{4.36}$$

$$\log\left(\frac{[Z_0]}{\sum_{\beta \in \mathcal{S}}[Z_\beta]_{\mathrm{eq}} + [Z_0]}\right)[Z_0] \approx -\sum_{\beta \in \mathcal{S}}[Z_\beta]_{\mathrm{eq}}, \tag{4.37}$$

using a Taylor expansion [10]. This implies that the chemical potential of $Z_\alpha$ depends only on its concentration

$$\mu_\alpha([Z_\alpha]_{\mathrm{eq}}) \approx \mu_\alpha^\circ(T) + RT\log[Z_\alpha]_{\mathrm{eq}}, \tag{4.39}$$

by rescaling the standard chemical potentials according to

$$\mu_\alpha^\circ(T) := \hat{\mu}_\alpha^\circ(T) - RT\log[Z_0], \tag{4.40}$$

since $[Z_0]$ is a constant quantity. Hence, for ideal dilute solutions, the free energy reads

$$G([\boldsymbol{Z}]_{\mathrm{eq}}) = \sum_{\alpha \in \mathcal{S}}(\mu_\alpha([Z_\alpha]_{\mathrm{eq}}) - RT)[Z_\alpha]_{\mathrm{eq}} + \hat{\mu}_0^\circ(T)[Z_0]. \tag{4.41}$$

*Mathematical Remark.* Note that i) the last term in Eq. (4.41), i.e., $\hat{\mu}_0^\circ(T)[Z_0]$, represents a constant shift of the Gibbs free energy and will be neglected hereafter, and ii)

$$\mu_\alpha([Z_\alpha]_{\mathrm{eq}}) = \frac{\partial G([\boldsymbol{Z}]_{\mathrm{eq}})}{\partial[Z_\alpha]_{\mathrm{eq}}}. \tag{4.42}$$

Equilibrium thermodynamics dictates that $[\boldsymbol{Z}]_{\mathrm{eq}}$ is a global minimum of $G$. Thus,

$$\lim_{\varepsilon \to 0}\frac{G([\boldsymbol{Z}]_{\mathrm{eq}} + \varepsilon\boldsymbol{S}_\rho) - G([\boldsymbol{Z}]_{\mathrm{eq}})}{\varepsilon} = 0, \tag{4.43}$$

for every $\rho$ or equivalently,

$$\Delta_\rho G([\boldsymbol{Z}]_{\mathrm{eq}}) := \boldsymbol{\mu}([\boldsymbol{Z}]_{\mathrm{eq}}) \cdot \boldsymbol{S}_\rho = 0, \tag{4.44}$$

where $\boldsymbol{\mu}([\boldsymbol{Z}]_{\mathrm{eq}}) = (\dots, \mu_\alpha([Z_\alpha]_{\mathrm{eq}}), \dots)_{\alpha \in \mathcal{S}}^{\mathrm{T}}$. Equation (4.44) can be rewritten as

$$-\boldsymbol{\mu}^\circ(T) \cdot \boldsymbol{S}_\rho = RT\log([\boldsymbol{Z}]_{\mathrm{eq}}) \cdot \boldsymbol{S}_\rho, \tag{4.45}$$

with $\boldsymbol{\mu}^\circ(T) = (\dots, \mu_\alpha^\circ(T), \dots)_{\alpha \in \mathcal{S}}^{\mathrm{T}}$.

*Mathematical Remark.* Equation (4.43) accounts for all possible changes of the concentrations that are consistent with the conservation laws. An alternative derivation of Eq. (4.44) uses Lagrange multipliers [11].

---

[10] By defining $\varepsilon = \sum_{\beta \in \mathcal{S}}[Z_\beta]_{\mathrm{eq}}$, we use the Taylor expansion

$$\log\left(\frac{[Z_0]}{\varepsilon + [Z_0]}\right) = \varepsilon\left[\frac{\varepsilon + [Z_0]}{[Z_0]}\frac{(-[Z_0])}{(\varepsilon + [Z_0])^2}\right]_{\varepsilon=0} + O(\varepsilon^2), \tag{4.38}$$

truncated at the first order.

[11] Given the Lagrangian function

$$\mathcal{L}([\boldsymbol{Z}]_{\mathrm{eq}}, \boldsymbol{f}) = G([\boldsymbol{Z}]_{\mathrm{eq}}) - \sum_\lambda f_\lambda C^\lambda([\boldsymbol{Z}]_{\mathrm{eq}}), \tag{4.46}$$

### 4.4.3    Equilibrium Local Detailed Balance

By combining Eqs. (4.32) and (4.45), we obtain the local detailed balance condition

$$RT \log \frac{k_\rho^+}{k_\rho^-} = -\boldsymbol{\mu}^\circ(T) \cdot \boldsymbol{S}_\rho \,. \tag{4.51}$$

*Physical Remark.* Our derivation of the local detailed balance is only based on the assumption of thermodynamic consistency, namely, the rate equation (4.7) for closed CRNs must admit an equilibrium steady state $[\boldsymbol{Z}]_{\mathrm{eq}}$ which corresponds to the global minimum of the (equilibrium) Gibbs free energy (4.41).

*Remark.* Note the difference between detailed balanced CRNs discussed in Subs. 4.3.2, namely, the existence of a well defined equilibrium steady state, and the local detailed balance condition derived here, namely, the correspondence between the kinetic constants and the standard chemical potentials.

### 4.4.4    Non-Equilibrium Local Detailed Balance

We now generalize the local detailed balance to nonequilibrium conditions. To do so, we assume that all degrees of freedom other than concentrations are always equilibrated. The temperature $T$ is set by the solvent playing the role of a thermal reservoir, the pressure $p$ is set by the environment the system is exposed to, and diffusion processes equilibrate on a much faster time scale than the chemical reactions thus maintaining the chemical species homogeneously distributed. This physically means that CRNs would be an equilibrated mixture in absence of reactions for every concentration $[\boldsymbol{Z}]$. The dynamics is merely interconverting the mixture $[\boldsymbol{Z}](t)$ into another $[\boldsymbol{Z}](t + \mathrm{d}t)$. In this way, thermodynamic state functions, namely, the Gibbs free energy and the chemical potentials, can be specified by their equilibrium form (in Eqs. (4.41) and (4.39), respectively) but expressed in terms of nonequilibrium concentrations $[\boldsymbol{Z}]$:

$$G([\boldsymbol{Z}]) = \sum_{\alpha \in \mathcal{S}} (\mu_\alpha([Z_\alpha]) - RT)[Z_\alpha] \,, \tag{4.52}$$

with

$$\mu_\alpha([Z_\alpha]) = \mu_\alpha^\circ(T) + RT \log[Z_\alpha] \,. \tag{4.53}$$

We analogously introduce

$$\Delta_\rho G([\boldsymbol{Z}]) := \boldsymbol{\mu}([\boldsymbol{Z}]) \cdot \boldsymbol{S}_\rho \neq 0 \,, \tag{4.54}$$

where

$$C^\lambda([\boldsymbol{Z}]_{\mathrm{eq}}) = \boldsymbol{\ell}^\lambda \cdot [\boldsymbol{Z}]_{\mathrm{eq}} - \overline{L}^\lambda \,, \tag{4.47}$$

with $\{\overline{L}^\lambda\}$ being the actual values of the conserved quantities and $\boldsymbol{f} = (\dots, f_\lambda, \dots)$ being the Lagrange multipliers, the global minimum of $G$ consistent with the conservation laws satisfies

$$\frac{\partial \mathcal{L}([\boldsymbol{Z}]_{\mathrm{eq}}, \boldsymbol{f})}{\partial f_\lambda} = -C^\lambda([\boldsymbol{Z}]_{\mathrm{eq}}) = 0 \,, \tag{4.48}$$

$$\frac{\partial \mathcal{L}([\boldsymbol{Z}]_{\mathrm{eq}}, \boldsymbol{f})}{\partial [Z_\alpha]_{\mathrm{eq}}} = \mu_\alpha([Z_\alpha]_{\mathrm{eq}}) - \sum_\lambda f_\lambda \ell_\alpha^\lambda = 0 \,, \tag{4.49}$$

with $\ell_\alpha^\lambda$ being the $\alpha$ entry of $\boldsymbol{\ell}^\lambda$. The condition in Eq. (4.49) can be rewritten as

$$\boldsymbol{\mu}([\boldsymbol{Z}]_{\mathrm{eq}}) = \sum_\lambda f_\lambda \boldsymbol{\ell}^\lambda \,, \tag{4.50}$$

which implies Eq. (4.44) because of Eq. (4.27).

which is in general different from zero in nonequilibirum conditions.

Thus, by summing $-RT \log[\boldsymbol{Z}] \cdot \boldsymbol{S}_\rho$ (with $[\boldsymbol{Z}]$ the nonequilibrium concentrations) on both sides of Eq. (4.51), and using Eqs. (4.9) and (4.54), we can formulate the local detailed balance in terms of a flux-force relation:

$$RT \log \frac{r_\rho^+([\boldsymbol{Z}])}{r_\rho^-([\boldsymbol{Z}])} = -\Delta_\rho G([\boldsymbol{Z}]),\tag{4.55}$$

where $\Delta_\rho G$ plays the role of the thermodynamic force driving the reaction. Indeed, if $\Delta_\rho G < 0$, then $r_\rho^+ > r_\rho^-$ and reaction $\rho$ proceeds in the forward direction $J_\rho > 0$. On the other hand, if $\Delta_\rho G > 0$, then $r_\rho^+ < r_\rho^-$ and reaction $\rho$ proceeds in the backward direction $J_\rho < 0$.

## 4.5    Thermodynamics of Closed CRNs

We now formulate the first and the second law of thermodynamics for closed CRNs. We will use the second law and the conservation laws to prove that the concentrations will eventually reach equilibrium, i.e., $\lim_{t \to +\infty}[\boldsymbol{Z}](t) = [\boldsymbol{Z}]_\text{eq}$, for every initial condition $[\boldsymbol{Z}](0)$.

### 4.5.1    First Law

We introduce the enthalpy $H$, quantifying the "internal" energy of CRNs [12], as

$$H([\boldsymbol{Z}]) := \frac{\partial(G([\boldsymbol{Z}])/T)}{\partial 1/T} = \boldsymbol{h}^\circ(T) \cdot [\boldsymbol{Z}],\tag{4.57}$$

where $\boldsymbol{h}^\circ(T) = (\dots, h_\alpha^\circ(T), \dots)_{\alpha \in \mathcal{S}}^\text{T}$ and

$$h_\alpha^\circ(T) := \frac{\partial(\mu_\alpha^\circ(T)/T)}{(\partial 1/T)} = \frac{\partial(\mu_\alpha([Z_\alpha])/T)}{(\partial 1/T)}\tag{4.58}$$

is the standard molar enthalpy of $Z_\alpha$. By taking the time derivative of $H([\boldsymbol{Z}])$ according to the rate equation (4.7), we obtain the nonequilibrium formulation of the first law of thermodynamics for closed CRNs:

$$\frac{\mathrm{d}}{\mathrm{d}t}H([\boldsymbol{Z}]) = \boldsymbol{h}^\circ(T) \cdot \mathbb{S}\boldsymbol{J}([\boldsymbol{Z}]) =: \dot{Q}.\tag{4.59}$$

Since the heat exchange with the thermal reservoir is the only mechanism to exchange energy, we recognize the heat flux $\dot{Q}$ on the rightmost-hand side of Eq. (4.59).

### 4.5.2    Second Law

We introduce the entropy $S$ as

$$S([\boldsymbol{Z}]) := -\frac{\partial G([\boldsymbol{Z}])}{\partial T} = \sum_{\alpha \in \mathcal{S}}(s_\alpha([Z_\alpha]) + R)[Z_\alpha],\tag{4.60}$$

---

[12]The enthalpy $H$ is actually the Legendre transform of the internal energy $U$. The natural variables of $U$ are entropy $S$, volume $V$, and concentrations $[\boldsymbol{Z}]$, and enthalpy is defined as

$$H(S, p, [\boldsymbol{Z}]) := [U(S, V, [\boldsymbol{Z}]) + pV]_{V = V(S, p, [\boldsymbol{Z}])},\tag{4.56}$$

with $V(S, p, [\boldsymbol{Z}])$ the function expressing the volume in terms of entropy $S$, pressure $p$, and concentrations $[\boldsymbol{Z}]$.

with

$$s_\alpha([Z_\alpha]) := s_\alpha^\circ(T) - R\log[Z_\alpha]\,, \tag{4.61}$$

$$s_\alpha^\circ(T) := -\partial\mu_\alpha^\circ(T)/\partial T\,, \tag{4.62}$$

being the molar entropy and the standard molar entropy of $Z_\alpha$, respectively. The time derivative of $S$ according to the rate equation (4.7) reads

$$\frac{\mathrm{d}}{\mathrm{d}t}S([\boldsymbol{Z}]) = \sum_{\alpha\in\mathcal{S}} s_\alpha([Z_\alpha])\frac{\mathrm{d}}{\mathrm{d}t}[Z_\alpha] \tag{4.63a}$$

$$= \boldsymbol{s}([\boldsymbol{Z}])\cdot\mathbb{S}\boldsymbol{J}([\boldsymbol{Z}]) \tag{4.63b}$$

$$= T^{-1}(\boldsymbol{h}^\circ(T) - \boldsymbol{\mu}([\boldsymbol{Z}]))\cdot\mathbb{S}\boldsymbol{J}([\boldsymbol{Z}]) \tag{4.63c}$$

with $\boldsymbol{s}([\boldsymbol{Z}]) = (\ldots, s_\alpha([Z_\alpha]), \ldots)^{\mathrm{T}}_{\alpha\in\mathcal{S}}$ and $\boldsymbol{s}([\boldsymbol{Z}]) = T^{-1}(\boldsymbol{h}^\circ(T) - \boldsymbol{\mu}([\boldsymbol{Z}]))$ [13]. By recognizing the (reversible) entropy flow

$$\dot{S}_\mathrm{e} := T^{-1}\boldsymbol{h}^\circ(T)\cdot\mathbb{S}\boldsymbol{J}([\boldsymbol{Z}]) = \frac{\dot{Q}}{T}\,, \tag{4.65}$$

representing the (reversible) entropy changes in the environment, and the entropy production rate

$$\dot{\Sigma} := -T^{-1}\boldsymbol{\mu}([\boldsymbol{Z}])\cdot\mathbb{S}\boldsymbol{J}([\boldsymbol{Z}]) \geq 0\,, \tag{4.66}$$

we obtain the nonequilibrium formulation of the second law of thermodynamics for closed CRNs:

$$\frac{\mathrm{d}}{\mathrm{d}t}S([\boldsymbol{Z}]) = \dot{S}_\mathrm{e} + \dot{\Sigma}\,. \tag{4.67}$$

Note that, by using Eq. (4.54), the entropy production rate can be written as

$$\dot{\Sigma} = -T^{-1}\sum_{\rho\in\mathcal{R}}\Delta_\rho G([\boldsymbol{Z}])J_\rho([\boldsymbol{Z}]) \geq 0\,, \tag{4.68}$$

which, by using the local detailed balance condition (4.55), can be rewritten in a manifestly non-negative form

$$\dot{\Sigma} = R\sum_{\rho\in\mathcal{R}}\underbrace{(r_\rho^+([\boldsymbol{Z}]) - r_\rho^-([\boldsymbol{Z}]))\log\frac{r_\rho^+([\boldsymbol{Z}])}{r_\rho^-([\boldsymbol{Z}])}}_{=-\Delta_\rho G([\boldsymbol{Z}])J_\rho([\boldsymbol{Z}])/(RT)} \geq 0\,. \tag{4.69}$$

We so notice that each term $-\Delta_\rho G([\boldsymbol{Z}])J_\rho([\boldsymbol{Z}])$ of the sum in Eq. (4.68) is greater than or equal to zero: the thermodynamic forces and currents are always aligned, namely, $-\Delta_\rho G > 0$ (resp. $-\Delta_\rho G < 0$) if and only if $J_\rho > 0$ (resp. $J_\rho < 0$).

*Remark.* The entropy production rate vanishes only at equilibrium, namely, when $\Delta_\rho G([\boldsymbol{Z}]_\mathrm{eq}) = 0$ and $J_\rho([\boldsymbol{Z}]_\mathrm{eq}) = 0$ for every reaction $\rho$.

---

[13]Indeed, $\boldsymbol{h}^\circ(T) = \partial(\mu_\alpha/T)/(\partial 1/T)$ implies

$$h_\alpha^\circ(T) = \frac{1}{T}\frac{\partial\mu_\alpha}{\partial 1/T} + \mu_\alpha\frac{\partial 1/T}{\partial 1/T} \tag{4.64a}$$

$$= \frac{1}{T}\frac{\partial\mu_\alpha}{\partial T}\frac{\partial T}{\partial 1/T} + \mu_\alpha\frac{\partial 1/T}{\partial 1/T} \tag{4.64b}$$

$$= \frac{1}{T}s_\alpha(-T^2) + \mu_\alpha \tag{4.64c}$$

proving that $\boldsymbol{s}([\boldsymbol{Z}]) = T^{-1}(\boldsymbol{h}^\circ(T) - \boldsymbol{\mu}([\boldsymbol{Z}]))$.

### 4.5.3    Thermodynamic Potential

We now show that the Gibbs free energy in Eq. (4.52) is the proper thermodynamic potential of closed CRNs, namely, it monotonically decreases during the dynamics and it is lower bounded by its equilibrium value given in Eq. (4.41).

First, we examine the time derivative of $G([\boldsymbol{Z}])$ and we obtain

$$\frac{\mathrm{d}}{\mathrm{d}t}G([\boldsymbol{Z}]) = \sum_{\alpha \in \mathcal{S}} \mu_\alpha([Z_\alpha]) \frac{\mathrm{d}}{\mathrm{d}t}[Z_\alpha] \tag{4.70a}$$

$$= \boldsymbol{\mu}([\boldsymbol{Z}]) \cdot \mathbb{S}\boldsymbol{J}([\boldsymbol{Z}]) \tag{4.70b}$$

and, by using the definition of the entropy production rate in Eq. (4.66),

$$\frac{\mathrm{d}}{\mathrm{d}t}G([\boldsymbol{Z}]) = -T\dot{\Sigma} \leq 0\,, \tag{4.71}$$

which physically means that the dynamics dissipates the Gibbs free energy.

*Remark.* Equation (4.71) is an alternative formulation of the second law (4.67) for closed CRNs.

Second, we examine the difference $G([\boldsymbol{Z}]) - G([\boldsymbol{Z}]_{\mathrm{eq}})$ (with $G([\boldsymbol{Z}])$ and $G([\boldsymbol{Z}]_{\mathrm{eq}})$ given in Eqs. (4.52) and (4.41), respectively). To do so, we start by inspecting Eq. (4.44). We notice that the vector of the equilibrium chemical potentials belongs to the cokernel of the stoichiometric matrix. Hence, it can be written as a linear combination of conservation laws (defined in Eq. (4.27)):

$$\boldsymbol{\mu}([\boldsymbol{Z}]_{\mathrm{eq}}) = \sum_\lambda f_\lambda \boldsymbol{\ell}^\lambda\,, \tag{4.72}$$

with $\{f_\lambda\}$ some coefficients. Consequently,

$$\boldsymbol{\mu}([\boldsymbol{Z}]_{\mathrm{eq}}) \cdot [\boldsymbol{Z}]_{\mathrm{eq}} = \sum_\lambda f_\lambda \underbrace{\boldsymbol{\ell}^\lambda \cdot [\boldsymbol{Z}]_{\mathrm{eq}}}_{=L^\lambda} \tag{4.73a}$$

$$= \sum_\lambda f_\lambda \underbrace{\boldsymbol{\ell}^\lambda \cdot [\boldsymbol{Z}]}_{=L^\lambda} \tag{4.73b}$$

$$= \boldsymbol{\mu}([\boldsymbol{Z}]_{\mathrm{eq}}) \cdot [\boldsymbol{Z}]\,. \tag{4.73c}$$

This implies that

$$\boldsymbol{\mu}([\boldsymbol{Z}]) \cdot [\boldsymbol{Z}] - \boldsymbol{\mu}([\boldsymbol{Z}]_{\mathrm{eq}}) \cdot [\boldsymbol{Z}]_{\mathrm{eq}} = RT \sum_{\alpha \in \mathcal{S}} [Z_\alpha] \log \frac{[Z_\alpha]}{[Z_\alpha]_{\mathrm{eq}}}\,, \tag{4.74}$$

and we can write the difference $G([\boldsymbol{Z}]) - G([\boldsymbol{Z}]_{\mathrm{eq}})$ as a relative entropy (or Kullback–Leibler divergence) for non-normalized distributions:

$$G([\boldsymbol{Z}]) - G([\boldsymbol{Z}]_{\mathrm{eq}}) = RT\mathcal{D}([\boldsymbol{Z}]||[\boldsymbol{Z}]_{\mathrm{eq}}) \geq 0\,, \tag{4.75}$$

with

$$\mathcal{D}([\boldsymbol{Z}]||[\boldsymbol{Z}]_{\mathrm{eq}}) := \sum_{\alpha \in \mathcal{S}} \left( [Z_\alpha] \log \frac{[Z_\alpha]}{[Z_\alpha]_{\mathrm{eq}}} - ([Z_\alpha] - [Z_\alpha]_{\mathrm{eq}}) \right) \geq 0\,. \tag{4.76}$$

Note that $\mathcal{D}([\boldsymbol{Z}]||[\boldsymbol{Z}]_{\mathrm{eq}}) \geq 0$ because of the logarithmic inequality $\log x \leq x - 1$ (when $x = [Z_\alpha]/[Z_\alpha]_{\mathrm{eq}}$).

*Mathematical Remark.* Equations (4.71) and (4.75) prove that the Gibbs free energy plays the role of a Lyapunov function for the closed system dynamics.

## 4.6 Thermodynamics of Open CRNs

We now generalize the results obtained in Sec. 4.5 to open CRNs. Before doing that, we first introduce the thermodynamic meaning of the exchange currents entering the rate equation (4.7).

### 4.6.1 Chemostats

In these lecture notes, we consider CRNs coupled to chemostats. This might be interpreted as if each exchanged species $Z_\alpha$ is involved in a chemical reaction (labeled $\rho_\alpha^{\mathrm{ex}}$) like

$$Y_\alpha \; \xrightleftharpoons{\;\rho_\alpha^{\mathrm{ex}}\;} \; Y_\alpha^{\mathrm{ex}} , \tag{4.77}$$

with the corresponding chemostats $Y_\alpha^{\mathrm{ex}}$. These reactions are assumed to be always equilibrated. This thermodynamically means that the chemical potential of each exchanged species is controlled by the corresponding chemostats:

$$\mu_\alpha([Y_\alpha]) = \mu_\alpha^{\mathrm{ex}} , \tag{4.78}$$

for every $\alpha \in \mathcal{S}_Y$.

*Remark.* The chemical potentials of the exchanged species $\boldsymbol{\mu}_Y = (\ldots, \mu_\alpha, \ldots)_{\alpha \in \mathcal{S}_Y}^{\mathrm{T}}$ are thus externally controlled parameters like their concentrations $[\boldsymbol{Y}]$.

### 4.6.2 First and Second Law

When the chemical species are exchanged with the environment, by using the rate equation (4.7), the first law (4.59) becomes

$$\frac{\mathrm{d}}{\mathrm{d}t} H([\boldsymbol{Z}]) = \dot{Q} + \sum_{\alpha \in \mathcal{S}_Y} h_\alpha^\circ(T) I_\alpha . \tag{4.79}$$

Therefore, while the expression of the second law (4.67) remains the same, the entropy flow (4.65) becomes

$$\dot{S}_{\mathrm{e}} = \frac{\dot{Q}}{T} + \sum_{\alpha \in \mathcal{S}_Y} s_\alpha([Z_\alpha]) I_\alpha . \tag{4.80}$$

The second term on the right-hand side of Eq. (4.79) (resp. Eq. (4.80)) accounts for the molar energy (resp. entropy) exchanged with the environment through the exchange of chemical species.

The formulation of the second law in Eq. (4.71) becomes

$$\frac{\mathrm{d}}{\mathrm{d}t} G([\boldsymbol{Z}]) = -T\dot{\Sigma} + \sum_{\alpha \in \mathcal{S}_Y} \mu_\alpha([Z_\alpha]) I_\alpha , \tag{4.81}$$

where

$$\dot{w}_{\mathrm{chem}} := \sum_{\alpha \in \mathcal{S}_Y} \mu_\alpha([Z_\alpha]) I_\alpha . \tag{4.82}$$

is the chemical work rate quantifying the free energy exchanged with the environment through the exchange of chemical species. By providing free energy, the chemical work can balance dissipation and maintain CRNs out of equilibrium.

*Physical Remark.* These formulations of the first and second laws do not provide any precise guideline to understand which mechanisms (if any) maintain CRNs out of equilibrium (besides the exchange processes with the chemostats). To identify these mechanisms, we need to use the conservation laws.

### 4.6.3 Broken Conservation Laws

In open CRNs, some of the quantities defined in Eq. (4.28) are not conserved anymore. The corresponding conservation laws are said to be *broken*. The other conservation laws, which still correspond to conserved quantities, are said to be *unbroken*. To systematically identify broken and unbroken conservation laws, we proceed as follows.

First, we consider the $n_\mathrm{u} \leq n_X$ (linearly independent) left-null eigenvectors of the (sub)stoichiometric matrix $\mathbb{S}^X$, identified by the index $\lambda_\mathrm{u}$:

$$\boldsymbol{\ell}_X^{\lambda_\mathrm{u}} \cdot \mathbb{S}^X = 0 \,, \tag{4.83}$$

where the subscript $X$ stresses that $\boldsymbol{\ell}_X^{\lambda_\mathrm{u}}$ has $n_X$ entries (corresponding to the internal species). The vectors

$$\boldsymbol{\ell}^{\lambda_\mathrm{u}} := \begin{pmatrix} \boldsymbol{\ell}_X^{\lambda_\mathrm{u}} \\ \mathbf{0}_Y \end{pmatrix} \,, \tag{4.84}$$

with $\mathbf{0}_Y$ the vector with $n_Y$ null entries, are unbroken conservation laws. Indeed,

$$\boldsymbol{\ell}^{\lambda_\mathrm{u}} \cdot \mathbb{S} = \boldsymbol{\ell}_X^{\lambda_\mathrm{u}} \cdot \mathbb{S}^X + \mathbf{0}_Y \cdot \mathbb{S}^Y = 0 \,, \tag{4.85}$$

because of Eq. (4.83), and the quantities

$$L^{\lambda_\mathrm{u}} = \boldsymbol{\ell}^{\lambda_\mathrm{u}} \cdot [\boldsymbol{Z}] \tag{4.86}$$

are conserved quantities in open CRNs:

$$\frac{\mathrm{d}}{\mathrm{d}t} L^{\lambda_\mathrm{u}} = \boldsymbol{\ell}^{\lambda_\mathrm{u}} \cdot \frac{\mathrm{d}}{\mathrm{d}t}[\boldsymbol{Z}] = \underbrace{\boldsymbol{\ell}^{\lambda_\mathrm{u}} \cdot \mathbb{S}\boldsymbol{J}([\boldsymbol{Z}])}_{=0} + \underbrace{\mathbf{0}_Y \cdot \boldsymbol{I}^Y}_{=0} = 0 \,, \tag{4.87}$$

where we used Eq. (4.7).

Second, we use the set $\{\boldsymbol{\ell}^{\lambda_\mathrm{u}}\}$ of unbroken conservation laws to express the set of conservation laws $\{\boldsymbol{\ell}^\lambda\}$ according to

$$\{\boldsymbol{\ell}^\lambda\} = \{\boldsymbol{\ell}^{\lambda_\mathrm{u}}\} \cup \{\boldsymbol{\ell}^{\lambda_\mathrm{b}}\} \tag{4.88}$$

where $\{\boldsymbol{\ell}^{\lambda_\mathrm{b}}\}$ is the set of $n_\mathrm{b}$ broken conservation laws (with $n_\ell = n_\mathrm{u} + n_\mathrm{b}$). Indeed, each vector $\boldsymbol{\ell}_Y^{\lambda_\mathrm{b}} = (\ldots, \ell_\alpha^{\lambda_\mathrm{b}}, \ldots)_{\alpha \in \mathcal{S}_Y}^\mathrm{T}$ (collecting the entries of $\boldsymbol{\ell}^{\lambda_\mathrm{b}}$ corresponding to the exchanged species) satisfies $\boldsymbol{\ell}_Y^{\lambda_\mathrm{b}} \neq \mathbf{0}_Y$, otherwise $\boldsymbol{\ell}^{\lambda_\mathrm{b}}$ would be an unbroken conservation law. Hence, the quantities

$$L^{\lambda_\mathrm{b}} = \boldsymbol{\ell}^{\lambda_\mathrm{b}} \cdot [\boldsymbol{Z}] \tag{4.89}$$

are in general not conserved (and their variation is solely due to the exchanges with the chemostats):

$$\frac{\mathrm{d}}{\mathrm{d}t} L^{\lambda_\mathrm{b}} = \boldsymbol{\ell}^{\lambda_\mathrm{b}} \cdot \frac{\mathrm{d}}{\mathrm{d}t}[\boldsymbol{Z}] = \underbrace{\boldsymbol{\ell}^{\lambda_\mathrm{b}} \cdot \mathbb{S}\boldsymbol{J}([\boldsymbol{Z}])}_{=0} + \underbrace{\boldsymbol{\ell}_Y^{\lambda_\mathrm{b}} \cdot \boldsymbol{I}^Y}_{\neq 0} \neq 0 \,. \tag{4.90}$$

*Example.* When the species P, W, S, and F of the CRN (4.3) are exchanged, the conservation laws $\boldsymbol{\ell}^\mathrm{S}$ and $\boldsymbol{\ell}^\mathrm{F}$ in Eq. (4.30) are broken, while the conservation law $\boldsymbol{\ell}^\mathrm{E}$ in Eq. (4.30) is unbroken. In Fig. 4.3, we show that the moieties corresponding to the conservation laws $\boldsymbol{\ell}^\mathrm{S}$ and $\boldsymbol{\ell}^\mathrm{F}$ are exchanged with chemostats.

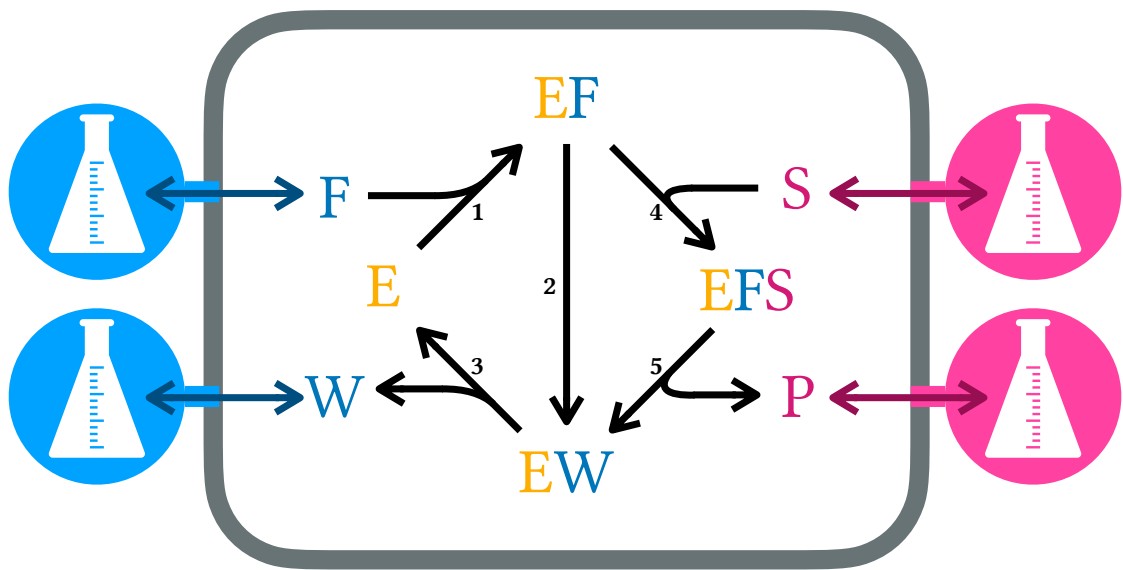

Figure 4.3: Pictorial illustration of the CRN (4.3) where the moieties corresponding to the conservation laws $\boldsymbol{\ell}^{\mathrm{S}}$ and $\boldsymbol{\ell}^{\mathrm{F}}$ in Eq. (4.30) are exchanged with chemostats. Crucially, the moiety $\boldsymbol{\ell}^{\mathrm{S}}$ (resp. $\boldsymbol{\ell}^{\mathrm{F}}$) is exchanged with the purple (resp. blue) chemostats exchanging S and P (resp. F and W).

### 4.6.4   Force and Potential Exchanged Species

In open CRNs the same moiety (the fragment of molecules identified by a conservation law) can be exchanged with more than one chemostat (as shown in Fig. 4.3), which can be identified by using the broken conservation laws.

We split the set of exchanged species $\{Y_\alpha\}$, and the corresponding set of indexes $\mathcal{S}_Y$, into two disjoint subsets: the set of $n_{Y_{\mathrm{p}}}$ *potential* species $\{Y_\alpha\}$ with $\alpha \in \mathcal{S}_{Y_{\mathrm{p}}}$, and the set of $n_{Y_{\mathrm{f}}}$ *force* species $\{Y_\alpha\}$ with $\alpha \in \mathcal{S}_{Y_{\mathrm{f}}}$. The former is defined as the smallest subset of exchanged species such that all the vectors $\{\boldsymbol{\ell}^{\lambda_{\mathrm{b}}}\}$ — independently of the specific representation — satisfy

$$\boldsymbol{\ell}_{Y_{\mathrm{p}}}^{\lambda_{\mathrm{b}}} \neq \mathbf{0}_{Y_{\mathrm{p}}} , \tag{4.91}$$

with $\boldsymbol{\ell}_{Y_{\mathrm{p}}}^{\lambda_{\mathrm{b}}} = (\dots, \ell_\alpha^{\lambda_{\mathrm{b}}}, \dots)_{\alpha \in \mathcal{S}_{Y_{\mathrm{p}}}}^{\mathrm{T}}$ (collecting the entries of $\boldsymbol{\ell}^{\lambda_{\mathrm{b}}}$ corresponding to the potential species), and $\mathbf{0}_{Y_{\mathrm{p}}}$ being the vector with $n_{Y_{\mathrm{p}}}$ null entries.

*Mathematical Remark.* The vectors $\{\boldsymbol{\ell}_{Y_{\mathrm{p}}}^{\lambda_{\mathrm{b}}}\}$ are linearly independent. Indeed, if they were linearly dependent, there would a representation of $\{\boldsymbol{\ell}^{\lambda_{\mathrm{b}}}\}$ such that

$$\boldsymbol{\ell}_{Y_{\mathrm{p}}}^{\lambda_{\mathrm{b}}} = \mathbf{0}_{Y_{\mathrm{p}}} , \tag{4.92}$$

for at least one $\lambda_{\mathrm{b}}$. This, together with $\mathcal{S}_{Y_{\mathrm{p}}}$ being the smallest subset such that Eq. (4.91) holds, implies that the number of broken conservations is equal to the number of potential species, i.e., $n_{Y_{\mathrm{p}}} = n_{\mathrm{b}}$.

*Mathematical Remark.* By applying the same splitting to the $(n_{\mathrm{b}} \times n_s)$ matrix $\mathbb{L}^{\mathrm{b}}$, collecting all the broken conservation laws as row vectors, we get

$$\mathbb{L}^{\mathrm{b}} = (\mathbb{L}_X^{\mathrm{b}}, \mathbb{L}_{Y_{\mathrm{f}}}^{\mathrm{b}}, \mathbb{L}_{Y_{\mathrm{p}}}^{\mathrm{b}}) , \tag{4.93}$$

where $\mathbb{L}_{Y_{\mathrm{p}}}^{\mathrm{b}}$ is a $(n_{\mathrm{b}} \times n_{Y_{\mathrm{p}}})$ square and invertible matrix (since the vectors $\{\boldsymbol{\ell}_{Y_{\mathrm{p}}}^{\lambda_{\mathrm{b}}}\}$ are linearly independent).

*Mathematical Remark.* By applying the same splitting to the stoichiometric matrix, we get

$$\mathbb{S} = \begin{pmatrix} \mathbb{S}^X \\ \mathbb{S}^{Y_{\mathrm{f}}} \\ \mathbb{S}^{Y_{\mathrm{p}}} \end{pmatrix}. \tag{4.94}$$

By removing all rows of the (sub)stoichiometric matrix $\mathbb{S}^{Y_{\mathrm{p}}}$ from $\mathbb{S}$, the only linearly-independent left null eigenvectors of the resulting matrix satisfy [14]:

$$\begin{pmatrix} \boldsymbol{\ell}_X^{\lambda_{\mathrm{u}}} \\ \mathbf{0}_{Y_{\mathrm{f}}} \end{pmatrix}, \tag{4.95}$$

with $\mathbf{0}_{Y_{\mathrm{f}}}$ being the vector with $n_{Y_{\mathrm{f}}}$ null entries.

*Physical Remark.* Every time a potential species is exchanged with a chemostat, a new conservation law is broken and a new moiety is exchanged with that chemostat. When a force species is exchanged with a chemostat, no new conservation laws are broken and that chemostat exchanges a moiety with the CRN that was already exchanged with another chemostat, thus establishing a moiety-flux between two (or more) chemostats.

*Remark.* The splitting of the exchanged species in force and potential species is not unique. Different choices have different physical interpretation.

*Example.* Possible sets of potential species of the CRN (4.3) when the species P, W, S, and F are exchanged are $\{\mathrm{P}, \mathrm{W}\}$ implying

$$\boldsymbol{\ell}_{Y_{\mathrm{p}}}^{\mathrm{S}} = \begin{array}{c} \mathrm{P} \\ \mathrm{W} \end{array}\begin{pmatrix} 1 \\ 0 \end{pmatrix} \quad \text{and} \quad \boldsymbol{\ell}_{Y_{\mathrm{p}}}^{\mathrm{F}} = \begin{array}{c} \mathrm{P} \\ \mathrm{W} \end{array}\begin{pmatrix} 0 \\ 1 \end{pmatrix}, \tag{4.96}$$

or $\{\mathrm{P}, \mathrm{F}\}$ implying

$$\boldsymbol{\ell}_{Y_{\mathrm{p}}}^{\mathrm{S}} = \begin{array}{c} \mathrm{P} \\ \mathrm{F} \end{array}\begin{pmatrix} 1 \\ 0 \end{pmatrix} \quad \text{and} \quad \boldsymbol{\ell}_{Y_{\mathrm{p}}}^{\mathrm{F}} = \begin{array}{c} \mathrm{P} \\ \mathrm{F} \end{array}\begin{pmatrix} 0 \\ 1 \end{pmatrix}, \tag{4.97}$$

or $\{\mathrm{S}, \mathrm{W}\}$ implying

$$\boldsymbol{\ell}_{Y_{\mathrm{p}}}^{\mathrm{S}} = \begin{array}{c} \mathrm{W} \\ \mathrm{S} \end{array}\begin{pmatrix} 0 \\ 1 \end{pmatrix} \quad \text{and} \quad \boldsymbol{\ell}_{Y_{\mathrm{p}}}^{\mathrm{F}} = \begin{array}{c} \mathrm{W} \\ \mathrm{S} \end{array}\begin{pmatrix} 1 \\ 0 \end{pmatrix}, \tag{4.98}$$

or $\{\mathrm{S}, \mathrm{F}\}$ implying

$$\boldsymbol{\ell}_{Y_{\mathrm{p}}}^{\mathrm{S}} = \begin{array}{c} \mathrm{S} \\ \mathrm{F} \end{array}\begin{pmatrix} 1 \\ 0 \end{pmatrix} \quad \text{and} \quad \boldsymbol{\ell}_{Y_{\mathrm{p}}}^{\mathrm{F}} = \begin{array}{c} \mathrm{S} \\ \mathrm{F} \end{array}\begin{pmatrix} 0 \\ 1 \end{pmatrix}. \tag{4.99}$$

Let us consider the species $\{\mathrm{S}, \mathrm{F}\}$ as potential species. This physically means that when the species S (resp. F) is exchanged the conservation law $\boldsymbol{\ell}^{\mathrm{S}}$ (resp. $\boldsymbol{\ell}^{\mathrm{F}}$) is broken. If afterwards, also the species P (resp. W) is exchanged, no new conservation law is broken, but the moiety corresponding to the conservation law $\boldsymbol{\ell}^{\mathrm{S}}$ (resp. $\boldsymbol{\ell}^{\mathrm{F}}$) is now exchanged with two different chemostats (see Fig. 4.3). Notice that the matrix $\mathbb{L}^{\mathrm{b}}$ now reads

$$\mathbb{L}^{\mathrm{b}} = \begin{array}{c} \boldsymbol{\ell}^{\mathrm{S}} \\ \boldsymbol{\ell}^{\mathrm{F}} \end{array}\begin{array}{c} \mathrm{E} \quad \mathrm{EF} \quad \mathrm{EW} \quad \mathrm{EFS} \quad \mathrm{P} \quad \mathrm{W} \quad \mathrm{S} \quad \mathrm{F} \\ \left( \begin{array}{cccc|cc|cc} 0 & 0 & 0 & 1 & 1 & 0 & 1 & 0 \\ 0 & 1 & 1 & 1 & 0 & 1 & 0 & 1 \end{array} \right). \end{array} \tag{4.100}$$

where the vertical lines split $\mathbb{L}^{\mathrm{b}}$ into $\mathbb{L}_X^{\mathrm{b}}$, $\mathbb{L}_{Y_{\mathrm{f}}}^{\mathrm{b}}$, and $\mathbb{L}_{Y_{\mathrm{p}}}^{\mathrm{b}}$.

---

[14] This can be proven by repeating the same reasoning we used to identify the unbroken conservation laws in Subs. 4.6.3. Every time we remove a row of $\mathbb{S}^{Y_{\mathrm{p}}}$ from $\mathbb{S}$, we break one conservation law because of Eq. (4.91) and the dimension of the cokernel of resulting matrix thus decreases.

### 4.6.5    Decomposition of the Entropy Production Rate

We now decompose the entropy production into a potential change and two work contributions. One work contribution accounts for the flows of moieties maintained across the CRN breaking the detailed balance condition. The other is due to time-dependent changes in the externally controlled chemostated concentrations. This decomposition constitutes a new formulation of the second law for open CRNs.

We start by rewriting the entropy production rate in Eq. (4.66) by explicitly accounting for the different sets of species (i.e., internal, force, and potential species):

$$T\dot{\Sigma} := -\big(\boldsymbol{\mu}_X \cdot \mathbb{S}^X + \boldsymbol{\mu}_{Y_\mathrm{f}} \cdot \mathbb{S}^{Y_\mathrm{f}} + \boldsymbol{\mu}_{Y_\mathrm{p}} \cdot \mathbb{S}^{Y_\mathrm{p}}\big)\boldsymbol{J} \geq 0\,, \tag{4.101}$$

where $\boldsymbol{\mu}_i = (\ldots, \mu_\alpha, \ldots)_{\alpha \in i}$ with $i = \{\mathcal{S}_X, \mathcal{S}_{Y_\mathrm{f}}, \mathcal{S}_{Y_\mathrm{p}}\}$, and we do not explicitly write the $[\boldsymbol{Z}]$ dependence of the chemical potentials and reaction currents for the sake of compactness.

*Remark.* Recall that the chemical potentials of the exchanged species, $\boldsymbol{\mu}_Y = (\boldsymbol{\mu}_{Y_\mathrm{f}}, \boldsymbol{\mu}_{Y_\mathrm{p}})^\mathrm{T}$, are controlled by the chemostats.

We then need to recognize that some of the moieties exchanged with the chemostats are stored in the internal species, while some other moieties are just transferred between chemostats via the CRN. To do so, we use the conservation laws. Given that

$$\mathbb{L}^\mathrm{b}\mathbb{S} = 0 = \mathbb{L}_X^\mathrm{b}\mathbb{S}^X + \mathbb{L}_{Y_\mathrm{f}}^\mathrm{b}\mathbb{S}^{Y_\mathrm{f}} + \mathbb{L}_{Y_\mathrm{p}}^\mathrm{b}\mathbb{S}^{Y_\mathrm{p}}\,, \tag{4.102}$$

and using that $\mathbb{L}_{Y_\mathrm{p}}^\mathrm{b}$ can be inverted (as discussed in 4.6.4), we obtain

$$\mathbb{S}^{Y_\mathrm{p}} = -\big(\mathbb{L}_{Y_\mathrm{p}}^\mathrm{b}\big)^{-1}\big(\mathbb{L}_X^\mathrm{b}\mathbb{S}^X + \mathbb{L}_{Y_\mathrm{f}}^\mathrm{b}\mathbb{S}^{Y_\mathrm{f}}\big)\,, \tag{4.103}$$

binding the variation of the number of molecules of the potential species to the other species. By inserting Eq. (4.103) in Eq. (4.101), we get

$$\begin{aligned} T\dot{\Sigma} = -\big(&\boldsymbol{\mu}_X \cdot \mathbb{S}^X + \boldsymbol{\mu}_{Y_\mathrm{f}} \cdot \mathbb{S}^{Y_\mathrm{f}} + \\ &- \boldsymbol{\mu}_{Y_\mathrm{p}} \cdot (\mathbb{L}_{Y_\mathrm{p}}^\mathrm{b})^{-1}(\mathbb{L}_X^\mathrm{b}\mathbb{S}^X + \mathbb{L}_{Y_\mathrm{f}}^\mathrm{b}\mathbb{S}^{Y_\mathrm{f}})\big)\boldsymbol{J}\,. \end{aligned} \tag{4.104}$$

By adding and subtracting $\boldsymbol{\mu}_{Y_\mathrm{p}} \cdot (\mathbb{L}_{Y_\mathrm{p}}^\mathrm{b})^{-1}\mathbb{L}_{Y_\mathrm{p}}^\mathrm{b}\mathbb{S}^{Y_\mathrm{p}} = \boldsymbol{\mu}_{Y_\mathrm{p}} \cdot \mathbb{S}^{Y_\mathrm{p}}$, the entropy production rate becomes

$$T\dot{\Sigma} = -\mathcal{F} \cdot \mathbb{S}\boldsymbol{J}\,, \tag{4.105}$$

where

$$\mathcal{F} = \big(\boldsymbol{\mu} \cdot \mathbb{I} - \boldsymbol{\mu}_{Y_\mathrm{p}} \cdot (\mathbb{L}_{Y_\mathrm{p}}^\mathrm{b})^{-1}\mathbb{L}^\mathrm{b}\big)^\mathrm{T} \tag{4.106}$$

and $\mathbb{I}$ is the identity matrix. Note the term $\boldsymbol{\mu}_{Y_\mathrm{p}} \cdot (\mathbb{L}_{Y_\mathrm{p}}^\mathrm{b})^{-1}\mathbb{L}^\mathrm{b}$ now appearing in Eq. (4.105) does not contribute to the entropy production rate since $\mathbb{L}^\mathrm{b}\mathbb{S} = 0$, but it is crucial to derive the proper thermodynamic potential of open CRNs as we show in the following.

To do so, we guess the following state function

$$\Psi([\boldsymbol{Z}]) = \mathcal{F} \cdot [\boldsymbol{Z}]\,, \tag{4.107}$$

whose time derivative

$$\frac{\mathrm{d}}{\mathrm{d}t}\Psi([\boldsymbol{Z}]) = \mathcal{F} \cdot \frac{\mathrm{d}}{\mathrm{d}t}[\boldsymbol{Z}] + \frac{\mathrm{d}}{\mathrm{d}t}\mathcal{F} \cdot [\boldsymbol{Z}] \tag{4.108}$$

according to the rate equation (4.7) reads

$$\begin{aligned} \frac{\mathrm{d}}{\mathrm{d}t}\Psi([\boldsymbol{Z}]) = &\mathcal{F} \cdot \mathbb{S}\boldsymbol{J} + \mathcal{F}_Y \cdot \boldsymbol{I}^Y + RT \sum_{\alpha \in \mathcal{S}} \frac{\mathrm{d}}{\mathrm{d}t}[Z_\alpha] \\ &- \frac{\mathrm{d}}{\mathrm{d}t}(\boldsymbol{\mu}_{Y_\mathrm{p}}) \cdot (\mathbb{L}_{Y_\mathrm{p}}^\mathrm{b})^{-1}\mathbb{L}^\mathrm{b}[\boldsymbol{Z}] \end{aligned} \tag{4.109}$$

with $\mathcal{F}_Y = \left(\boldsymbol{\mu}_Y \cdot \mathbb{I} - \boldsymbol{\mu}_{Y_\mathrm{p}} \cdot (\mathbb{L}_{Y_\mathrm{p}}^\mathrm{b})^{-1} \mathbb{L}_Y^\mathrm{b}\right)^\mathrm{T}$ and $\mathbb{L}_Y^\mathrm{b} = (\mathbb{L}_{Y_\mathrm{f}}^\mathrm{b}, \mathbb{L}_{Y_\mathrm{p}}^\mathrm{b})$.

To proceed, i) we express $-\mathcal{F} \cdot \mathbb{S}\boldsymbol{J} = T\dot{\Sigma}$ as a function of the other terms, ii) we recognize that $\mathcal{F}_Y \cdot \boldsymbol{I}^Y = \mathcal{F}_{Y_\mathrm{f}} \cdot \boldsymbol{I}^{Y_\mathrm{f}}$, and iii) we define $\mathcal{G}([\boldsymbol{Z}]) := \Psi([\boldsymbol{Z}]) - RT \sum_{\alpha \in \mathcal{S}}[Z_\alpha]$. We thus get

$$T\dot{\Sigma} = -\frac{\mathrm{d}}{\mathrm{d}t}\mathcal{G}([\boldsymbol{Z}]) + \dot{w}_\mathrm{nc} + \dot{w}_\mathrm{driv} \geq 0 \,. \tag{4.110}$$

Here, we introduced the new free energy $\mathcal{G}$

$$\mathcal{G}([\boldsymbol{Z}]) = \boldsymbol{\mu} \cdot [\boldsymbol{Z}] - RT \sum_{\alpha \in \mathcal{S}}[Z_\alpha] - \boldsymbol{\mu}_{Y_\mathrm{p}} \cdot (\mathbb{L}_{Y_\mathrm{p}}^\mathrm{b})^{-1} \mathbb{L}^\mathrm{b}[\boldsymbol{Z}] \tag{4.111}$$

which can be written using the Gibbs free energy as

$$\mathcal{G}([\boldsymbol{Z}]) = G([\boldsymbol{Z}]) - \boldsymbol{\mu}_{Y_\mathrm{p}} \cdot [\boldsymbol{m}] \,, \tag{4.112}$$

by using Eq. (4.52), and defining the concentration of the moieties as

$$[\boldsymbol{m}] = (\mathbb{L}_{Y_\mathrm{p}}^\mathrm{b})^{-1} \mathbb{L}^\mathrm{b}[\boldsymbol{Z}] \,. \tag{4.113}$$

Equation (4.112) is reminiscent of the grand canonical free energy in equilibrium thermodynamics: when passing from the canonical to the grand canonical ensemble, the grand canonical free energy is obtained from the Gibbs free energy by eliminating the energetic contribution due to the matter exchange with the environment. Here, something similar happens: we eliminate the energetic contribution due to the exchanged moieties, i.e., $\boldsymbol{\mu}_{Y_\mathrm{p}} \cdot [\boldsymbol{m}]$. This is the reason why $\mathcal{G}([\boldsymbol{Z}])$ is also called a *semi*-grand Gibbs free energy. Note that $\mathcal{G}([\boldsymbol{Z}])$ is a state function and therefore its time derivative vanishes at steady-state. Note also that if $\dot{w}_\mathrm{nc} = 0$ and $\dot{w}_\mathrm{driv} = 0$, Eq (4.110) simplifies to

$$\frac{\mathrm{d}}{\mathrm{d}t}\mathcal{G}([\boldsymbol{Z}]) = -T\dot{\Sigma}(t) \leq 0 \,. \tag{4.114}$$

Namely, the semigrand free energy is dissipated (decreasing monotonously in time) and CRNs relax towards an equilibrium state since $\mathcal{G}([\boldsymbol{Z}])$ is lower bounded by its equilibrium value $\mathcal{G}([\boldsymbol{Z}]_\mathrm{eq})$ (see 4.6.6).

The nonconservative work rate is given by

$$\dot{w}_\mathrm{nc} = \mathcal{F}_{Y_\mathrm{f}} \cdot \boldsymbol{I}^{Y_\mathrm{f}} \,, \tag{4.115}$$

and quantifies the energetic cost of maintaining fluxes of moieties between different chemostats through the CRN. These fluxes result from the fundamental nonconservative forces

$$\mathcal{F}_{Y_\mathrm{f}} = \left(\boldsymbol{\mu}_{Y_\mathrm{f}} \cdot \mathbb{I} - \boldsymbol{\mu}_{Y_\mathrm{p}} \cdot (\mathbb{L}_{Y_\mathrm{p}}^\mathrm{b})^{-1} \mathbb{L}_{Y_\mathrm{f}}^\mathrm{b}\right)^\mathrm{T} \,. \tag{4.116}$$

Indeed, $\dot{w}_\mathrm{nc}$ always vanishes when all the chemostatted species break a conservation law, i.e., $\mathcal{S}_Y = \mathcal{S}_{Y_\mathrm{p}}$. Only when there are some $\mathcal{S}_{Y_\mathrm{f}}$ species (i.e., when at least one moiety is exchanged with more than one chemostat), the forces (4.116) breaking the detailed balance condition and keeping the system out of equilibrium can emerge. This is the reason why the $\{Y_\alpha\}$ species with $\alpha \in \mathcal{S}_{Y_\mathrm{f}}$ are named *force* species, while the $\{Y_\alpha\}$ species with $\alpha \in \mathcal{S}_{Y_\mathrm{p}}$ are named *potential* species. The forces (4.116) can still vanish where there are force species. This happens when $\boldsymbol{\mu}_{Y_\mathrm{f}} \cdot \mathbb{I} = \boldsymbol{\mu}_{Y_\mathrm{p}} \cdot (\mathbb{L}_{Y_\mathrm{p}}^\mathrm{b})^{-1} \mathbb{L}_{Y_\mathrm{f}}^\mathrm{b}$. We stress that $\boldsymbol{\mu}_{Y_\mathrm{p}} \cdot (\mathbb{L}_{Y_\mathrm{p}}^\mathrm{b})^{-1} \mathbb{L}_{Y_\mathrm{f}}^\mathrm{b}$ encodes the values of chemical potential $\boldsymbol{\mu}_{Y_\mathrm{f}}$ at the equilibrium to which open CRNs would

relax if there were only potential species [15]. Hence, the fundamental nonconservative forces (4.116) result from the chemostats maintaining the chemical potentials of the forces species to different values with respect to the equilibrium ones.

*Remark.* The nonconservative forces $\mathcal{F}_{Y_{\mathrm{f}}}$ and corresponding currents $\boldsymbol{I}^{Y_{\mathrm{f}}}$ do not have to be aligned. This allows for free energy transduction.

The driving work rate $\dot{w}_{\mathrm{driv}}$ is specified as

$$\dot{w}_{\mathrm{driv}} = -\frac{\mathrm{d}}{\mathrm{d}t}\boldsymbol{\mu}_{Y_{\mathrm{p}}} \cdot [\boldsymbol{m}]\,, \tag{4.119}$$

and results for the time dependent manipulation of the chemical potentials $\boldsymbol{\mu}_{Y_{\mathrm{p}}}$. It quantifies the energetic cost of modifying the equilibrium to which open CRNs would relax in the absence of nonconservative forces and vanishes in autonomous systems.

### 4.6.6  Thermodynamic potential

We conclude here by showing that $\mathcal{G}([\boldsymbol{Z}])$ is always lower bounded by its equilibrium value $\mathcal{G}([\boldsymbol{Z}]_{\mathrm{eq}})$. To do so, we use a similar strategy to the one we used in Subs. (4.5.3).

First, we use that $\boldsymbol{\mu}([\boldsymbol{Z}]_{\mathrm{eq}})$ is still a linear combination of conservation laws (given in Eq. (4.72)) because of Eq. (4.44). However, only the unbroken conservation laws still define conserved quantites. Thus,

$$\boldsymbol{\mu}([\boldsymbol{Z}]_{\mathrm{eq}}) \cdot [\boldsymbol{Z}]_{\mathrm{eq}} = \sum_{\lambda_{\mathrm{u}}} f_{\lambda_{\mathrm{u}}} \underbrace{\boldsymbol{\ell}^{\lambda_{\mathrm{u}}} \cdot [\boldsymbol{Z}]_{\mathrm{eq}}}_{=L^{\lambda_{\mathrm{u}}}} + \sum_{\lambda_{\mathrm{b}}} f_{\lambda_{\mathrm{b}}} \boldsymbol{\ell}^{\lambda_{\mathrm{b}}} \cdot [\boldsymbol{Z}]_{\mathrm{eq}} \tag{4.120a}$$

$$= \sum_{\lambda_{\mathrm{u}}} f_{\lambda_{\mathrm{u}}} \underbrace{\boldsymbol{\ell}^{\lambda_{\mathrm{u}}} \cdot [\boldsymbol{Z}]}_{=L^{\lambda_{\mathrm{u}}}} + \sum_{\lambda_{\mathrm{b}}} f_{\lambda_{\mathrm{b}}} \boldsymbol{\ell}^{\lambda_{\mathrm{b}}} \cdot [\boldsymbol{Z}]_{\mathrm{eq}} \tag{4.120b}$$

$$= \boldsymbol{\mu}([\boldsymbol{Z}]_{\mathrm{eq}}) \cdot [\boldsymbol{Z}] + \sum_{\lambda_{\mathrm{b}}} f_{\lambda_{\mathrm{b}}} \boldsymbol{\ell}^{\lambda_{\mathrm{b}}} \cdot [\boldsymbol{Z}]_{\mathrm{eq}} - \sum_{\lambda_{\mathrm{b}}} f_{\lambda_{\mathrm{b}}} \boldsymbol{\ell}^{\lambda_{\mathrm{b}}} \cdot [\boldsymbol{Z}] \tag{4.120c}$$

where we summed and subtracted $\sum_{\lambda_{\mathrm{b}}} f_{\lambda_{\mathrm{b}}} \boldsymbol{\ell}^{\lambda_{\mathrm{b}}} \cdot [\boldsymbol{Z}]$.

Second, we use that the entries of the unbroken conservation laws corresponding to the exchanged species vanish:

$$\boldsymbol{\mu}_{Y_{\mathrm{p}}} = \sum_{\lambda_{\mathrm{u}}} f_{\lambda_{\mathrm{u}}} \underbrace{\boldsymbol{\ell}^{\lambda_{\mathrm{u}}}_{Y_{\mathrm{p}}}}_{=0} + \sum_{\lambda_{\mathrm{b}}} f_{\lambda_{\mathrm{b}}} \boldsymbol{\ell}^{\lambda_{\mathrm{b}}}_{Y_{\mathrm{p}}} = \boldsymbol{f}_{\mathrm{b}} \cdot \mathbb{L}^{\mathrm{b}}_{Y_{\mathrm{p}}}\,, \tag{4.121}$$

where $\boldsymbol{f}_{\mathrm{b}} := (\ldots, f_{\lambda_{\mathrm{b}}}, \ldots)^{\mathrm{T}}$. This implies that

$$\boldsymbol{\mu}_{Y_{\mathrm{p}}} \cdot [\boldsymbol{m}] = \boldsymbol{f}_{\mathrm{b}} \cdot \mathbb{L}^{\mathrm{b}}_{Y_{\mathrm{p}}}[\boldsymbol{m}] \tag{4.122}$$

$$= \boldsymbol{f}_{\mathrm{b}} \cdot \mathbb{L}^{\mathrm{b}}_{Y_{\mathrm{p}}} (\mathbb{L}^{\mathrm{b}}_{Y_{\mathrm{p}}})^{-1} \mathbb{L}^{\mathrm{b}}[\boldsymbol{Z}] \tag{4.123}$$

$$= \boldsymbol{f}_{\mathrm{b}} \cdot \mathbb{L}^{\mathrm{b}}[\boldsymbol{Z}] \tag{4.124}$$

$$= \sum_{\lambda_{\mathrm{b}}} f_{\lambda_{\mathrm{b}}} \boldsymbol{\ell}^{\lambda_{\mathrm{b}}} \cdot [\boldsymbol{Z}] \tag{4.125}$$

---

[15] To prove that $\boldsymbol{\mu}_{Y_{\mathrm{p}}} \cdot (\mathbb{L}^{\mathrm{b}}_{Y_{\mathrm{p}}})^{-1} \mathbb{L}^{\mathrm{b}}_{Y_{\mathrm{f}}}$ is the equilibrium value of the chemical potentials of the force species if only the potential species were chemostatted, we use that the entries of the unbroken conservation laws corresponding to the exchanged species vanish and thus

$$\boldsymbol{\mu}^{\mathrm{eq}}_{Y_{\mathrm{f}}} = \sum_{\lambda_{\mathrm{u}}} f_{\lambda_{\mathrm{u}}} \underbrace{\boldsymbol{\ell}^{\lambda_{\mathrm{u}}}_{Y_{\mathrm{f}}}}_{=0} + \sum_{\lambda_{\mathrm{b}}} f_{\lambda_{\mathrm{b}}} \boldsymbol{\ell}^{\lambda_{\mathrm{b}}}_{Y_{\mathrm{f}}} = \boldsymbol{f}_{\mathrm{b}} \cdot \mathbb{L}^{\mathrm{b}}_{Y_{\mathrm{f}}}\,, \tag{4.117}$$

with $\boldsymbol{f}_{\mathrm{b}} := (\ldots, f_{\lambda_{\mathrm{b}}}, \ldots)^{\mathrm{T}}$. This together with Eq. (4.121) and the existence of $(\mathbb{L}^{\mathrm{b}}_{Y_{\mathrm{p}}})^{-1}$, namely, $\boldsymbol{\mu}_{Y_{\mathrm{p}}} \cdot (\mathbb{L}^{\mathrm{b}}_{Y_{\mathrm{p}}})^{-1} = \boldsymbol{f}_{\mathrm{b}} \cdot \mathbb{I}$ leads to

$$\boldsymbol{\mu}^{\mathrm{eq}}_{Y_{\mathrm{f}}} = \boldsymbol{\mu}_{Y_{\mathrm{p}}} \cdot (\mathbb{L}^{\mathrm{b}}_{Y_{\mathrm{p}}})^{-1} \mathbb{L}^{\mathrm{b}}_{Y_{\mathrm{f}}}\,, \tag{4.118}$$

as stated in the main text.

where we used the definition of the concentrations of the moieties given in Eq. (4.113), and analogously

$$\boldsymbol{\mu}_{Y_{\mathrm{p}}} \cdot [\boldsymbol{m}_{\mathrm{eq}}] = \sum_{\lambda_{\mathrm{b}}} f_{\lambda_{\mathrm{b}}} \boldsymbol{\ell}^{\lambda_{\mathrm{b}}} \cdot [\boldsymbol{Z}]_{\mathrm{eq}} . \tag{4.126}$$

Third, we use Eqs. (4.120c), (4.125) and (4.126) in the difference $\mathcal{G}([\boldsymbol{Z}]) - \mathcal{G}([\boldsymbol{Z}]_{\mathrm{eq}})$ and we obtain

$$\mathcal{G}([\boldsymbol{Z}]) - \mathcal{G}([\boldsymbol{Z}]_{\mathrm{eq}}) = \mathcal{D}([\boldsymbol{Z}]||[\boldsymbol{Z}]_{\mathrm{eq}}) \geq 0 , \tag{4.127}$$

with $\mathcal{D}([\boldsymbol{Z}]||[\boldsymbol{Z}]_{\mathrm{eq}})$ defined in Eq. (4.76).

*Example. Autonomous and Detailed Balanced.* Consider the CRN (4.3) when only the species S and F are exchanged and their concentrations are maintained constants. Both species S and F are potential species. The semigrand free energy (4.112) thus reads

$$\mathcal{G}([\boldsymbol{Z}]) = G([\boldsymbol{Z}]) - \mu_{\mathrm{F}} L^{\mathrm{F}} - \mu_{\mathrm{S}} L^{\mathrm{S}} , \tag{4.128}$$

where $\mu_{\mathrm{F}}$ and $\mu_{\mathrm{S}}$ are the (constant) chemical potential of F and S, respectively. For this setup, the second law is given in Eq. (4.114) since there are no force species and the concentrations [S] and [F] are constant. This, together with $\mathcal{G}([\boldsymbol{Z}])$ being lower bounded by its equilibrium value, implies that the CRN is detailed balance and will eventually reach an equilibrium state despite being open. Note that, by using the thermodynamic theory we derived, we can predict that the CRN will reach an equilibrium state without solving the dynamics. The only information we need is encoded in the stoichiometric matrix, i.e., in the topology of the CRN.

*Example. Nonautonomous and Detailed Balanced.* Consider the CRN (4.3) when only the species S and F are exchanged and their concentrations are changed in time according to unspecified protocols. The semigrand free energy (4.112) is still given in Eq. (4.128), since there are no new potential species, but the second law now becomes

$$T\dot{\Sigma} = -\frac{\mathrm{d}}{\mathrm{d}t}\mathcal{G}([\boldsymbol{Z}]) + \dot{w}_{\mathrm{driv}} , \tag{4.129}$$

where

$$\dot{w}_{\mathrm{driv}} = -\left(\frac{\mathrm{d}}{\mathrm{d}t}\mu_{\mathrm{F}}\right) L^{\mathrm{F}} - \left(\frac{\mathrm{d}}{\mathrm{d}t}\mu_{\mathrm{S}}\right) L^{\mathrm{S}} . \tag{4.130}$$

The CRN is still detailed balance, but the time dependent manipulation of the concentrations [F] and [S] provides free energy, in the form of the driving work (4.130), which changes the equilibrium to which the CRN would relax and balances the dissipation so maintaining the CRN out of equilibrium.

*Example. Autonomous and Nondetailed Balanced.* Consider the CRN (4.3) when the species P, W, S and F are exchanged and their concentrations are maintained constant. Exchanging the species P and W does not break any additional conservation laws, but creates the following nonconservative forces (obtained by specializing Eq. (4.116)):

$$\mathcal{F}_{\mathrm{P}} = \mu_{\mathrm{P}} - \mu_{\mathrm{S}} \quad \text{and} \quad \mathcal{F}_{\mathrm{W}} = \mu_{\mathrm{W}} - \mu_{\mathrm{F}} . \tag{4.131}$$

Hence, the semigrand free energy (4.112) is still given in Eq. (4.128), since there are no new potential species, but the second law now becomes

$$T\dot{\Sigma} = -\frac{\mathrm{d}}{\mathrm{d}t}\mathcal{G}([\boldsymbol{Z}]) + \dot{w}_{\mathrm{nc}} , \tag{4.132}$$

with

$$\dot{w}_{\mathrm{nc}} = (\mu_{\mathrm{P}} - \mu_{\mathrm{S}})I_{\mathrm{P}} + (\mu_{\mathrm{W}} - \mu_{\mathrm{F}})I_{\mathrm{W}} . \tag{4.133}$$

The CRN is not detailed balance because of the nonconservative forces (4.131) and will reach a nonequilbrium steady state. Note again that, by using the thermodynamic theory we derived, we can predict that the CRN will reach a nonequilibrium steady state without solving the dynamics.

### 4.7 Conclusions

We derived here, in a self-contained way, a thermodynamic theory for deterministic open CRNs in ideal dilute solutions. Crucially, we used the conservation laws to derive the formulation of the second law in Eq. (4.110) which identifies the specific mechanisms (nonconservative forces and driving) that can balance dissipation and maintain open CRNs out of equilibrium.

The same formulation of the second law (4.110) can be derived for deterministic non-ideal CRNs [30], deterministic reaction-diffusion systems [69, 70], and stochastic CRNs in ideal dilute solutions [29].

# 5 Stochastic Chemical Reaction Networks

**Matteo Polettini.** *I illustrate results about the stochastic dynamics and thermodyamics of mass-action kinetics chemical reaction networks by simple examples and outputs of simulations. In particular I focus on the role of deficiency for stationary fluctuations.*

## 5.1 Introduction

This lecture is an attempt to provide an incremental bottom-up presentation of techniques, results and nomenclature to treat the stochastic dynamics and thermodynamics of mass-action kinetics (MAK) chemical reaction networks (CRNs). The topics addressed would be enough for at least a full master course ranging from linear algebra to metabolic networks and quite possibly machine learning (*ça va sans dire*). In the two-hours span that was gently offered by the organizers I decided to provide by examples some basic intuitions, leaving generalizations and proofs to the willingness of the student and to the literature.

## 5.2 Fishing for Poisson

Consider the chemical reaction

$$\emptyset \rightleftharpoons \mathrm{X}. \tag{5.1}$$

Let $\hat{X}(t)$ be the population (number of molecules present in the reactor) of species X at time $t$[16]. Apart from species $\emptyset$ that has fixed population $[\emptyset](t) = 1$ (possibly because it is chemostatted via osmosis or other mechanisms, see Lecture 4), this is a fluctuating quantity taking integer values $X \in \mathbb{N}$, due to what is sometimes called intrinsic or population noise (to distinguish it from outside disturbances). We can represent the population space as

$$0 \longrightarrow 1 \longrightarrow 2 \longrightarrow 3 \longrightarrow \ldots \tag{5.2}$$

MAK prescribes that the rate of injection of molecules is constant, while that of the ejection of molecules is proportional to the population

$$
\begin{aligned}
r_+(X) &= k_+ \\
r_-(X) &= k_- \, X.
\end{aligned} \tag{5.3}
$$

These are called reaction rates or velocities, while the $k_\pm$ are called rate constants.

Consider the probability $p_t(X) = p(\hat{X}(t) \equiv X)$ that at time $t$ the reactor contains $X$ molecules and define the (average) population-space current from $X - 1$ to $X$ as

$$j_t(X, X - 1) = r_+(X - 1)p_t(X - 1) - r_-(X)p_t(X), \tag{5.4}$$

with $j_t(0, -1) = 0$ The evolution of the probability is dictated by the Chemical Master Equation

$$\frac{d}{dt}p_t(X) = j_t(X, X - 1) - j_t(X + 1, X) \tag{5.5}$$

given some initial distribution $p_0$.

---

[16]In this lecture, differing from Lecture 4, we use capital italic letters $X$ for stochastic populations and small-case italic letters $x$ for deterministic concentrations in place of $[X]$).

```python
from random import uniform
from math import log

kp = 2          # rate constant forward
km = 1          # rate constant backward
x = 0           # initial population
t = 100000.     # total time
tau = 0         # time elapsed

h1, h2 = {}, {}    # output histograms

while tau < t:

  rp = kp        # calculate rate forward with MAK
  rm = km * x    # calculate rate backward with MAK

  r = rp + rm

  y1 = uniform(0,1)
  y2 = uniform(0,1)

  s = log(1/y1)/r # sample time by inversion rule

  h1[x] = h1.get(x, 0) + s     # count time spent at state
  h2[x] = h2.get(x,0) + 1      # count state occurrences

  x == x - 1 if y2 < rp / r else x == x + 1 # sample new state
  tau += s
```

Figure 5.1: Doob-Gillespie algorithm to generate the histograms of the total number of occurrences of a population and of the total time spent at a state and compare to theory. The output is shown in Fig. 5.2.

One simple question we can ask is what is the distribution of the population in the stationary limit $t \to \infty$. Such distribution must be invariant over time translations, thus it is obtained by setting Eq. (5.5) to zero. We obtain

$$0 = j_\infty(1,0) = j_\infty(2,1) = \dots \tag{5.6}$$

leading to the principle of detailed balance, which prescribes

$$k_+ p_\infty(X) = k_-(X+1)p_\infty(X+1) \tag{5.7}$$

yielding

$$p_\infty(X) = \alpha \frac{\left(\frac{k_+}{k_-}\right)^X}{X!}, \tag{5.8}$$

where $\alpha = p(0)$ is fixed by normalization $\sum_X p_\infty(X) = 1$. Noting that $x_\infty \equiv x(\infty) = k_+/k_-$ is the unique fixed point $\lim_{t \to \infty} x(t)$ of the corresponding deterministic rate equation (see Lecture 4)

$$\frac{dx(t)}{dt} = k_+ - k_- x(t) \tag{5.9}$$

we find that $\alpha = e^{-x_\infty}$. Therefore Eq. (5.8) is just the the Poisson distribution with parameter $x_\infty$.

We can plot the stationary distribution and compare it to the average time spent at state $X$ via a Monte-Carlo Markov Chain / Doob-Gillespie algorithm (see Lecture 1). A simple imperative approach with basic Python is in Fig. 5.1 Its output Fig. 5.2 compares the total time spent at states vs. the total number of times a state is visited against the true solution Eq. (5.8), showing that only the former is correct (confusion between these two observables is a frequent early mistake in such kind of coding).

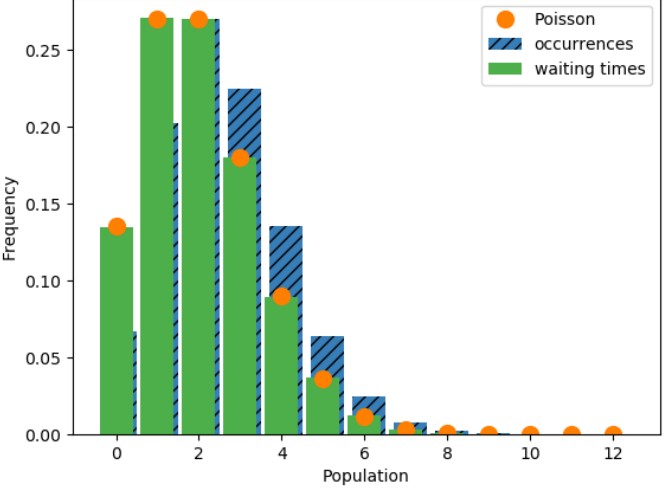

Figure 5.2: The output of algorithm 5.1. *Top histogram*: time spent at a state, normalized; *Bottom histogram*: number of times a state is visited, normalized; *Bullets*: values of the Poisson distribution.

The Doob-Gillespie algorithm produces the exact statistics for such kind of observables. However, notice that the estimation of $p_{t\to\infty}$ should be done on $n$ independent samples at large enough $t$. Here instead we are integrating along a single realization, hoping that sample averages equal time averages (ergodic principle). This assumption is not trivial and it lends itself to some errors that in principle should be evaluated or dealt with, in particular [73]:

- *Relaxation error*: we started counting the time spent at a state at $t = 0$, but in principle we should wait long-enough to let the system relax, and then we should start counting for long-enough. But how long is long?

- *Autocorrelation error*: the probability of being back to some state right after having been there is larger than being at a more distant state. This means that by counting all of the states visited we are favouring more probable ones ("the rich get richer"). To avoid this we should put a long-enough time between one sample and the next. But again, how long is long?

Of course all such petty issues are washed away by machine learning [17].

**Additional.**    Equation (5.5) is the backward Kolmogorov (Fokker-Panck) equation. One interesting question we can ask is: is there an analog of Brownian motion (Langevin equation) for chemical processes? That is, an equation directly expressed in terms of $\hat{X}(t)$?

Notice that we can write

$$\hat{X}(t) = \hat{X}_0 + \hat{N}_+(t) - \hat{N}_-(t) \tag{5.10}$$

where $\hat{X}_0$ is sampled with probability $p_0$ and $\hat{N}_\pm(t)$ are respectively the number of times reaction $\pm$ occurs.

Now let $\hat{N}(t)$ be a unit Poisson process with distribution $p(\hat{N}(t) \equiv \mathcal{N}) = e^{-t}t^{\mathcal{N}}/\mathcal{N}!$. Clearly the forward reaction is a Poisson process

$$\hat{N}_+(t) = \hat{N}(k_+ t) = \hat{N}\left(\int_0^t k_+ ds\right) = \hat{N}\left(\int_0^t k_+[\emptyset](s)ds\right) \tag{5.11}$$

where we made some dull manipulations. The backward reaction is not a homogeneous Poisson process because its rate depends on the number of molecules present in the reactor. However, inspired by the latter expression we are tempted to write

$$\hat{N}_-(t) = \hat{N}\left(\int_0^t k_- \hat{X}(s)ds\right). \tag{5.12}$$

This given, a wrong argument is that, given that the mean of a Poissonian with parameter $\lambda$ is $\lambda$, then the mean of Eq. (5.10) is

$$\langle \hat{X}(t)\rangle = \langle \hat{X}_0\rangle + k_+ t - \int_0^t k_- \langle \hat{X}(s)\rangle ds \tag{5.13}$$

---

[17]I am being overtly ironic about the way the machine learning hype is propagating into scientific narratives. Inference based on high-dimensional nonlinear functions is very fine-tuned to specific tasks, and in my opinion it is not poised as comes to the creativeness and rigour of scientific inquiry. However, there might exist interesting specific applications of such algorithms for CRNs, see passages of Ref. [74] for a review.

where $\langle \cdot \rangle$ is the average over samples. This equation in fact is just the integral solution of Eq. (5.9), which on the other hand can be obtained by multiplying by $X$ and summing over $X$ in the master equation Eq. (5.5).

This latter line of derivations is only true for linear systems, but it can be made more correct in terms of the mode in place of the mean in the large-scale limit, see e.g. Ref. [75] for a thorough explanation of random-time-change Poisson processes and their large-scale limits, that we will briefly introduce in Sec. 5.6, by means of a scaling parameter $\Omega$. Physicists often talk about "mean-field" behaviour by replacing

$$\langle X_t^2 \rangle \to \langle X_t \rangle^2 \tag{5.14}$$

which does reproduce the deterministic dynamics. However, this is a strong assumption that implies that the mean behaviour is representative of the process, in some sense. In fact, CRNs can display a very different behaviour among their stochastic and their deterministic counterparts. For example in Ref. [76] it has been shown that a so-called strongly endotactic CRN (whose deterministic dynamics falls into a closed domain) can have stochastic escape routes to infinity (and therefore no stationary distribution), while the deterministic dynamics stays confined in a compact set. This is a case where the mean and mode of the process deviate wildly (although it requires irreversible reactions).

## 5.3 Fishing for Poisson-like

Now consider

$$\emptyset \rightleftharpoons 2X \tag{5.15}$$

creating a complex $C_2 = 2X$ made of two interacting copies of species X out of the empty complex $C_1 = \emptyset$. The population space is

$$0 \quad 1 \quad 2 \quad 3 \quad 4 \quad \ldots \tag{5.16}$$

There are two disconnected components $\mathscr{C}_{\mathrm{even}} = 2\mathbb{N}$ and $\mathscr{C}_{\mathrm{odd}} = 2\mathbb{N} + 1$ (even and odd integers), called stoichiometric compatibility classes. Correspondingly, the parity of $\hat{X}(t)$ is conserved, given an initial even/odd number of molecules. Notice that this (broken) symmetry is not present in the corresponding deterministic rate equation $dx(t)/dt = k_+ - k_- x(t)^2$ (in Quantum Field Theory similar things are called anomalies). The stationary distribution now reads

$$p_\infty(X) = \alpha \frac{x_\infty{}^X}{X!} \left[ p_{\mathrm{even}} \mathbf{1}_{\mathscr{C}_{\mathrm{even}}}(X) + p_{\mathrm{odd}} \mathbf{1}_{\mathscr{C}_{\mathrm{odd}}}(X) \right] \tag{5.17}$$

where $\mathbf{1}_{\mathscr{A}}(a)$ is the indicator function[18] and $p_{\mathrm{even/odd}}$ are the normalized probabilities that the initial state is either even or odd, and $\alpha$ is again fixed by normalization yielding

$$\alpha = \left[ p_{\mathrm{even}} \cosh x_\infty ) + p_{\mathrm{odd}} \sinh x_\infty \right]^{-1}. \tag{5.18}$$

Notice that Eq. (5.17) looks Poissonian, but it is not because for a stochastic variable to be Poissonian it has to have the same expression over all the integers. Here they are split in two, and therefore we talk of Poisson-like or Poisson-form distribution.

---

[18]Namely, 1 if $a \in \mathscr{A}$ else 0. Quite conveniently, in the Doob-Gillespie algorithm shown in Fig. 5.1 the indicator function has not to be implemented explicitly because negative populations are never reached, given that the reaction $0 \to -1$ always has zero rate

## 5.4   Moieties and product-form

Now consider

$$X_2 + X_3 \rightleftharpoons 2X_1. \tag{5.19}$$

Each single realization of the reaction is a random event due to the collision of two molecules, e.g. of $X_2$ and $X_3$ forward, or of $X_1$ with itself backwards. MAK prescribes

$$\begin{aligned}
r_+(X_1, X_2, X_3) &= k_+ X_2 X_3 \\
r_-(X_1, X_2, X_3) &= k_- X_1 (X_1 - 1)
\end{aligned} \tag{5.20}$$

where the "$-1$" is due to the fact that there is one less molecule to bounce into (this correction is lost at the deterministic level).

The stoichiometric matrix

$$\mathbf{S} = \begin{pmatrix} +2 \\ -1 \\ -1 \end{pmatrix} \tag{5.21}$$

has a two-dimensional left-null space (obviously). As a meaningful basis of null co-vectors such that $\ell^\top \mathbf{S} = 0$ we prefer to choose ones that have small positive integer entries, for example $\ell_\text{mass}^\top = (1, 1, 1)$ (total mass conservation) and $\ell_\text{moie}^\top = (1, 2, 0)$. The former corresponds to mass conservation. Chemists call the latter a moiety, that is, a part of a molecule that has a name because it is also part of other molecules, in this case the two "disks" in $X_2$ and the one "disk" in $X_1$:

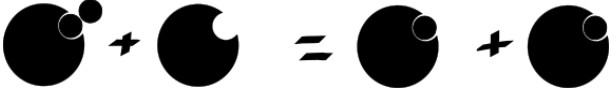

(mass is the ultimate moiety given that it is part of all molecules. Notice that we could have also chosen $\ell_\text{moie}^\top = (1, 0, 2)$, in which case the moiety would be the disk with a cavity). In general finding right or left null vectors of a stoichiometric matrix of chemical meaning is not just a purely theoretical exercise, but it entails some understanding of the chemistry behind the formalism. See Ref. [74] for some interesting recent advancements on the interpretation of moieties.

The associated deterministic system reads

$$\begin{aligned}
\frac{d}{dt} x_2(t) &= k_- x_1(t)^2 - k_+ x_2(t) x_3(t) \\
\lambda_\text{mass} &= x_1(t) + x_2(t) + x_3(t) \\
\lambda_\text{moie} &= x_1(t) + 2x_2(t)
\end{aligned} \tag{5.22}$$

where $\lambda_\text{mass}, \lambda_\text{moie}$ are the deterministically conserved quantities. Given these values, the above system has a unique fixed point $\boldsymbol{x}_\infty = \boldsymbol{x}(\infty) = \lim_{t \to \infty} \boldsymbol{x}(t)$. Notice that by the first equation the ratio

$$\frac{k_+}{k_-} = \frac{x_1(\infty)^2}{x_2(\infty) x_3(\infty)} \tag{5.23}$$

does not dependent explicitly on the values of the conserved quantities. However, in general each initial population will converge to a different fixed point parametrized by $\lambda_\text{mass}, \lambda_\text{moie}$.

The stochastic population space is composed of compatibility classes analogous to the even/odd classes in the previous example according to the values

$$
\begin{aligned}
L_{\text{mass}} &= X_1 + X_2 + X_3 \\
L_{\text{moie}} &= X_1 + 2X_2.
\end{aligned}
\tag{5.24}
$$

Visually, for $L_{\text{mass}} = 0$ (red dot), $L_{\text{mass}} = 1$ (green dots), $L_{\text{mass}} = 2$ (blue dots):

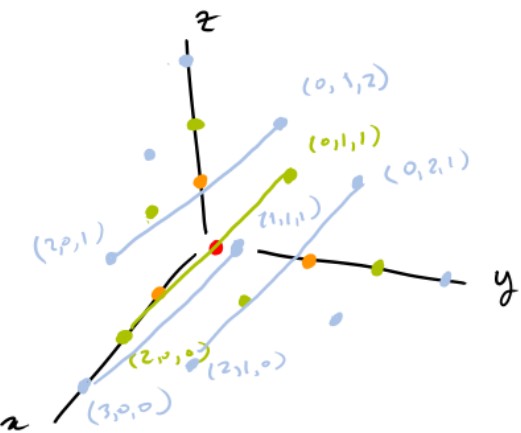

For higher masses and moieties each compatibility class is at most a finite interval

$$
(\overline{X}_1, \overline{X}_2, \overline{X}_3) \;\text{——}\; (\overline{X}_1 + 2, \overline{X}_2 - 1, \overline{X}_3 - 1) \;\text{——}\; \cdots \;\text{——}\; (\overline{X}_1 + 2N, \overline{X}_2 - N, \overline{X}_3 - N)
$$

ranging from either $\overline{X}_1 = 0, 1$ for some $N$ such that either $\overline{X}_2 = N$ or $\overline{X}_3 = N$. On each such subspace we can again use the principle of detailed balance

$$
k_-(X_1 + 2)(X_1 + 1)p_\infty(X_1 + 2, X_2 - 1, X_3 - 1) = k_+ X_2 X_3 p_\infty(X_1, X_2, X_3)
\tag{5.25}
$$

to propagate the solution to any population compatible with $(\overline{X}_1, \overline{X}_2, \overline{X}_3)$. We find for example

$$
\begin{aligned}
p_\infty(\overline{X}_1 + 4, \overline{X}_2 - 2, \overline{X}_3 - 2) &= \\
&= \frac{k_+}{k_-} \frac{(\overline{X}_2 - 1)(\overline{X}_3 - 1)}{(\overline{X}_1 + 4)(\overline{X}_1 + 3)} p_\infty(\overline{X}_1 + 2, \overline{X}_2 - 1, \overline{X}_3 - 1) \\
&= \left(\frac{k_+}{k_-}\right)^2 \frac{(\overline{X}_2 - 1)\overline{X}_2(\overline{X}_3 - 1)\overline{X}_3}{(\overline{X}_1 + 4)(\overline{X}_1 + 3)(\overline{X}_1 + 2)(\overline{X}_1 + 1)} p_\infty(\overline{X}_1, \overline{X}_2, \overline{X}_3) \\
&= \left[\frac{x_1(\infty)^2}{x_2(\infty)x_3(\infty)}\right]^2 \frac{\overline{X}_1!\overline{X}_2!\overline{X}_3!}{(\overline{X}_1 + 4)!(\overline{X}_2 - 2)!(\overline{X}_3 - 2)!} p_\infty(\overline{X}_1, \overline{X}_2, \overline{X}_3)
\end{aligned}
\tag{5.26}
$$

where in the second passage we used Eq. (5.23) and played a trick with factorials. We find

$$
p_\infty(\overline{X}_1 + 4, \overline{X}_2 - 2, \overline{X}_3 - 2) = \alpha \frac{x_1(\infty)^{\overline{X}_1 + 4}}{(\overline{X}_1 + 4)!} \frac{x_2(\infty)^{\overline{X}_2 - 2}}{(\overline{X}_2 - 2)!} \frac{x_3(\infty)^{\overline{X}_3 - 2}}{(\overline{X}_3 - 2)!}
\tag{5.27}
$$

where we played yet another trick by multiplying and dividing by $x_1(\infty)^{\overline{X}_1} x_2(\infty)^{\overline{X}_2} x_3(\infty)^{\overline{X}_3}$ and defining

$$
\alpha = \frac{\overline{X}_1!\overline{X}_2!\overline{X}_3!}{x_1(\infty)^{\overline{X}_1} x_2(\infty)^{\overline{X}_2} x_3(\infty)^{\overline{X}_3}} p_\infty(\overline{X}_1, \overline{X}_2, \overline{X}_3).
\tag{5.28}
$$

Proceeding like above once again to the next population one would find

$$p_\infty(\overline{X}_1 + 6, \overline{X}_2 - 3, \overline{X}_3 - 3) = \alpha \frac{x_1(\infty)^{\overline{X}_1+6}}{(\overline{X}_1 + 6)!} \frac{x_2(\infty)^{\overline{X}_2-3}}{(\overline{X}_2 - 3)!} \frac{x_3(\infty)^{\overline{X}_3-3}}{(\overline{X}_3 - 3)!}, \tag{5.29}$$

and so on. This kind of distribution is called product-form Poisson-like. Notice, quite interestingly, that while a solution of the related deterministic system enters the distribution, we could express this latter just in terms of $k_+/k_-$, as exemplified by the first passage of the derivation of Eq. (5.26). Thus the distribution does not depend on the deterministic conserved quantities $\lambda_{\text{mass}}, \lambda_{\text{moie}}$, and one can plug in any solution of the deterministic system. The computation of $\alpha$, containing all of the correlation between the species, is far from trivial.

## 5.5   One trivial cycle

Consider

$$\emptyset \overset{1}{\rightleftharpoons} X$$
$$\emptyset \overset{2}{\rightleftharpoons} X \tag{5.30}$$

where we assume that we can distinguish the two reactions. The population space is

$$0 \frown 1 \frown 2 \frown 3 \frown \ldots \tag{5.31}$$

By lumping together the two reaction rates into an effective $r_\pm(X) = r_{\pm 1}(X) + r_{\pm 2}(X)$ we re-obtain the first example in Sec. 5.2 with reaction rate constants $k_\pm = k_{\pm 1} + k_{\pm 2}$, thus from a kinetic point of view this example has nothing to offer and the stationary population is the Poissonian with parameter $(k_{+1} + k_{+2})/(k_{-1} + k_{-2})$.

Here we are rather interested in dynamical behaviour. Because we can distinguish two different mechanisms, when the system has relaxed to the stationary distribution there will be a net stationary circulation of currents

$$0 \frown 1 \frown 2 \frown 3 \frown \ldots \tag{5.32}$$

In Lecture 2 the cycle affinities were defined as the log-product of ratio of forward and backward rates of a process along any cycle. Here we would have to consider all of the infinite cycles, but we have that populations cancel out and the affinity is the same for all cycles

$$\log \frac{k_{+1}X k_{-2}(X + 1)}{k_{+2}(X + 1)k_{-1}X} = \log \frac{k_{+1}k_{-2}}{k_{+2}k_{-1}} =: \mathcal{A}. \tag{5.33}$$

For a chemical interpretation, we can think of $\emptyset$ to actually be species that are chemostatted, e.g.

$$Y_1 + Y_2 \overset{1}{\rightleftharpoons} X$$
$$Y_3 \overset{2}{\rightleftharpoons} X \tag{5.34}$$

where all $[Y_1], [Y_2], [Y_3]$ are assumed to be kept fixed in time (see Lecture 4). Notice that this only amounts to a rewriting the rate constants $k_{+1} = k'_{+1}[Y_1][Y_2]$ and

$k_{+2} = k'_{+2}[Y_3]$ on the assumption that when the chemostatted species are held to 1 (or to some standard chemical potential, on which we do not delve into), detailed balance holds $k'_{+1}/k'_{-1} = k'_{+2}/k'_{-2}$. In this case the affinity reads

$$\mathcal{A} = \log[Y_1] + \log[Y_2] - \log[Y_3] \tag{5.35}$$

and it dictates the direction of transport of the chemostatted chemicals through the environment as a cycle is performed

$$Y_1 + Y_2 \to Y_3, \tag{5.36}$$

which is reminiscent of the school's logo.

Now, defining the time-integrated current $\hat{J}(t) = \hat{N}_+(t) - \hat{N}_-(t)$ as the total number of reactions $+1$ minus the total number of reactions $-1$ occuring up to time $t$, that is, the number of all the transitions of the kind

$$\begin{array}{ccccc} & \overset{+1}{\frown} & \overset{+1}{\frown} & \overset{+1}{\frown} & \overset{+1}{\frown} \\ 0 & 1 & 2 & 3 & \dots, \end{array} \tag{5.37}$$

minus that of transitions

$$\begin{array}{ccccc} & \overset{-1}{\frown} & \overset{-1}{\frown} & \overset{-1}{\frown} & \overset{-1}{\frown} \\ 0 & 1 & 2 & 3 & \dots \end{array} \tag{5.38}$$

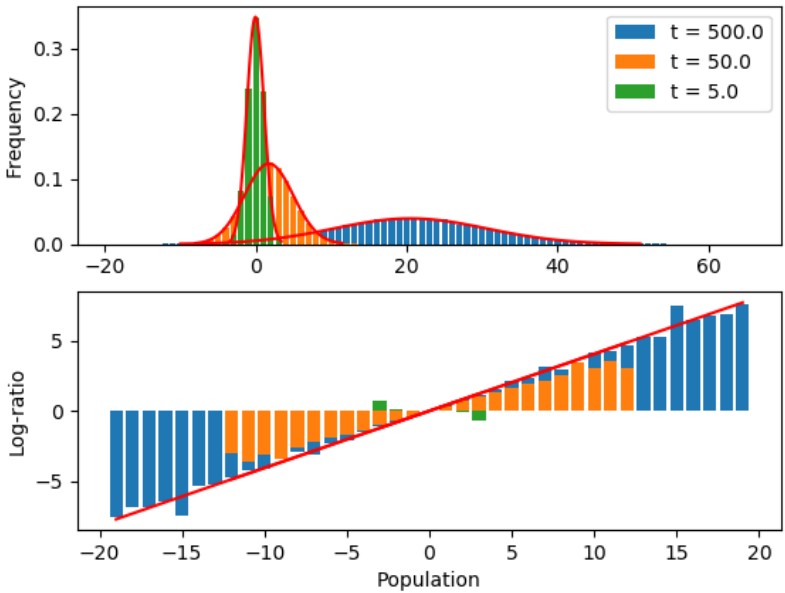

Figure 5.3: For $k_{+1} = 1, k_{-1} = 2, k_{+2} = 3, k_{-2} = 4$, initial population $\hat{X}(0) = 0$, number of samples 50000, for three values of the final time, histograms of the currents are their Gaussian fit, and log-ratios of positive to negative currents' probabilities.

one question we can ask is whether the following well-known fluctuation relation (see e.g. [77]) is satisfied:

$$\log \frac{p(\hat{J}(t) \equiv \mathcal{J})}{p(\hat{J}(t) \equiv -\mathcal{J})} \overset{?}{=} \mathcal{A}\mathcal{J}. \tag{5.39}$$

We plot the left-hand side quantity in the bottom panel of Fig. 5.3, for various values of time. First we notice that at short times there is a systematic bias due to dependence on the initial population. So we need to wait long enough to claim that a fluctuation relation is satisfied. But even then, is this relation of statistical significance[19]? Notice in fact that, from the first panel in Fig. 5.3, the distribution of the currents appears to be reasonably Gaussian, as we expect by the onset of the Central Limit Theorem. Thus the linearity here appears to be just a statistical artifact, and it is hard to claim a truly nonequilibrium phenomenon.

One way out of this conundrum may be to sample from an initial distribution such that the fluctuation relation is satisfied at all times, if it exists. One might be tempted to speculate that such initial distribution should be the stationary one. The top panel of Fig. 5.4 hints that the initial distribution that yields the fluctuation relation at all times is the one obtained by first letting the system relax to the equilibrium distribution of reaction 2 only, such that the population space is

---

[19]How can we claim to observe a nonequilibrium behaviour? For this to be, we would need to develop some sound statistical test of the fluctuation relation not being explained by a null hypothesis representing the linear regime.

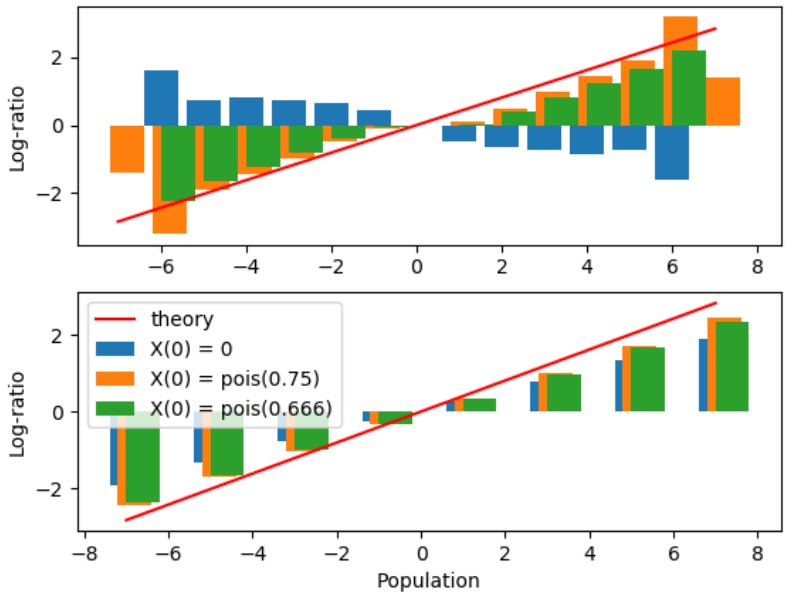

Figure 5.4: *Top panel:* Same rates as above, number of samples 5000000, final time $t = .4$, the log-ratio of forward to backward probabilities for the initial population $\hat{X}_0$: (blue) equal to 0 with certainty; (yellow) sampled with Poissonian with parameter $k_{+2}/k_{-2}$; (green) sampled with Poissonian with parameter $(k_{+2} + k_{+1})/(k_{-1} + k_{-2})$. *Bottom panel:* Same as above, but with stopping time the total number of reactions $\hat{N}_{+1} + \hat{N}_{-1} = 8$.

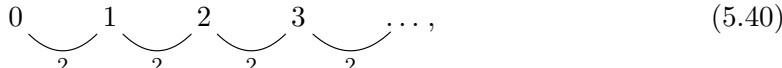

$$0 \underbrace{\phantom{xx}}_{2} 1 \underbrace{\phantom{xx}}_{2} 2 \underbrace{\phantom{xx}}_{2} 3 \underbrace{\phantom{xx}}_{2} \dots, \tag{5.40}$$

and then at time $t = 0$ turn on the other transitions and count the currents. The general proof for generic graphs is in [78], but this does not generally translate to CRNs so well.

A second way out is to consider fluctuation relations at stopping time, as in Ref. [3]. For example, in the bottom panel of Fig. 5.4 we stop the process exactly after a fixed number reactions $\hat{N}_{+1} + \hat{N}_{-1}$ have occurred, and it appears that convergence of the fluctuation relation is slightly faster.

## 5.6   One less trivial cycle: Schlögl model

Take

$$\emptyset \overset{1}{\rightleftharpoons} X$$
$$2X \overset{2}{\rightleftharpoons} 3X \tag{5.41}$$

with asymmetric population space

$$0 \overset{1}{\frown} 1 \overset{1}{\frown} 2 \underset{2}{\overset{1}{\oversetfrown{\frown}}} 3 \underset{2}{\overset{1}{\oversetfrown{\frown}}} \dots \tag{5.42}$$

To compute the stationary distribution we can lump together reactions to obtain the effective rates

$$r_+(X) = k_{+1}\Omega + k_{+2}\frac{X(X-1)}{\Omega}, \quad \text{for } X \geq 1$$
$$r_-(X) = k_{-1}X + k_{-2}\frac{X(X-1)(X-2)}{\Omega^2}, \quad \text{for } X \geq 2 \tag{5.43}$$

where we introduced a scaling parameter $\Omega$ to approach the deterministic limit (the reasoning behind the scaling is that by rescaling $X = \Omega x$ we want to reproduce a deterministic equation such as Eq. (5.9), therefore rate constants need to scale consistently, see also Lecture 8). Notice that these latter lumped reaction rates are not MAK.

State $X = 2$ is pivotal so we can compute the stationary distribution in the same way as above by propagating from $\alpha = p_\infty(2)$:

$$p_\infty(0) = \alpha \left( \frac{k_{-1}}{k_{+1}\Omega} \right)^2 \tag{5.44}$$

$$p_\infty(1) = \alpha \frac{k_{-1}}{k_{+1}\Omega} \tag{5.45}$$

$$p_\infty(X) = \alpha \prod_{X>2} \frac{k_{+1}\Omega + k_{+2}X(X-1)/\Omega}{k_{-1}X + k_{-2}X(X-1)(X-2)/\Omega^2}, \qquad \text{for } X > 2. \tag{5.46}$$

This distribution is definitely not Poisson-like (see plots in Fig. 5.5). In fact, this is a well-studied toy model (see e.g. [77,79,80]) due to Schlögl to study bistability and critical points. The Schlögl model is a special case of a the broader class of networks that have non-zero deficiency, a concept that we will explain below in greater detail. For now, just be aware that in this case the deficiency is $\delta = 1$ because the system obviously has a cycle,

but the network of complexes depicted in Eq. (5.41) has no "visible" cycle. Eventually we can make the cycle visible by turning from a representation in terms of a graph to one in

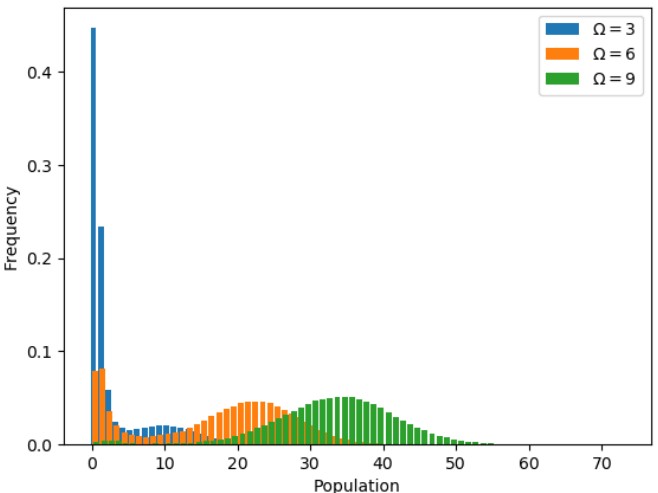

Figure 5.5: The stationary distribution for various values of the volume, for $k_{+1} = 0.5$, $k_{-1} = 3$, $k_{+2} = 4.6$, $k_{-2} = 1$.

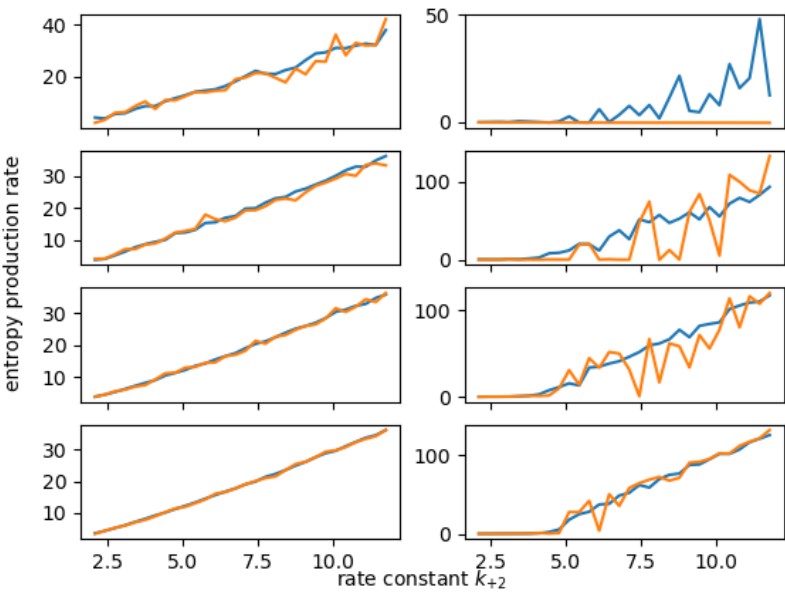

Figure 5.6: The mean (blue more regular curve) and most probable (yellow more zig-zagged curve) entropy production rate for the simple model (on the left) with rescaled rates $k_+ \to k_+\Omega$, $k_- \to k_-$ and the Schlögl model (on the right), $k_{+1} = .5$, $k_{-1} = 3$, $k_{+2} = 1$ as a function of $k_{-2}$, for four different values of the volume $\Omega = 4, 4^2, 4^3, 4^4$ (top to bottom), $\hat{X}(0) = 0$, final time $t = 10.0$ and $n = 10$ samples.

terms of a hypergraph (aka Petri net):

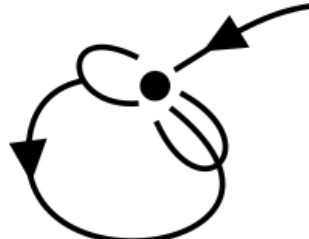

Here the bullet is species X, the edge coming into X from the blue is reaction 1 and the other hyperedge with three tails and two tips corresponds to reaction 2. Notice that in this example there are an equal number of incoming arrows as outcoming arrows from X (the bullet), thus indeed the two reactions form an (hyper-)cycle.

Let us now turn to the time-integrated current's statistics. I tried hard to come up with a plot of a figure analogous to Fig. 5.3 in the critical range of parameters. Importantly, the distributions I obtain are not bimodal, as one might have expected. Rather, they have a very fat tail. This is well-known and explained by the methods of large deviations in Lecture 6. However, it is very difficult to obtain negative currents and thus a significant fit for the fluctuation relation. Alas, once again we cannot display the fluctuation relation by numerical simulations, for different reasons than the ones pointed out in the previous example.

Instead, in Fig. 5.6 we compare the most probable value of the entropy production rate $\mathcal{A}\hat{J}(t)/t$ and its mean for the simple model studied above (on the left), and for the Schlögl model (on the right), for three different values of the volume and several values of rate $k_{+2}$. We observe that for the simple model basically there is not discrepancy between the two. This is due to the fact that for systems with zero deficiency noise does not systematically contribute to dissipation [81], and in fact all of the mean stationary currents coincide with their deterministic counterparts. Instead, in the Schlögl model we see a huge discrepancy between the mean and mode behaviour (in particular there appear to be two more likely values), that only vanishes away in the large volume limit.

## 5.7  Zero deficiency

The CRN

$$X_2 + X_3 \overset{1}{\rightleftharpoons} 2X_1$$
$$2X_1 \overset{2}{\rightleftharpoons} X_2 \tag{5.47}$$
$$X_2 \overset{3}{\rightleftharpoons} X_2 + X_3$$

looks significantly more complicated than the ones above, but is it really? The complexes are $(C_1, C_2, C_3) = (X_2 + X_3, 2X_1, X_2)$ and we can represent the network as the graph

$$
\begin{array}{l}
C_2 \\
\big| \quad \diagdown \\
C_3 \!\!-\!\!-\!\! C_1
\end{array}
\tag{5.48}
$$

Both the parity of $X_1$ and the moiety $X_1 + 2X_2$ are conserved. Each stoichiometric compatibility class $\mathscr{C}$ looks like

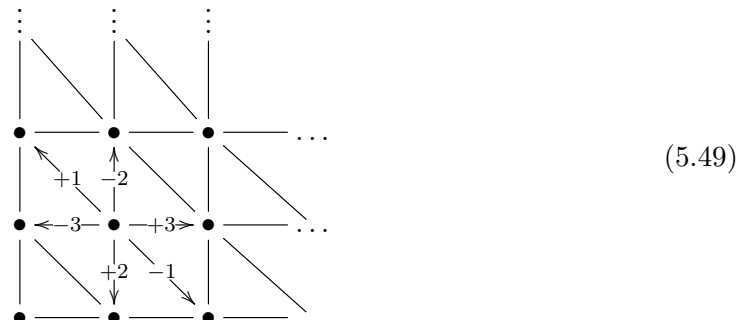

$$(5.49)$$

Notice that this space is very symmetrical, and that it is basically an infinite copy-paste of the above network of complexes 5.48. This is not always the case, as already observed in the Schlögl model.

The stoichiometric matrix reads

$$
\mathbf{S} = \begin{array}{c} \\ X_1 \\ X_2 \\ X_3 \end{array}
\begin{array}{ccc} {\scriptstyle 1} & {\scriptstyle 2} & {\scriptstyle 3} \end{array}
\left( \begin{array}{ccc}
+2 & -2 & 0 \\
-1 & +1 & 0 \\
-1 & 0 & +1
\end{array} \right) = \mathbf{KD}
\tag{5.50}
$$

where

$$
\mathbf{D} = \begin{array}{c} \\ C_1 \\ C_2 \\ C_3 \end{array}
\begin{array}{ccc} {\scriptstyle 1} & {\scriptstyle 2} & {\scriptstyle 3} \end{array}
\left( \begin{array}{ccc}
-1 & 0 & +1 \\
+1 & -1 & 0 \\
0 & +1 & -1
\end{array} \right)
\tag{5.51}
$$

is the incidence matrix of the complete graph whose vertices are the complexes and

$$
\mathbf{K} = \begin{array}{c} \\ X_1 \\ X_2 \\ X_3 \end{array}
\begin{array}{ccc} {\scriptstyle C_1} & {\scriptstyle C_2} & {\scriptstyle C_3} \end{array}
\left( \begin{array}{ccc}
0 & 1 & 1 \\
2 & 0 & 0 \\
0 & 1 & 0
\end{array} \right)
\tag{5.52}
$$

is sometimes referred to as Kirchhoff matrix, telling which species belong to which complexes. The quantity

$$
\delta = \dim \ker \mathbf{S} - \dim \ker \mathbf{D}
\tag{5.53}
$$

is the deficiency briefly mentioned above, where for a graph $\dim \ker \mathbf{S}$ is the number of independent cycles of the graph (see Lecture 2). Then, roughly, the deficiency is the number of independent stoichiometric cycles that cannot be visualized as cycles in the graph of complexes (but can eventually be visualized as cycles in the hypergraph).

In this case the deficiency turns out to be zero. Then several important results apply. In particular [82], the corresponding deterministic system has a unique fixed point $\boldsymbol{x}(\infty)$ subject to the deterministic conservation law $\ell = x_1(\infty) + 2x_2(\infty)$ found by solving the continuity equations

$$
\overbrace{k_{+1}x_2(\infty)x_3(\infty) - k_{-1}x_1(\infty)^2}^{\iota_1} = \overbrace{k_{+3}x_2(\infty) - k_{-3}x_2(\infty)x_3(\infty)}^{\iota_2} = \overbrace{k_{+2}x_1(\infty)^2 - k_{-2}x_2(\infty)}^{\iota_3}
\tag{5.54}
$$

where the overbraces define the stationary deterministic currents. Notice that once again there is no deterministic analogue of the conservation of parity. A proof that such fixed points are globally attractive is under review since seven years [83].

The stationary distribution of the master equation is found by solving the continuity equation at each node of the population network

$$\sum_{\rho \in \mathscr{R}} j_\infty(\rho, X_1, X_2, X_3) = 0, \tag{5.55}$$

where $\rho$ labels reactions and $\mathscr{R} = \{\pm 1, \pm 2, \pm 3\}$. Let us write this explicitly:

$$
\begin{aligned}
0 = &+ k_{+1} X_2 X_3 p_\infty(X_1, X_2, X_3) - k_{-1}(X_1 + 2)(X_1 + 1) p_\infty(X_1 + 2, X_2 - 1, X_3 - 1) \\
&+ k_{-1} X_1(X_1 - 1) p_\infty(X_1, X_2, X_3) - k_{+1}(X_2 + 1)(X_3 + 1) p_\infty(X_1 - 2, X_2 + 1, X_3 + 1) \\
&+ k_{+2} X_1(X_1 - 1) p_\infty(X_1, X_2, X_3) - k_{-2}(X_2 + 1) p_\infty(X_1 - 2, X_2 + 1, X_3) \\
&+ \text{etc.}
\end{aligned}
\tag{5.56}
$$

Clearly, solving this by hand is not feasible, but the important ACK theorem [84] stipulates that, since this system has zero deficiency, its stationary distribution is a product-form Poisson-like distribution with density

$$\alpha \frac{x_1(\infty)^{X_1}}{X_1!} \frac{x_2(\infty)^{X_2}}{X_2!} \frac{x_3(\infty)^{X_3}}{X_3!} \tag{5.57}$$

on any stoichiometric compatibility class. Let us check that this works by plugging into the above Eq. (5.56). After some tedious work one obtains:

$$
\begin{aligned}
0 = &\left[ k_{+1} x_2(\infty) x_3(\infty) - k_{-1} x_1(\infty)^2 \right] \frac{x_1(\infty)^{X_1} x_2(\infty)^{X_2 - 1} x_3(\infty)^{(X_3 - 1)}}{X_1!(X_2 - 1)!(X_3 - 1)!} \\
&+ \overbrace{\left[ k_{-1} x_1(\infty)^2 - k_{+1} x_2(\infty) x_3(\infty) \right]}^{-\iota_1} \frac{x_1(\infty)^{X_1 - 2} x_2(\infty)^{X_2} x_3(\infty)^{X_3}}{(X_1 - 2)! X_2! X_3!} \\
&+ \overbrace{\left[ k_{+2} x_1(\infty)^2 - k_{-2} x_2(\infty) \right]}^{\iota_2} \frac{x_1(\infty)^{X_1 - 2} x_2(\infty)^{X_2} x_3(\infty)^{X_3}}{(X_1 - 2)! X_2! X_3!} \\
&+ \text{etc.}
\end{aligned}
\tag{5.58}
$$

where we recognized the deterministic stationary currents defined in Eq. (5.54). But then notice that since $\iota_1 = \iota_2 = \iota_3$ and the prefactor is the same, the second and third terms cancel out. The same can be proven for all the other terms, which cancel out cyclically.

Notice that, as anticipated, the stationary distribution does not depend on the deterministic conserved quantities, and that we did not need to specify the stoichiometric compatibility class (that would be necessary to compute the normalization). For networks with non-zero deficiency, if one tries to plug in the multi-Poisson-like distribution in the generator of the Chemical Master Equation one obtains that the prefactors do not match as nicely, and terms do not cancel out. The above procedure is, in fact, the blueprint of the original proof of the ACK theorem.

## 5.8  Deficiency and response

First we start with the simple linear network

$$
\begin{aligned}
X_1 &\overset{1}{\rightleftharpoons} X_2 \\
X_2 &\overset{2}{\rightleftharpoons} X_3 \\
X_3 &\overset{3}{\rightleftharpoons} X_1
\end{aligned}
\tag{5.59}
$$

(clockwise orientation for rate constants $k_+$, counterclockwise for $k_-$). The unique deterministic fixed points $\boldsymbol{x}(\infty)$ such that $\mathbf{1} \cdot \boldsymbol{x}(\infty) = \ell_{\text{mass}}$ are found by applying the Markov Chain matrix-tree theorem (see Lecture 1). Having deficiency $\delta = 0$ the stationary distribution is product-form Poisson-like with stoichiometric subspaces labelled by $\mathbf{1} \cdot \boldsymbol{X} = L_{\text{mass}}$. As mentioned above, there is no relation between the choice of $\ell_{\text{mass}}$ and $L_{\text{mass}}$. The system is nonequilibrium with cycle affinity

$$\mathcal{A} = \log \frac{k_{+1}k_{+2}k_{+3}}{k_{-1}k_{-2}k_{-3}}. \tag{5.60}$$

We assume the above reactions to be hidden from the observer. We want to study how they affect the behaviour of an additional observable reaction

$$X_1 + 2X_2 \overset{4}{\rightleftharpoons} 3X_3. \tag{5.61}$$

Overall, the system has stoichiometric matrix

$$\mathbf{S} = \begin{pmatrix} -1 & 0 & +1 & -1 \\ +1 & -1 & 0 & -2 \\ 0 & +1 & -1 & +3 \end{pmatrix}. \tag{5.62}$$

The observable reaction respects the conservation law, and thus by a similar reasoning as in §5.2 overall the system has two chemical cycles (right-null vectors). The graph in the space of complexes is

$$\begin{array}{ccc}
C_1 \overset{1}{\rule{1cm}{0.4pt}} C_2 \\
\left. 2 \right| \quad \diagup 3 \qquad\qquad C_4 \overset{4}{\rule{1cm}{0.4pt}} C_5 \\
C_3
\end{array} \tag{5.63}$$

with incidence matrix

$$\mathbf{D} = \begin{pmatrix} -1 & 0 & +1 & 0 \\ +1 & -1 & 0 & 0 \\ 0 & +1 & -1 & 0 \\ 0 & 0 & 0 & -1 \\ 0 & 0 & 0 & +1 \end{pmatrix}. \tag{5.64}$$

The hypergraph instead is

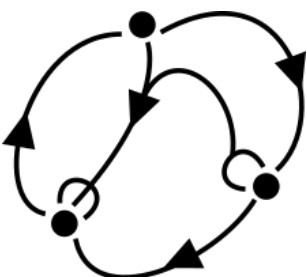

and there are two independent linear combinations of hyperedges such that an equal number of arrows go into and out of every node, e.g.

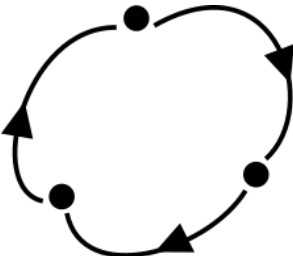
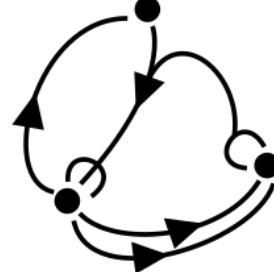

corresponding to the null eigenvectors of the stoichiometric matrix

$$
\begin{pmatrix} +1 \\ +1 \\ +1 \\ 0 \end{pmatrix}, \qquad \begin{pmatrix} 0 \\ -2 \\ +1 \\ +1 \end{pmatrix}. \tag{5.65}
$$

Notice that, differing from the first, the second is not a null vector of the incidence matrix, while it is of the stoichiometric matrix. Therefore the deficiency is $\delta = 1$. While graph cycles admit a simple physical interpretation in terms of stuff that is conserved throughout, hypercycles do not allow such simple visuals. An attempt to construct a dictionary and geometric interpretation is in Ref. [20].

Now let $a = \log k_{+4}/k_{-4}$ be a parameter regulating the intensity of the observable reaction, let $\hat{J}(t)$ be the its time-integrated current, and let

$$
j(a) = \lim_{t \to \infty} \frac{\langle \hat{J}(t) \rangle}{t} \tag{5.66}
$$

be the stationary mean current (notice that in general this is different from the deterministic current $\iota$ because the system has nonzero deficiency). We are interested in the relationship between the integrated current's scaled variance (as a function of $a$)

$$
\kappa(a) := \lim_{t \to \infty} \frac{\langle \hat{J}(t)^2 \rangle - \langle \hat{J}(t) \rangle^2}{t} \tag{5.67}
$$

and the current's response to a perturbation of $a$

$$
\rho(a) := \frac{\partial j(a)}{\partial a}. \tag{5.68}
$$

In the first panel of Fig. 5.7 we plot the mean current $j(a)$ and in the second the response-to-twice-variance $\kappa(a)/2\rho(a)$ for several choices of rate constants and $L_{\text{mass}} = 6$. Different data series correspond to different values of $a$. Different data points along a series correspond to different values of the affinity $\mathcal{A}$. The picture suggests that for the value $a = a^*$ such that the mean current is $j(a^*) = 0$ the following fluctuation-dissipation relation is satisfied

$$
\kappa(a^*) = 2\rho(a^*). \tag{5.69}
$$

This latter is a milestone of close-to-equilibrium statistical mechanics. However, here it would be displayed far from equilibrium ($\mathcal{A} \neq 0$) in states characterized by vanishing observable currents – called stalling instead of equilibrium – whose observable currents vanish, but that have non-observable flows.

**Additional:** The fluctuation-dissipation relation above can be easily derived if for some appropriate initial distribution the following integral fluctuation relation holds:

$$
1 = \left\langle e^{(a^*-a)\hat{J}} \right\rangle (a) \tag{5.70}
$$

for all values of $a$, in which case the exponent is a candidate for a physically meaningful measurement of (partial) entropy production. Notice that in general the average itself depends on $a$. In fact, by taking the first derivative with respect to $a$ and evaluating at

$a = a^*$ we find

$$0 = \frac{d}{da} \left\langle e^{(a^*-a)\hat{J}} \right\rangle (a) \bigg|_{a=a^*}$$

$$= \left[ \frac{\partial}{\partial a} \left\langle e^{(a^*-a)\hat{J}} \right\rangle (a) - \left\langle \hat{J} e^{(a^*-a)\hat{J}} \right\rangle (a) \right] \bigg|_{a=a^*}$$

$$= \left[ \frac{\partial}{\partial a} 1 - \left\langle \hat{J} e^{(a^*-a)\hat{J}} \right\rangle (a) \right] \bigg|_{a=a^*}$$

$$= -\langle \hat{J} \rangle (a^*) \tag{5.71}$$

where in the second passage we used the integral fluctuation relation above. This confirms that the average current at $a = a^*$ stalls. By taking the second total derivative with

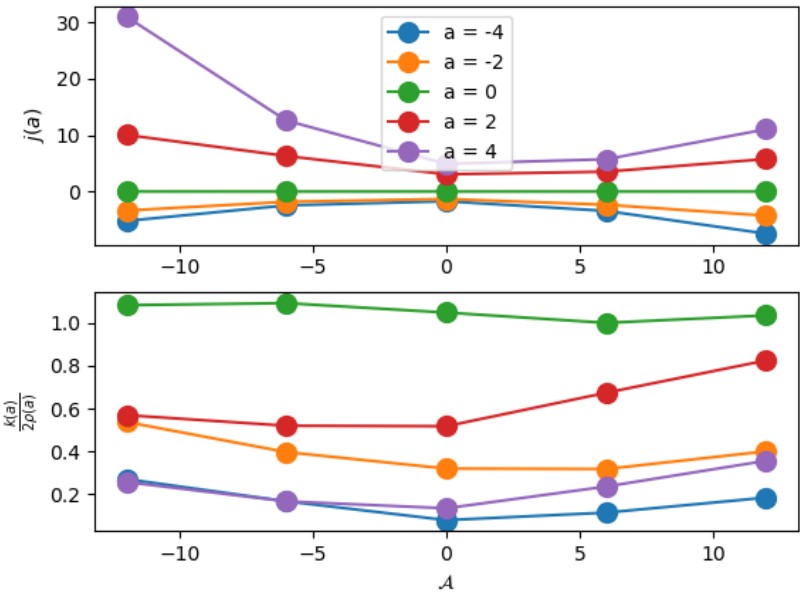

Figure 5.7: Top panel: the average current for trajectories stopping at time $t = 50$, 10000 samples, values of the hidden rates $k_{+1} = k_{+2} = k_{+3} = \exp \mathcal{A}/6$ and $k_{-1} = k_{-2} = k_{-3} = \exp -\mathcal{A}/6$ and of the observable rates $k_{+4} = \exp a/2$, $k_{-4} = \exp -a/2$ as a function of the affinity $\mathcal{A}$, for different values of $a$. Bottom panel: under the same conditions, ratio of the response over twice the variance of the current. For this subset of rates the stalling value is $a^* = 0$, which follows from the following argument. First notice that the stationary distribution of the hidden network is independent of the $k$'s. Hence the value of the rate constants $k_{\pm 4}$ for which the observable current vanishes is independent of the other rate constants. So we can choose hidden symmetric rates $k_{+1} = k_{-1}$ etc. such that $\mathcal{A} = 0$ for which stalling is equilibrium, and then detailed balance prescribes $k_{+4} = k_{-4}$.

respect to $a$ we find

$$
\begin{aligned}
0 &= \frac{d^2}{da^2} \left\langle e^{(a^*-a)\hat{J}} \right\rangle (a) \Big|_{a=a^*} \\
&= \frac{d}{da} \left[ \frac{\partial}{\partial a} \left\langle e^{(a^*-a)\hat{J}} \right\rangle (a) - \left\langle \hat{J} e^{(a^*-a)\hat{J}} \right\rangle (a) \right] \Big|_{a=a^*} \\
&= \left[ \frac{\partial^2}{\partial a^2} \left\langle e^{(a^*-a)\hat{J}} \right\rangle (a^*) - 2 \frac{\partial}{\partial a} \left\langle \hat{J} e^{(a^*-a)\hat{J}} \right\rangle (a) + \left\langle \hat{J}^2 e^{(a^*-a)\hat{J}} \right\rangle (a) \right] \Big|_{a=a^*} \\
&= -2 \frac{\partial}{\partial a} \langle \hat{J} \rangle (a^*) + \langle \hat{J}^2 \rangle (a^*)
\end{aligned}
\tag{5.72}
$$

which is, upon taking the long-time limit, Eq. (5.69).

So, why didn't we just show computationally the validity of the integral fluctuation relation, instead of considering first-order response? In Fig. 5.8 we show that an estimator of the right-hand side of Eq. (5.70) becomes extremely noisy outside of the linear regime. Adding samples tends to linearize the expression around the stalling value $a^* = 0$, but this is a particularly slow process (in terms of samples). In particular, there is a systematic bias towards small values further away from stalling due to the fact that most probably the current is directed in the direction of the effective affinity, making the exponential very small. However, very rare events of a large current opposite to the effective affinity can occur, producing the spikes above the expected value of 1. This clash between most typical behaviour and fat tails makes convergence very problematic (see e.g. [85]).

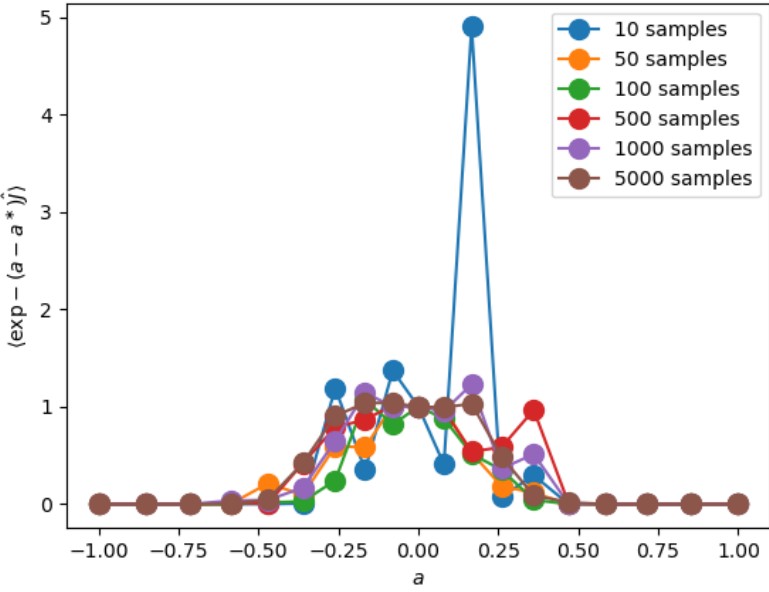

Figure 5.8: For initial population $(2, 2, 2)$, time $t = 50$, internal rates $k_{+1} = k_{+2} = k_{+3} = 2$, $k_{-1} = k_{-2} = k_{-3} = 1$ and observable rates $k_{+4} = \exp a/2$, $k_{-4} = \exp -a/2$, the integral fluctuation relation estimator as a function of the stalling parameter $a$, and for several values of the total number of samples.

## 5.9   Considerations

While preparing this lecture a couple of interesting questions arose that may be worth investigating in the future.

In Sec. 5.5 it was shown that the fluctuation relation can be recovered at all times if an appropriate initial probability distribution is prepared: the question remains open in what sense and under which conditions this distribution has physical sense (that is, it is operationally realizable) or is just a mathematical artifact. There, there is only one current in the system. For more currents, as in Sec. 5.8, there is preliminary work in progress by the Author, extending the formalism of Ref. [86], indicating that in certain circumstances the integral fluctuation relation can be recovered.

Here and there it was suggested that properties of the CRN were linked to some sort of symmetry of the population space. The language of symmetries being that of groups (e.g. the population space of $\emptyset \rightleftharpoons 2X$ must be related to $\mathbb{Z}_2$, the cyclic group of order two) I wonder whether there could be a systematic characterization of deficiency in terms of group theory.

It is informally known that the experimental observability of the (integral) fluctuation relation is problematic, in particular as it comes to deciding whether a system is far from equilibrium. There exist algorithms that fix the problem computationally (e.g. by such techniques as cloning, adaptive sampling, etc.), but in my view there are foundational issues when it comes to experiments with very little sample size. Statistics (delineation of a null hypothesis, p-value estimation, power study etc.) could help in the choice and design. Here it was suggested that response out of stalling states may be the proper ground for making statistically significant claims. Another option is to resort to stopping times different than the external clock time $t$.

# 6   Dynamical Large Deviations: at long times

**William D. Piñeros and Vivien Lecomte.**[20]*Rare events represent anomalous realizations in the dynamics of stochastic systems. Due to their potentially significant impact, investigating and understanding rare events is crucial across various disciplines, including physics, living systems and climatology. This lecture serves as an introduction to the core principles of large deviation theory (LDT), offering an easily comprehensible outline of its formal structure and its applications. We further motivate its use through the specific case of diffusive systems in the large-size and long-time limits where we explore current fluctuations. We also introduce and explain the use of a population dynamics algorithm via a discrete, Markovian picture, and show how it allows us to explicitly evaluate relevant related LDT functions (so-called rate functions).*

## 6.1   Introduction: Why large deviations?

Rare events are by definition anomalous situations that fall outside typical behavior. These events can encompass anything from harmless incidents such as a fortunate winning streak in a game of dice to significantly influential occurrences, such as the abrupt formation of crystals due to nucleation in a supercooled liquid, a sliding layer of snow triggering an avalanche, or large-scale natural disasters causing irreversible changes to a landscape. These motivate one to find and employ some systematic and practical framework for estimating their probability of occurrence. In terms of probability distributions, the focus lies not only on determining the most probable state, which typically centers around the mean, but also on examining its behavior up to its tails. The question consists in assessing the likelihood of a stochastic process fluctuating from its usual state to a particular rare value.

The theory of large deviations (LDT) answers precisely this question and provides us with a framework that allows to systematically estimate the probability of observing a particular rare event relative to its most likely value. In particular, LDT can provide an asymptotic estimation of such probabilities in the limit of very large observation windows. For instance, consider a coin toss game where we count the number of heads after $n$ coin tosses. LDT then allows us to estimate the likelihood of achieving any (possibly large) number of heads, in the asymptotics of large $n$ coin tosses. Certainly, in this particular example, one can explicitly calculate probabilities for all values of $n$ (as we will see). The strength of LDT, however, lies in scenarios where such explicit computations are generally unfeasible but become viable in the limit large number of occurrences.

LDT has expanded its range of applications beyond its original mathematical framework, serving as a relevant tool to characterize rare events across various fields. For example, in statistical physics, LDT allows one to investigate dynamical phase transitions (DPTs), which occur between phases that are characterized by different values of dynamical observables (such as flows or activities). Consider for instance a lattice one-dimensional system of particles with the ability to move either left or right (provided their target site is empty): values of particle current significantly lower than the average are characterized by a "jammed" phase with the formation of a cluster of particles. Similar DPTs have been described through the application of LDT to the distribution of entropy production in systems such as active Brownian particles, with a phase transition marked by a collective particle alignment [87]. Furthermore, LDT may aid in uncovering DPTs in other systems, such as chemical reaction networks or conformational changes in energy-driven biomolecules, by identifying rare dynamical regimes of their dynamics. At larger

---

[20]VL was the lecturer; WP wrote the chapter.

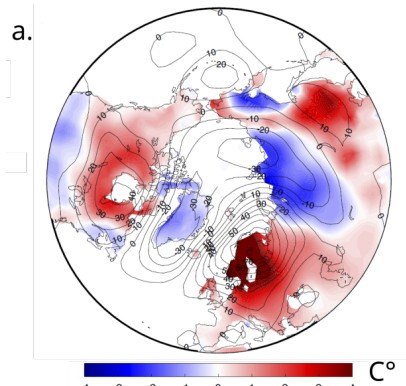 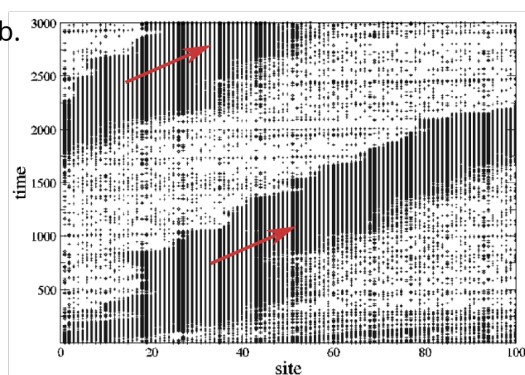

Figure 6.1: **a**. Temperature deviation in the northern hemisphere over a 90 days window. Large deviations in temperature were obtained from an LDT algorithm biased towards achieving a heat wave (rare event). Figure from Ref. [88]. **b**. A one dimensional lattice hopping model where particles tend to jump right (figure adapted Ref. [96]). Conditioning the system to present a low time-integrated particle currents, induces the emergence of density fronts (red arrows).

scales, in the area of climatology and geophysical modeling, LDT has gained prominence in evaluating climate anomalies, including sustained, long-term heat waves across spatially extended regions (see Fig. 6.1 and Ref. [88]).

These lecture notes are intended as an extremely simple introduction to the notion of large-deviation scaling. The interested reader may continue with textbook references on the subject, for instance Refs. [89–95].

## 6.2   Basics and formalism

Having introduced some of the motivation behind LDT we now present its formal settings. Consider a stochastic process and a time-additive observable $A$ defined on some observation window duration $t$ as:

$$A = \int_0^t dt' \, \alpha(t') \tag{6.1}$$

where $\alpha(t')$ depends on an underlying stochastic dynamics at time $t'$. Examples of such quantities can be obtained from any noisy phenomena such as a temperature series in a climate model over some period of observation, and arise naturally in physical systems when calculating quantities like work (integrated power) and other forms of integrated currents (e. g. the entropy production in a non-equilibrium system).

Of course, given the stochastic nature of the system, values of $A$ are generally not fixed but fluctuate around a steady value that represents its more likely outcome. One may therefore want to characterize the probability distribution $P(A, t)$ of $A$ at time $t$ that describes the rare events of interest. The LDT then tells that, in the limit of large $t$, $P(A, t)$ scales as

$$P(A = at, t) \asymp e^{-tI(a)} \, , \tag{6.2}$$

where $\asymp$ is shorthand to denote a the logarithmic equivalent. In other words:

$$\lim_{t \to \infty} \frac{1}{t} \log P(A = at, t) = -I(a) \, , \tag{6.3}$$

and $I(a)$ is a so-called rate function. If the rate function $I(a)$ reaches its minimum in $\bar{a}$, then $\bar{a}$ is the typical value of $A/t$. As its name implies, it quantifies the rate at which the probability of observing a value $A$ deviates (exponentially rarely) from that of its typical value $\bar{a}\, t$. Fig. 6.2a provides a generic example for the behavior of such rate functions.

To better illustrate these concepts and help highlight the origin of the scaling form of Eq. (6.2) for the rate function we now turn to two basic examples.

### 6.2.1   A coin tossing game

Consider a fair coin toss game where a player wins a prize based on the number $F$ of heads obtained after $N$ trials. Thus $F$ in this case is the additive observable of interest and $N$ is the duration of our window of observation. This means that we are interested in the quantity

$$F = \sum_{i=1}^{N} \delta_{H,x_i} \qquad (6.4)$$

where $\delta$ is the Kronecker delta and $x_i \in \{T, H\}$ indicates the value tails or heads respectively for the random value $x_i$ of the process at instance $i$.

Since the coin is fair, the average of $F$ is $N/2$; its second cumulant is also proportional to $N$. This implies that small deviations of $F$ from its average scale as $\sqrt{N}$ ; in fact, the central limit theorem states that the distribution of $(F - N/2)/\sqrt{N}$ is Gaussian; in other words:

$$\lim_{N \to \infty} P\big(F = \tfrac{N}{2} + \sqrt{N}\, \delta\hat{F}\big) = \frac{1}{\sqrt{2\pi\sigma^2}} e^{-\frac{1}{2}\frac{\delta\hat{F}^2}{\sigma^2}} \qquad (6.5)$$

for some variance $\sigma > 0$ independent of $N$.

The large-deviation approach focuses on deviations of $F$ that are far more rare than those normal deviations. Namely, we are interested in the probability $P(F/N = f)$ that the time average $F/N$ achieves some value $f$. In our coin toss problem, this can be solved by representing the number of heads as per a binomial distribution. Explicitly

$$P(F/N = f) = \frac{1}{2^N} \frac{N!}{(fN)![(1-f)N]!} \ . \qquad (6.6)$$

In the large-$N$ limit, taking the log on both sides of Eq. (6.6) and apply Stirling's expansion $\ln N! = N \ln N - N + O(\ln N)$, one finds

$$\ln P(F/N = f) \approx -N \ln 2 - Nf \ln f - N(1-f)\ln(1-f) \qquad (6.7)$$
$$P(F/N = f) \approx e^{-NI(f)} \ ,$$

where we identified the rate function of this process as $I(f) \equiv \ln 2 + f \ln f + (1-f)\ln(1-f)$. This scaling is an example of the LDT scaling of Eq. (6.2). It is represented in Fig. 6.2b. As expected, the most likely fraction of heads is $\bar{f} = 1/2$, which is the point where $I(f)$ is zero and reaches its minimum. Importantly, the rate function $I(f)$ describes fluctuations of $F/N$ that go beyond the Gaussian fluctuations around its average value.

### 6.2.2   Gaussian Sums

Consider now a sum $A$ of $N$ independent and identically distributed (i.i.d.) Gaussian random variables $\alpha$ with individual mean values $\mu$ and standard deviations $\sigma$:

$$A = \sum_{i=1}^{N} \alpha_i \ , \qquad (6.8)$$

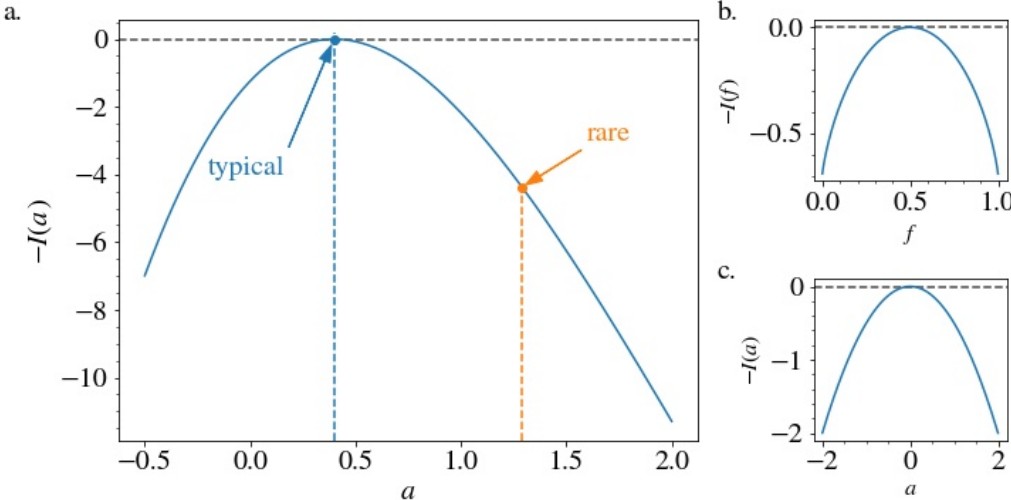

Figure 6.2: a. A schematic rate function $I(a)$ illustrating the typical event (blue dashed line) and a rare event of an observable $a$ (orange dashed line). The magnitude of $I(a)$ (circle markers) dictates the likelihood, via $P \approx^{-tI(a)}$, of observing a rare event relative to the typical. Hence larger magnitudes indicate lower probability. b. Rate function in Eq. (6.8) for the binomial example. The most typical value falls on $f = 0.5$ where $I(f) = 0$ c. Rate function for the Gaussian process example which yields a simple parabola.

where $\alpha_i$ represents the individual realization of the random variable. As before, $N$ is the duration of the window of observation and $A$ the additive observable. We are interested in finding values of $a = A/N$ which represent possibly atypical values of $A$ away from its most likely value $N\mu$. Using that the sum of $N$ Gaussian i.i.d. variables yields another Gaussian variable with effective mean $\mu' = N\mu$ and square standard deviation $\sigma'^2 = N\sigma^2$, one has

$$P(A/N = a) = \sqrt{\frac{N}{2\pi\sigma^2}} e^{-\frac{N(a-\mu)^2}{2\sigma^2}} \ . \tag{6.9}$$

$$\frac{1}{N} \ln P(A/N = a) = -\frac{(a-\mu)^2}{2\sigma^2} - \frac{1}{2N} \ln(2\pi\sigma^2) + \frac{1}{2N} \ln N \ ,$$

and taking the large-$N$ limit one finds

$$\lim_{N\to\infty} \frac{1}{N} \ln P(A/N = a) = -\frac{(a-\mu)^2}{2\sigma^2} \tag{6.10}$$

$$= I(a) \ .$$

Note that terms sublinear on $N$ drop out and we end up with the rate function $I(a)$ which in this case is just quadratic. In other words we have shown that $P(A/N = a)$ presents again the LDT scaling $\asymp e^{-NI(a)} = e^{-N\frac{(a-\mu)^2}{2\sigma^2}}$. The most likely value $\bar{a} = \mu$ is the location of the minimum of the rate function.

In this specific example, the central-limit theorem and the LDT yield the same Gaussian expression for the large-$N$ behavior of the distribution of $A$, but this is in general not the case, as we have seen in the coin toss example above.

## 6.3 Application: Large Deviations in Diffusive Systems

We now move on to examples of physical relevance to model stochastic phenomena, and consider as a paradigmatic application the case of diffusive dynamics. Chiefly, we are

concerned with a physical system – discrete, particle based, or continuum – coupled to a thermal bath, and/or whose internal micro-dynamics gives rise to an inherent noise that drives fluctuations in the observables of the system. For instance, in the classic example of a 1D overdamped Brownian particle coupled to a thermal bath, one may model the dynamics of its position $X$ as

$$\dot{X} = \mu F + \sqrt{2D}\xi , \tag{6.11}$$

where $F$ represents a deterministic forcing in the system (possibly depending on $X$), and $\xi$ is a Gaussian white noise with zero mean and correlation $\langle \xi(t)\xi(t') \rangle = \delta(t' - t)$. This noise represents for instance fluctuations coming from a thermal bath. The probability density $\rho(x,t)$ of its configurational density is governed by the Fokker–Planck equation

$$\partial_t \rho = - \partial_x \left( \mu F \rho - D\partial_x \rho \right) \tag{6.12}$$
$$= - \partial_x j ,$$

where we identified the probability current $j \equiv \mu F\rho - D\partial_x\rho$. One is then typically interested in quantifying the fluctuations of time-additive observables $A$ such as the work or the total entropy production. They depend on the level of non-equilibrium drive in the system, measured by $j$. This implies that knowledge of the fluctuations of $j$ informs us on the fluctuations of such time-additive observables, and therefore gives access to the associated distributions of $P(A,t)$. Identifying the distribution of $j$, via the dynamics of $\rho$, is therefore of great interest, but also necessitates the solution of the associated Fokker–Planck equation, which may be typically unfeasible in large and interacting systems.

We can alternatively attempt to coarse-grain over small details and consider a macroscopic picture in a steady-state where dynamics is determined by a very large number of particles evolving in time over diffusive time scales. This procedure allows one to transform a discrete (or particle) representation into a continuum limit, for large systems. Their dynamics may be externally driven or coupled to thermodynamic gradients and can describe in or out of equilibrium regimes. The usefulness of this approach, called Macroscopic Fluctuation Theory (MFT, see Ref. [97] for a review), is then that it provides us with a more general description of the non-equilibrium dynamics through the joint fluctuations of the densities and currents in the system. These latter may in part be captured by a joint-probability distribution function $P(\rho, j)$ and therefore neatly encapsulate all relevant information about rare events.

Concretely, consider as an example a one-dimensional system coupled to two distinct thermal reservoirs and without external driving. This may represent some substance, (material, particles) confined in one dimension but subject to a thermal gradient which generates a density current. As before, we are interested in characterizing the system activity and its fluctuations through a current observable $Q$ integrated over a given time interval. In particular we are interested in estimating $P(Q,t)$ which, in the large-size limit $N \gg 1$ and long-time limit $t \to \infty$ LDT allows one to estimate as $P(Q,t) \asymp e^{-tNI(Q)}$, where $I(Q)$ is the rate function associated with fluctuations in $Q$. As we will see, it is possible to estimate $I(Q)$ from knowledge of the fluctuations in $\rho$ and $j$ which represent the underlying dynamics.

We first define the time-integrated current as

$$Q = \int_0^t dt' \int_0^1 dx \, j(x,t') , \tag{6.13}$$

where $j$ represents the change of density in the system as given by the following dynamics

$$\partial_t \rho = -\partial_x j \tag{6.14}$$
$$j = -D\partial_x \rho + \sqrt{2\sigma}\eta . \tag{6.15}$$

Here $\eta$ is a white noise term i.e. $\langle \eta(x,t)\eta(x',t') \rangle = \frac{1}{N}\delta(x-x')\delta(t-t')$ and $\sigma$ is its associated diffusion coefficient. Following the MFT, this noise term is considered to capture the underlying microscopic dynamics in this large time and particle limit. Furthermore, the Gaussian form of the noise allows one to immediately estimate the probability of observing a given trajectory of noise $\eta$ as

$$P(\eta) = e^{-\frac{N}{2} \int_0^t dt' \int_0^1 dx \ \eta(x,t')^2} \ . \tag{6.16}$$

Then, rearranging from Eq. (6.15) we see that we can rewrite $\eta = \frac{j+D\partial_x\rho}{\sqrt{2\sigma}}$ and thus obtain a relation for the joint probability distribution of $\rho$ and $j$ as

$$P(\rho,j) = e^{-\frac{N}{2}\int_0^t dt' \int_0^1 dx \frac{(j+D\partial_x\rho)^2}{\sigma}} \tag{6.17}$$

$$= e^{-tN\, I(\rho,j)} \ ,$$

where we identify $I \equiv \frac{1}{t}\int_0^t dt' \int_0^1 dx \frac{(j+D\partial_x\rho)^2}{\sigma}$ as a joint rate function of $\rho$ and $j$ given the large limit assumptions in $N$ and $t$, and where $j$ and $\rho$ are constrained to verify the continuity equation $\partial_t\rho + \partial_x j = 0$.

Next, we wish to estimate the likelihood of observing a given fluctuation of $Q$ through knowledge of $I(Q)$. Here, one expects that knowing the joint rate function of $\rho$ and $j$, and through the dependence of $Q$ on these variables, that we will be able to estimate fluctuations on $Q$ as well. Indeed, LDT allows one to map, or transform from a rate function $I(a)$ to another $I(b)$ in a mathematical procedure known as a contraction if there exists a function, or constraint that relates $a$ to $b$. Briefly, if $b = f(a)$ where $f$ is some function, then the contraction is defined as

$$I(b) = \min_{a:\, b=f(a)} I(a) \ , \tag{6.18}$$

where by an abuse of notation we denote by the same symbol the rate functions of the random variables $a$ and $b$. This result follows from a so-called saddle point analysis which states that for integrals of the form $\int dx \, e^{-r\, I(x)} \asymp e^{-r\, \min I(x)}$ as $r \to \infty$. Thus, for an exponentially weighted probability distribution, one seeks to pick the most likely event fulfilling some constraint which amounts to finding values that minimize the argument. Its namesake follows in the case where the linking function is many-to-one so that it leads to a net reduction, or contraction in the number of variables. More rigorous derivations of these results go beyond the scope of this lecture but are available from reviews in the literature.

Using this procedure we then finally obtain our desired estimates of $P(Q,t)$ via $I(Q)$. Namely (denoting for short $\int dt' \int dx = \int_0^t dt' \int_0^1 dx$),

$$I(Q) = \min_{j:\, Q=\int dt' \int dx\, j} \ \min_{\rho:\, \partial_t\rho+\partial_x j=0} I(\rho,j) \tag{6.19}$$

$$= \min_{j:\, Q=\int dt' \int dx\, j} \ \min_{\rho:\, \partial_t\rho+\partial_x j=0} \int_0^t dt' \int_0^1 dx \, \frac{(j+D\partial_x\rho)^2}{2\sigma} \ ,$$

where the double minimization arises due to the bivariance of $I(\rho,j)$ with respect to the densities and currents, and the additional expressions underneath represent constraints as per Eq. (6.13) and Eq. (6.14). Thus, fluctuations in $Q$ are driven by the most likely fluctuations in $\rho$ and $j$ that coincide in achieving the given value of $Q$.

We can therefore appreciate how the use of LDT allows us to forego sampling through all possible realizations of a dynamics in a system, and instead provides a direct means to characterize observable fluctuations of a quantity of interest via knowledge of its corresponding rate functions. Our original goal of estimating the likelihood of rare events is then reduced to finding rate functions which we now address next.

## 6.4   Biased dynamics: making rare dynamics typical

Rare events are, by definition, rare and hence not necessarily observable within some feasible window of time in numerical approaches. Instead we may condition, or choose relevant trajectories, such that a particular value of an observable $A/t = a$ is obtained upon averaging. This is equivalent to working in a conditioned probability distribution of the form

$$P(x; A/t = a) = \int \mathcal{D}(X)\, p(X = x)\, \delta\big(\tfrac{1}{t}A(X) - a\big),  \tag{6.20}$$

where $\mathcal{D}(X)$ represents the path integral over all realizations of trajectories $X$. Such conditioned probabilities might seem familiar as they are the dynamical analogy to a microcanonical ensemble in equilibrium statistical systems, where an arbitrary $A$, rather than energy, now plays the constraining variable. Naturally, just as in the microcanonical ensemble, such conditioned ensemble can also be used to obtain other observables $\mathcal{O}$ of interest under the condition that $A/t = a$. Thus one may obtain averages of $\mathcal{O}$ as

$$\begin{aligned}
\bar{\mathcal{O}}(a) &= \frac{\int \mathcal{D}(x) P(x; A/t = a) \mathcal{O}(x)}{\int \mathcal{D}(x) P(x; A/t = a)} \\
&= \frac{\langle \mathcal{O}\, \delta(\tfrac{1}{t}A(X) - a)\rangle}{\langle \delta(\tfrac{1}{t}A(X) - a)\rangle}
\end{aligned}  \tag{6.21}$$

where $\mathcal{D}(x)$ denotes a path integral over all conditioned paths. Here, $\bar{\mathcal{O}}(a)$ represents the average of observable $\mathcal{O}$ over trajectories conditioned to $A/t = a$.

However, much like the microcanonical ensemble, rather than working with ensembles conditioned on $A/t = a$, it is often easier and more practical to instead have a modified ensemble where trajectory probabilities are biased by a weight $e^{\lambda A}$ where $\lambda$ is a fixed parameter. In fact, if $\mathcal{O}$ does not behave exponentially in time, it is possible to show that such procedure respects expectation values so that this 'microcanonical' average $\bar{\mathcal{O}}$ may now be re-expressed as

$$\bar{\mathcal{O}}(a) = \frac{\langle \mathcal{O}(X) e^{\lambda A(X)}\rangle}{\langle e^{\lambda A(X)}\rangle},  \tag{6.22}$$

for a well-chosen $\lambda$, conjugated to $a$. Thus we can reinterpret this biased ensemble as a 'canonical' dynamical ensemble where the sum in the denominator may be interpreted as analogous to a dynamical partition function, and $\lambda$ plays the role of a conjugate variable to $A$ akin to the conjugate relation of temperature and energy in thermodynamics. In fact, the dynamical partition function $\langle e^{\lambda A(X)}\rangle$ implicitly encodes for all relevant statistical measurements of the system by recognizing: it is also the moment generating function (MGF) of the original probability density $P(X)$. That is, by noting that

$$\langle e^{\lambda A}\rangle = \int \mathcal{D}(X) P(X) e^{\lambda A}  \tag{6.23}$$

it follows that,

$$\frac{\partial^m}{\partial \lambda^m} \int \mathcal{D}(X) P(X) e^{\lambda A}\bigg|_{\lambda=0} = \int \mathcal{D}(X) A^m P(X) e^{\lambda A}\bigg|_{\lambda=0} = \langle A^m\rangle,  \tag{6.24}$$

so that the average, variances and higher order statistical measures of the biased ensemble are indeed readily accessible from the derivatives of the MGF in $\lambda = 0$. Furthermore, if we now consider the log of Eq. (10.34), known as the cumulant generating function (CGF), we

may then obtain mean-centered moments $\langle (A - \langle A \rangle)^2 \rangle$ of the distribution, which therefore provides more relevant information of the deviations of a quantity from its average value.

In fact, it turns out that knowledge of the CGF allows us to compute the rate function $I(a)$ directly in the long-time limit as follows. In particular, it can be shown that

$$\lim_{t \to \infty} \frac{1}{t} \ln \langle e^{\lambda A} \rangle = \psi(\lambda) \tag{6.25}$$

where $\psi(\lambda, A)$ is known as the scaled cumulant generating function (SCGF). Then, following a result in LDT known as the Gärtner–Ellis theorem, we can now obtain the rate function as

$$I(a) = \max_{\lambda} \lambda a - \psi(\lambda) \ . \tag{6.26}$$

Notice that this extremalization principle gives precisely the value $\lambda$ conjugated to $a$ that was needed in Eq. (6.22). We can understand this result briefly by supposing that $P(A/t = a) \asymp e^{-tI(a)}$ and evaluating the MGF as

$$\begin{aligned} \langle e^{\lambda A} \rangle &= \int da \, P(A = at) e^{\lambda a t} \\ &= \int da \, e^{t(\lambda a - I(a))} \\ &\asymp e^{\max_{\lambda} t(\lambda a - I(a))} \ , \end{aligned} \tag{6.27}$$

where the last line follows from the saddle-point evaluation for $t \to \infty$. The result in Eq. (6.25) then follows from taking the log on both sides of Eq. (6.28) and letting $t \to \infty$. Thus the likelihood of fluctuations of $A$ can be obtained from the limiting calculation of $\psi(\lambda)$ alone.

In practice, different analytical and numerical methods are available in obtaining $\psi$ for actual systems. However, the general challenge lies in obtaining accurate averages of the observables for increasing values of the bias $\lambda$. In what follows we will motivate a numerical approach to calculate $\psi$ in Markovian systems as inspired in a population dynamics algorithm.

### 6.4.1 Population dynamics algorithm: a Markov jump process motivation

Consider a Markov process defined by a finite number of configurations $\{x\}$ and whose dynamics are determined by transition rates $r(x|x')$ to go from state $x'$ to $x$. For example, in a discrete graph system these configurations could correspond to the number of nodes and the transition rate the probability of jumping from one node to another. The dynamics of the probability distribution of states $p(x, t)$ is then given by the master equation

$$\partial_t p(x, t) = \sum_{x' \neq x} \left[ r(x|x') p(x', t) - r(x'|x) p(x, t) \right] \ , \tag{6.28}$$

and which we can interpret as the difference between a *gain* term, from states $x'$ going to $x$, and a *loss* term from states leaving $x$.

Now consider an additive observable $A$ which can be written over a time history of $K$ jumps as

$$A = \sum_{0 \leq k \leq K-1} \alpha_{x_k x_{k+1}} \ , \tag{6.29}$$

where $\alpha_{x_k x_{k+1}}$ are individual contributions along the history path of successively visited states $\{x_k\}_{0 \le k \le K}$. The quantity $\alpha_{xx'}$ represents the amount by which $A$ is increased during a jump from state $x$ to state $x'$. For instance, in a 1D particle system, taking $\alpha = \pm 1$ depending on whether a particle jumps towards right/left will yield for $A$ a time-integrated current. Our goal is to compute the likelihood of observing fluctuations of this observable through knowledge of its corresponding SCGF $\psi(\lambda)$.

We may achieve this by first considering a joint probability distribution $p(x, A, t)$ of the observable and the states as

$$\partial_t p(x, A, t) = \sum_{x' \ne x} \left[ r(x|x') p(x', A - \alpha_{x'x}, t) - r(x'|x) p(x, A, t) \right] . \tag{6.30}$$

where the $-\alpha_{x'x}$ term accounts for the fact that we are coming from the previous value of $A$ at the $x'$ state. Next we are interested in biasing dynamics with respect to specific values $A$ which, as discussed in the previous section, can be achieved by exponentially weighing the corresponding probability distribution $p(x, A, t)$. This then corresponds to the biased 'canonical' ensemble expressed as

$$\hat{p}(x, \lambda, t) = \sum_A e^{\lambda A} p(x, A, t) , \tag{6.31}$$

where $\lambda$ is the conjugate variable with $A$ and determines the 'strength' of the biasing. We can then multiply by the corresponding weights $e^{\lambda A}$ in the Eq. (6.30), sum over all $A$ and find the desired biased, joint dynamics

$$\partial_t \hat{p}(x, \lambda, t) = \sum_{x' \ne x} \left[ e^{-\lambda \alpha_{x',x}} r(x|x') \hat{p}(x', \lambda, t) - r(x'|x) \hat{p}(x, \lambda, t) \right] . \tag{6.32}$$

Finally using the result in Eq. (6.25) we recognize that

$$\sum_x \hat{p}(x, \lambda, t) = \langle e^{\lambda A} \rangle \underset{t \to \infty}{\asymp} e^{t \psi(\lambda)} , \tag{6.33}$$

so that our task involves evaluating the biased dynamics of Eq. (6.32) and obtaining the expectation values with respect to $p(x, \lambda, t)$ and hence $\psi(\lambda)$. Note further that the exponential modification of the rates means the probability is no longer conserved in Eq. (6.32) and instead we are now dealing with an open system.

Analytically, such calculation is equivalent to solving a maximum eigenvalue problem of the form $R_\lambda |\hat{p}\rangle = e^{t\psi(\lambda)} |\hat{p}\rangle$, where $|\hat{p}\rangle$ represents the eigenvector and $R_\lambda$ an evolution operator with matrix entries $R_{x,y} = e^{\lambda \alpha_{yx}} r(x|y) - \delta_{x,y} r(x)$ (with $r(x) = \sum_y r(y|x)$ is the escape rate). However, for more general cases where closed solutions are not tractable, we can instead consider a numerical approach that, as we will now demonstrate, amounts to a population dynamics algorithm with respect to $\lambda$ and $x$.

First let us then re-write Eq. (6.32) as

$$\partial_t \hat{p}(x, \lambda, t) = \sum_{x' \ne x} r_\lambda(x|x') \hat{p}(x', \lambda, t) - r_\lambda(x) \hat{p}(x, \lambda, t) + \delta r_\lambda(x) \hat{p}(x, \lambda, t) , \tag{6.34}$$

where

$$r_\lambda(x'|x) = e^{\lambda \alpha_{x'x}} r(x'|x) \tag{6.35}$$

$$r_\lambda(x) = \sum_{x' \ne x} r_\lambda(x'|x) \tag{6.36}$$

$$\delta r_\lambda(x) = r_\lambda(x) - r(x) , \tag{6.37}$$

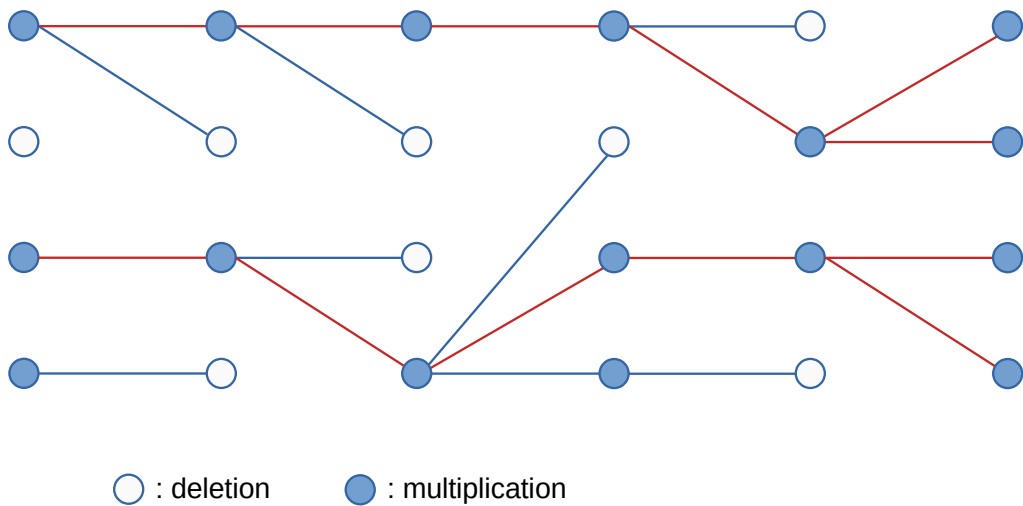

○ : deletion    ● : multiplication

Figure 6.3: Schematic of a population dynamics evolution with a biased selection rule for a system with $N_c = 4$. Clones displaying larger biases towards a desired event are multiplied, while those with average or below performance are deleted. The reproduction success i.e. population deletion/multiplication is controlled by the bias parameter $\lambda$. Surviving trajectories (highlighted here in red) are those representative of the biased ensemble.

and are, respectively, the biased transition rates from $x'$ to $x$, the net biased escape rate from $x$, and the difference between the original and modified escape rates in the biased ensemble. Importantly, we can recognize the first terms in Eq. (6.34) as corresponding to a *conserved* dynamics of rates $r_\lambda$, plus an additional term $\delta r_\lambda$ representing *gain* or *loss* in the probability depending on sign.

Furthermore, the Feynman–Kac formula tells that Eq. (6.34) implies

$$\langle e^{\lambda A} \rangle = \langle e^{\int_0^t dt' \, \delta r_\lambda(x(t'))} \rangle_\lambda \tag{6.38}$$

where the subscript in $\lambda$ indicates the biased ensemble (i.e. of transition rates $r_\lambda$), so that calculation of $\psi(t)$, from its long time limit relation in Eq. (6.33), may be now be reinterpreted as a history average over $\delta r_\lambda$. Given that this term is precisely the injection/loss of probability in the dynamics, we can interpret this procedure as a population dynamics algorithm where population members, all evolving in parallel in the biased dynamics, play the role of probability outcomes to be removed (probability loss) or replicated (probability gain) as they evolve in time.

The idea to represent probability loss or gain comes from similar quantum-mechanical problems (finding the ground-state energy of an operator using diffusion Monte-Carlo methods [98]) and was applied to varied problems in statistical mechanics [99] and mathematics [100]. Its application to compute SCGFs was put forward in [96, 101].

Concretely, we can now formulate a population dynamics algorithm as follows. First, define $Y \equiv e^{\int_0^t dt' \, \delta r_\lambda(x(t'))}$ and discretize the integral in Eq. (6.38) as $Y = \prod_{t_k=0}^{t} Y(x_k)^{\Delta t'}$ where $Y(x_k)^{\Delta t'} \equiv e^{\delta t'(r_\lambda(x_k)-r(x_k))}$, and $x_k$ represent configurations at a given time $t_k$. Then, starting from a very large population of $N_0$ clones all evolving in parallel under the biased dynamics:

- Compute $Y(x_k)^{\Delta t}$ during an interval $\Delta t$

- Clone $n_i(x_k)$ is pruned or replicated with rate $Y(x_k)^{\Delta t}$. For instance, given random

a variable $\epsilon$ uniformly distributed on $[0, 1]$, make $y = \lfloor Y(x_k)^{\Delta t'} + \epsilon \rfloor$ copies of the clone. This then alters the population by a factor $F_t = \frac{N+y-1}{N}$

- Evolve new modified population for the next interval $\Delta t$ and repeat process up to $t$.

A schematic of this population dynamics with selection is illustrated in Fig. 6.3. The SCGF is then evaluated from the change in population as

$$\psi(\lambda, t) \asymp \tfrac{1}{t} \ln \langle Y \rangle_\lambda \qquad (6.39)$$
$$= \tfrac{1}{t} \ln \prod_t F_t \ .$$

Therefore, $\psi(t)$ can ultimately be interpreted as the exponential rate of growth or decrease in a population. However, in practice such cloning procedure may lead to diverging or vanishing populations which could be undesirable from a numerical point of view. Instead, one may also choose to maintain a constant population throughout the evolution of the system by different selection rules that prune/copy clones from a statistically weighted chance with respect to the entire population. Specific examples of these rules, as well as other choice of cloning regimes, are for instance described in the review [102].

Altogether, through this cloning algorithm we are therefore able to sample for biased values of an observable $A$ for any Markovian system of choice. Such procedure is formally exact in the limit of large number of clones, and simulation times, and has been used successfully to probe for phase transitions in both discrete and particle based systems. However, achieving large parameter limits can be challenging in practice, and finite-size and finite-time scaling analysis can be helpful [103, 104].

# 7 Dynamical Large Deviations: at large size

**Alexandre Lazarescu.** *In this chapter, we focus on large deviations at large size and finite time. Using examples of population models, and particularly chemical networks, we will see how far we can push the large deviations principle to obtain information on not only time averages, but also precise time evolutions of large systems, using path integrals.*

## 7.1 Dynamical large deviations: Formalism

In the previous chapter, we saw how the large deviations principle can be applied to describe fluctuations of time-averaged observables in a Markovian process, when time becomes large. In this section, we focus on a different scaling: we do not put any restrictions on time, but we assume the system to be very large. This large-size limit is the same as the one we take in equilibrium statistical mechanics in order to reach the thermodynamic limit; here, we merely consider the distributions to be time-dependent.

If we focus on interacting particle models, there are essentially two ways to reach that limit: either look at particles on a fixed network, and increase the number of particles until there is a macroscopic number of particles on each site, or consider particles on a lattice or in continuous space, and rescale space until the density of particles at every point is finite. The first class of systems is generally called *population models*, and includes chemical networks and certain ecological models, and the second is often referred to as *diffusion models* (see Section 6.3), even if the resulting dynamics is sometimes ballistic. In this lecture, we will focus on populations, but we will provide a few formulas relating to diffusions when appropriate. In short, we will explain where formula (6.17) comes from, and what we can do with it.

Our approach will be the following: we first set up a few explicit population models where we can take a large size limit; we then present the fundamental building bloc of dynamical large deviations: the instantaneous displacement distribution, and its generating function; we use said building blocks to construct path integrals; from those path integrals, we derive equations of motion, both in a Lagrangian and a Hamiltonian formalism; we make a brief aside to present the similarities and differences between those formalisms within statistical mechanics, quantum mechanics, and classical mechanics; and, finally, we describe the long time behaviour of those path integrals.

The majority of objects and relations that we will present in this lecture are quite general: they will involve generic Markov processes, with some static observables (densities) and some dynamical observables (currents), and a relation between the two (continuity equation). This abstraction makes our framework quite powerful, but also rather formal. It is then useful to first consider a few concrete cases, where explicit calculations can be made, and pictures can be drawn.

### 7.1.1 Population models

As is customary, we start with independent particles. Consider $N$ independent random walkers governed by a process $R$ (as defined in Section 1.2). They live on a fixed finite network with sites $x$, and at every time $t$ we denote the occupancies of every site as $n_x(t)$, and the corresponding densities as $\rho_x \equiv n_x/N$.

The jump process $R$ is Markovian and performs transitions between sites $(x \to y)$ with rates $r_{yx}$, which can be translated into transitions between occupancies, where occupancy

vectors transform as $\{n_z\} \to \{n_z - \delta_{z,x} + \delta_{z,y}\}$ with rate

$$R_{yx}^{(N)} = n_x r_{yx} = N\rho_x r_{yx} \equiv N w_{yx}(\rho). \tag{7.1}$$

We can count the number of jumps between each pair of neighbours at any time (which we will denote with a generic # symbol for "number"), over a short period, i.e. between times $t$ and $t + \delta t$. This yields a current observable

$$\lambda_{yx} \equiv \frac{\#[x \to y]_t^{t+\delta t}}{N \delta t} \tag{7.2}$$

which is a random variable taking a time-dependent value for each realisation. Note that the typical value of this random variable is related to the semi-currents defined in Section 1.4, in the case of independent particles (there is simply a factor $N$ between them).

This current observable completely determines the evolution of the density observable: the change in density at any site is the difference between all incoming and outgoing currents. This can be formally written using a graph divergence $\nabla$ (which is the opposite of the incidence matrix $D$ defined in Section 2.2) such that we have the usual continuity equation $\dot{\rho} = -\nabla \cdot \lambda$. This matrix can be written as

$$\nabla_{z,(y,x)} = \delta_{z,x} - \delta_{z,y} \tag{7.3}$$

and acts on edges $(y, x)$ to produce a vector on sites $z$.

We can make things slightly more complex by changing the type of transitions that are allowed in terms of occupancies, by removing or adding various numbers of particles from any number of sites. If we rename sites as species, what we get is a chemical network.

Consider thus a chemical system of volume $V$ with species $x$. We collect the number of particles of each species at time $t$ into a composition vector $n_x(t)$, and we define the corresponding concentrations as $\rho_x = n_x/V$. In order to transform groups of particles into other groups of particles (i.e. to perform chemical reactions), it is useful to first determine which groups can appear: we define complexes $\gamma$ with stoichiometry $\nu_x^\gamma$ (the number of particles of species $x$ which appear in complex $\gamma$). Note that those stoichiometric coefficients are sometimes indexed by the label and direction of a reaction instead (as in Section 4.2).

Given this, we can cast reactions as being transformations between complexes $(\gamma \to \gamma')$, with the corresponding change of occupancy $\{n_z\} \to \{n_z - \nu_z^\gamma + \nu_z^{\gamma'}\}$. We call $\gamma$ the reactants, and $\gamma'$ the products, as is customary. Note that we treat forward and backward reactions as different transformations, because we don't assume micro-reversibility. The simplest transition rates we can then consider are called *mass action rates*, which contain a reaction constant $K$ which we scale as $k$ times a suitable power of $V$, and a combinatorial factor counting the number of ways to choose the reactants:

$$R_{\gamma'\gamma}^{(N)} = \left(\prod_x \frac{n_x!}{[n_x - \nu_x^\gamma]!}\right) K_{\gamma'\gamma} = V\rho^{\nu^\gamma} k_{\gamma'\gamma} \equiv V w_{\gamma'\gamma}(\rho). \tag{7.4}$$

Note the similarity in structure with the previous simpler case (7.1), which was also made of a combinatorial factor $n_x$ multiplied by a constant term tied to the type of transition $r_{yx}$. This case can be recovered if all reactions involve only one reactant and one product, like for enzyme conformation changes, for instance.

As for independent particles, we can define a (chemical) current observable by counting the number of times a transformation is made between two instants:

$$\lambda_{\gamma'\gamma} \equiv \frac{\#[\gamma \to \gamma']_t^{t+\delta t}}{V \delta t}. \tag{7.5}$$

And just as before, this current observable completely determines the evolution of the concentration of each species in the system: the change in concentration is the difference between the number of particles created and the number of particles destroyed, where each number can be determined by the number of occurrences of each reaction, multiplied by the stoichiometric coefficient of the species in that reaction. This can be written in terms of a continuity equation $\dot{\rho} = -\nabla \cdot \lambda$ which involves a chemical divergence $\nabla$ which is essentially twice the stoichiometric matrix 4.4:

$$\nabla_{z,(\gamma',\gamma)} = \nu_z^\gamma - \nu_z^{\gamma'}. \tag{7.6}$$

This structure can be generalised, and will be the basis of our abstract analysis: we will always require our systems to have a well-defined volume $V$, states $\rho$, currents $\lambda$, state-dependent rates $R(\rho) \sim Vw(\rho)$, and a divergence operator $\nabla$ such that the continuity equation $\dot{\rho} = -\nabla \cdot \lambda$ translates currents into changes of state. The appropriate state and current observables are often obvious from the definition of the model. If the currents do not completely determine $\dot{\rho}$, it often means that some parts of the process were forgotten (either from missing transitions or an inappropriate bunching of transitions). A suitable set of currents will be called *fundamental currents*, and we should note that it is not unique: a transition can always be duplicated and its rate split in halves, without changing the process.

Note that, due to common usage, we will use the same symbol $\nabla$ for gradients, acting on state vectors and producing edge vectors. Also due to usage, the gradient will be defined as minus the transpose of the divergence, i.e. for instance

$$\nabla_{(y,x),z} = -\delta_{z,x} + \delta_{z,y}. \tag{7.7}$$

The distinction between the two symbols will be made through the fact that products in edge-space are denoted by a dot, as in $\nabla \cdot \lambda$, whereas products in state-space are not, such as $\nabla \rho$.

### Example

Let us conclude this part with a concrete example: the replicator, involving a single species $A$ with occupancy $n$ and three reactions

$$A \to 2A \quad \text{with rate} \quad an \qquad\qquad\qquad \text{such that} \quad w_{21}(\rho) = a\rho \tag{7.8}$$
$$2A \to A \quad \text{with rate} \quad bn(n-1)/V \qquad\quad \text{such that} \quad w_{12}(\rho) = b\rho^2 \tag{7.9}$$
$$A \to \varnothing \quad \text{with rate} \quad cn \qquad\qquad\qquad \text{such that} \quad w_{01}(\rho) = c\rho. \tag{7.10}$$

We can define three complexes $\gamma_1 = A$, $\gamma_2 = 2A$, $\gamma_0 = \varnothing$, with stoichiometries $\nu^{(1)} = 1$, $\nu^{(2)} = 2$, $\nu^{(0)} = 0$.

In this case, the divergence is a one-line matrix $\nabla = [1-2, 2-1, 1-0] = [-1, 1, 1]$ in basis $\{A \to 2A, 2A \to A, A \to \varnothing\}$.

This example will reappear regularly as illustration in the rest of the lecture.

### 7.1.2   Displacement distribution at large volume

Having defined the necessary mathematical objects, we can now look at the dynamics of our process. We know the rates of every possible transition, each associated to a current observable. Looking at the joint statistics of all currents $\lambda(t)$ starting from a given state $\rho(t)$ is in general complicated: the probabilities of that collection of jumps depends on the order in which they are performed (the individual processes of the transition matrix do not commute). However, if the size $V$ is large and all occupancies $n_x$ scale with $V$, a small displacement $\delta\rho$ will not affect the rates much: the processes become effectively independent. This is the most crucial approximation we make, and it relies on a specific scaling of the dynamics: typical occupancies must scale as $V$, and the number of occurrences of each transition during $\delta t$ starting from a typical state must scale as $V\delta t$, so that the continuity equation $\dot{\rho} = -\nabla \cdot \lambda$ is finite.

We are still considering population models with Markovian dynamics (hence with Poisson-distributed waiting times, called Poisson clocks, for each transition), and we want to describe the statistics of whole trajectories of the system. Since a trajectory is a sequence of small steps, a good starting point is to look at the distribution of those steps, i.e. the distribution of the rate of displacement of $\rho$, from any starting point, at any time, over a small time step $\delta t$. We have

$$\mathrm{P}_{\delta t}(\dot{\rho}|\rho) = \mathrm{P}\left[\rho(t + \delta t) = \rho + \delta t\dot{\rho} \mid \rho(t) = \rho\right] \tag{7.11}$$

with

$$\mathrm{P}_{\delta t}(\dot{\rho}|\rho) = \sum_{\lambda:\dot{\rho}=-\nabla\cdot\lambda} \mathrm{P}_{\delta t}(\lambda|\rho), \tag{7.12}$$

i.e. the probability of a displacement in $\rho$ is the sum of probabilities of all currents that produce that specific displacement. Those are, in principle, joint probabilities of currents, which are complicated. However, given the scaling discussed above, and given the set of possible transitions $(x \to y)$, we have in fact a quasi-independence

$$\mathrm{P}_{\delta t}(\lambda|\rho) \approx \prod_{x,y} \mathrm{P}_{\delta t}(\lambda_{yx}|\rho). \tag{7.13}$$

For each transition $x \to y$, observing a flux $\lambda_{yx}$ means that the Poisson clock of that transition, with rate $Vw_{yx}(\rho)$, must ring $\delta t V\lambda_{yx}$ times during a time $\delta t$, with probability

$$\mathrm{P}_{\delta t}(\lambda_{yx}|\rho) = \frac{\left(\delta t V w_{yx}(\rho)\right)^{\delta t V \lambda_{yx}}}{[\delta t V \lambda_{yx}]!}\mathrm{e}^{-\delta t V w_{yx}(\rho)}. \tag{7.14}$$

In the limit $V \to \infty$, $\delta t \to 0$, $V\delta t \to \infty$, i.e. large volume, short time, large number of jumps, we can perform a Stirling approximation:

$$\mathrm{P}_{\delta t}(\lambda_{yx}|\rho) \asymp \mathrm{e}^{-\delta t V L_{yx}(\lambda_{yx};\rho)} \quad \text{with} \quad L_{yx}(\lambda_{yx};\rho) = \lambda_{yx}\ln\left(\frac{\lambda_{yx}}{w_{yx}(\rho)}\right) - \lambda_{yx} + w_{yx}(\rho). \tag{7.15}$$

At this stage, we have entered large deviations territory: the function $L_{yx}$ is a rate function for one of the currents, conditioned on the state from which it is produced. From here on, we will combine LDFs to construct other LDFs and SCGFs, and mainly play around *inside the exponentials*. Most of the calculations and formulas will be made on those objects, which are all logarithms of probabilities, and translated back into probabilities only at the end.

Multiplying the probabilities for each transition, we get the joint LDF for all the currents, which we call *detailed Lagrangian* for reasons which will become clear later:

$$
P_{\delta t}(\lambda|\rho) \asymp e^{-N\delta t L(\lambda;\rho)} \quad \text{with} \quad L(\lambda;\rho) = \sum_{x,y} \lambda_{yx} \ln\left(\frac{\lambda_{yx}}{w_{yx}(\rho)}\right) - \lambda_{yx} + w_{yx}(\rho). \quad (7.16)
$$

Since a displacement is a contraction of currents (i.e. a linear combination), we can use the contraction principle to obtain the LDF of $\dot\rho$, which we call the *standard Lagrangian*

$$
P_{\delta t}(\dot\rho|\rho) \asymp e^{-N\delta t \mathcal{L}(\dot\rho;\rho)} \quad \text{with} \quad \mathcal{L}(\dot\rho;\rho) = \inf_{\lambda:\dot\rho=-\nabla\cdot\lambda}\left[L(\lambda;\rho)\right]. \quad (7.17)
$$

In most cases, we are not able to perform the minimisation explicitly, so that this object remains implicitly defined, unlike the detailed Lagrangian which is explicit.

### Example

For the replicator, we have three currents $\lambda_{21}$, $\lambda_{12}$, $\lambda_{01}$, related respectively to the transitions with rate constants $a$, $b$ and $c$, and we can compute

$$
L(\lambda;\rho) = \lambda_{21}\ln\left(\frac{\lambda_{21}}{a\rho}\right) + \lambda_{12}\ln\left(\frac{\lambda_{12}}{b\rho^2}\right) + \lambda_{01}\ln\left(\frac{\lambda_{01}}{c\rho}\right) - \lambda_{21} - \lambda_{12} - \lambda_{01} + a\rho + b\rho^2 + c\rho. \quad (7.18)
$$

The way those currents contribute to the change in concentration, encoded in the divergence operator defined above, is $\dot\rho = \lambda_{21} - \lambda_{12} - \lambda_{01}$. In this simple one-dimensional case, the minimisation of $L$ on the currents conditioned on a fixed value of $\dot\rho$ can be performed explicitly, by substituting $\lambda_{21}$ by $\dot\rho + \lambda_{12} + \lambda_{01}$ and looking for the values of $\lambda_{12}$ and $\lambda_{01}$ where $L$ is minimal. We leave it to the overzealous reader to check that the result is

$$
\mathcal{L}(\dot\rho;\rho) = \dot\rho \ln\left(\frac{\dot\rho + \sqrt{\dot\rho^2 + 4a\rho(b\rho^2 + c\rho)}}{2a\rho}\right) - \frac{\dot\rho^2 - (2a\rho - \sqrt{\dot\rho^2 + 4a\rho(b\rho^2 + c\rho)})^2}{4}. \quad (7.19)
$$

The same result can be obtained much more easily using Hamiltonians rather than Lagrangians, which we explain next.

We can alternatively do computations at the level of SCGFs, via the generating function over all possible numbers of jumps $k = \delta t V \lambda_{yx}$ for a given transition $(x \to y)$, with a conjugate variable $f_{yx}$:

$$
G_{yx}(f_{yx}) = \sum_k e^{kf_{yx}} P_{\delta t}(k/\delta t V|\rho) = \sum_k e^{kf_{yx}} \frac{(\delta t V w_{yx}(\rho))^k}{k!} e^{-\delta t V w_{yx}(\rho)} = e^{\delta t V w_{yx}(\rho)(e^{f_{yx}}-1)}. \quad (7.20)
$$

We can express it in terms of a SCGF:

$$
G_{yx}(f_{yx}) \asymp e^{\delta t V H_{yx}(f_{yx};\rho)} \quad \text{with} \quad H_{yx}(f_{yx};\rho) = w_{yx}(\rho)(e^{f_{yx}} - 1). \quad (7.21)
$$

Multiplying all the independent generating functions, we get the joint SCGF, which we call *detailed Hamiltonian*:

$$
G(f) \asymp e^{\delta t y x H(f;\rho)} \quad \text{with} \quad H(f;\rho) = \sum_{x,y} w_{yx}(\rho)(e^{f_{yx}} - 1). \quad (7.22)
$$

Contracting at the level of the SCGF means specifying the variable to a more restricted value, which is much easier to perform than a minimisation: for instance, it might involve specifying a vector of independent variables to a value depending on only one parameter, rather than finding the optimal value of $L$ under a single linear constraint involving all variables. In the present case, contracting from currents $\lambda$ to displacements $\dot{\rho} = -\nabla \cdot \lambda$ translates to specifying the dynamical variable $f$ to the gradient of a state variable $u$, i.e. $f = \nabla u$. We obtain the SCGF of displacements, which we call the *standard Hamiltonian*:

$$\boxed{\tilde{G}(u) \asymp e^{\delta t V \mathcal{H}(u;\rho)} \quad \text{with} \quad \mathcal{H}(u;\rho) = H(\nabla u;\rho),} \tag{7.23}$$

where $u$ is the variable conjugate to the displacement rate $\dot{\rho}$, as can be determined by writing the consistent Legendre scalar product

$$u\dot{\rho} = -u(\nabla \cdot \lambda) = (\nabla u) \cdot \lambda = f \cdot \lambda. \tag{7.24}$$

**Example:**
For the replicator, we have

$$H(f;\rho) = a\rho(e^{f_{21}} - 1) + b\rho^2(e^{f_{12}} - 1) + c\rho(e^{f_{01}} - 1). \tag{7.25}$$

Specifying the entries of $f$ as

$$\begin{bmatrix} f_{21} \\ f_{12} \\ f_{01} \end{bmatrix} = \nabla u = \begin{bmatrix} u \\ -u \\ -u \end{bmatrix}, \tag{7.26}$$

where we remember that the gradient is *minus* the transpose of the divergence. Injecting this into $H$ yields the standard Hamiltonian

$$\mathcal{H}(u;\rho) = a\rho(e^u - 1) + b\rho^2(e^{-u} - 1) + c\rho(e^{-u} - 1) = \left(a\rho - (b\rho^2 + c\rho)e^{-u}\right)(e^u - 1), \tag{7.27}$$

whose Legendre transform can be taken relatively easily, to obtain the result found above in (7.19).

### 7.1.3   A few common functions you may encounter

This section is a short list of Lagrangians and Hamiltonians of common classes of models, for reference. They are all written at the most detailed level, where everything is explicit.

First, independent walkers on a network. Sites $x$, densities $\rho_x$, transition rates $r_{yx}$, currents $\lambda_{yx}$. The detailed Lagrangian and Hamiltonian are:

$$L(\lambda;\rho) = \sum_{x,y} \lambda_{yx} \ln\left(\frac{\lambda_{yx}}{r_{yx}\rho_x}\right) - \lambda_{yx} + r_{yx}\rho_x \quad , \quad H(f;\rho) = \sum_{x,y} r_{yx}\rho_x(e^{f_{yx}} - 1). \tag{7.28}$$

Note that, due to the independence of the particles, the Hamiltonian is linear in $\rho$. We sometimes refer to such cases as *linear processes*.

Second, a chemical network. Species $x$, complexes $\gamma$, concentrations $\rho_x$, reaction constants $k_{\gamma'\gamma}$, currents $\lambda_{\gamma'\gamma}$, stoichiometries $\nu_x^\gamma$. The detailed Lagrangian and Hamiltonian are:

$$L(\lambda;\rho) = \sum_{\gamma,\gamma'} \lambda_{\gamma'\gamma} \ln\left(\frac{\lambda_{\gamma'\gamma}}{\rho^{\nu^\gamma} k_{\gamma'\gamma}}\right) - \lambda_{\gamma'\gamma} + \rho^{\nu^\gamma} k_{\gamma'\gamma} \quad , \quad H(f;\rho) = \sum_{\gamma,\gamma'} \rho^{\nu^\gamma} k_{\gamma'\gamma} (\mathrm{e}^{f_{\gamma'\gamma}} - 1)$$

$$(7.29)$$

The Hamiltonian is no longer linear, unless the stoichiometries are such that the particles are independent. The structure is however very much the same as above, because in both cases the distribution of individual currents is Poissonian.

Third, independent diffusions. Particles in continuous space $\mathbb{R}^d$, with positions $x$, local density $\rho(x)$, drift field $V$, diffusion matrix $D$, local currents $j(x)$. The detailed Lagrangian and Hamiltonian are:

$$L(j;\rho) = \iint (j - V\rho + D\nabla\rho) \cdot \frac{D^{-1}}{4\rho} (j - V\rho + D\nabla\rho) \, \mathrm{d}x\mathrm{d}y,$$

$$H(f;\rho) = \iint f \cdot (Df\rho + V\rho - D\nabla\rho) \, \mathrm{d}x\mathrm{d}y \qquad (7.30)$$

Note that, once again, the Hamiltonian is linear in $\rho$.

Fourth, interacting diffusions, such as the famous WASEP. Same parameters as above, but with an extra function: the mobility $\sigma(\rho)$, which is no longer proportional to $\rho$. The detailed Lagrangian and Hamiltonian are:

$$L(j;\rho) = \iint (j - V\sigma(\rho) + D\nabla\rho) \cdot \frac{D^{-1}}{4\sigma(\rho)} (j - V\sigma(\rho) + D\nabla\rho) \, \mathrm{d}x\mathrm{d}y,$$

$$H(f;\rho) = \iint f \cdot (Df\sigma(\rho) + V\sigma(\rho) - D\nabla\rho) \, \mathrm{d}x\mathrm{d}y \qquad (7.31)$$

The Hamiltonian is no longer linear, unless the mobility $\sigma(\rho)$ is proportional to $\rho$, i.e. unless the particles are independent. The structure is however the same as above, with $L$ being quadratic in $j$ and $H$ being quadratic in $f$, because in both cases the distribution of individual currents is Gaussian.

Finally, the Gaussian approximation of a chemical network, related to the Chemical Langevin Equation (CLE). Same parameters as the second example, but here the currents are bidirectional and written as $j_{\gamma'\gamma}$. The detailed Lagrangian and Hamiltonian are:

$$L(j;\rho) \sim \sum_{\gamma,\gamma'} \frac{\left(j_{\gamma'\gamma} - \left(\rho^{\nu^\gamma} k_{\gamma'\gamma} - \rho^{\nu^{\gamma'}} k_{\gamma\gamma'}\right)\right)^2}{2\left(\rho^{\nu^\gamma} k_{\gamma'\gamma} + \rho^{\nu^{\gamma'}} k_{\gamma\gamma'}\right)},$$

$$H(f;\rho) \sim \sum_{\gamma,\gamma'} f_{\gamma'\gamma} \left(\frac{f_{\gamma'\gamma}}{2}\left(\rho^{\nu^\gamma} k_{\gamma'\gamma} + \rho^{\nu^{\gamma'}} k_{\gamma\gamma'}\right) + \left(\rho^{\nu^\gamma} k_{\gamma'\gamma} - \rho^{\nu^{\gamma'}} k_{\gamma\gamma'}\right)\right) \qquad (7.32)$$

Those functions are simply the second order approximation of the functions above, around typicality ($\lambda_{\gamma'\gamma} = \rho^{\nu^\gamma} k_{\gamma'\gamma}$ and $f = 0$). This translates the fact that this is a Gaussian approximation. This approximation is quite bad for any observable which cares about rare events, such as entropy production, or any marker of irreversibility.

### 7.1.4 Path integrals

Having described the little steps, we can combine them to get whole paths. For a finite time step $\delta t$, the probability of a path is a simple product

$$P_t[\{\rho_k\}] = \prod_{k=1}^{K} P_{\delta t}(\dot\rho_k|\rho_k). \tag{7.33}$$

Keeping only the initial and final conditions fixed, we can write a finite transition probability in terms of these paths, and also express the path probabilities in terms of currents rather than displacements, by introducing the continuity equation as a constraint. We get

$$P_t(\rho_K|\rho_0) = \sum_{\{\rho_k\}} P_t[\{\rho_k\}] = \sum_{\{\rho_k\}} \prod_{k=0}^{K-1} P_{\delta t}(\rho_{k+1}|\rho_k)$$

$$= \sum_{\{\lambda_k\}} \prod_{k=0}^{K-1} P_{\delta t}(\lambda_k|\rho_k)\delta\left(\rho_{k+1} - \rho_k - \nabla\cdot\lambda\delta t\right), \tag{7.34}$$

A $\delta t \to 0$ limit will turn this expression into a path integral, where the dynamical variable is either the displacement rate $\dot\rho$, or the currents $\lambda$, with an extra continuity constraint. This constraint, as well as the initial and final condition, can be implemented by delta functions.

$$P_t\left(\rho_t|\rho_0\right) \asymp \int e^{-V\int_0^t \mathcal{L}(\dot\rho;\rho)\mathrm{d}\tau}\delta\left(\rho(t) - \rho_t\right)\delta\left(\rho(0) - \rho_0\right)\,\mathrm{d}[\rho(\tau)]$$

$$\asymp \int e^{-V\int_0^t L(\lambda;\rho)\mathrm{d}\tau}\delta[\dot\rho + \nabla\cdot\lambda]\delta\left(\rho(t) - \rho_t\right)\delta\left(\rho(0) - \rho_0\right)\,\mathrm{d}[\lambda(\tau)]. \tag{7.35}$$

More generally, we can write the expectation of an observable $\mathcal{O}_t$ at time $t$ starting from a distribution $P_0$, which we both write at the large deviations scale in terms of two rate functions $U$ and $\theta$

$$P_0(\rho) \asymp e^{-VU(\rho)} \quad \text{and} \quad \mathcal{O}_t(\rho) \asymp e^{-V\theta(\rho)}. \tag{7.36}$$

We get the following path integral as our generic object of study (many objects of interest can be written in that form):

$$\langle\mathcal{O}_t\rangle_{P_0} \asymp \int e^{-V\left(\int \mathcal{L}(\dot\rho(\tau);\rho(\tau))\mathrm{d}\tau + U(\rho_0) + \theta(\rho_t)\right)}\,\mathrm{d}[\rho(\tau)]$$

$$\asymp \int e^{-V\left(\int L(\lambda(\tau);\rho(\tau))\mathrm{d}\tau + U(\rho_0) + \theta(\rho_t)\right)}\,\delta[\dot\rho + \nabla\cdot\lambda]\,\mathrm{d}[\lambda(\tau)]. \tag{7.37}$$

As stated earlier, we will mostly work inside the exponentials. The arguments of those two last exponentials are called *actions* (respectively the *standard action* and the *detailed action*), and they have a lot of structure, which we will look into next.

## 7.2 Properties of the action

Looking at the expressions above, we must remember that $V$ is large, so that the path with least action exponentially dominates the path integral. We will describe this dominating path with the usual tools of analytical mechanics, and we will see why we called our LDFs Lagrangians and our SCGFs Hamiltonians.

### 7.2.1  Equations of motion

The least action path can be characterised through a functional differentiation. For the standard action

$$\mathcal{S}[\dot\rho(\tau), \rho_0] = \int_{\tau=0}^{t} \mathcal{L}(\dot\rho(\tau); \rho(\tau))\mathrm{d}\tau + U(\rho_0) + \theta(\rho_t), \tag{7.38}$$

the differentiation of the action with respect to $\rho(\tau)$ at every time $\tau$, with an infinitesimal displacement $\delta\rho(\tau)$, gives:

$$\int_{\tau=0}^{t} (\partial_\rho\mathcal{L}\ \delta\rho(\tau) + \partial_{\dot\rho}\mathcal{L}\ \delta\dot\rho(\tau))\ \mathrm{d}\tau + \partial_\rho U(\rho_0)\delta\rho_0 + \partial_\rho\theta(\rho_t)\delta\rho_t = 0. \tag{7.39}$$

It is preferable to get rid of the term $\delta\dot\rho(\tau)$, which we can do through an integration by parts, which yields

$$\int_{\tau=0}^{t} \left(\partial_\rho\mathcal{L} - \frac{\mathrm{d}}{\mathrm{d}t}\partial_{\dot\rho}\mathcal{L}\right)\delta\rho(\tau)\ \mathrm{d}\tau + \partial_{\dot\rho}\mathcal{L}\ \delta\rho(t) - \partial_{\dot\rho}\mathcal{L}\ \delta\rho(0) + \partial_\rho U(\rho_0)\delta\rho_0 + \partial_\rho\theta(\rho_t)\delta\rho_t = 0. \tag{7.40}$$

Since the expression on the left-hand side needs to vanish regardless of the choice of displacement, the coefficients of $\delta\rho(\tau)$ must vanish separately at every time. Cancelling the first integrated term will yield the standard Euler-Lagrange equation, which justifies our calling $\mathcal{L}$ a standard Lagrangian:

$$\boxed{\partial_\rho\mathcal{L} - \frac{\mathrm{d}}{\mathrm{d}t}\partial_{\dot\rho}\mathcal{L} = 0.} \tag{7.41}$$

Canceling the boundary terms fix the boundary conditions:

$$\boxed{\partial_{\dot\rho}\mathcal{L}(\dot\rho_0; \rho_0) = \partial_\rho U(\rho_0) \quad \text{and} \quad \partial_{\dot\rho}\mathcal{L}(\dot\rho_t; \rho_t) = -\partial_\rho\theta(\rho_t).} \tag{7.42}$$

Note that, in the case where the initial or final condition is fixed to a single state $\tilde\rho$, the corresponding equation becomes irrelevant and the boundary condition should be replaced by $\rho_0 = \tilde\rho$ or $\rho_t = \tilde\rho$.

For the detailed action, the continuity constraint is enforced through a Lagrange multiplier $\mu$ (e.g. from a Fourier transform of the delta function):

$$S_{\lambda,\mu}[\dot\rho(\tau), \rho_0] = \int_{\tau=0}^{t} L(\lambda(\tau); \rho(\tau))\mathrm{d}\tau + U(\rho_0) + \theta(\rho_t) + \int_{\tau=0}^{t} \mu(\tau)(\dot\rho(\tau) + \nabla\cdot\lambda(\tau))\mathrm{d}\tau. \tag{7.43}$$

The minimisation of the integral terms with respect to the three independent variables $\rho$, $\lambda$ and $\mu$ yields three independent parts:

$$\delta\left[L + \mu(\dot\rho + \nabla\cdot\lambda)\right] = \left[\partial_\rho L\ \delta\rho + \mu\ \delta\dot\rho\right] + \left[\partial_\lambda L\cdot\delta\lambda + \mu\nabla\cdot\delta\lambda\right] + \left[\dot\rho + \nabla\cdot\lambda\right]\delta\mu = 0. \tag{7.44}$$

Each part has to vanish separately, yielding three equations. As above, we can handle the first $\partial_\rho L\ \delta\rho + \mu\ \delta\dot\rho = 0$ through an integration by parts on time, whose boundary terms will end up in the boundary conditions. The second $\partial_\lambda L\cdot\delta\lambda + \mu\nabla\cdot\delta\lambda = 0$ can be handled by transferring $\nabla$ to the left, as a gradient, i.e. applying $\mu\nabla\cdot\delta\lambda = (-\nabla\mu)\cdot\delta\lambda$. These two manipulations yield two equations, in addition to the continuity equation from the third term $\dot\rho + \nabla\cdot\lambda = 0$:

$$\partial_\rho L = \dot\mu \quad \text{and} \quad \partial_\lambda L = \nabla\mu. \tag{7.45}$$

The second equation allows us to get rid of $\mu$ in any equation, at the cost of applying $-\nabla$ to the rest of the equation. For instance, combining the two equations above yields the detailed Euler-Lagrange equation

$$\boxed{\nabla \partial_\rho L + \frac{\mathrm{d}}{\mathrm{d}t} \partial_\lambda L = 0.} \tag{7.46}$$

Moreover, the boundary terms from the partial integration of $\mu \, \delta\dot\rho$, combined with the differentials of $U_0$ and $\theta_t$, give the boundary conditions

$$\mu(0) = \partial_\rho U(\rho_0) \quad \text{and} \quad \mu(t) = -\partial_\rho \theta(\rho_t) \tag{7.47}$$

so that

$$\partial_\lambda L(\lambda_0; \rho_0) = \nabla \partial_\rho U(\rho_0) \quad \text{and} \quad \partial_\lambda L(\lambda_t; \rho_t) = -\nabla \partial_\rho \theta(\rho_t). \tag{7.48}$$

Note that, being LDFs, both Lagrangians vanish with zero derivative at their most probable values:

$$\boxed{\mathcal{L}(\dot\rho^\star(\rho); \rho) = L(\lambda^\star(\rho); \rho) = 0 \quad \text{or} \quad \partial_{\dot\rho}\mathcal{L}(\dot\rho^\star; \rho) = \partial_\lambda L(\lambda^\star; \rho) = 0} \tag{7.49}$$

**Example** No example here, calculations are too messy. Everything is nicer with Hamiltonians.

The Hamiltonians are the SCGFs of the Lagrangians, i.e. their Legendre transforms: in the detailed case,

$$H(f; \rho) = \sup_\lambda \left[ f \cdot \lambda - L(\lambda; \rho) \right] \quad \text{and} \quad L(\lambda; \rho) = \sup_f \left[ f \cdot \lambda - H(f; \rho) \right] \tag{7.50}$$

so that, if $L$ is differentiable and convex with respect to $\lambda$,

$$\boxed{H(f; \rho) + L(\lambda; \rho) = f \cdot \lambda \quad \text{with} \quad f = \partial_\lambda L \quad \text{or} \quad \lambda = \partial_f H,} \tag{7.51}$$

and in the standard case,

$$\mathcal{H}(u; \rho) = \sup_u \left[ u\dot\rho - \mathcal{L}(\dot\rho; \rho) \right] \quad \text{and} \quad \mathcal{L}(\dot\rho; \rho) = \sup_{\dot\rho} \left[ u\dot\rho - \mathcal{H}(u; \rho) \right], \tag{7.52}$$

so that, if $\mathcal{L}$ is differentiable and convex with respect to $\dot\rho$,

$$\boxed{\mathcal{H}(u; \rho) + \mathcal{L}(\dot\rho; \rho) = u\dot\rho \quad \text{with} \quad u = \partial_{\dot\rho}\mathcal{L} \quad \text{or} \quad \dot\rho = \partial_u \mathcal{H}.} \tag{7.53}$$

This automatically justifies their name, but let us still check that they verify nice equations.

The equations for optimal paths in terms of Hamiltonians can be found thus: first, from the Legendre transform, we have the force-flux relations

$$\lambda = \partial_f H \quad \text{or} \quad \dot\rho = \partial_u \mathcal{H}. \tag{7.54}$$

Second, we rewrite the Euler-Lagrange equation in terms of the Hamiltonian. For this, we first need to differentiate the Legendre relation with respect to $\rho$ to get

$$\partial_\rho \mathcal{H} + \partial_\rho \mathcal{L} + \partial_{\dot\rho}\dot\rho \, \partial_{\dot\rho}\mathcal{L} = u \, \partial_\rho\dot\rho = \partial_{\dot\rho}\mathcal{L} \, \partial_\rho\dot\rho \tag{7.55}$$

so that we immediately get the Hamilton equations (where we recall the force-flux relation as one of the two equations):

$$\dot{u} = -\partial_\rho \mathcal{H} \quad \text{with} \quad \dot{\rho} = \partial_u \mathcal{H}, \tag{7.56}$$

and in the same way

$$\dot{f} = \nabla \partial_\rho H \quad \text{with} \quad \dot{\rho} = -\nabla \cdot \partial_f H. \tag{7.57}$$

The boundary conditions simply translate to

$$u(0) = \partial_\rho U(\rho_0) \quad \text{and} \quad u(t) = -\partial_\rho \theta(\rho_t). \tag{7.58}$$

and

$$f(0) = \nabla \partial_\rho U(\rho_0) \quad \text{and} \quad f(t) = -\nabla \partial_\rho \theta(\rho_t) \tag{7.59}$$

Both Hamiltonians have a few useful properties:

- $H$ and $\mathcal{H}$ are conserved by the dynamics.

- $H(0; \rho) = 0$ and $\mathcal{H}(0; \rho) = 0$.

- $H$ is convex in $u$ and $\mathcal{H}$ is convex in $f$.

**Example**
for the replicator, we get

$$\dot{u} = -a(e^u - 1) - 2b\rho(e^{-u} - 1) - c(e^{-u} - 1) \tag{7.60}$$

with $\dot{\rho} = a\rho e^u - b\rho^2 e^{-u} - c\rho e^{-u}$,

$$[\dot{f}_{21}, \dot{f}_{12}, \dot{f}_{01}] = \left( a(e^{f_{21}} - 1) + 2b\rho(e^{f_{12}} - 1) + c(e^{f_{01}} - 1) \right) [-1, 1, 1] \tag{7.61}$$

with $\dot{\rho} = a\rho e^{f_{21}} - b\rho^2 e^{f_{12}} - c\rho e^{f_{01}}$.

One last thing we can do to be complete in our parallel with analytical mechanics is to examine the evolution of the static LDF at time $t$, i.e. the function $g_t(\rho_t)$ which is the logarithm of the probability of states at a fixed time $t$, starting, for instance, from an initial condition $\rho_0$ at $t = 0$. This LDF is precisely the action of the optimal path between $(\rho_0, t = 0)$ and $(\rho_t, t)$, i.e.

$$g_t(\rho_t) = \int_{0,\rho_0}^{t} \mathcal{L}(\dot{\rho}^\star; \rho^\star) d\tau = \int_{0,\rho_0}^{t} L(\lambda^\star; \rho^\star) d\tau \equiv S_t^\star(\rho_t, \rho_0). \tag{7.62}$$

Differentiating with respect to time $t$, and being careful to propagate the action over the infinitesimal time difference, we get the expected Hamilton-Jacobi equation

$$\partial_t g_t(\rho) = -\mathcal{H}(\partial_\rho g_t(\rho); \rho). \tag{7.63}$$

Note: varying the initial time instead yields a second Hamilton-Jacobi equation, with opposite signs. The computations are left as an exercise for the enthusiastic reader.

### 7.2.2    What it means

At this point, it is worth taking a step back from the mathematics to look at the physics of the systems we typically want to describe. For a system of particles diffusing in a fluid, the randomness in the current of particles has a very specific source: the collisions between the particles and the fluid molecules. At every point in time, these collisions exert a net force per unit time on the particles, which is random, and whose distribution is related to the momentum distribution of the fluid molecules (as characterised by their temperature) but is independent of the properties of the particles. Those properties only come into play when the random force produced by the fluid is translated into a random current of particles, depending in particular on their mobility (i.e. the ease with which they move in a medium).

The point here is the following: the dynamical variable $f$ of the detailed Hamiltonian can be interpreted as the *random force* produced by the medium. The relation between that force and the resulting current, called the *force-flux relation*, is simply

$$\boxed{\lambda = \partial_f H(f; \rho)} \tag{7.64}$$

up to a possible shift of the reference $f = 0$. In particular, when there are fixed external forces acting on the particles as well, they will appear additively with $f$ in this relation. This is particularly clear for systems with detailed balance, where the force deriving from the free energy of a system turns up as a special value of $f$. In more general settings, this can instead be taken as the definition of a force, being the quantity conjugate to currents through Legendre transforms.

Finally, the variable $u$ of the standard Hamiltonian can be interpreted as (minus) a random potential, in situations where all random forces derive from potentials.

## 7.3    Long time phenomenology

The framework we have presented is quite versatile: all kinds of observables can be analysed, or at least expressed in terms of solutions to known equations. It is however often complicated to get explicit results. One thing that usually makes things simpler is to look at long times, where systems converge to characteristic behaviours that are more clearly interpretable. For simplicity, we focus solely on the transition probability from $\rho_0$ to $\rho_t$, as an example to get an intuition on what features of the model are relevant for its long time behaviour.

### 7.3.1    Hamiltonian flow

Let us first recall the optimal value of the action over a finite time $t$, which tells us which path is the most likely, and what its probability is. It is useful to write it in terms of the Hamiltonian:

$$S_t^\star(\rho_t, \rho_0) \equiv \lim_{V \to \infty} -\frac{1}{V} \ln\left(\mathrm{P}_t(\rho_t | \rho_0)\right) = \inf_\rho \left[\int_{0;\rho_0}^{t;\rho_t} \mathcal{L}(\dot\rho; \rho)\mathrm{d}\tau\right] = \int_{0;\rho_0}^{t;\rho_t} \mathcal{L}(\dot\rho^\star; \rho^\star)\mathrm{d}\tau$$

$$= -t\mathcal{H}^\star + \int_{0;\rho_0}^{t;\rho_t} u^\star \, \dot\rho^\star \, \mathrm{d}\tau \tag{7.65}$$

where the asterisks indicate optimal values. The path we need to plug into the integral is a solution to the Hamilton equations with the correct duration $t$.

General calculations are difficult in this case, but a simple example can teach us a lot.

**Example**

Let us look once again at the replicator. Since this is a 1D model, the solutions to the Hamilton equations are simply the level lines of the Hamiltonian, with appropriate boundary conditions:

$$\mathcal{H}(u; \rho) = \left(a\rho - (b\rho^2 + c\rho)\mathrm{e}^{-u}\right)(\mathrm{e}^u - 1) = \mathrm{cst}. \tag{7.66}$$

We can easily study those trajectories by drawing the phase portrait of this Hamiltonian. The boundary conditions will correspond to vertical lines $\rho =$ constant, and we still need to select the trajectory (or trajectories) which join those two lines in the prescribed duration $t$. All of this is represented on the figure below, for three different durations, along with the shape of the trajectory $\rho(t)$ as a function of time.

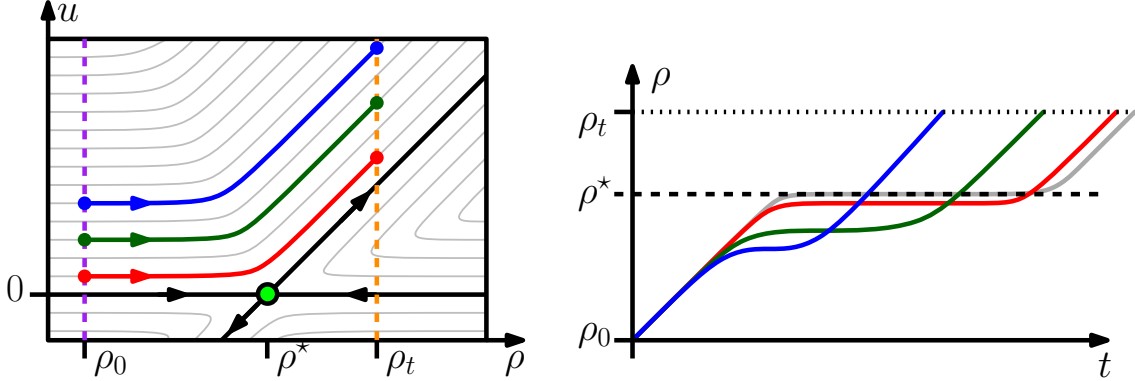

Figure 7.1: Left: sketch of Hamiltonian trajectories close to a fixed point, for increasing times (blue to green to red). Right: evolution of $\rho(t)$ for the same trajectories.

### 7.3.2 Convergence of trajectories

The duration of each trajectory does not only depend on its length in phase-space: for trajectories of the same length, those that are nearer a fixed point will have lower velocities and thus take a longer time to perform. A trajectory with given boundary conditions but a very long duration will therefore spend an extensive time in the vicinity of the fixed point, with finite *boundary layers* connecting it to the initial and final condition. This means that, despite the Hamiltonian nature of the dynamics, the trajectories converge to the critical point thanks to the confining boundary conditions, making that fixed point an *attractor*. The right plot of 7.1 represents $\rho(t)$ for several trajectories with the same boundary conditions but different durations, and the grey one represents the infinite time limit (where the middle flat section can be extended to arbitrarily long times). Each plot exhibits a plateau at a value of $\rho$ converging to $\rho^\star$, and a boundary layer on each side, which are constrained between $\rho_0$ and $\rho_t$.

Although this is just a fairly trivial example with a single scalar state variable, it is somewhat representative of typical stochastic processes, due to a few generic properties of their Hamiltonians, or certain reasonable assumptions. Here is an overview of some relevant considerations:

- The manifold $u = 0$ is stable under the dynamics, and contains the deterministic evolution of the system, including all the attractors of that evolution. This restricted dynamics $\dot{\rho} = \partial_u \mathcal{H}(0; \rho)$ is not Hamiltonian on its own, and is allowed to converge even though the appropriate boundary conditions are just $\rho(0) = \rho_0$.

- For the large deviations principle to hold, we need to ensure that the trajectories we consider stay away from the boundaries of phase-space (e.g. from $n = 0$ in the example above, since we need $n$ to be of order $V$). A reasonable assumption to guarantee that is to consider only systems that are *globally stable*, meaning that their deterministic dynamics are such that they always go back towards a given compact region of phase-space. In one dimension, this means that $\dot{\rho}$ must be positive for $\rho$ small enough, and negative for $\rho$ large enough. In general, this implies that all heteroclines (i.e. trajectories going from one attractor to another) outside that region are stable manifolds, so the system cannot diverge towards infinity in any direction, and the Poincaré-Bendixon theorem then guarantees that said region contains at least one stable attractor. With density-type boundary conditions for the Hamilton equations, this guarantees convergence in the long time limit.

- In many simple and physically motivated models, all attractors belong to the $u = 0$ manifold. This means that the value of $\mathcal{H}^\star$ in the principal function (7.65) will necessarily be 0. There can also be closed trajectories with $\mathcal{H} \neq 0$, which are not attractors of the deterministic dynamics, but all known examples are such that $\mathcal{H} < 0$ so that they are not selected by the infimum.

- Those attractors are typically connected in two ways: through the $u = 0$ manifold, which contains the deterministic trajectories between them, and through another manifold $u^\star(\rho)$ such that $\mathcal{H}(u^\star(\rho); \rho) = 0$, which contains less probable trajectories called *instantons*, of which we give an example below.

To summarise, the long-time behaviour of the Hamilton equations for stochastic processes is mostly characterised by the attractors of the deterministic dynamics along with their stable and unstable manifolds, which all verify $\mathcal{H} = 0$. In practice, the only type of attractor that is manageable with current methods is fixed points (and some simple limit cycles).

### 7.3.3 Computations of the extinction time in the replicator

To illustrate those last few points, let us answer a specific long time question: what is the probability for the replicator to go extinct in the long run, i.e. to reach density $\rho = 0$. To make things easier, we can take $c$ very small : this parameter was here to avoid having an absorbing state, but here we don't mind.

The special manifolds of $\mathcal{H}$ verify

$$\mathcal{H}(u; \rho) = 0 \quad \Rightarrow \quad u = 0 \quad \text{or} \quad u = \ln\left(\frac{b\rho^2 + c\rho}{a\rho}\right) \sim \ln\left(\frac{b\rho}{a}\right). \tag{7.67}$$

The probability rate to reach $\rho = 0$ along the escape trajectory, starting from the steady state $\rho = a/b$, is then given in terms of the corresponding action

$$S = \int u(\rho)\dot{\rho}\,\mathrm{d}\tau = \int_{a/b}^0 u(\rho)\mathrm{d}\rho = \frac{a}{b} \tag{7.68}$$

so that

$$\mathrm{P}_t(\rho_t = 0|\rho_0 = a/b) \asymp \mathrm{e}^{-Va/b} \tag{7.69}$$

so that the extinction timescale is its inverse, $\tau = \mathrm{e}^{Va/b}$, and is exponentially large in $V$.

We can also compare this result to what we would get in the Chemical Langevin Equation approximation, to see how good of an approximation it is. The approximate Hamiltonian is

$$\mathcal{H}(u;\rho) \sim u\left(a\rho - b\rho^2 - c\rho + \frac{u}{2}\left(a\rho + b\rho^2 + c\rho\right)\right) \tag{7.70}$$

so that

$$\mathcal{H}(u;\rho) = 0 \quad \Rightarrow \quad u = 0 \quad \text{or} \quad u = 2\frac{b\rho + c - a}{b\rho + c + a} \sim 2\frac{b\rho - a}{b\rho + a}. \tag{7.71}$$

The probability rate to reach $\rho = 0$ along the escape trajectory, starting from the steady state $\rho = a/b$, is given in terms of the corresponding action

$$S = \int_{a/b}^{0} u(\rho)\mathrm{d}\rho = \frac{a}{b}(\ln(4) - 1) \approx 0.386\frac{a}{b}, \tag{7.72}$$

so that

$$\mathrm{P}_t(\rho_t = 0|\rho_0 = a/b) \asymp (4/e)^{-Va/b} \tag{7.73}$$

so that the extinction timescale is its inverse, $\tau = (4/e)^{Va/b}$. This is exponentially longer than the real value.

## 7.4   Path integral formalisms in physics: a comparison

To conclude, let us compare the path integral formalism we have described here with its equivalents within classical and quantum mechanics. There are many similarities in structure, but the specificities of each version make all the difference.

Let us first recap the relevant characteristics of the formalism in statistical mechanics, which is built starting from the Lagrangian:

- The Lagrangian is the logarithm of a classical probability distribution: it is positive and vanishes in the most likely state. The Lagrangian is physical, whereas the Hamiltonian is effective (it has no physical interpretation, as it is merely a generating function).

- The Hamiltonian vanishes along the most likely trajectory, and is constant along other local extrema, but other trajectories that do not verify the Hamilton equations are possible due to statistical fluctuations.

- The dynamical variable of the Hamiltonian is a random force or a potential.

- The boundary conditions of the Hamilton equations are mixed, allowing for convergence even with conservative dynamics.

In contrast, the formalism for quantum mechanics is built starting from the Hamiltonian:

- The Hamiltonian is the logarithm of a quantum probability distribution: it is positive and vanishes in the most likely state. It is physically interpretable as an energy, whereas the Lagrangian is effective.

- The Lagrangian vanishes along the typical trajectory, but other trajectories that do not verify the Euler-Lagrange equations are possible due to quantum fluctuations.

- The dynamical variable of the Hamiltonian is a physical momentum.

- The boundary conditions of the Hamilton equations are full initial conditions: trajectories cannot converge in time due to unitary dynamics.

In analytical mechanics, the starting point is the conservation in time of the total energy of a system:

- The Hamiltonian is an energy, and is usually positive. Its conservation in time leads to the Hamilton equation. Only the trajectories that verify the Hamilton equations are physical.

- The Lagrangian is effective, and is merely a convenient object with which to write the equations of motion of the system. Only the trajectories verifying the Euler-Lagrange equations are physical, as there are no fluctuations.

- The dynamical variable of the Hamiltonian is a physical momentum.

- The boundary conditions of the Hamilton equations are full initial conditions: trajectories cannot converge in time due to conservative dynamics.

For more details on large deviation theory and its applications in physics, along with illustrations, exercices, and moderate ranting about the definitional nightmare that is thermodynamics, the reader may refer to *Large deviations in statistical mechanics: from free energies to path integrals*, a set of lecture notes by myself (Alexandre Lazarescu), which will appear on arXiv some day.

# 8    Metastability In and Out of Equilibrium

**Gianmaria Falasco and Massimo Bilancioni.**[21] *Metastability indicates the existence of long-lived, yet quasi-stationary states, possibly maintained by continuous energy dissipation. Examples are the logic state of an electronic circuit and the homeostatic state of a biochemical reaction network. We review the definition of metastability based on the spectral theory of the generator of a Markov process. We present two prototypical examples representing jump processes in the large-size limit and diffusive dynamics in the small-temperature limit. For the latter, we introduce the WKB expansion of the Fokker-Planck equation, the quasi-potential and its relation with the life-time of metastable states. For detailed balance dynamics, we retrieve the Arrhenius law from the standpoint of large deviations theory. For dynamics subjected to nonconservative forces, we show the relation between the life-time of metastable states and dissipation.*

## 8.1    Introduction

### 8.1.1    Intuitive idea

The existence of various dynamical regimes, widely separated by distinct timescales is called metastability [105]. In the simplest case, metastability manifests itself when a process seems stationary for a very long time before jumping to a new seemingly stationary state. Such states are local minima of the free energy for nondissipative systems [106], i.e. whose dynamics are detailed balanced. What are they, when detailed balance is not satisfied? How to characterize metastability in terms that are intrinsically dynamic and make no *a priori* reference to geometric concepts such as free energy landscapes? These questions are crucial far from equilibrium where the free energy and the stationary probability are no longer related to each other in a simple way.

### 8.1.2    Examples

To set the stage we list some notable examples of metastability the reader might be familiar with. Detailed balanced systems include:

- the Ising model (in contact with a single thermal bath) below its critical temperature. Upwards and downwards magnetizations are the two metastable states.

- Carbon under standard conditions in the diamond state. It will eventually relax to a lower free energy, i.e graphite.

- A supercooled liquid, i.e. a liquid cooled fast below the melting point in the absence of crystal nucleators, e.g. impurities. It will eventually solidify [107].

Metastable dynamics that are not detailed balanced comprise

- Climate. It has (at least) two metastable states, warm and snowball—paleoclimatology identified 2 full glaciations and tens of ice-ages. Transitions are induced by (weather) noise and (astronomical) time-periodic factors [108].

- Chemical reaction networks: gene regulatory networks have multiple phenotypic (or epigenetic) states which are metastable—each state is characterized by the activation of different gene patterns, resulting in different protein concentrations. Intrinsic (i.e. finite copy numbers) and extrinsic (i.e. variability) noise cause transitions [109].

---

[21]GF was the lecturer and wrote this chapter; MB was the angel.

- Electronic circuits: in a bit-storage element (CMOS inverters) 0 and 1 states are metastable. Thermal noise induces transitions, resulting in computation errors [110].

## 8.2 Metastability for Markov processes

The phenomenon is understood within the spectral theory of the Markovian generator [111–114], which is summarized as follows. The dynamical equation for the probability density vector $|P_t\rangle$ at time $t$ is

$$\partial_t |P_t\rangle = R |P_t\rangle \tag{8.1}$$

where the operator $R$ is the generator of the stochastic process. Think of $R$ as the irreducible rate matrix [22] for a Markov jump process on a finite state space of dimension $N$. States can be labelled by $n$ such that $\langle n|P_t\rangle = P(n,t)$. The scalar product is standard matrix multiplication.

### 8.2.1 Spectral decomposition

The time evolution of the probability distribution can be expanded in the right eigenvectors $\left|\psi_\alpha^{(r)}\right\rangle$ of the operator $R$, with eigenfunctions $\psi_\alpha^{(r)}(n) = \left\langle n\middle|\psi_\alpha^{(r)}\right\rangle$, defined by the equations

$$R \left|\psi_\alpha^{(r)}\right\rangle = \lambda_\alpha \left|\psi_\alpha^{(r)}\right\rangle \qquad\qquad \left\langle\psi_\alpha^{(l)}\right| R = \lambda_\alpha \left\langle\psi_\alpha^{(l)}\right|. \tag{8.2}$$

Therefore, we can write

$$|P_t\rangle = \sum_\alpha b_\alpha \left|\psi_\alpha^{(r)}\right\rangle e^{\lambda_\alpha t}, \tag{8.3}$$

where $b_\alpha = \left\langle\psi_\alpha^{(l)}\middle|P_0\right\rangle$ is the overlap of the initial condition with the left eigenvector $\left\langle\psi_\alpha^{(l)}\right|$. Since $R$ is not Hermitian in general [23], left and right eigenvectors are different but still form an orthonormal basis

$$\left\langle\psi_\alpha^{(l)}\middle|\psi_{\alpha'}^{(r)}\right\rangle = \delta_{\alpha\alpha'} \qquad\qquad I = \sum_\alpha \left|\psi_\alpha^{(r)}\right\rangle \left\langle\psi_\alpha^{(l)}\right|, \tag{8.4}$$

and the eigenvalues $\lambda_\alpha$ are not necessarily real. However, since $R$ is an irreducible stochastic matrix [24], by the Perron-Frobenius theorem [25] its eigenvalue $\lambda_0 = 0$ with largest real part

- is nondegenerate,

- is associated to the left eigenfunction $\left\langle\psi_0^{(l)}\right| = \langle-|$, i.e.

$$0 = \langle-| R, \tag{8.5}$$

  with $\langle-|n\rangle$ constant,

- and to a right eigenfunction with positive components, which is the stationary probability distribution, i.e. $\left|\psi_0^{(r)}\right\rangle = |P_\infty\rangle$.

Hence, $R$ has $N$ nonpositive eigenvalues $\lambda_\alpha$ which can be ordered by their real part $\mathrm{Re}(\lambda_\alpha) \geq \mathrm{Re}(\lambda_{\alpha+1})$.

---

[22]The graph is connected, all states can be reached and so the dynamics is ergodic.

[23]This means that in general it cannot even be diagonalized, it can only be put in a Jordan normal form. This happens if, for some values of the system's parameters (called exceptional points), there are degenerate eigenvalues. We assume that the smaller eigenvalues are not degenerate.

[24]Actually, $e^R$ is a stochastic matrix, since it has only positive entries and $\langle-|e^R = \langle-|$ because of probability normalization.

[25]Or its infinite-dimensional generalization, the Krein-Rutman theorem.

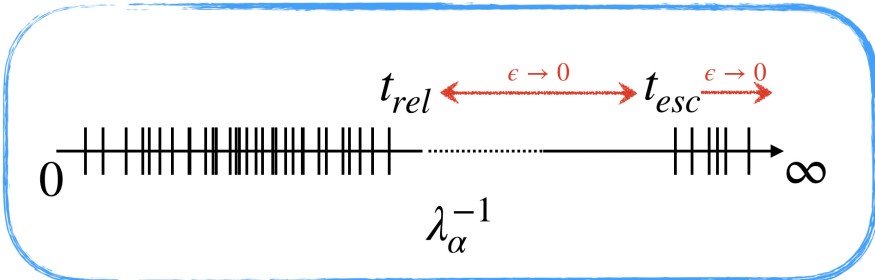

Figure 8.1: Sketch of the spectrum of $R$ (inverse of the eigenvalues) for a system displaying metastability.

### 8.2.2 Dynamical identification of metastable states

We say that the generator depends on a parameter $\epsilon$ which we can imagine as the temperature, the inverse size of the system, etc.. Metastability appears if there exist eigenvalues $\lambda_1, \ldots, \lambda_M$ whose real part goes to $0 = \lambda_0$ as $\epsilon \to 0$, while all others $\lambda_\alpha$ with $\alpha > M$ stay finite. We can then identify two well separated timescales:

- $t_{esc} = \min(\lambda_1^{-1}, \ldots, \lambda_M^{-1})$ is the shortest escape time out of metastable states

- $t_{rel} = \max(\lambda_{M+1}^{-1}, \ldots, \lambda_N^{-1})$ the longest time to relax within a metastable state.

The gap in the spectrum, whose inverse is related to $t_{esc} - t_{rel}$, corresponds to a diverging time scale separating the (fast relaxation) dynamics within metastable states from the slow (jump dynamics) between them.

This intuitive picture can be made precise. It can be shown [111] that on the intermediate timescale $t_{rel} \ll t \ll t_{esc}$ one can find

- a basis of $M + 1$ right eigenvectors $|\rho_i\rangle$ which are positive normalized, stationary, i.e $R|\rho_i\rangle \simeq 0$, nonzero only on non-overlapping regions of the phase space;

- a basis of $M + 1$ left eigenvectors $\langle q_i|$ such that $\langle q_i| R \simeq 0$ and satisfying the ortogonality condition $\langle q_i|\rho_j\rangle \simeq \delta_{ij}$.

Basically, $\rho_i(n)$ are the probability distributions of the metastable states and $q_i(n)$ are nonzero only on their basin of attraction (decaying rapidly to zero outside). Moreover, $q_i(n)$ gives the probability of reaching the metastable state $i$ starting from $n$ on the intermediate timescale. It can be used to define the transition state out of the metastable state $i$, which is located at $q_i(n) = 1/2$. They are called commitor functions [115]. Note that they both are linear combinations of the eigenvectors $\left\langle \psi_\alpha^{(l)} \right|$ and $\left| \psi_\alpha^{(r)} \right\rangle$, which allow us to write the evolution operator as a $e^{tR} \simeq \sum_i |\rho_i\rangle \langle q_i|$ on the intermediate timescale [26].

Ultimately, if $\epsilon \to 0$ at finite $t$, the system probability can only converge to those $\rho_i$ that are selected by the initial condition, i.e. with $\langle q_i|P_0\rangle \neq 0$. Therefore, the long-time limit and the small-$\epsilon$ limit do not commute, because if $t \to \infty$ at any finite $\epsilon$ the probability distribution converges surely to the unique $\left| \psi_0^{(r)} \right\rangle$. Historically, this fact is known as Keizer's paradox [80, 116].

---

[26] Everywhere in this subsection the symbol $\simeq$ means that we ignore terms of order $e^{t\lambda_\alpha}$ with $\alpha > M$.

### 8.2.3   Example: the Schlögl model

We consider the Schlögl model [80,117], introduced to describe first order phase transitions far from equilibrium. It describes the stochastic dynamics of the copy number $n$ of a chemical species X, subject to the chemical reactions

$$2\,\mathrm{X} + \mathrm{A} \underset{k_{-1}}{\overset{k_{+1}}{\rightleftharpoons}} 3\,\mathrm{X} \;; \quad \mathrm{B} \underset{k_{-2}}{\overset{k_{+2}}{\rightleftharpoons}} \mathrm{X} \;, \tag{8.6}$$

taking place in a well-mixed container of volume $V$. We assume that their kinetics follow the mass-action law, valid for dilute ideal mixtures. The species A and B are kept at constant concentration by chemical reservoirs. If $V$ is large in comparison to the molecular scale, its inverse can be taken to be the small parameter $\epsilon = 1/V$. Therefore, in the macroscopic limit we introduce the intensive variable $x = n/V = n\epsilon$ corresponding to the concentration of species X, and the scaled (by $V$) transition rates

$$r_1(x) = k_{+1}ax^2, \quad r_{-1}(x) = k_{-1}x^3, \quad r_2(x) = k_{+2}b, \quad r_{-2}(x) = k_{-2}x. \tag{8.7}$$

By expanding the chemical master equation, multiplying by $x$ and integrating, one arrives at the rate equation of chemistry

$$\dot{x} = r_1(x) - r_{-1}(x) + r_2(x) - r_{-2}(x). \tag{8.8}$$

in the limit $\epsilon \to 0$ in which the probability peaks around the most likely value $\langle x \rangle$.

In figure (8.2) we show for the inverse of the volume $\epsilon = 1/30 \simeq 3.3 \times 10^{-2}$ the first two right and left eigenfunctions obtained numerically with Mathematica. Clearly, from their linear combination we can obtain the (positive, normalized) metastable states and the committors (1 on the respective attractor). Partial overlapping of the metastable states and the discrepancies with the $\epsilon = 0$ predictions are due to the finite value of $\epsilon$.

In figure (8.3) we show the absolute value of the difference between consecutive leading eigenvalues. Even at the moderately small value $\epsilon = 1/30$ it is apparent that $\lambda_1 \to 0$ while the other eigenvalues stay finite.

### 8.2.4   Detailed balance vs. nondetailed balance dynamics

The first important point is that a system with a convex energy function $\mathcal{U}$ can develop metastable states when it is subjected to nonpotential forces. Such states will require continuous energy dissipation to exist (see examples above). A second point is that in the absence of detailed balance $R$ is not Hermitian, so the eigenvalue $\lambda_\alpha$ can have a nonzero imaginary part. Thus, a metastable state can be periodic, e.g. a stable limit cycle is reached as $\epsilon \to 0$ [118,119]. Nevertheless, we do not explicitly consider such periodic metastable states hereafter, or more general quasi-periodic and chaotic ones. In general, $R$ is symmetrizable only if detailed balance holds. In this case the leading right eigenfunction is the equilibrium Gibbs distribution, e.g. $\psi_0^{(r)}(n) \propto e^{-\beta\mathcal{U}(n)}$ in the canonical ensemble at inverse temperature $\beta$, such that a change of basis defines the symmetric operator $R_{sym} = e^{\beta\mathcal{U}/2}Re^{-\beta\mathcal{U}/2} = R_{sym}^\dagger$.

## 8.3   Overdamped Langevin dynamics: low temperature limit

The general framework for the description of metastability can be applied to systems with a continuous state space [120], which we analyze more explicitly in the following by using large deviations theory [94]. We consider the Langevin equation for $x \in \mathbb{R}^d$,

$$\dot{x} = F(x) + \sqrt{2\epsilon}\xi \tag{8.9}$$

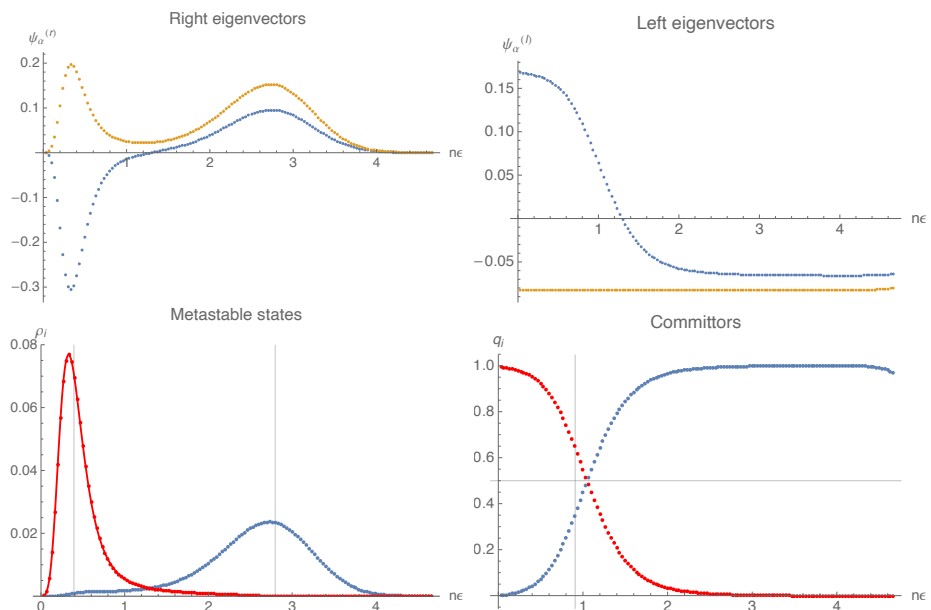

Figure 8.2: For $\epsilon^{-1} = 30$, first two right and left eigenvectors (in (a) and (b), respectively), and their linear combination giving the metastable states (c) and the committors (d). Vertical grey lines in the panels indicate the stable (lower left) and unstable (lower right) fixed points of the deterministic dynamics (8.8), corresponding to $\epsilon = 0$. The horizontal grey line in (d) is at $q_i = 1/2$ and singles out the transition state. Numerical evaluation is obtained by approximating $R$ with a $150 \times 150$ matrix (finite size effects are also visible at $n\epsilon \simeq 5$). Parameters are $k_1 a = 4.1, k_{-1} = 1, k_2 b = 1, k_{-2} = 4$.

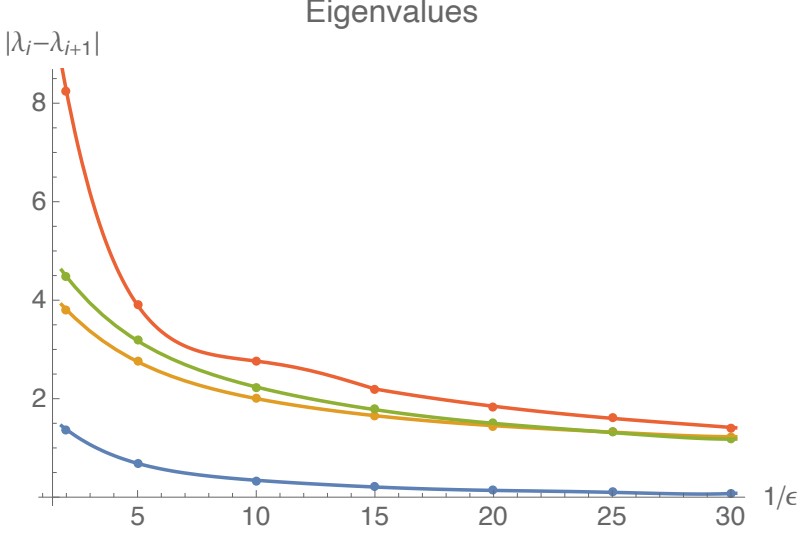

Figure 8.3: For $\epsilon = 1/30$, absolute value of the difference between the first 5 consecutive eigenvalues. System's parameter as before.



Figure 8.4: Example of a drift field composed by an underlying double well free energy potential $\mathcal{U}(x)$ and a superimposed nonequilibrium forcing $f(x)$. The arrows represent schematically the drift's direction and intensity.

with (weak) noise, Gaussian with zero mean and correlation matrix $\langle \xi(t)\xi(0)\rangle = D(x)\delta(t)$ [27]. The drift field has stable and unstable zeros, denoted $x_i^*$ and $x_\nu$, respectively. Namely, $F(x_i^*) = 0 = F(x_\nu)$ and with (assume nonsingular) Jacobian matrix $\nabla F(x)$ having only negative eigenvalues in $x_i^*$ (resp. at least a positive eigenvalue in $x_\nu$). Namely, $x_i^*$ are stabled fixed points of (8.9), while $x_\nu$ are saddle points.

We specify the drift field as a conservative part coming from an underlying free energy landscape $\mathcal{U}(x)$ and a nonequilibrium forcing part $f$ (Fig. 8.4):

$$F(x) = -D(x) \cdot \nabla \mathcal{U}(x) + f(x). \tag{8.10}$$

We take an autonomous dynamics leading to a stationary probability density $P_\infty$ and current $J_\infty$, i.e. the stationary solution of the Fokker-Planck

$$\partial_t P(x,t) = -\nabla \cdot J(x,t) = -\nabla \cdot [F(x)P(x,t) - D(x)\epsilon\nabla P(x,t)]. \tag{8.11}$$

Detailed balance dynamics corresponds to $f = 0$, in which case $x_i^*$ are the local minima of the energy $\mathcal{U}$ and $x_\nu$ its saddles. This setup can describe, for example, interacting colloidal particles under a shear flow at low temperature[28].

### 8.3.1 Deterministic limit and typical fluctuations

For $\epsilon = 0$ the dynamics are deterministic, namely, $P(x,t) = \delta(x - x(t))$ where $x(t)$ is the solution of

$$\dot{x}(t) = F(x(t)). \tag{8.12}$$

Equation (8.12) describes the relaxation to the stable fixed point $x_i^*$, where $i$ is selected by the initial condition $x(0) = x_0$. The dynamics are therefore nonergodic, with the state space partitioned into the various $i = 0, \ldots, M$ basins of attactions.

To describe the low-$\epsilon$ regime, it is therefore natural to use the WKB ansatz

$$P(x,t) = W_\epsilon(x,t)e^{-\frac{1}{\epsilon}I(x,t)}, \tag{8.13}$$

---

[27]All choices of stochastic calculus are equivalent to leading order in $\epsilon$. We will use Ito later on when we do calculations at leading order.

[28]in the sense that $k_B T$ is small in comparison to the typical interaction energy and forcing.

with $W_\epsilon(x,t)$ subexponential in $\epsilon$. In the language of large deviations theory,

$$I(x,t) = -\lim_{\epsilon \to 0} \epsilon \ln P(x,t), \tag{8.14}$$

is called rate function and concentration of probability happens at speed $1/\epsilon$. The logarithmic equality (8.14) is usually written $P(x,t) \asymp e^{-\frac{1}{\epsilon}I(x,t)}$. Plugging (8.13) into (8.11) we obtain at the leading order

$$-\partial_t I(x,t) = H(x, \nabla I(x,t)) \tag{8.15}$$

with Hamiltonian

$$H(x,\pi) = F(x) \cdot \pi + \pi \cdot D(x) \cdot \pi. \tag{8.16}$$

Note that the Hamiltonian is the cumulant generating function of a Gaussian noise with mean $F(x)$ and covariance matrix $D(x)$. Equation (8.15) is a Hamilton-Jacobi equation for the action function $I(x,t)$ and conjugate momentum $\pi = \nabla I$. We can recast the problem in terms of the equivalent Hamiltonian equations

$$\dot{x} = F(x) + 2D(x) \cdot \pi, \quad \dot{\pi} = -\nabla F(x) \cdot \pi + \pi \cdot \nabla D(x) \cdot \pi. \tag{8.17}$$

These equations describe the most likely trajectories followed by the system as $\epsilon \to 0$. They also follow from the minimization of the action functional [29]

$$\mathcal{A}[\{x(t), \pi(t)\}_0^\tau] = \int_0^\tau dt (\pi(t) \cdot \dot{x}(t) - H(x(t), \pi(t))). \tag{8.18}$$

From lecture 6 we know that such action determines the leading order probability of paths $\{x(t)\}_0^\tau$ as[30]

$$\text{Prob}[\{x(t)\}_0^\tau | x_0] \asymp \int \mathcal{D}\pi e^{-\frac{1}{\epsilon}\mathcal{A}[\{x(t), \pi(t)\}_0^\tau]}. \tag{8.19}$$

The stationary rate function $I_\infty(x) = -\lim_{\epsilon \to 0} \epsilon \ln P_\infty(x)$ is obtained by solving

$$0 = H(x, \nabla I_\infty(x)), \tag{8.20}$$

which means that we look for solution of (8.17) on the submanifold $H = 0$. If $f \neq 0$, $I_\infty$ generally has nondifferentiable points, so that (8.20) should be solved within each basin on attraction, giving a local rate function [124].

### 8.3.2  Instantons and escape rates

From the probability current introduced in (8.11), we can also derive another decomposition of the drift field

$$F(x) = -D(x)\nabla I_\infty(x) + v(x) \tag{8.21}$$

---

[29]The boundary conditions are determined by the initial distribution $P(x,0)$ and possibly from a constrain at the time of interest $\tau$.

[30]This can be derived in a number of more or less rigorous ways. For example, starting from the Langevin equation and the Gaussian path probability of the noise $\{\xi(t)\}_{t=0}^\tau$ as in the MSRDJ formalism [121–123], or à la Feynman by time-slicing a formal solution of the Fokker-Planck equation [18] .

that does not make reference to the underlying microscopic details (energy and forcing), but rather uses dynamically emergent quantities, i.e the stationary rate function and the probability velocity

$$v(x) = \lim_{\epsilon \to 0} \frac{J_\infty(x)}{P_\infty(x)} \qquad\qquad v \cdot \nabla I_\infty = 0. \qquad (8.22)$$

The last property follows immediately from the substitution of (8.21) into (8.20). Note that $v(x)$ is zero for detailed balance dynamics (since $J_\infty$ is zero). Therefore, we check that

$$I_\infty = \mathcal{U} \quad \text{if} \quad f = 0 \qquad\qquad (8.23)$$

namely, the equilibrium canonical (Gibbs) distribution is the stationary probability distribution for detailed balance dynamics. It is an important fact that with detailed balance the subexponential prefactor $W$ is independent of the state $x$. It is a mere normalization constant, as can be checked by direct substitution into (8.11). This remains true for $f \neq 0$ for $\nabla \cdot f = 0$. In this case, the stationary probability remains Gibbsian even if $J_\infty \neq 0$. This kind of nondetailed balance dynamics is known to accelerate relaxation, i.e. the absolute value of the first nonzero eigenvalue $|\lambda_1|$ increases in the presence of a divergenceless $f \neq 0$ [125, 126].

   This decomposition is useful to analyze the two main sets of solutions of (8.17), determined by the boundary conditions. We see that $\pi = 0$ is a solution giving the noiseless dynamics (8.12) which we have already seen as a relaxation to the stable fixed point $x_i^*$. Typical fluctuations (called optimal trajectories or instantons) are characterized by $\pi \neq 0$. One particular family of such trajectories is that happening on the manifold of Hamiltonian $H = 0$: from (8.20) we see that they correspond to $\pi(t) = \nabla I_\infty(x(t))$. Plugging that into the first of (8.17) we obtain

$$\dot{x} = F(x) + 2D(x) \cdot \nabla I_\infty(x) = D(x) \cdot \nabla I_\infty(x) + v(x). \qquad (8.24)$$

These dynamics take place in a 'reversed' rate function but with the same currents. They are trajectory, called instantons, that start arbitrarily close to the stable fixed point $x_i^*$ and reach $x_\nu$ in an arbitrarily long time [31]. Notably, if $f = 0$ we find that deterministic dynamics and typical trajectories are one the time-reversed of the other, which is the content of detailed balance dynamics specialized to this set of trajectories.

   If we use (8.20) in (8.19), and the insight that such instanton takes infinite time to escape a fixed point, we obtain the long-time transition probability

$$\lim_{t \to \infty} \text{Prob}(x_i^* \to x) \asymp e^{-\frac{1}{\epsilon} \int_{x(0)=x_i^*}^{x(\infty)=x} dt\, \dot{x}(t) \cdot \nabla I_\infty(x(t))} = e^{-\frac{1}{\epsilon}[I_\infty(x) - I_\infty(x_i^*)]} \qquad (8.25)$$

Note that the trajectory could be continued from $x_\nu$ to the other stable fixed point $x_{j\neq i}^*$ without changing the result: the action along any relaxation trajectory is zero [32].

   It can be proved [127] that for $\epsilon \to 0$ the escape time from a basin of attraction through the saddle $x_\nu$ is exponentially distributed with rate [128]

$$k_\nu \asymp \lim_{t \to \infty} \text{Prob}(x_i^* \to x_\nu) \asymp e^{-\frac{1}{\epsilon}[I_\infty(x_\nu) - I_\infty(x_i^*)]}. \qquad (8.26)$$

---

[31] This is because both $\nabla I_\infty(x)$ and $v(x)$ tend to zero close to the zeros of $F$ ($I_\infty$ by definition, and $v$ as a consequence of (8.21)).

[32] We could also start from any other point in the basin of attraction of $x_i^*$ without changing the result: For very long times we would first relax to $x_i^*$ (with a zero-action relaxation trajectory) and the follow the instanton.

As a consequence, the mean first passage time to reach the unstable fixed point $x_\nu$ from the basin of attraction of $x_i^*$ equals the inverse of the escape rate (8.26). This boils down to the Arrhenius law when $f = 0$; from eq. (8.23), it is

$$k_\nu^{eq} \asymp e^{-\frac{1}{\epsilon}[\mathcal{U}(x_\nu) - \mathcal{U}(x_i^*)]}. \tag{8.27}$$

### 8.3.3 Thermodynamic bounds on the rate function and the escape rates

For $f$ arbitrary, we make use of the following relations

- The decomposition of the scaled entropy flow rate (in the reservoir) into the adiabatic and nonadiabatic component 3: $\dot{\sigma} = \dot{\sigma}_{ad} + \dot{\sigma}_{na}$ [129, 130]. The entropy flow rate can be obtained by the log ratio of probabilities for the forward and backward trajectories conditioned to start on a given state, together with the drift field decomposition (8.21),

$$\sigma = \epsilon \ln \frac{\text{Prob}[\{x(t)\}_0^\tau | x_0]}{\text{Prob}[\{\tilde{x}(t)\}_0^\tau | x_\tau]} = \int_0^\tau dt \dot{x} \cdot D^{-1} \cdot F = \int_0^\tau dt \dot{x} \cdot D^{-1} \cdot (-D \cdot \nabla I_\infty + v). \tag{8.28}$$

- The positivity of adiabatic entropy production rate along relaxation and instanton:

$$\dot{\sigma}_{ad} = \dot{x} \cdot D^{-1} \cdot v = (\pm D \nabla I_\infty + v) \cdot D^{-1} \cdot v = v \cdot D^{-1} \cdot v \geq 0. \tag{8.29}$$

- the fact that the nonadiabatic entropy flow rate equals $\dot{\sigma}_{na} = -\dot{x} \cdot \nabla I_\infty = -\frac{d}{dt} I_\infty(x(t))$ along any trajectory.

Integrating the entropy production rate along the relaxation ($\sigma_{\nu \to i}$) and the instanton ($\sigma_{i \to \nu}$) and using the relation (8.26) between the rate function difference and the transition rate, we obtain

$$-\sigma_{\nu \to i} \leq \lim_{\epsilon \to 0} \epsilon \ln \kappa_\nu \leq \sigma_{i \to \nu}. \tag{8.30}$$

With detailed balance, $\sigma_{\nu \to i} = -\sigma_{i \to \nu} = \mathcal{U}(x_\nu) - \mathcal{U}(x_i^*)$ because relaxation and instanton are the time-reversed of each other [131].

Moreover, the above equality holds even at first order in $f \to 0$, that is, around detailed balance [26]. Indeed, setting $I_\infty = \mathcal{U} + g$ (with $O(g) = O(f)$) in (8.20) and using the decomposition of the drift field we can obtain, neglecting terms $O(f)$ and higher

$$-\nabla \mathcal{U} \cdot D \cdot \nabla g = f \cdot \nabla \mathcal{U}. \tag{8.31}$$

Here, the left-hand side is the time derivative of the correction $g$ along a relaxation trajectory with $f = 0$ and the right-hand side is minus the entropy flow rate along the same trajectory. Therefore, integrating over an infinite time from $x(0) = x_\nu$ to $x(\infty) = x_i^*$ we see that the correction $g$ to the energy is given by $\sigma_{\nu \to i} = -\sigma_{i \to \nu}$, and (8.30) holds as an equality in view of (8.26).

## 8.4 Jump processes in occupation-number space: macroscopic limit

The large deviations formalism can be applied to processes described by master equations that admit a deterministic limit. For example, in a state space coordinatized by the occupation number $n$ (individuals of a species in population dynamics, molecule number in chemical reactions, charge number in electronic circuits, photon number per frequency

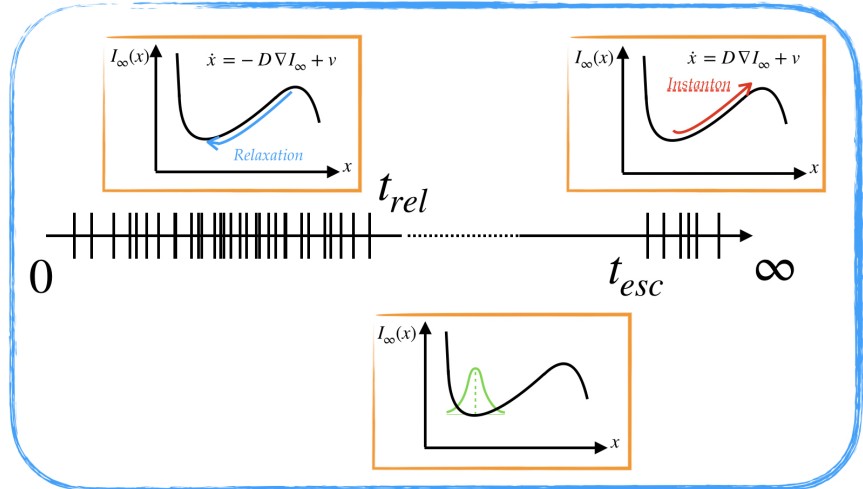

Figure 8.5: Sketch of the typical dynamics on the three different timescales: short-time relaxation towards a local minimum of the stationary rate function, small (Gaussian) fluctuations around a stable fixed point, long-time transition to another attractor with rate $k_\nu$.

in scattering) there might exists a scale parameter (typically an inverse system size) $\epsilon$ such that $n\epsilon = x$ defines a finite continuous variable as $\epsilon \to 0$. If the transition matrix $R$ scales like $1/\epsilon$, we incur in a weak-noise limit similar to the one presented before. The Schlögl model given before is a particular example of this class of systems.

Here, we highlight only the main differences with respect to overdamped diffusion. From the WKB ansatz inserted into the Master equation we arrive at (8.15) with the Hamiltonian that is now the cumulant generating function of Poisson noise [33]

$$H(x, \pi) = \sum_\rho r_\rho(x)(e^{\Delta_\rho \cdot \pi} - 1) \tag{8.32}$$

where $r_\rho(x)$ is the macroscopic (i.e. scaled by $1/\epsilon$) transition rate from state $x$ and $\Delta_\rho$ is the jump size of transition $\rho$.

- The resulting Hamiltonian equations are [132]

$$
\begin{aligned}
\dot{x} &= \sum_\rho \Delta_\rho r_\rho(x) e^{\Delta_\rho \cdot \pi} \\
\dot{\pi} &= -\sum_\rho \nabla r_\rho(x) \left( e^{\Delta_\rho \cdot \pi} - 1 \right),
\end{aligned}
\tag{8.33}
$$

Only close to a stable fixed point (or time depedent-attractor) these equations can be linearized in $\pi$ and $x - x_i^*$ to obtain (the linear version of) (8.17). Clearly, this approximation is valid only for times $t \ll t_{esc} \asymp k_\nu^{-1}$ and correspond to van Kampen's system size expansion [133].

- The deterministic dynamics has a *nonlinear* gradient part plus the 'orthogonal' probability velocity

$$\dot{x} = -A(x) \cdot \nabla I_\infty + v \tag{8.34}$$

---

[33]Recall that a Markov jump process is characterized by a (conditional) exponential waiting time distribution, or equivalently, a (conditional) Poisson distribution of number of jumps per site.

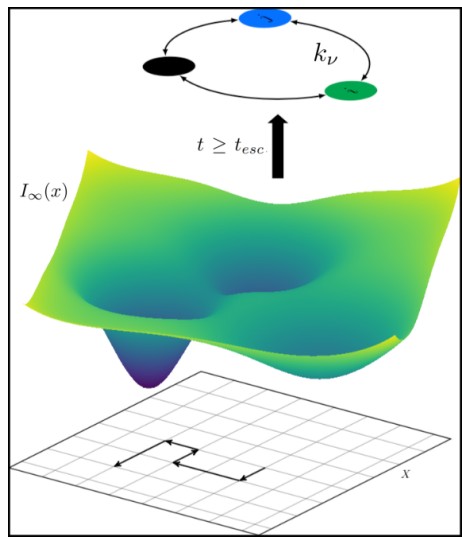

Figure 8.6: On very long times the dynamics can be coarse-grained into a discrete jump process from one attractor to the other

with $A(x) = \int_0^1 d\theta(1-\theta)\partial_\pi^2 H(c,\pi)|_{\pi=\theta\nabla I_\infty}$ a nonnegative diffusion matrix depending on the rate function itself.

- The instanton is related to the relaxation by considering the adjoint dynamics which acts on single transition rates as $r_\rho(x) \mapsto r_{-\rho}(x)e^{-\Delta_\rho \cdot \nabla I_\infty(x)}$. This in general does not boil down to flipping the sign of the rate function and leaving $v$ untouched.

- The bounds (8.30) remain valid.

## 8.5   Markov jump process on the space of attractors

On very long times of the order $t \geq t_{esc}$ we can describe the dynamics as a jump process from one attractor to the other with transition rates $k_\nu$ and $p^{(i)}(t)$ the probability of being within the attractor $i$ (Fig. 8.6). Here $\nu$ labels all the transitions in both directions, counted as positive and negative, respectively. Because of the previous result (8.27), local detailed balance on the rates $k_\nu$ holds trivially when $f = 0$ but also when $f$ is small, in the form [26]

$$\frac{k_\nu}{k_{-\nu}} = e^{-[\mathcal{U}(o(-\nu))-\mathcal{U}(o(\nu))+\sigma_{o(\nu)\to o(-\nu)}]}. \tag{8.35}$$

where $o(\pm\nu)$ is the origin of the transition trough $\pm\nu$ and $\sigma_{x_i^*\to x_j^*}$ is the entropy flow the most probable path between $x_i^*$ and $x_j^*$. As mentioned above, solutions of (8.20) only give the rate function within each basin of attraction $I_\infty^{(i)}(x)$, which is determined up to a constant $\alpha_i$. Intuitively, this is because to obtain $I^{(i)}$ we first took the limit $\epsilon \to 0$, before $t \to \infty$, thus the system is stuck within a basin of attraction. Such constants have to be fixed relative to each other. They give the relative weight of each attractor, i.e. $p_i$, obtained by solving the master equation over the space of attractors [124, 128]. The rate function is finally obtained by taking at each $x$ the maximum $I_\infty^{(i)}(x) + \alpha_i$ over $i$.

### 8.5.1  Example: the Schlögl model (cont.)

The system is bistable, so we can write the long time Markov jump dynamics in terms of a 2X2 stochastic matrix

$$\begin{pmatrix} -k_1 & k_{-1} \\ k_1 & -k_{-1} \end{pmatrix} \tag{8.36}$$

whose eigenvalues are 0 and $-(k_1 + k_{-1})$. One can check [80] that they correspond to the first two eigenvalues of the generator of the Schlögl model.

## 8.6  Conclusions

We conclude by listing the differences that emerged above between detailed balance and nondetailed balance dynamics.

- **Geometric vs dynamic view:** in detailed balance systems, it is possible to understand metastability in terms of the underlying free energy landscape. This geometrical perspective is lost in nondetailed balance systems where metastability may be entirely due to nonconservative forces. In this case its understanding is based on the dynamics: one looks at the behavior of the smallest system's eigenvalues.

- **Generator $R$:** in the detailed balance case, the generator of the stochastic process is symmetrizable. This means that all of its eigenvalues are real. In the nondetailed balance case, this is no longer true and one may have complex eigenvalues leading to relaxation with oscillations, limit cycles or more complex time-dependent attractors.

- **Rate function**: $I_\infty(x) = \mathcal{U}(x)$ for detailed balance dynamics. When nonconservative forces are introduced, the relation is modified. Close to detailed balance, the correction to $\mathcal{U}$ is given by the entropy dissipated by the nonconservative force on the typical trajectory.

- **Instanton**: the instanton is the time-reverse of the relaxation for detailed balance systems and their associated entropy productions are equal and opposite $\sigma_{\nu \to i} = -\sigma_{i \to \nu}$. When nonconservative forces are present, the instanton and the relaxation may follow different trajectories, and an inequality holds for the entropy productions $-\sigma_{\nu \to i} \leq \sigma_{i \to \nu}$ (at first order in $f \to 0$ it becomes the above equality).

- **Coarse-grained dynamics**: on longer time-scales, $t \geq t_{esc}$, the dynamics resembles a jump process among mesostates. For detailed balance systems a local detailed balance condition emerges on the mesoscopic rates. The dissipation is entirely due to jumps among attractors, whereas in the nondetailed balance case, there is a contribution to the dissipation coming from the microscopic currents inside each basin of attraction.

# 9   Martingale Approach for First-Passage Problems

**Izaak Neri and Adarsh Raghu.**[34]   *We explore nonequilibrium thermodynamics for first-passage processes with martingale theory. We use the gambler's ruin problem as a unifying theme.*

## 9.1   Introduction

First-passage problems describe processes that have a finite, random termination time. Examples are escape problems — such as, a particle that escapes from a bounded domain or a system that escapes from a metastable state (i.e., Kramers' escape problem) — and diffusion controlled reactions. For processes in equilibrium, the time it takes for a particle to escape a bounded domain decreases exponentially as a function of the energy barrier that separates it from the outside world, as described by the van't Hoff-Arrhenius law [2, 134, 135].

The focus of these notes is on first-passage problems in nonequilibrium scenarios, where the underlying stochastic process does not adhere to detailed balance; see Fig. 9.1 for illustrations of two examples of escape problems in nonequilibrium, statistical physics. In nonequilibrium scenarios, we are often interested in the escape problem of a macroscopic current out of a bounded interval. An instance of such a scenario is observed in molecular motors, like kinesin-1, which convert free energy into mechanical work while attached to a biofilament. Due to the polarity of biofilaments, the exerted force on the filament is biased towards one of its ends. Notably, as the filament has a finite length, the motor's work required to transport a cargo to one of the filament's ends is evaluated at a random time.

As thermodynamicists, our primary goal is to establish thermodynamic connections between quantities that characterize first-passage processes, such as splitting probabilities and mean first-passage times, and thermodynamic quantities, such as the entropy production rate. Our focus is on obtaining universal relations that are applicable to a wide range of stochastic processes far from thermal equilibrium, similar to the van't Hoff-Arrhenius law that holds for equilibrium processes [135].

To analyze stochastic processes that are far from thermal equilibrium, we employ stochastic thermodynamics [35, 136, 137]. However, instead of analyzing stochastic thermodynamics over ensembles of trajectories that terminate at a fixed time, we investigate ensembles of trajectories that terminate at a trajectory-dependent termination time (i.e., a random termination time). Recent works [138–140] have demonstrated that *martingale theory* significantly aids in studying thermodynamics at random termination times leading to the discovery of several universal relations in nonequilibrium thermodynamics; see also the earlier work Ref. [141] on the use of martingale in stochastic thermodynamics.

In particular, we review two results in this context. Firstly, we review the second law of thermodynamics at stopping times [140, 142, 143] that extends the classical second law of thermodynamics, which is valid for processes terminating at a fixed time, to processes that terminate at a random time. Secondly, we review a recently discovered relation that connects the splitting probability of a current with its mean first-passage time and the rate of dissipation in a nonequilibrium process [144–147]. This relation can be interpreted as a trade-off between uncertainty (expressed as the ruin probability $p_-$), speed (quantified by the mean first passage time $\langle \hat{T} \rangle$), and thermodynamic cost (quantified by the rate of dissipation $\sigma$). We explain these results by making an analogy with the gambler's

---

[34]IN was the lecturer and AR was the angel. Both authors wrote the chapter.

ruin problem, as studied originally by Blaise Pascal, and then extending this problem to thermodynamic setups.

The organization of the Lecture Notes is as follows: Section 9.2 provides a review of the gambler's ruin problem, which serves as a basic illustration of a first-passage problem with two boundaries. In this section, we solve the gambler's ruin problem through the classical approach that relies on difference equations. Section 9.3 revises some essentials from martingale theory, and in Sec. 9.4, we utilise martingales to solve the gambler's ruin problem. In Sec. 9.5, we introduce various thermodynamic versions of the gambler's ruin problem and explain how to utilise martingale theory to solve some of them. These lecture notes conclude with a discussion of some of the questions that arose during the lectures at the workshop.

## 9.2 Gambler's ruin problem

The gambler's ruin problem was developed in 1656 by Blaise Pascal [148] and is often used in textbooks to introduce the first-passage problem and the mathematical tools that come with it, see e.g. Chapter XIV of Feller's introductory textbook book on probability theory [149]. Furthermore, from a physics perspective it can be seen as an elementary example of a first-passage problem in a nonequilibrium process.

First, we briefly discuss the history of this problem, which is interesting in itself, as it is part of the Scientific Revolution that took place in the 17th century. Second, we frame the problem and provide its standard solution using difference equations.

### 9.2.1 A brief history of the problem

Although probability theory is commonly used in our everyday reasoning, its mathematical foundations were established relatively late. Nevertheless, during the 17th century's Scientific Revolution the development of probability theory was bound to happen due to various circumstances.

In this period there was a surge of interest in gambling amongst the wealthy and noble classes in Europe leading to the establishment of gambling houses, such as, the Ridotto in Venice (1638). Several puzzles in probability theory had appeared in the literature and were discussed by prominent thinkers of the time, notably, the problem of points that was known from Luca Paccioli's work "Summa de Arithmetica, Geometria, Proportioni et Proportionalità" (Summary of Arithmetic, Geometry, Proportion and Proportionality), published in 1494. Blaise Pascal got interested in probability problems through his interactions with Antoine Gombaud, who was a French writer (also known as Chevalier de Méré), but also an amateur mathematician and a gambler. Against this backdrop, Pascal developed the gambler's ruin problem, two years after his famous correspondence with Pierre de Fermat on the problem of points in 1654 [150].

The first in print version of the gambler's ruin problem appeared soon after in Cristiaan Huygens' 1660 paper on probability entitled "*Van Rekeningh in Spelen van Geluck*" (On calculations for games on chance) [151], and a year later it was translated by his mathematics teacher Schooten into Latin as "*De Ratiociniis in Ludo Aleae*", and was published in Schooten's "*Exercitationum Mathematicarum*"; note that Huygens visited Paris for the first time in 1655 — a few months after he identified the rings of Saturn and discovered Saturn's moon Titan, — where he met with French scholars and became aware of their interest in probability theory. At the end of Huygens' paper he presents five problems to the reader, the last of which became better known as the gambler's ruin problem. Huygens' formulation goes as follows:

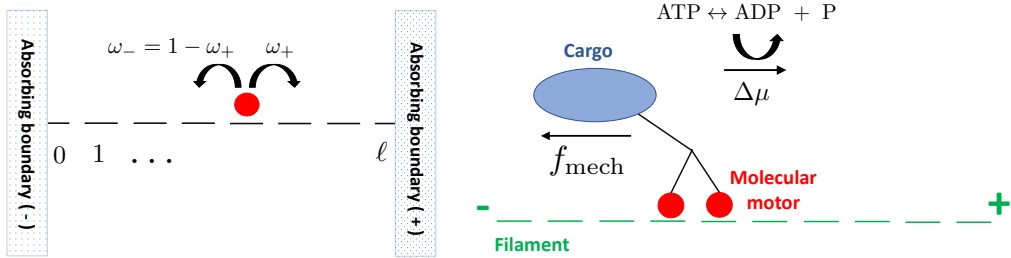

Figure 9.1: Examples of (nonequilibrium) processes with a finite, random termination time. *Gambler's ruin problem* [Left Panel]: a particle represents the wealth of gambler A and moves on a finite segment until it reaches either the "−"-end (indicating that gambler A has lost their stake) or the "+"-end (indicating that gambler B has lost their stake). The ratio of the odds of the two gamblers is denoted by $\omega_+/\omega_-$. *Molecular motor bound to a filament* [Right Panel]: A two-headed molecular motor, such as kinesin-1, attaches to a filament, after which it converts the free energy $\Delta\mu$, obtained from the hydrolysis of adenosine triphosphate (ATP) into adenosine diphosphate (ADP) and an inorganic phosphate (P), into mechanical work ($f_{\text{mech}}$), yielding a stochastic motion biased towards the filament's plus end (+). When the motor reaches one of its end points, then it detaches from the filament, and its enzymatic activity stops.

"*A and B have both 12 coins and play a game of chance with three dice under these conditions: if a throw shows 11 pips, then A must give a coin to B; however, if a throw shows 14 pips, then B must give a coin to A; the game is won by the player that has first all the coins.*". Huygens also reveals the solution:"*The odds of A with respect to B are 244140625 to 282429536481.*"

Huygens' publication generated significant interest in probability theory, notably from Jacob Bernoulli. In 1713, eight years after Bernoulli's death, his famous work "*Ars Conjectandi*" was published. In this book, he reformulates the gamblers' ruin problem in a way that closely ressembles its modern version, viz., gambler $A$ has $m$ coins, gambler $B$ has $n$ coins, and the odds of $A$ winning to $B$ or $a$ to $b$. Jacob Bernoulli shows that the chance for $A$ to win the game, and thus $B$ to lose the game, is [152]

$$p_+^{\text{A}} = p_-^{\text{B}} = \frac{a^n(a^m - b^m)}{a^{m+n} - b^{m+n}}. \tag{9.1}$$

Equation (9.1) implies that for $b > a$ and large values of $m$, the ruin probability $p_-^{\text{B}}$ decays exponentially as a function of $n$, i.e.,

$$p_-^{\text{B}} = \exp\left(n \ln \frac{a}{b}\right)\left(1 + O\left(\exp\left(m \ln \frac{a}{b}\right)\right)\right) \tag{9.2}$$

where $O$ represents the big $O$ notation. Formulas similar to Eqs. (9.1) and (9.2) reoccur in the present Lecture Notes.

### 9.2.2 Problem statement and solution

For either gambler A or B, the gambler's ruin problem can be seen as a first-passage problem in which a particle hops on a segment until it reaches one of two endpoints, after which the process terminates, as depicted in the left Panel of Fig. 9.1. If the position of

the random walker, denoted by $\hat{X}_n \in [\ell] = \{0, 1, \ldots, \ell\}$ with $n \in \mathbb{N}$ a discrete time index, represents the stakes of gambler A, then the absorbing boundary at the origin corresponds to the event where gambler A loses all of its stakes, while the other endpoint represents the event where gambler B loses all their stakes in the game. Note that the main quantities of interest, such as the profit of gambler A, are ensemble averages of $\hat{X}$ evaluated at the random termination time $\hat{T}$.

Let us formalise the above. Consider a Markov process $\hat{X}_n \in [\ell]$ governed by the transition matrix

$$\mathbb{P}(\hat{X}_n = x | \hat{X}_{n-1} = y) = \omega_+ \delta_{x,y-1} + \omega_- \delta_{x,y+1} \tag{9.3}$$

for $y \in [\ell] \setminus \{0, \ell\}$, with

$$\omega_+ = 1 - \omega_-, \tag{9.4}$$

with absorbing boundary conditions

$$p(\hat{X}_n = x | \hat{X}_{n-1} = 0) = \delta_{x,0} \quad \text{and} \quad p(\hat{X}_n = x | \hat{X}_{n-1} = \ell) = \delta_{x,\ell}, \tag{9.5}$$

and with initial state $\hat{X}_0 = x_0 \in [\ell]$. When the particle reaches the boundaries $x = 0$ or $x = \ell$, then the process terminates, in the sense that the particle stops moving, and hence we have a process of finite, random termination time given by

$$\hat{T} = \min\left\{n \in \mathbb{N} \cup \{0\} : \hat{X}_n = \ell \text{ or } \hat{X}_n = 0\right\}. \tag{9.6}$$

Hence, each realisation $\hat{X}_0^{\hat{T}} = \left\{\hat{X}_0, \hat{X}_1, \ldots, \hat{X}_{\hat{T}}\right\}$ has a different duration $\hat{T}$ and $\hat{X}_{\hat{T}} \in \{0, \ell\}$.

Classically, the gambler's ruin problem is solved with difference equations [149]. In what follows, we illustrate the solution for $\omega_+ \neq 1/2$, and we refer to Ref. [149] for more details, including the derivation for $\omega_+ = 1/2$. Specifically, we determine the main observables of interest, viz., the probability of ruin $p_x^-$, the average profit $\langle \hat{X}_{\hat{T}} \rangle$, the average termination time $\langle \hat{T} \rangle$, and the generating functions of the termination time $\hat{T}$ at the negative boundary, $g_x^-$, and at the positive boundary, $g_x^+$.

**Probability of ruin.** The probability of ruin,

$$p_x^- := \mathbb{P}\left(\hat{X}_{\hat{T}} = 0 | \hat{X}_0 = x\right) \tag{9.7}$$

solves the difference equation

$$p_x^- = \omega_+ p_{x+1}^- + \omega_- p_{x-1}^- \tag{9.8}$$

with boundary conditions $p_\ell^- = 0$ and $p_0^- = 1$. The solution to these equations are unique, as follows from the Maximum Principle for harmonic functions, see Ref. [153] or Theorem 2.1 in [154].

First we consider the Eq. (9.8) without boundary conditions. In this case, $p_x^-$ should be seen as a function of $x$ rather than the splitting probability in the gambler's ruin problem. For functions of the form $p_x^- = \alpha^x$, where $\alpha \in \mathbb{R}^+$, we obtain from Eq. (9.8) that $\alpha = \omega_+ \alpha^2 + \omega_-$. For $\omega_+ \neq 1/2$, this quadratic equation contains two solutions, namely, $\alpha = 1$ and $\alpha = \omega_-/\omega_+$. Using the linearity of the difference Eq. (9.8), we obtain

$$p_x^- = c_1 + c_2 \left(\frac{\omega_-}{\omega_+}\right)^x, \tag{9.9}$$

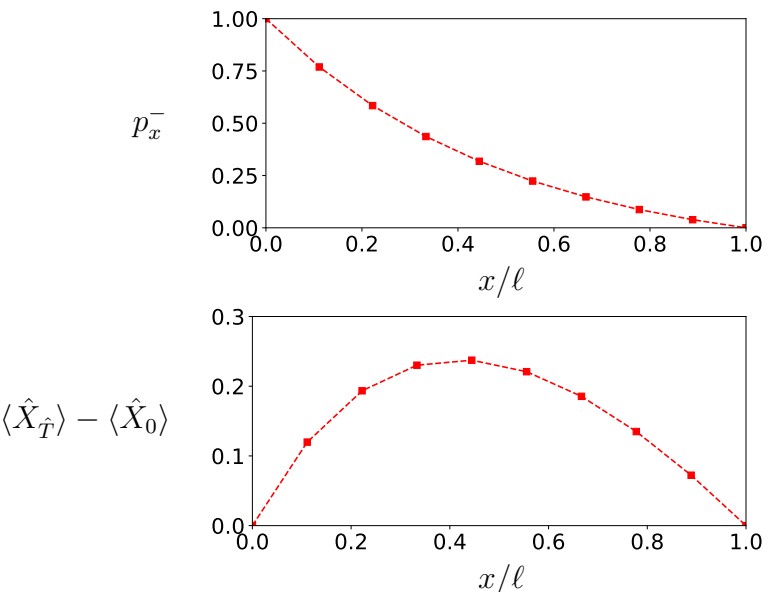

Figure 9.2: Top: Ruin probability $p_x^-$ as a function of $x/\ell$ for parameters $\ell = 10$, $\omega_- = 0.55$. Bottom: The expected profit $\langle \hat{X}_{\hat{T}} | \hat{X}_0 = x \rangle$ in the gambler's ruin problem for the same parameters $\ell = 10$, $\omega_- = 0.55$.

where $c_1$ and $c_2$ are arbitrary constants fixed by the boundary conditions. Second, we impose the boundary conditions on $p_x^-$ to obtain the splitting probability

$$p_x^- = \frac{(\omega_-/\omega_+)^\ell - (\omega_-/\omega_+)^x}{(\omega_-/\omega_+)^\ell - 1} \tag{9.10}$$

for the gambler's ruin problem. We have plotted $p_x^-$ as a function of $x/\ell$ in the top Panel of Fig. 9.2. Notice that Eq. (9.10) is equivalent to Bernoulli's formula Eq. (9.1) when identifying $n = x$, $m = \ell - x$, $b = \omega_+$, and $a = \omega_-$.

**Profit at the termination time.**    The average profit $\langle \hat{X}_{\hat{T}} \rangle - \langle \hat{X}_0 \rangle$ at the termination time is given by

$$\langle \hat{X}_{\hat{T}} \rangle - \langle \hat{X}_0 \rangle = p_x^+(\ell - x) - x p_x^- = \frac{\ell - x - (\omega_-/\omega_+)^x(\ell - x(\omega_-/\omega_+)^{\ell-x})}{1 - (\omega_-/\omega_+)^\ell}. \tag{9.11}$$

In the bottom panel of Fig. 9.2, we plot $\langle \hat{X}_{\hat{T}} \rangle$ as a function of $x/\ell$. Note that the profit is nonnegative for $\omega_+ \geq \omega_-$, a fact that we demonstrate in Sec. 9.4 with martingale theory.

**Mean first-passage time.**    The mean duration of the process,

$$t_x := \langle \hat{T} | \hat{X}_0 = x \rangle, \tag{9.12}$$

solves

$$t_x = \omega_+ t_{x+1} + \omega_- t_{x-1} + 1 \tag{9.13}$$

with boundary conditions $t_0 = 0$ and $t_\ell = 0$. Solving these difference equations for $\omega_+ \neq 1/2$ towards $t_x$, we obtain [149]

$$t_x = -\frac{x}{\omega_+ - \omega_-} + \frac{\ell}{\omega_+ - \omega_-} \frac{1 - (\omega_-/\omega_+)^x}{1 - (\omega_-/\omega_+)^\ell}. \tag{9.14}$$

**Generating function.** The fluctuations of the termination time at the negative boundary are determined by the generating function

$$g_x^-(s) := \sum_{n=0}^{\infty} \mathbb{P}(\hat{T} = n, \hat{X}_{\hat{T}} = 0 | \hat{X}_0 = x) s^n, \tag{9.15}$$

which solves the difference equation

$$g_x^-(s) = \omega_+ s g_{x+1}^-(s) + \omega_- s g_{x-1}^-(s) \tag{9.16}$$

with boundary conditions $g_0^-(s) = 1$ and $g_\ell^-(s) = 0$. Solving towards $g_x(s)$, we obtain for $\omega_+ \neq 1/2$ [149]

$$g_x^-(s) = \left(\frac{\omega_-}{\omega_+}\right)^x \frac{\lambda_+^{\ell-x}(s) - \lambda_-^{\ell-x}(s)}{\lambda_+^{\ell}(s) - \lambda_-^{\ell}(s)} \tag{9.17}$$

where

$$\lambda_\pm(s) := \frac{1 \pm \sqrt{1 - 4\omega_- \omega_+ s^2}}{2\omega_+ s}. \tag{9.18}$$

Analogously, for

$$g_x^+(s) := \sum_{n=0}^{\infty} \mathbb{P}(\hat{T} = n, \hat{X}_{\hat{T}} = \ell | \hat{X}_0 = x) s^n, \tag{9.19}$$

we find

$$g_x^+(s) = \frac{\lambda_+^x(s) - \lambda_-^x(s)}{\lambda_+^{\ell}(s) - \lambda_-^{\ell}(s)}. \tag{9.20}$$

Note that setting $s = 0$, we recover $g_x^+(0) = p_x^+$ and $g_x^-(0) = p_x^-$.

## 9.3 Martingales: Definition, Represenations, and Properties

As noted by Jean-André Ville in his PhD thesis [155], martingales provide an alternative framework to solve the gambler's ruin problem. We provide here a brief overview of martingale theory, and in the next section we discuss how to use martingales to solve the gambler's ruin problem.

### 9.3.1 Definition

Let $\hat{X}_m \in \mathscr{X}$ be a stochastic process taking values in a finite set $\mathscr{X}$, and let $m \in \mathbb{N}$ be a discrete time index. We denote a sequence of (random) variables by

$$\hat{X}_0^n := (\hat{X}_0, \hat{X}_1, \ldots, \hat{X}_n) \tag{9.21}$$

Let $\mathbb{P}$ be the probability measure describing the statistics of $X$, and we denote its path probability by

$$p(x_0^n) := \mathbb{P}(\hat{X}_0^n = x_0^n). \tag{9.22}$$

For simplicity, we assume in these lecture notes that both the phase space $\mathscr{X}$ and time $n$ are discrete, but all the concepts can be extended into a continuous setting, see Ref. [143].

A martingale $\hat{M}_n\left(\hat{X}_0^n\right)$, relative to $\hat{X}$ and $\mathbb{P}$, is a real-valued process defined on $\hat{X}_0^n$ for which it holds that [156]

$$\langle |\hat{M}_m| \rangle < \infty \tag{9.23}$$

and

$$\langle \hat{M}_n | \hat{X}_0^m \rangle = \hat{M}_m \tag{9.24}$$

for all $m \in [n] := \{0, 1, \ldots, n\}$. Here the averages $\langle \cdot \rangle$ are taken with respect to the probability measure $\mathbb{P}$. Using the definition of conditional probabilities, we get

$$\langle \hat{M}_n | \hat{X}_0^m \rangle = \sum_{x_{m+1}^n \in \mathscr{X}^{m-n}} \frac{p(x_0^n)}{p(x_0^m)} M_n(x_0^n). \tag{9.25}$$

We require the first condition, Eq. (9.23), as it guarantees that the conditional expectation $\langle \hat{M}_n | \hat{X}_0^m \rangle$ exists, see e.g., Theorem 10.1.1 in [157]. The second condition (9.24) is the defining property of the martingale. If we replace the condition (9.24) by the inequality

$$\langle \hat{M}_n | \hat{X}_0^m \rangle \geq \hat{M}_m \tag{9.26}$$

then we speak of a submartingale process.

We give a brief historical overview on martingales. The martingale condition Eq. (9.24), appears in Paul Lévy's book [158] published in 1937 on the sums of random variables as a technical condition required to extend the classical central limit theorem to sums of dependent random variables. However, Paul Lévy does not define martingales and does not study their properties. It is, around the same time, Jean-André Ville who introduces stochastic proceses satisfying the condition Eq. (9.24) in his PhD thesis [155], and he coined them martingales. In chapter V of his thesis, Ville proves several properties of martingales and uses them to study, amongst others, the gambler's ruin problem. Developments of martingale theory accelerated after the second world war, mainly after the publication of Joseph Doob's book on stochastic processes [159]; Doob, aware of Jean-Ville's work, borrows its name. Several important results in martingale theory are named after Doob, such as, Doob's regularity theorem [160], which plays an important role in defining stochastic processes on spaces of right-continuous trajectories, the martingale convergence theorems [159], and Doob's stopping theorems [159], on which we will rely in this short review.

Nowadays, martingales are models for stocks prices under the efficient market hypothesis [161], and in physics martingales describe important functionals in nonequilibrium, thermodynamics, such as, the exponentiated negative entropy production [138, 140–142] and the exponential of the housekeeping heat [162].

### 9.3.2 Martingale representations

We discuss two important examples of processes that are martinagles.

- *Nonnegative martingales:* Let $\hat{M}_n$ be a nonnegative martingale relative to $\hat{X}$ and $\mathbb{P}$, and additionally we assume that $\langle \hat{M}_0 \rangle = 1$. Then $\hat{M}_n$ is the ratio of two path probabilities. Indeed, if we define

$$\mathbb{Q}(\hat{X}_0^n = x_0^n) := \hat{M}_n(x_0^n)\mathbb{P}(\hat{X}_0^n = x_0^n), \tag{9.27}$$

then $\mathbb{Q}(\hat{X}_0^n = x_0^n) \geq 0$ and

$$\sum_{x_0^n \in \mathscr{X}^n} \mathbb{Q}(\hat{X}_0^n = x_0^n) = \langle \hat{M}_n \rangle = 1. \tag{9.28}$$

The converse is also true. If $q(x_0^n) := \mathbb{Q}(\hat{X}_0^n = x_0^n)$ is a path probability that is absolutely continuous with respect to $p(x_0^n) := \mathbb{P}(\hat{X}_0^n = x_0^n)$, i.e., $p(x_0^n) = 0 \Rightarrow q(x_0^n) = 0$, then

$$\hat{M}_n := q(\hat{X}_0^n)/p(\hat{X}_0^n) \tag{9.29}$$

exists and is a martingale. Indeed, $\langle|\hat{M}_n|\rangle = \langle\hat{M}_n\rangle = 1$, and for all $0 \leq m \leq n$ it holds that

$$
\begin{aligned}
&\langle\hat{M}_n|\hat{X}_0, \hat{X}_1, \ldots, \hat{X}_m\rangle \\
&= \sum_{x_{m+1}^n \in \mathscr{X}^{m-n}} \mathbb{P}(\hat{X}_{m+1}^n = x_{m+1}^n|\hat{X}_0^m) \frac{q\left(\hat{X}_0^m, x_{m+1}^n\right)}{p(\hat{X}_0^m, x_{m+1}^n)} \\
&= \sum_{x_{m+1}^n \in \mathscr{X}^{m-n}} \frac{p(\hat{X}_0^m, x_{m+1}^m)}{p(X_0^m)} \frac{q\left(\hat{X}_0^m, x_{m+1}^n\right)}{p(\hat{X}_0^m, x_{m+1}^n)} \\
&= \frac{q(\hat{X}_0^m)}{p(\hat{X}_0^m)} = M_m,
\end{aligned}
\tag{9.30}
$$

where we have used $p(\hat{X}_0^m, x_{m+1}^n)$ to denote the path probability of the sequence $(\hat{X}_0, \hat{X}_1, ..., \hat{X}_m, x_{m+1}, ..., x_n)$. Notice that in the last line we have used the marginalisation properties

$$
\sum_{x_{m+1}^n \in \mathscr{X}^{m-n}} q(\hat{X}_0^m, x_{m+1}^m) = q(\hat{X}_0^m) \tag{9.31}
$$

of path probabilities.

- *Harmonic functions:*  Let $\hat{X} \in \mathscr{X}$ be a Markov process with transition matrix

$$
\mathbf{R}_{xy} := \mathbb{P}\left(\hat{X}_{n+1} = x|\hat{X}_n = y\right), \tag{9.32}
$$

and let $\mathbf{h}(x)$ be a left eigenvector of $\mathbf{R}$ associated with the unit eigenvalue, i.e.,

$$
\mathbf{h}(y) = \sum_{x \in \mathscr{X}} \mathbf{h}(x)\mathbf{R}_{xy}. \tag{9.33}
$$

It then holds that

$$
\hat{h}(\hat{X}_n) := \mathbf{h}(\hat{X}_n) \tag{9.34}
$$

is martingale. Indeed,

$$
\langle\hat{h}(\hat{X}_n)|\hat{X}_{n-1}\rangle = \sum_{x \in \mathscr{X}} \mathbf{h}(x)\mathbf{R}_{x\hat{X}_{n-1}} = \hat{h}(\hat{X}_{n-1}). \tag{9.35}
$$

Analogously,

$$
\langle\hat{h}(\hat{X}_n)|\hat{X}_{n-2}\rangle = \sum_{x_1, x_2 \in \mathscr{X}} \mathbf{h}(x_2)\mathbf{R}_{x_2 x_1}\mathbf{R}_{x_1, \hat{X}_{n-1}} = \hat{h}(\hat{X}_{n-2}) \tag{9.36}
$$

and so forth.

The functions $h$ are also called *harmonic functions*. Note that these should not be confused with the right eigenvectors

$$
\mathbf{p}_\infty(x) = \sum_{y \in \mathscr{X}} \mathbf{R}_{xy}\mathbf{p}_\infty(y), \tag{9.37}
$$

which represent stationary distributions. For ergodic processes, there exists exactly one stationary distribution, and correspondingly, also exactly one harmonic function, viz., $\mathbf{h}(x) = c$ for all $x \in \mathscr{X}$, where $c$ is an arbitrary constant independent of

$x$. Hence, to have a nontrivial harmonic functions we need to make the process nonergodic.

A possibility to make the process nonergodic is to introduce absorbing states. Specifically, if we set

$$\mathbf{h}(x) = 1 \quad \text{if} \quad x \in \mathscr{X}_+ \tag{9.38}$$

and

$$\mathbf{h}(x) = 0 \quad \text{if} \quad x \in \mathscr{X}_-, \tag{9.39}$$

then $\mathbf{h}(x)$ is the probability to reach $\mathscr{X}_+$ before reaching $\mathscr{X}_-$ when the initial state $X_0 = x$. Hence, splitting probabilities are examples of harmonic functions, and the difference Eq. (9.8) in Sec. 9.2.2 is equivalent to the Eq. (9.35).

### 9.3.3 Doob's optional stopping theorem

We end the first part of this lectures with Doob's optional stopping theorem, which is the main theorem we will be using.

Doob's optional stopping theorem states the following, see Theorem 4.1.1. in Ref. [163].

**Theorem 1** *Let $\hat{M}_n$ be a martingale with respect to $\hat{X}$ and $P$, and let $\hat{T}$ be a stopping time with respect to $\hat{X}$ and $P$. If one of the the following two conditions holds, viz.,*

- *$\hat{T} \leq n$ for some $n$;*

- *or, $\hat{T} < \infty$ and there exists some $c \in \mathbb{R}^+$ such that $|\hat{M}_n| \leq c$ for all $n \leq \hat{T}$;*

*then*

$$\langle \hat{M}_{\hat{T}} \rangle = \langle \hat{M}_0 \rangle. \tag{9.40}$$

If the abovementioned conditions do not apply, then Doob's optional stopping theorem is in general not valid. For example, for an unbiased random walk $X_n$ with $\omega_+ = \omega_- = 1/2$, which is a martingale, and for $\hat{T} = \min \left\{ n \geq 0 : \hat{X}_n = x_+ \right\}$, it holds that $\langle \hat{X}_{\hat{T}} \rangle = x_+ \neq \langle \hat{X}_0 \rangle$, and thus Doob's optional stopping theorem does not apply. One way to bound $\hat{M}_n$ for $n \leq \hat{T}$ is by defining $\hat{T}$ as the escape time from a bounded set.

For submartingales $M_n$, it holds analogously that $\langle \hat{M}_{\hat{T}} \rangle \geq \langle \hat{M}_0 \rangle$.

## 9.4 Martingale solution to the gambler's ruin problem

We now solve the gambler's ruin problem with martingales.

### 9.4.1 The average profit

To illustrate the use of martingale theory, let us first consider the average profit $\langle \hat{X}_{\hat{T}} \rangle$, which is nonnegative as we have seen from explicitly deriving the formula Eq. (9.11). Using martingale theory, this result is seen as a direct consequence of the submartingale property of $\hat{X}_n$. Indeed, since for $\omega_+ \geq \omega_-$,

$$\langle \hat{X}_n | \hat{X}_0^m \rangle \geq \hat{X}_m, \tag{9.41}$$

the process $\hat{X}_n$ is a submartingale, and hence Doob's optional stopping theorem for submartingales applies, yielding

$$\langle \hat{X}_{\hat{T}} \rangle \geq \langle \hat{X}_0 \rangle. \tag{9.42}$$

Thus, the profit $\hat{X}_{\hat{T}} - \hat{X}_0$ is on average nonnegative. Notably, this result holds for any stopping strategy $\hat{T}$ as long as $\hat{X}_n$ is bounded for all times $n < \hat{T}$ (i.e., the total amount of money is finite). Conversely, when $\omega_- \leq \omega_+$, then

$$\langle \hat{X}_{\hat{T}} \rangle \leq \langle \hat{X}_0 \rangle, \tag{9.43}$$

and the gambler is losing money, no matter which betting strategy, determined by $\hat{T}$, is used.

### 9.4.2 Family of martingale processes

To obtain the splitting probabilities and the generating function of $\hat{T}$, we introduce a family of martingales.

First, we define the process $\hat{X}_n$ as in Sec. 9.2.2 with the transition matrix Eq. (9.3) but without absorbing boundary conditions, i.e., $\hat{X} \in \mathbb{Z}$. The process

$$\hat{Z}_n := \exp\left( z\hat{X}_n + nf(z) \right), \tag{9.44}$$

with

$$f(z) := -\ln[\omega_+ \exp(z) + \omega_- \exp(-z)] \tag{9.45}$$

is then a martingale with respect to $\hat{X}$ and $\mathbb{P}$ for all values $z \in \mathbb{R}$.

Indeed, condition (9.24) follows from

$$
\begin{aligned}
&\langle \hat{Z}_n | \hat{X}_0, \hat{X}_1, \ldots, \hat{X}_m \rangle \\
&= \sum_{x \in \mathbb{Z}} \exp(zx + nf(z)) \mathbb{P}(\hat{X}_n = x | \hat{X}_0^m) \\
&= \sum_{n_+=0}^{n-m} \exp\left( z(\hat{X}_m + n_+ - (n - m - n_+)) + nf(z) \right) \binom{n-m}{n_+} \omega_+^{n_+} \omega_-^{n-m-n_+} \\
&= \exp\left( z\hat{X}_m + mf(z) \right) \exp\{((n-m)f(z))\} \sum_{n_+=0}^{n-m} \binom{n-m}{n_+} \left( e^z \omega_+ \right)^{n_+} \left( e^{-z} \omega_- \right)^{n-m-n_+} \\
&= \hat{Z}_m \left( \frac{\omega_+ e^z + \omega_- e^{-z}}{\exp(-f(z))} \right)^{n-m} = \hat{Z}_m, 
\end{aligned}
\tag{9.46}
$$

where $n_+$ represents the number of forward steps taken by $\hat{X}$ when moving from $\hat{X}_m$ towards $\hat{X}_n$.

### 9.4.3 Splitting probabilities

We use the martingales $\hat{Z}_n$ to determine the splitting probabilities in the gambler's ruin problem. Using Doob's optional stopping theorem (Theorem 1) for $\hat{Z}_n$ and the stopping time $\hat{T} = \min\left\{ n \geq 0 : \hat{X}_n \notin (0, \ell) \right\}$, we get

$$p_x^+ \langle \exp\left( z\ell + \hat{T}f(z) \right) | \hat{X}_{\hat{T}} = \ell \rangle + p_x^- \langle \exp\left( \hat{T}f(z) \right) | \hat{X}_{\hat{T}} = 0 \rangle = \exp(zx_0). \tag{9.47}$$

Setting

$$z = \ln\left( \frac{1-q}{q} \right), \tag{9.48}$$

the nontrivial root of $f(z)$ (i.e., $f(z) = 0$ and $z \neq 0$), Eq. (9.47) can be solved together with

$$p_x^+ + p_x^- = 1, \tag{9.49}$$

yielding Eq. (9.10) for the gambler's ruin probability.

### 9.4.4 Generating functions

To obtain the generating functions $g_x^+(s)$ and $g_x^-(s)$, we consider Eq. (9.47) at the two roots $z = z_\pm(s)$ that solve the equation

$$f(z_\pm) = \ln(s). \tag{9.50}$$

This yields the two equations

$$\exp(z_\pm \ell)g_x^+(s) + g_x^-(s) = \exp(z_\pm x) \tag{9.51}$$

where $g_x^\pm$ are the generating functions of $\hat{T}$ defined in Sec. 9.2.2. Solving Eq. (9.50) towards $z_\pm$ gives

$$z_\pm = \ln \lambda_\pm(s) \tag{9.52}$$

where $\lambda_\pm(s)$ are as defined in Eq. (9.18). Solving the Eqs. (9.51) towards $g_x^\pm$ yields Eqs. (9.17) and (9.20).

## 9.5 Thermodynamic versions of the gambler's ruin problem

We define thermodynamic versions of the gambler's ruin problem, which account for the attributes of nonequilibrium processes. As nonequilibrium processes are distinguished by currents $\hat{J}_n$ with a non-zero average value, $\langle \hat{J}_n \rangle > 0$, we focus on first-passage problems of currents $\hat{J}_n$ in Markov processes $\hat{X}_n$. Note that currents $\hat{J}_n$ are functionals defined on the trajectories $\hat{X}_0^n$, and hence they are in general nonMarkovian processes, which complicates the analysis of the corresponding first-passage problem. Nevertheless, using martingale theory we will be able to solve several first-passage problems of currents in nonequilibrium processes $X$.

### 9.5.1 General setup

Let $\hat{X}_n \in \mathscr{X}$, with $n \in \mathbb{N}$ and $\mathscr{X}$ a discrete set, be a stochastic process whose statistics described by $\mathbb{P}$ are Markovian with transition matrix $\mathbf{R}$, see Eq. (9.32). We assume that $\hat{X}$ is irreducible and that there exists a unique stationary, probability mass function $p_\infty(x)$ that solves Eq. (9.37) for all $x \in \mathscr{X}$. In other words, the process is ergodic, see Theorem 4.1 in Ref. [154]. In addition, we assume that $\mathbf{R}_{xy} > 0$ whenever that $\mathbf{R}_{yx} > 0$ so that the time-reversed Markov process exists.

An empirical integrated current $\hat{J}$ is a stochastic process of the form

$$\hat{J}_n := \sum_{x,y \in \mathscr{E}} c_{xy}\hat{J}_n(y|x) \tag{9.53}$$

where the empirical edge current

$$\hat{J}_n(y|x) := \hat{N}_n(y|x) - \hat{N}_n(x|y) \tag{9.54}$$

is the difference between the number of jumps $\hat{N}_n(y|x)$ in $\hat{X}_0^n$ from $x$ to $y$ and the number of jumps $\hat{N}_n(x|y)$ from $y$ to $x$ in the same trajectory, $c_{xy} \in \mathbb{R}$ are constant coefficients, and $\mathscr{E}$ is the set of pairs $(x,y)$ for which $\mathbf{R}_{xy} \neq 0$. In a stationary process, we denote the average current by

$$j(y|x) = \mathbf{R}_{yx}\mathbf{p}_\infty(x) - \mathbf{R}_{xy}\mathbf{p}_\infty(y). \tag{9.55}$$

The gambler's ruin problem in $\hat{J}$ reads

$$\hat{T}_J := \min \left\{ n \geq 0 : \hat{J}_n \notin (-J_-, J_+) \right\}. \tag{9.56}$$

where the thresholds $J_-, J_+ \geq 0$, and we assume, without loss of generality, that $\langle \hat{J}_n \rangle > 0$. The probability of ruin is then defined by

$$p_J^- := \mathbb{P}\left( \hat{J}_{\hat{T}_J} \leq -J_- \right). \tag{9.57}$$

The thermodynamic cost of a process can be quantified in terms of the stochastic entropy production [35, 136, 137]

$$\hat{S}_n := \hat{S}^{\mathrm{sys}}(\hat{X}_n) - \hat{S}^{\mathrm{sys}}(\hat{X}_0) + \hat{S}_n^{\mathrm{env}} \tag{9.58}$$

with

$$\hat{S}^{\mathrm{sys}}(\hat{X}_n) = -\ln \mathbf{p}_\infty(\hat{X}_n) \tag{9.59}$$

the Shannon entropy capturing the information theoretic content of the state $\hat{X}_n$, and

$$\hat{S}_n^{\mathrm{env}} := \frac{1}{2} \sum_{(x,y)\in\mathscr{E}} \ln \frac{\mathbf{R}_{xy}}{\mathbf{R}_{yx}} \hat{J}_n(y|x) \tag{9.60}$$

is the environment entropy change. The environment entropy is based on the principle of local detailed balance [25], which can be seen as a stochastic version of the local equilibrium concept [164]. At stationarity, the average rate of dissipation is

$$\sigma := \langle \hat{S}_n \rangle / n = \frac{1}{2} \sum_{(x,y)\in\mathscr{E}} \mathbf{R}_{xy} \mathbf{p}_\infty(y) \ln \frac{\mathbf{R}_{xy}}{\mathbf{R}_{yx}}. \tag{9.61}$$

In what follows, we discuss the gambler's ruin problem for three examples of currents $\hat{J}$.

### 9.5.2 Gambler's ruin problem for entropy production

The first case we consider is for $\hat{J} = \hat{S}$, the stochastic entropy production. The stopping problem for entropy production reads

$$\hat{T}_S = \min\left\{ n \geq 0 : \hat{S}_n \notin (-S_-, S_+) \right\}. \tag{9.62}$$

This case is relevant for two reasons: (i) in the uni-cyclic case, the macroscopic current is asymptotically proportional to the entropy production; (ii) using the stopping problem $\hat{T}_{\hat{S}}$ will establish relationships that resemble fluctuation relations for entropy production, albeit with regard to characteristics of entropy production at random times.

The calculation of the ruin probability $p_S^-$ and the average thermodynamic cost $\langle \hat{S}_{\hat{T}_S} \rangle$ is facilitated by the observation that the process $\exp\left(-\hat{S}_n\right)$ is a martingale [138, 141, 143]. Indeed,

$$e^{-\hat{S}_n} = \frac{\tilde{p}\left( \hat{X}_0^n \right)}{p\left( \hat{X}_0^n \right)} \tag{9.63}$$

is the ratio of the two path probabilities [35, 136, 137]

$$p\left( \hat{X}_0^n \right) = \mathbf{p}_\infty(\hat{X}_0) \prod_{i=1}^n \mathbf{R}_{\hat{X}_i \hat{X}_{i-1}} \tag{9.64}$$

and

$$\tilde{p}\left( \hat{X}_0^n \right) = \mathbf{p}_\infty(\hat{X}_0) \prod_{i=1}^n \tilde{\mathbf{R}}_{\hat{X}_i \hat{X}_{i-1}} \tag{9.65}$$

with

$$\tilde{\mathbf{R}}_{xy} = \mathbf{R}_{yx} \frac{\mathbf{p}_\infty(x)}{\mathbf{p}_\infty(y)}. \tag{9.66}$$

Hence, according to Eq. (9.30), $e^{-\hat{S}(t)}$ is a martingale. Note that

$$\tilde{p}\left(\Theta_n\left(\hat{X}_0^n\right)\right) = p\left(\hat{X}_0^n\right), \tag{9.67}$$

where $\Theta_n\left(\hat{X}_0^n\right) = (\hat{X}_n, \hat{X}_{n-1}, \dots, \hat{X}_0)$ is the time-reversed trajectory, and hence $\tilde{p}$ describes the statistics of an observer that goes backwards in time.

Using Doob's optional stopping theorem, see Sec. 9.3.3, we find that

$$\langle e^{-\hat{S}_{\hat{T}_S}} \rangle = 1, \tag{9.68}$$

which is a version of the Integral Fluctuation Theorem at Stopping Times [140]. Further, using that

$$\mathbb{P}\left(\hat{T}_S < \infty\right) = p_S^+ + p_S^- = 1, \tag{9.69}$$

we obtain

$$p_S^+ \langle e^{-\hat{S}_{\hat{T}_S}} \rangle_+ + p_S^- \langle e^{-\hat{S}_{\hat{T}_S}} \rangle_- = 1. \tag{9.70}$$

If $\hat{S}_{\hat{T}_S} \in \{-S_-, S_+\}$, which includes the continuum limit, then Eqs. (9.69) and (9.70) yield

$$p_S^- = \frac{e^{S_+} - 1}{e^{S_+ + S_-} - 1} \tag{9.71}$$

for the ruin probability. Notice that Eq. (9.71) can be identified with Bernoulli's formula Eq. (9.1) when we set $S_+ = m \ln(b/a)$ and $S_- = n \ln(b/a)$, with $a, b \in (0, 1)$ arbitrary numbers. Analogously, setting $S_+ = (\ell - x) \ln(\omega_+/\omega_-)$ and $S_- = x \ln(\omega_+/\omega_-)$ in Eq. (9.71), we obtain the Eq. (9.10) for the ruin probability in the gambler's ruin problem. For processes that are non continuous, we get the inequality [140]

$$p_S^- \le \frac{1}{e^{S_-} - e^{-S_+}} \tag{9.72}$$

for the probability of ruin.

Interestingly, the results Eqs. (9.71) and (9.72) for $p_-$ allow us to precisely quantify the negative fluctuations of the entropy production in terms of its infimum value

$$\hat{S}_{\text{inf}} := \inf_{n \ge 0} \hat{S}_n. \tag{9.73}$$

Indeed, observe that

$$\lim_{S_+ \to \infty} p_S^- = \mathbb{P}\left(\hat{S}_{\text{inf}} \le -S_-\right). \tag{9.74}$$

Therefore, Eq. (9.72) yields in the limit of $S^+ \gg 1$ the inequality

$$\mathbb{P}\left(\hat{S}_{\text{inf}} \le -s\right) \le \exp(-s) \tag{9.75}$$

such that on average

$$\langle \hat{S}_{\text{inf}} \rangle \ge -1, \tag{9.76}$$

with equalities attained in the continuum limit. The inequality (9.75) readily implies the bound

$$\mathbb{P}\left(\hat{S}_n \le -s\right) \le \exp(-s) \tag{9.77}$$

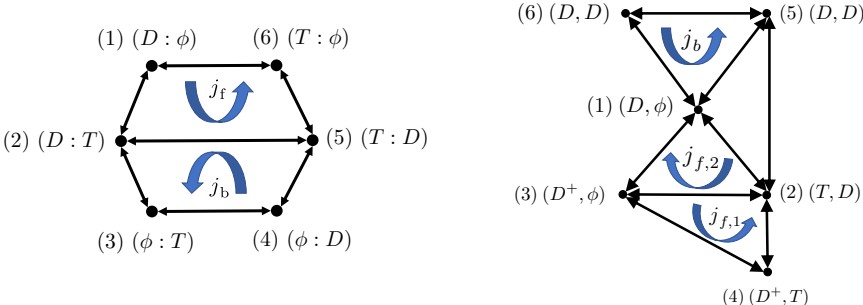

Figure 9.3: Two examples of stochastic processes that model the dynamics of two-headed molecular motors, such as, kinesin-1. The motor states (D:T), (T:$\phi$), and so forth, denote the chemical states of the rear and front motor head, viz., T stands for ATP bound, D for ADP bound, D' for ADP-P bound, and $\phi$ for nucleotide free. Left Panel: Model taken from Ref. [165] with three cycle currents, corresponding to forward motion ($j_{\mathrm{f}}$), backward motion ($j_{\mathrm{b}}$), and a current with no motion yielding a futile cycle [165]. Notably, the motor position along the biofilament is given by the edge current $\hat{J}_n^{2\to5}$. Right Panel: Molecular motor model of Ref. [166]. In this case, there are two cycles corresponding with forward motion (with currents $j_{\mathrm{f},1}$ and $j_{\mathrm{f},2}$, respectively), a cycle yielding backward motion (with current $j_{\mathrm{b}}$), and several futile cycles.

that follows from the integral fluctuation relation $\langle \exp\left(-\hat{S}_n\right)\rangle = 1$ [35]. This illustrates how martingale methods can be used to obtain bounds on negative fluctuations of entropy production that are tighter than those obtained from fixed-time fluctuation relations in stochastic thermodynamics.

Furthermore, if $\hat{S}_{\hat{T}_S} \in \{-S_-, S_+\}$, then we obtain

$$\langle \hat{S}_{\hat{T}_S}\rangle = p_S^+ S_+ - p_S^- S_- = \frac{S_+(e^{S_-}-1)-S_-(1-e^{-S_+})}{e^{S_-}-e^{-S_+}} \tag{9.78}$$

for the average thermodynamic cost at the termination time. Setting $S_+ = (\ell-x)\ln(\omega_+/\omega_-)$ and $S_- = x\ln(\omega_+/\omega_-)$ in Eq. (9.78), yields Eq. (9.11) for the average profit at the termination time. More generally, we have the inequality

$$\langle \hat{S}(\hat{T}_S)\rangle \geq S_+ - \frac{S_+ + S_-}{e^{S_-}-e^{-S_+}} \geq 0, \tag{9.79}$$

which is a specific instance of the second law of thermodynamics at stopping times [140, 142]. This law states that on average the entropy production increases, even when we stop the process at a random stopping time. Key for the second law of thermodynamics at stopping times to hold is that the random time $\hat{T}$ obeys causality, i.e., it is a functional of the trajectory $X_0^{\hat{T}}$ up to the stopping time.

### 9.5.3 Gambler's ruin problem for edge currents

In general, currents of interest in nonequilibrium processes do not match the entropy production. Indeed, processes typically exhibit multiple currents, and the stochastic entropy production is a specific linear combination of those currents, see Eqs. (9.58-9.60). Therefore, in the present section, we consider the simplest example of a current that is not the stochastic entropy production, viz., an edge current $\hat{J} = \hat{J}(y|x)$. The corresponding

stopping problem is given by

$$\hat{T}(y|x) = \min\left\{n \geq 0 : \hat{J}_n(y|x) \notin (-J_-, J_+)\right\}. \tag{9.80}$$

Note that in general macroscopic currents are not edge currents, but nevertheless, there exist interesting examples of processes for which this is the case. An example is the position of two-headed molecular motors bound to a biofilament as described by the model developed in Ref. [165], whose transition graph is shown in the Left Panel of Fig. 9.3. In this model, the position of the motor along the filament is given by the edge current $\hat{J}_n(5|2)$.

From the solution of the gambler's ruin problem for $\hat{S}$, we have learned that it is sufficient to find a martingale associated with $\hat{J}_n(y|x)$ to solve its gambler's ruin problem. However, $\exp\left(-a\hat{J}_n(y|x)\right)$ is not a martingale, except for the trivial case of $a = 0$. Nevertheless, as shown in Ref. [147], we can find a martingale that is asymptotically equivalent to $\exp\left(-a\hat{J}_n(y|x)\right)$, i.e., up to terms of the order $O_n(1)$, for a certain value of $a$. Indeed, the process

$$\hat{M}_n(y|x) := \frac{p_\infty^{(x,y)}(\hat{X}_0)q_\infty(\hat{X}_n)}{{}_\infty^{(x,y)}(\hat{X}_n)p_\infty(\hat{X}_0)} \exp\left(-a^*(y|x)\hat{J}_n(y|x)\right) \tag{9.81}$$

is a martingale, where

$$a^*(y|x) := \ln \frac{\mathbf{R}_{yx}\,\mathbf{p}_\infty^{(x,y)}(x)}{\mathbf{R}_{xy}\,\mathbf{p}_\infty^{(x,y)}(y)} \tag{9.82}$$

is the so-called effective affinity [86, 167, 168], where $p_\infty^{(x,y)}$ is the stationary state of the Markov chain $(\hat{X}, \mathbb{P}^{(x,y)})$ obtained from $(\hat{X}, \mathbb{P})$ by removing the edge $(x, y)$ from the Markov transition graph, and where $q_\infty$ is the stationary state of an auxiliary process $(\hat{X}, \mathbb{Q})$ with the transition matrix $\mathbf{V}$ with entries

$$\mathbf{V}_{uv} = \begin{cases} \frac{\mathbf{p}_\infty^{(x,y)}(u)}{\mathbf{p}_\infty^{(x,y)}(v)}\mathbf{R}_{vu}, & \text{if} \quad (u, v) \in \mathscr{E} \setminus \{(x,y),(y,x)\}, \\ \mathbf{R}_{uv} & \text{if} \quad (u, v) \in \{(x,y),(y,x)\}. \end{cases} \tag{9.83}$$

In the Appendix 9.7, we perform a calculation analogous to the one in Ref. [147] for continuous time Markov chains showing that $\hat{M}_n(y|x)$ is a Radon-Nikodym derivative process of the form Eq. (9.29), and hence a martingale with respect to $\hat{X}$ and $\mathbb{P}$.

Consequently, Doob's optional stopping theorem in Sec. 9.3.3 applies to the martingale $\hat{M}_n(y|x)$ and the stopping time $\hat{T}(y|x)$, yielding

$$\langle \hat{M}_{\hat{T}(y|x)}(y|x)|\hat{X}_0 = x_0\rangle = \langle \hat{M}_0(y|x)|\hat{X}_0 = x_0\rangle = \frac{\mathbf{q}_\infty(x_0)}{\mathbf{p}_\infty(x_0)}. \tag{9.84}$$

Defining the splitting probabilities

$$p_{(y|x)}^\pm(x_0) = \mathbb{P}\left(\hat{J}_{\hat{T}(y|x)}(y|x) = J_\pm | X_0 = x_0\right), \tag{9.85}$$

which obey

$$p_{(y|x)}^+(x_0) + p_{(y|x)}^-(x_0) = 1, \tag{9.86}$$

we find from Eqs.(9.84) and (9.86) that

$$p_{(y|x)}^-(x_0) = \frac{\exp\left(a^*(y|x)J_+\right)\zeta(x_0) - 1}{\exp\left(a^*(y|x)(J_- + J_+)\right)\zeta(x) - 1}, \tag{9.87}$$

where

$$\zeta(u) = \frac{\mathbf{p}_\infty^{(x,y)}(y)\mathbf{q}_\infty(u)}{\mathbf{p}_\infty^{(x,y)}(u)\mathbf{q}_\infty(y)}, \quad \text{for} \quad u \in \mathscr{X}. \tag{9.88}$$

Analogously to the study of the infimum statistics of entropy production, Ref. [147] uses Eq. (9.87) to determine the probability mass function of the infimum $\hat{J}_{\inf}(y|x)$ of $\hat{J}(y|x)$ and finds that it is geometrically distributed; this follows readily from taking the limit $J_+ \to \infty$, see also Ref. [169]. Remarkably, this property is unique to edge currents can thus be used to identify whether a measured current $\hat{J}$ is an edge current; note that we assume here that the observer can measure $\hat{J}$ but not $\hat{X}$.

Since the thermodynamic interpretation of the stationary distributions $p_\infty$, $p_\infty^{(x,y)}$, and $q_\infty$ is not entirely clear, we take the limits $J_+ \gg 1$ and $J_- \gg 1$ in Eq. (9.87), with the ratio $J_+/J_-$ fixed. In this limit, the formula (9.87) simplifies into

$$p_{(y|x)}^-(x_0) = \exp\left(-a_{(y|x)}^* J_-(1 + o_{J_{\min}}(1))\right) \tag{9.89}$$

with $J_{\min} = \min\{J_-, J_+\}$. Additionally, using the Wald-like equality [170, 171],

$$\langle \hat{T} \rangle = \frac{J_+}{j(y|x)}(1 + o_{J_{\min}}(1)), \tag{9.90}$$

we get that

$$|\log p_{(y|x)}^-(x_0)| = \frac{J_-}{J_+} a^*(y|x)\langle \hat{T} \rangle j(y|x)(1 + o_{J_{\min}}(1)). \tag{9.91}$$

The Eq. (9.91) has an appealing thermodynamic meaning when realising that [86, 168]

$$a^*(y|x)j(y|x) \leq \sigma, \tag{9.92}$$

yielding [145]

$$|\log p_{(y|x)}^-(x_0)| \leq \frac{J_-}{J_+}\langle \hat{T} \rangle \sigma(1 + o_{J_{\min}}(1)). \tag{9.93}$$

The Eq. (9.93) expresses a trade-off between speed ($\langle \hat{T} \rangle$), dissipation ($\sigma$), and uncertainty ($|\log p_-|$) in a nonequilibrium process $\hat{X}$. It is a universal inequality in the sense that it applies to arbitrary Markov chains $\hat{X}$ and, importantly, it remains valid for generic currents $\hat{J}$ as we discuss in the next subsection.

### 9.5.4   Gambler's ruin problem for generic currents

Although edge currents play a significant role in certain specific models — such as for the position of a molecular motor described by the model illustrated in the Left Panel of Fig. 9.3 — it is vital to study the first-passage problem of general currents in nonequilibrium processes, as macroscopic currents are typically not edge currents. This can be seen in the model illustrated in the Right Panel of Fig.9.3, which is an alternative model for a two-headed molecular motor that takes into consideration multiple pathways by which the motor can move forwards or backwards.

Therefore, we consider in this section the gambler's ruin problem $\hat{T}_J$, described by Eq. (9.56), for a general current $\hat{J}$ of the form given by Eq. (9.53).

It should be noted that the derivations we present in this section are based on continuous time Markov chains, as this is the setup considered in Ref. [145]. Therefore, we use the time index $t \in \mathbb{R}^+$ instead of $n \in \mathbb{N}$. Nevertheless, several results for the gambler's ruin problem reviewed here also apply to discrete time Markov chain; we will come back on this matter in more detail.

Ergodic, continuous time, Markov chains defined on a finite set $\mathscr{X}$ satisfy a large deviation principle for the current, see Ref. [94, 172], i.e., in the limit of $t \gg 1$ it holds that

$$p_{\hat{j}_t/t}(u) = \exp\left(-t\mathcal{J}(u)(1 + o_t(1))\right) \tag{9.94}$$

where $\mathcal{J}(u) \geq 0$ is the large deviation function defined for $u \in \mathbb{R}$. The large deviation function satisfies the bound

$$\mathcal{J}(u) \leq \frac{\sigma}{4}\left(\frac{u}{j} - 1\right)^2 \tag{9.95}$$

which was obtained in Refs. [173, 174] (where $j$ is the average current). Using this inequality, Ref. [145] shows that

$$|\ln p_J^-| \leq \frac{J_-}{J_+}\langle \hat{T}_J \rangle \sigma(1 + o_{J_{\min}}(1)). \tag{9.96}$$

hods generically for currents $\hat{J}$ of the form Eq. (9.53). Note that even though the large deviation bound Eq. (9.95) does not apply in discrete time Markov chains, the inequality Eq. (9.96) applies in the discrete time setting (see previous section for a derivation for edge currents). The reason for this is that the ruin probability $p_-$ is conserved under the mapping from discrete to continuous time Markov chains discussed in Ref. [175].

The inequality Eq. (9.96) expresses the trade-off between speed ($\langle \hat{T} \rangle$), dissipation ($\sigma$), and uncertainty ($|\log p_-|$), and is interesting from these three perspectives. Let us discuss these three perspectives below.

The inequality Eq.(9.96) has implications for the speed of processes, as measured by $\langle \hat{T} \rangle$. It suggests that far from equilibrium, processes can be faster than their equilibrium counterparts, but with a cost in terms of either dissipation or uncertainty. This is evident when considering the nonequilibrium version of Kramer's escape problem, as discussed in [145]. Near equilibrium, the inequality reduces to a van't Hoff-Arrhenius law-like equality. However, driving the process away from equilibrium results in a reduction of $\langle \hat{T} \rangle$, which is captured by the inequality Eq.(9.96).

Second, we discuss the inequality from the viewpoint of uncertainty, quantified with $|\log p_J^-|$. In general, the quantity $p_J^-$ decreases exponentially in the threshold $\ell_-$, i.e.,

$$p_J^- = \exp(a^* J_-(1 + o_{J_{\min}}(1))). \tag{9.97}$$

One way to approach the challenge of determining $a^*$ in experiments is to use the inequality (9.96), which provides a bound on the quantity. Specifically, the inequality implies that $a^*$ is less than or equal to $\langle \hat{T}_J \rangle \sigma/J_+$, up to small corrections that are negligible for large $J_{\min}$. This means that by measuring the rate of dissipation and the mean first-passage time, we can estimate $a^*$. This approach is particularly useful as $p_-$ decreases rapidly with $J_-$, making it difficult to directly determine $a^*$ in experiments.

Lastly, we analyze the inequality from the perspective of dissipation $\sigma$. In nonequilibrium processes, fluctuations of currents carry information about the amount of dissipation. Therefore, the inequality (9.96) can be interpreted as $\sigma \geq \sigma_{\mathrm{FPR}}(1 + o_{J_{\min}}(1))$, where $\sigma_{\mathrm{FPR}}$ is an estimator of dissipation given by

$$\sigma_{\mathrm{FPR}} = \frac{J_+}{J_-}\frac{|\ln p_J^-|}{\langle \hat{T}_J \rangle}. \tag{9.98}$$

This estimator combines the splitting probability $p_J^-$ and the mean first-passage time $\langle \hat{T}_J \rangle$, and can be used to bound the rate of dissipation from below. We can compare this

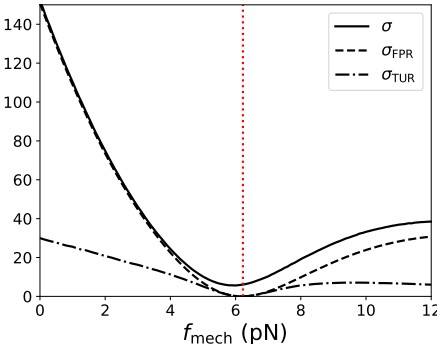

Figure 9.4: Rate of dissipation $\sigma$ and its lower bounds $\sigma_{\mathrm{FPR}}$ and $\sigma_{\mathrm{TUR}}$ plotted as a function of the mechanical force exerted $f_{\mathrm{mech}}$ exerted on the motor. The model is the one of Fig. 9.3, and the parameters chosen are physiological relevant and described in Ref. [147] [data is taken from Figure 5 in Ref. [147]].

approach with the alternative inequality $\sigma \geq \sigma_{\mathrm{TUR}}(1 + o_{J_{\min}}(1))$, where [176]

$$\sigma_{\mathrm{TUR}} = 2\frac{\langle \hat{T}_J \rangle}{\langle \hat{T}_J^2 \rangle - \langle \hat{T}_J \rangle^2} \tag{9.99}$$

is the thermodynamic uncertainty ratio, that holds for continuous time Markov chains [176]; notice that comparing with (9.96) the $\sigma_{\mathrm{TUR}}$ quantifies the uncertainty with the coefficient of variation, $(\langle \hat{T}_J^2 \rangle - \langle \hat{T}_J \rangle^2)/\langle \hat{T}_J \rangle^2$ instead of $|\ln p_J^-|$. In this regard, rather surprisingly, it was found in [146] that $\sigma_{\mathrm{FPR}}$ is a better estimator than $\sigma_{\mathrm{TUR}}$ for noncontinuous processes that run far from thermal equilibrium; see Fig. 9.4 for the standard molecular motor example. As a final remark on this, note that the thermodynamic uncertainty bound (9.99) does not apply in discrete time, whereas the first-passage ratio bound (9.96) is valid in discrete time.

We end the paper with discussing the thermodynamic cost $\langle \hat{S}(\hat{T}_J) \rangle$ for the gambler's ruin problem of a generic current. It holds generically that [140, 142]

$$\langle \hat{S}_{\hat{T}_J} \rangle \geq 0 \tag{9.100}$$

for all $\hat{J}$, and in fact, the inequality $\langle \hat{S}_{\hat{T}} \rangle \geq 0$ holds for any stopping time $\hat{T}$. We coined this relation the second law of thermodynamics at stopping times [140, 142]. This law expresses that an observer cannot anticipate negative fluctuations of entropy production, even if it is "infinitely" smart and has full knowledge of the trajectory's history [143]. It will be interesting to further explore the connection betweenn the second law of thermodynamics at stopping times and Maxwell demons , see e.g. Refs. [142, 177–179].

## 9.6    Discussion

We discuss some of the questions that were raised by the participants of the workshop.

- Q1:"*Thermodynamics methods have exploited the martingale approach to the gamblers ruin problem but not the difference equations approach. Hence, can we use the difference method to solve the gambler's ruin problem in stochastic thermodynamics?*"

  **Answer:** This should in principle be possible, but since it has not been explored much so far, it should be rather taken as a suggestion for future research, than a question. One of the purposes of the present lecture notes is to precisely draw attention to these kind of interesting problems.

- Q2:"*The trade-off inequality (9.96) is reminiscent of the thermodynamic uncertainty relation for first-passage times (9.99). Why is in the simulations of Figure 9.4 the former tighter than the latter?*

  **Answer:** It is important to note that the two inequalities are distinct. The thermodynamic uncertainty relation for first-passage times solely relies on the statistics of $\hat{T}_J$ at the positive boundary $J_+$ and is agnostic to the negative boundary $J_-$. In contrast, inequality (9.96) is based on the probability of reaching the negative boundary, and it seems that these negative fluctuations are critical for estimating entropy production.

- Q3:"*The Eq. (9.96) is an equality when you set $\hat{J} = \hat{S}$. Why should we care, as in general currents $\hat{J}$ are not proportional to $\hat{S}$?:*

  **Answer:** An equality places more strict limitations on the behavior of nonequilibrium systems than an inequality. For example, the Jarzysnki equality is considered a stronger result than the second law of thermodynamics. Although, in general, the equality in Eq. (9.96) is not realized, we can demonstrate that it is achieved when $\hat{J} = \hat{S}$, which is an intriguing feature. This has practical implications, such as when using $\sigma_{\mathrm{FPR}}$ to estimate dissipation, as it means the estimator is unbiased if $\hat{J} = \hat{S}$. An estimator lacking this attribute is unable to accurately estimate $\sigma$ even with complete knowledge of $\hat{S}$.

- *Q4: To what extent does the martingale formalism for $e^{-\hat{S}_t}$ apply to diffusion processes (or Langevin processes)?*:

  **Answer:** Demonstrating the martingale property of $e^{-\hat{S}}$ is more challenging for diffusion processes. It is immediate that $e^{-\hat{S}}$ is a local martingale, see [139], but martingality requires an additional assumption, called the Novikov condition, see Ref. [143]. Alternatively, it is sufficient to show the integral fluctuation relation $\langle e^{-\hat{S}_t} \rangle = 1$, which together with the local martingale property of $e^{-\hat{S}_t}$ implies that $e^{-\hat{S}_t}$ is a martingale. In the physics literature, deriving $\langle e^{-\hat{S}_t} \rangle = 1$ is usually avoided with the (sensible) argument that the Radon-Nikodym derivative between the forward and backward measures exists, in which case the latter equality is immediate [41]. Nevertheless, identifying whether for a given diffusion process the ratio between the forward and backward measures exists is a priori not clear and a challenging problem.

## 9.7    Appendix: Martingality of Eq. (9.81)

We show that $\hat{M}_n(y|x)$, as defined in Eq. (9.81), is a martingale. First, in Sec. 9.7.1, we show that the matrix $\mathbf{V}$, as defined in Eq. (9.83), defines a Markov chain. Second, in Sec. 9.7.2, we show that $\hat{M}_n(y|x)$ is the ratio of two path probabilities, and thus a martingale according to the derivations in Sec. 9.3.2.

### 9.7.1    Validity of the Markov Transition Matrix Eq. (9.83)

We show that the matrix $\mathbf{V}_{uv}$, given by Eq. (9.83), defines a Markov chain. Since it is clear that all the elements are nonnegative, it remains to show that

$$\sum_{u \in \mathscr{X}} \mathbf{V}_{uv} = 1 \tag{9.101}$$

for all $v \in \mathscr{X}$. For $v \in \mathscr{X} \setminus \{x, y\}$, Eq. (9.101) is equivalent to

$$\sum_{u \in \mathscr{X}} \mathbf{p}_\infty^{(x,y)}(u) \mathbf{R}_{vu} = \mathbf{p}_\infty^{(x,y)}(v), \tag{9.102}$$

which holds as $\mathbf{p}_\infty^{(x,y)}(u)$ is the stationary state of the Markov chain $(\hat{X}, \mathbb{P}^{(x,y)})$, obtained from $(X, \mathbb{P})$ by removing the $(x, y)$ edge from the set $\mathscr{E}$. Note that the removal of an edge yields a transition matrix $\mathbf{R}^{(x,y)}$ with $\mathbf{R}_{xy}^{(x,y)} = \mathbf{R}_{yx}^{(x,y)} = 0$ and the diagonal elements are adjusted to preserve normalization so that

$$\mathbf{R}_{xx}^{(x,y)} = \mathbf{R}_{yx} + \mathbf{R}_{xx} \quad \text{and} \quad \mathbf{R}_{yy}^{(x,y)} = \mathbf{R}_{xy} + \mathbf{R}_{yy}. \tag{9.103}$$

Therefore, for $v \in \mathscr{X} \setminus \{x, y\}$ it holds that $\mathbf{R}_{uv}^{(x,y)} = \mathbf{R}_{uv}, \forall u \in \mathscr{X}$, and hence Eq. (9.102) is the steady state equation.

We are left to verify Eq. (9.101) for $v = x$ and $v = y$. We discuss here $v = x$, and leave $v = y$ as an exercise for the reader. For $v = x$, Eq. (9.101) reads

$$\sum_{u \in \mathscr{X} \setminus \{x,y\}} \frac{\mathbf{p}_\infty^{(x,y)}(u)}{\mathbf{p}_\infty^{(x,y)}(x)} \mathbf{R}_{xu} + \mathbf{R}_{yx} + \mathbf{R}_{xx} = 1, \tag{9.104}$$

which simplifies into

$$\sum_{u \in \mathscr{X} \setminus \{x,y\}} \mathbf{p}_\infty^{(x,y)}(u) \mathbf{R}_{ux} + \mathbf{R}_{xx}^{(x,y)} \mathbf{p}_\infty^{(x,y)}(x) = \mathbf{p}_\infty^{(x,y)}(x) \tag{9.105}$$

after using Eq. (9.103). Since Eq. (9.105) is the steady state equation for $\mathbf{p}_\infty^{(x,y)}$, it holds as an equality, which is what we were meant to show.

### 9.7.2 The process in Eq. (9.81) is ratio of two path probabilities

We now show that $\hat{M}_n(y|x)$, as defined in Eq. (9.81), is a ratio of two path probabilities, and hence a martingale. The calculation follows along the same lines as the one presented in Sec. 5 of Ref. [147] for continuous time Markov chains.

Specifically, we show that

$$\hat{M}_n(y|x) = \frac{r(\hat{X}_0^n)}{p(\hat{X}_0^n)} \tag{9.106}$$

where $p(\hat{X}_0^n)$ is the path probability

$$p(\hat{X}_0^n) = \mathbf{p}_\infty(\hat{X}_0) \prod_{j=1}^{n} \mathbf{R}_{\hat{X}_j \hat{X}_{j-1}}, \tag{9.107}$$

and

$$r(\hat{X}_0^n) = \mathbf{q}_\infty(\hat{X}_0) \prod_{j=1}^{n} \tilde{\mathbf{V}}_{\hat{X}_j \hat{X}_{j-1}} \tag{9.108}$$

is an alternate path probability, where $\mathbf{q}_\infty(u)$ is the stationary state of the Markov chain $\mathbf{V}$, and

$$\tilde{\mathbf{V}}_{uv} = \mathbf{V}_{vu} \frac{\mathbf{q}_\infty(u)}{\mathbf{q}_\infty(v)} \tag{9.109}$$

is its corresponding time-reversed Markov chain. Because of the definition of $\mathbf{V}$ in Eq. (9.83), we find the expression

$$
\tilde{\mathbf{V}}_{uv} = \begin{cases} \frac{\mathbf{p}_\infty^{(x,y)}(v)\mathbf{q}_\infty(u)}{\mathbf{p}_\infty^{(x,y)}(u)\mathbf{q}_\infty(v)}\mathbf{R}_{uv}, & \text{if} \quad (u,v) \in \mathscr{E} \setminus \{(x,y),(y,x)\}\,, \\ \frac{\mathbf{q}_\infty(u)}{\mathbf{q}_\infty(v)}\mathbf{R}_{vu} & \text{if} \quad (u,v) \in \{(x,y),(y,x)\}\,. \end{cases}
\tag{9.110}
$$

Substituting the expressions (9.107), (9.108), and (9.110) into Eq. (9.106) we obtain the right-hand side of Eq. (9.81), which we were meant to show.

# 10 Quantum Thermodynamics

**Ariane Soret and Vasco Cavina**[35] *This lecture aims at providing the basic concepts to understand ongoing research in the field of quantum thermodynamics. Starting from the foundations of the statistical approach to quantum mechanics, we present the notion of quantum measurement as a tool to compute the full statistics of thermodynamic quantities (heat, work and entropy) in closed quantum systems. This formalism is then used to discuss the laws of thermodynamics, the fluctuation theorems and the Jarzynski equality for closed quantum systems. We then extend the discussion to open systems, where we focus on master equation approaches. Sections 2 and 3 are mainly based on the references [180] and [181], while the end of section 3 is based on [28]. Section 4 presents recent developments [182].*

## 10.1 Introduction

The initial development of quantum mechanics was motivated by a thermodynamics problem: blackbody radiation. Experiments carried out during the 19th century evidenced that the spectrum of the radiation emitted by a blackbody depended exclusively on its temperature. Thanks to rapid improvements in experimental methods, providing increasingly accurate data, by the end of the 19th century Max Planck was able to derive an empirical equation describing the blackbody spectrum with great accuracy. Planck later tried to find a theoretical explanation for his equation and succeeded in 1900, by applying Ludwig Boltzmann theory of gases [183]. Boltzmann's approach relied on a mathematical trick: dividing the energy spectrum in small units, which would then be taken infinitely small in order to recover a continuous energy distribution. Planck realized that the energy spectrum had to be divided in units $\epsilon = \hbar\nu$, with $\nu$ the frequency at which the molecules of the blackbody resonated, and $\hbar$ - later known as the Planck constant - a quantity which could not be taken arbitrarily small. Five years later, Albert Einstein took the reasoning one crucial step further, by arguing that the radiation field was in fact constituted of individual quanta (photons) of energy $\hbar\nu$ [184]. Einstein showed that this approach fully reconciled Maxwell's theory of electromagnetism (which predicts a divergence of the black body spectrum in the ultraviolet regime – the so-called ultra violet catastrophe) with the blackbody radiation spectrum. This was the starting point of the theory of quanta, which later on led to the development of quantum mechanics by Bohr [185], then Schrödinger [186], Heisenberg [187], Born [188] and Dirac [189]. Quantum thermodynamics appeared as a field of research a few decades later, to study the emergence of thermodynamics in quantum systems [190, 191], and reconnecting with the historical starting point of quantum physics. Even more recently, the focus in the field was put in the study of out of equilibrium systems. For further reading on the history of quantum physics and thermodynamics, see [192].

This course provides an overview of the mathematical foundations of quantum thermodynamics, and presents modern methods used in the field today as well as recent developments.

Quantum mechanics is an inherently probabilistic theory, in the sense that a full description of a quantum object requires to perform a series of measurements on copies of that object, prepared in identical conditions. Consider for example the polarization of a photon: if we measure it using an interferometer, the result will be either horizontal or vertical, but the photon was actually in a linear combination of the two. In order to fully describe the photon, we would therefore need to repeat the experiment several times on a

---

[35]AS was the lecturer and wrote this chapter; VC was the angel.

set of photons prepared in the same way and acquire information on the initial state using the statistics of our series of measurements. Despite it being intrinsically probabilistic, the framework of quantum statistics differs strongly from classical statistics. This has to do with the fact that the classical statistical physics theory, built on probability spaces, is incompatible with the fundamental principles of quantum mechanics, that a system is fully described only by its wave function on a Hilbert space. We will explain this point in the first part of the course.

## 10.2   Mathematical framework for quantum statistics

This chapter contains reminders of quantum mechanics, and introduces the mathematical framework of quantum statistics. We state the main postulates on which quantum mechanics is built, define random variables, and introduce the concept of density matrix. We begin with two fundamental postulates of quantum mechanics:

> **Postulate 1:** A quantum system is described by a state vector, or wave function, $|\psi\rangle$, defined on a Hilbert space $\mathcal{H}$.

> **Postulate 2:** Measurable quantities, or *observables*, are represented by linear, self-adjoint operators acting on the Hilbert space $\mathcal{H}$.

Every Hilbert space is naturally equipped with an inner product (also known as scalar product). The inner product between two vectors $|\psi\rangle, |\phi\rangle$ of the Hilbert space is noted $\langle\psi|\phi\rangle$, and the norm of a vector $|\psi\rangle$ is given by $||\psi|| = \sqrt{\langle\psi|\psi\rangle}$. Throughout this course, we will consider only *separable* Hilbert spaces, i.e., Hilbert spaces with a finite or infinite and countable basis $\{\phi_\alpha\}$, s.t. $\langle\phi_\alpha|\phi_\beta\rangle = \delta_{\alpha\beta}$. Any vector $|\psi\rangle$ then admits a unique decomposition

$$|\psi\rangle = \sum_\alpha \langle\psi|\phi_\alpha\rangle|\phi_\alpha\rangle . \tag{10.1}$$

An operator $\hat{M}$ is *self-adjoint* if it is equal to its adjoint $\hat{M}^\dagger$, which is the operator satisfying $\langle\hat{M}\psi|\phi\rangle = \langle\psi|\hat{M}^\dagger\phi\rangle$ for all $|\psi\rangle, |\phi\rangle$.

Example: For a Hilbert space of finite dimension, we can represent the operators by matrices in the coordinate representation induced by a selected basis. The inner product coincides with the standard inner product on $\mathbb{C}$. In this case, the matrix associated to the adjoint operator $\hat{M}^\dagger$ is the conjugate transpose of the matrix associated to $\hat{M}$.

The time evolution of the state of a quantum system is described by a time dependent wave function $|\psi(t)\rangle$.

> **Postulate 3:** For *isolated systems* (i.e., systems which exchange neither energy nor particles with the environment), the state vector $|\psi(t)\rangle$ evolves in time according to the Schrödinger equation,
>
> $$i\hbar\frac{d|\psi(t)\rangle}{dt} = \hat{H}|\psi(t)\rangle \tag{10.2}$$
>
> where $\hat{H}$ is a time independent operator – the system's Hamiltonian – and $\hbar$ is the Planck's constant. If the system is submitted to external forces (e.g. an electromagnetic field), and if the dynamics of the system can be described as a time dependent Hamiltonian $\hat{H}(t)$, then the system is said to be *closed* [180], and its state vector satisfies
>
> $$i\hbar\frac{d|\psi(t)\rangle}{dt} = \hat{H}(t)|\psi(t)\rangle . \tag{10.3}$$

The solution of (10.3) takes the form

$$|\psi(t)\rangle = \hat{U}(t, t_0)|\psi(t_0)\rangle \tag{10.4}$$

where $\hat{U}(t, t_0)$ is the so called *propagator* or *time-evolution* operator, satisfying

$$
\begin{aligned}
i\hbar\frac{\partial \hat{U}(t,t_0)}{\partial t} &= \hat{H}(t)\hat{U}(t, t_0) \\
\hat{U}(t_0, t_0) &= \mathbf{I} : \text{initial condition}.
\end{aligned}
\tag{10.5}
$$

$\hat{U}(t, t_0)$ is a unitary operator: $\hat{U}^\dagger(t, t_0)\hat{U}(t, t_0) = \hat{U}(t, t_0)\hat{U}^\dagger(t, t_0) = \mathbf{I}$. If the Hamiltonian $\hat{H}$ is time-independent, then $\hat{U}(t, t_0)$ is given by

$$\hat{U}(t, t_0) = e^{-i(t-t_0)\hat{H}/\hbar}. \tag{10.6}$$

If $\hat{H}(t)$ is time dependent, then

$$\hat{U}(t, t_0) = \mathcal{T}_\leftarrow \left[ e^{-i\int_0^t ds\hat{H}(s)/\hbar} \right] \tag{10.7}$$

where $\mathcal{T}_\leftarrow$ is the time-ordering operator.

An important theorem for quantum measurement and quantum statistics is the *spectral theorem*,

**Spectral Theorem (discrete version[36]):** For every self-adjoint operator $\hat{M}$ with a discrete spectrum $\{\mu_n\}$, there exists a unique spectral family $\{\hat{\Pi}_n\}$ of projection operators s.t. $\hat{M} = \sum\limits_n \mu_n\hat{\Pi}_n$, with $\hat{\Pi}_n = \sum\limits_k |\psi_{n,k}\rangle\langle\psi_{n,k}|$ where $\{|\psi_{n,k}\rangle\}_k$ is an eigenbasis of the eigenspace associated to the eigenvalue $\mu_n$. $\square$

The projection operators are orthogonal to each other: $\hat{\Pi}_n\hat{\Pi}_m = \delta_{mn}\hat{\Pi}_n$, and satisfy the completeness relation: $\sum\limits_n \hat{\Pi}_n = \mathbf{I}$.

**Statistical approach to quantum mechanics**   Let us now introduce the framework to describe quantum mechanics from a statistical point of view. In the statistical interpretation of quantum mechanics, every state $|\psi\rangle$ represents an ensemble $\mathcal{E}$ of identically prepared systems, $\mathcal{E} = \{S_1, ..., S_N\}$ The result of any measurement process of $\mathcal{E}$, associated to an observable as stated in postulate 2, can be interpreted in a consistent way as an ensemble of possible outcomes as state in the following postulate.

Consider an observable $\hat{M}$.

> **Postulate 4 (Born):** The outcomes of the measurements of $\hat{M}$ on $|\psi\rangle$ represent a real valued random variable $M$. The mean value of $M$, noted $\langle M\rangle$, and its variance $\text{var}(M)$, are then given by
>
> $$
> \begin{aligned}
> \langle M\rangle &= \langle\psi|\hat{M}|\psi\rangle \\
> \text{Var}(M) &= \langle M^2\rangle - \langle M\rangle^2 = \langle\psi|\hat{M}^2|\psi\rangle - \langle\psi|\hat{M}|\psi\rangle^2.
> \end{aligned}
> \tag{10.8}
> $$

One could also consider a *statistical mixture* of a set of ensembles $\mathcal{E}_\alpha$, with weights $w_\alpha$ s.t. $\sum\limits_\alpha w_\alpha = 1$. Such a mixture can be achieved, for instance, by choosing $N_\alpha$ systems

---

[36]This is the discrete version of the spectral theorem; we only introduce the discrete version since, in this course, we focus on operators with discrete spectra. See section 2.1 in [180] for the continuous version.

from $\mathcal{E}_\alpha$; the weights $w_\alpha$ would then be $w_\alpha = N_\alpha/N$ with $N = \sum_\alpha N_\alpha$. The measurements of an observable $\hat{M}$ over the statistical mixture yields a random variable with an average $\langle M \rangle = \sum_\alpha w_\alpha \langle \psi_\alpha | \hat{M} | \psi_\alpha \rangle$.

The above description can be conveniently reformulated using the *density matrix*:

$$\hat{\rho} = \sum_\alpha w_\alpha |\psi_\alpha\rangle\langle\psi_\alpha| \tag{10.9}$$

We can then write

$$\langle M \rangle = \mathrm{Tr}\left[\hat{M}\hat{\rho}\right] \tag{10.10}$$

$$\mathrm{var}(M) = \mathrm{Tr}\left[\hat{M}^2\hat{\rho}\right] - \mathrm{Tr}\left[\hat{M}\hat{\rho}\right]^2. \tag{10.11}$$

A few properties of the density matrix:

- self-adjoint: $\hat{\rho}^\dagger = \hat{\rho}$;

- positive: $\hat{\rho} \geq 0$ (positive eigenvalues);

- normalization: $\mathrm{Tr}[\hat{\rho}] = 1$;

- $\mathrm{Tr}[\hat{\rho}^2] \leq \mathrm{Tr}[\hat{\rho}]$, with equality iff $\hat{\rho}$ is a *pure state* (i.e., $\hat{\rho} = |\psi\rangle\langle\psi|$).

Finally, notice that from (10.2) and (10.9), we deduce that, for a closed system, the density matrix evolves in time according to the equation

$$\hbar\frac{d\hat{\rho}(t)}{dt} = -i[\hat{H}(t), \hat{\rho}(t)]. \tag{10.12}$$

This is the Liouville or Liouville-von Neumann equation, whose solution is given by

$$\hat{\rho}(t) = \hat{U}(t, t_0)\hat{\rho}(t_0)\hat{U}^\dagger(t, t_0), \tag{10.13}$$

where $\hat{\rho}(t_0)$ is the initial state of the density matrix at time $t_0$ and where the propagator $\hat{U}(t, t_0)$ was given in (10.7).

*Remark: difference between quantum and classical statistics:* We point out that the formalism introduced above for quantum statistics automatically rules out deterministic variables, or dispersion free ensembles, i.e., ensembles such that every operator $\hat{X}$ is associated to a random variable $X$ with zero variance: $\mathrm{var}(X) = 0$ (which is possible for classical stochastic states). Indeed, let's assume that such an ensemble exists. Then, taking $\hat{M} = |\psi\rangle\langle\psi|$ with $|\psi\rangle$ a unit vector, the condition $\mathrm{var}(M) = 0$ becomes $\langle\psi|\hat{\rho}|\psi\rangle = \langle\psi|\hat{\rho}|\psi\rangle^2$, which implies that $\langle\psi|\hat{\rho}|\psi\rangle = 0$ or $\langle\psi|\hat{\rho}|\psi\rangle = 1$ for all unit vectors, hence $\hat{\rho} = 0$ or $\hat{\rho} = \mathbf{I}$, which is incompatible with the normalization property $\mathrm{Tr}[\hat{\rho}] = 1$.

**Tensor products**    We will be interested in studying the energy exchanges between a system and its environment. According to the postulate 1, the system and the environment can both be described within their own Hilbert space, what happens when there is a coupling between the two? The total Hilbert space of system and environment is described by the following postulate.

> **Postulate 5:** A set of different quantum systems defined in different Hilbert spaces is described within a global Hilbert space obtained by taking the *tensor product* of the individual Hilbert spaces.

The tensor product of two Hilbert spaces $\mathcal{H}_A$ and $\mathcal{H}_B$ is noted $\mathcal{H}_A \otimes \mathcal{H}_B$. It is a Hilbert space to which is associated a bilinear map

$$
\begin{aligned}
\mathcal{H}_A, \mathcal{H}_B &\quad \rightarrow \mathcal{H}_A \otimes \mathcal{H}_B \\
|\psi_A\rangle, |\psi_B\rangle &\quad \mapsto |\psi_A\rangle \otimes |\psi_B\rangle
\end{aligned}
\tag{10.14}
$$

Let's note $\mathcal{H}_{tot} = \mathcal{H}_A \otimes \mathcal{H}_B$. A basis for $\mathcal{H}_{tot}$ can be obtained from the bases of $\mathcal{H}_A$ and $\mathcal{H}_B$, respectively $\{|\phi_i^A\rangle\}_i$ and $\{|\phi_i^B\rangle\}_i$, by taking the tensor products of the basis vectors: the set $\{|\phi_i^A\rangle \otimes |\phi_j^B\rangle\}_{i,j}$ is a basis of $\mathcal{H}_{tot} = \mathcal{H}_A \otimes \mathcal{H}_B$. Any vector $|\psi\rangle$ in $\mathcal{H}_{tot}$ can therefore be written in the form $|\psi\rangle = \sum_{i,j} c_{i,j} |\phi_i^A\rangle \otimes |\phi_j^B\rangle$ with $c_{i,j} \in \mathbb{C}$.

Similarly, any operator acting on $\mathcal{H}_{tot}$ can be written as a linear combination of product operators of the form $\hat{M}_A \otimes \hat{M}_B$, where $\hat{M}_A, \hat{M}_B$ act respectively on $\mathcal{H}_A$ and $\mathcal{H}_B$. Such product operators act on product states like

$$
(\hat{M}_A \otimes \hat{M}_B)(|\psi_A\rangle \otimes |\psi_B\rangle) = (\hat{M}_A|\psi_A\rangle) \otimes (\hat{M}_B|\psi_B\rangle).
\tag{10.15}
$$

If the density matrix $\hat{\rho}$ of the total system+bath is of the form $\hat{\rho} = \hat{\rho}_A \otimes \hat{\rho}_B$, the system and bath are said to be *uncorrelated*. In this case, the expectation values of product operators are given by $\langle \hat{M}_A \otimes \hat{M}_B \rangle = \langle \hat{M}_A \rangle \langle \hat{M}_B \rangle$.

The *reduced density matrix* of the system or bath can be obtained from the total density matrix by performing a partial trace: $\hat{\rho}_A = \text{Tr}_B[\hat{\rho}]$ and $\hat{\rho}_B = \text{Tr}_A[\hat{\rho}]$.

**Separable states and entangled states**    An important feature of quantum mechanics is *entanglement*. To understand it, it is convenient to use the

**Schmidt decomposition theorem:** For any state $|\psi\rangle \in \mathcal{H}_A \otimes \mathcal{H}_B$, there exist orthogonal bases – the Schmidt bases – $\{|\chi_i^A\rangle\}$ and $\{|\chi_i^B\rangle\}$ of $\mathcal{H}_A$ and $\mathcal{H}_B$ respectively, such that

$$
|\psi\rangle = \sum_i c_i |\chi_i^A\rangle \otimes |\chi_i^B\rangle.
\tag{10.16}
$$

$\square$

Consider now a vector state $|\psi\rangle$ of $\mathcal{H}_{tot}$. If $|\psi\rangle$ can be written in the form $|\psi_A\rangle \otimes |\psi_B\rangle$, it is a *product state*; otherwise it is an *entangled* state. A statistical mixture of states of $\mathcal{H}_A \otimes \mathcal{H}_B$ is generically described, using the Schmidt decomposition, by a density matrix of the form,

$$
\hat{\rho} = \sum_\alpha w_\alpha \left( \sum_i c_i^\alpha |\chi_i^{A,\alpha}\rangle \otimes |\chi_i^{B,\alpha}\rangle \right) \left( \sum_j c_j^{\alpha*} \langle \chi_j^{A,\alpha}| \otimes \langle \chi_j^{B,\alpha}| \right).
\tag{10.17}
$$

A state is then said *separable* if $\hat{\rho}$ can be written as a probability distribution over uncorrelated states, i.e.,

$$
\hat{\rho} = \sum_\alpha w_\alpha \hat{\rho}_A^\alpha \otimes \hat{\rho}_B^\alpha.
\tag{10.18}
$$

Otherwise, the state is entangled. Notice that, if $\hat{\rho}$ is a pure state, i.e., $\hat{\rho} = |\psi\rangle\langle\psi|$ where $|\psi\rangle$ is the vector state of the system, then $\hat{\rho}$ is separable iff $|\psi\rangle$ is a product state. Entanglement is a purely quantum feature: entangled states lead to *quantum* correlation functions, which cannot be described using product states. On the other hand, separable states can yield classical correlations. For an illustration of these differences, see the Einstein-Podolsky-Rosen (EPR) paradox [193] and Bell's inequalities [194].

## 10.3 Quantum measurement and thermodynamic quantities

In this section, we introduce the notion of quantum measurement, and define thermodynamic quantities. Note that the theory of quantum measurement is still an active field of research, with fundamental questions regarding the nature of the wave function and decoherence yet to be answered. We will use the Copenhagen interpretation of quantum mechanics: the wave function is a mathematical object which yields probabilities for the outcomes of measurements, and measuring a system involves a brutal change ("collapse") of the wave function. Further reading about the content of this chapter: chapter 2 in [180], chapter 1 in [181].

### 10.3.1 The projection postulate

As explained in the previous section, the outcome of the measurement of an observable $\hat{M}$ is given by $\text{Tr}[\hat{M}\hat{\rho}]$. But performing a measurement on a quantum system also perturbs the system itself. This fact is formalized by the *projection postulate*:

> **Postulate 6 (projection postulate):** Consider the measurement of an observable $\hat{M}$, with a spectral decomposition $\hat{M} = \sum_n \mu_n \hat{\Pi}_n$, on a system described by $\hat{\rho}$.
>
> Then, the sub-ensemble of systems where the result $\mu_n$ was observed is described by the density matrix
>
> $$\hat{\rho}'(\mu_n) = \frac{\hat{\Pi}_n \hat{\rho} \hat{\Pi}_n}{\text{Tr}[\hat{\Pi}_n \hat{\rho}]} \ . \tag{10.19}$$
>
> $\hat{\rho}'(\mu_n)$ is called the *conditional* post measurement density matrix. We could choose to re-mix the post measurement matrices with the probability weights $p_n = \text{Tr}[\hat{\Pi}_n \hat{\rho}]$ of each possible outcome, to obtain the *unconditional* density matrix
>
> $$\hat{\rho}' = \sum_n p_n \hat{\rho}'(\mu_n) = \sum_n \hat{\Pi}_n \hat{\rho} \hat{\Pi}_n \ . \tag{10.20}$$

In general, $\hat{\rho}' \neq \hat{\rho}$. The equality holds, however, if $[\hat{M}, \hat{\rho}] = 0$.

*Remark:* It is possible to describe the quantum measurement (10.19) without resorting to the projection postulate, using auxiliary systems ("copies" of the system to measure); see chapter 1 in [181] for details.

### 10.3.2 Two-point measurement method

We now introduce the two-point measurement method, which is a convenient tool to study the variations and fluctuations of thermodynamic quantities. For further reading see [28].

Consider a (possibly time dependent) observable $\hat{M}(t) = \sum_n \mu_n(t) \hat{\Pi}_n(t)$. We now perform two measurements of $\hat{M}(t)$, at times $t = 0$ and $t$. The joint probability to observe the value $\mu_l(0)$ at $t = 0$ and $\mu_n(t)$ at $t$ is, using the Born postulate,

$$P[\mu_n(t), \mu_l(0)] = \text{Tr}[\hat{\Pi}_n(t)\hat{U}(t,0)\hat{\Pi}_l(0)\hat{\rho}(0)\hat{\Pi}_l(0)\hat{U}^\dagger(t,0)\hat{\Pi}_n(t)] \tag{10.21}$$

where $\hat{\rho}(0)$ is the density matrix of the system at time $t = 0$, and where we used the projection postulate combined with the evolution from the initial state obtained by integrating the Liouville equation (10.12). For energy observables, we will mainly be interested in the probability to observe a fluctuation $\Delta\mu$; this probability distribution is given by

$$p(\Delta\mu) = \sum_{\mu_l(0), \mu_n(t)} P[\mu_n(t), \mu_l(0)]\delta(\Delta\mu - (\mu_n(t) - \mu_l(0))) \ . \tag{10.22}$$

To study the statistics of $p(\Delta\mu)$, it is useful to study its Fourier transform, called the characteristic function:

$$G(\lambda, t) := \int_{-\infty}^{+\infty} e^{i\lambda\Delta\mu} p(\Delta\mu) \, d\Delta\mu \qquad (10.23)$$

where $\lambda \in \mathbb{R}$ is called a *counting field*. Note that $G(\lambda, t)$ is time dependent, since $\Delta\mu$ corresponds to a fluctuation observed between the times $t = 0$ and $t$; strictly speaking, we should write $G(\lambda, t, 0)$ or use a more general notation $G(\lambda, t_f, t_i)$ with $t_i, t_f$ the initial and final measurement times. The moments of $\Delta\mu$ are then given by the derivatives of $G(\lambda, t)$ in $\lambda = 0$:

$$\langle \Delta\mu^n \rangle = (-i)^n \partial_\lambda^n G(\lambda, t)|_{\lambda=0}. \qquad (10.24)$$

It is convenient to re-write $G(\lambda, t)$ as the trace of a "tilted" density matrix $\hat{\rho}^\lambda(t)$ [28], defined in the following way. First, we introduce the "dressed" evolution operator $\hat{U}_\lambda(t, 0)$ (the evolution operator $\hat{U}(t, 0)$ "dressed" with the counting field $\lambda$),

$$\hat{U}_\lambda(t, 0) := e^{i\hat{M}(t)\lambda/2} \hat{U}(t, 0) e^{-i\hat{M}(0)\lambda/2}. \qquad (10.25)$$

The tilted density matrix $\hat{\rho}^\lambda(t)$ is then defined as

$$\hat{\rho}^\lambda(t) \quad := \hat{U}_\lambda(t, 0) \hat{\bar{\rho}}(0) \hat{U}^\dagger_{-\lambda}(t, 0) \qquad (10.26)$$

where $\hat{\bar{\rho}}(0)$ is the diagonal part of $\hat{\rho}(0)$ in the eigenbasis of $\hat{M}$. Let's show that

$$G(\lambda, t) = \text{Tr}[\hat{\rho}^\lambda(t)]. \qquad (10.27)$$

Substituting Eq. (10.22) in Eq. (10.34), and using the fact that $\hat{\Pi}_n(t)^2 = \hat{\Pi}_n(t)$ and that $\sum_{\mu_n(t)} e^{i\lambda\mu_n(t)}\hat{\Pi}_n(t) = e^{i\lambda\hat{M}(t)}$, we obtain

$$G(\lambda, t) = \sum_{\mu_l(0), \mu_n(t)} e^{i\lambda(\mu_n(t)-\mu_l(0))} \text{Tr}[\hat{\Pi}_n(t)\hat{U}(t, 0)\hat{\Pi}_l(0)\hat{\rho}(0)\hat{\Pi}_l(0)\hat{U}^\dagger(t, 0)]$$

$$\qquad (10.28)$$

$$= \sum_{\mu_l(0)} e^{-i\lambda\mu_l(0)} \text{Tr}[e^{i\lambda\hat{M}(t)/2}\hat{\Pi}_n(t)\hat{U}(t, 0)\hat{\Pi}_l(0)\hat{\rho}(0)\hat{\Pi}_l(0)\hat{U}^\dagger(t, 0)e^{i\lambda\hat{M}(t)/2}]$$

Finally, using $\sum_{\mu_l(0)} e^{-i\lambda\mu_l(0)}\hat{\Pi}_l(0)\hat{\rho}(0)\hat{\Pi}_l(0) = e^{-i\lambda\hat{M}(0)/2}\hat{\bar{\rho}}(0)e^{-i\lambda\hat{M}(0)/2}$, we obtain the equality (10.27).

### 10.3.3   Heat, work and internal energy

Using the two point measurement with counting fields method presented above, we can study the fluctuations of the variations of heat, work and internal energy.

We consider the case of a quantum system $A$ coupled to $N$ baths. We allow the system Hamiltonian $\hat{H}_A(t)$ to be time dependent, to describe possible external forces, while the Hamiltonian of the baths reads $\sum_{\alpha=1}^{N} \hat{H}_\alpha$ with $\hat{H}_\alpha$ the free Hamiltonian of the $\alpha$-th bath. The total Hamiltonian is

$$\hat{H}(t) = \hat{H}_A(t) + \sum_{\alpha=1}^{N} \hat{H}_\alpha + \sum_{\alpha=1}^{N} \hat{V}_\alpha(t), \qquad (10.29)$$

where $\hat{V}(t) = \sum\limits_{\alpha=1}^{N} \hat{V}_\alpha(t)$ is the coupling Hamiltonian between the baths and the system. The total system consisting of the system $A$ and the baths is closed according to the definition in the postulate 3. Note that, rigorously speaking, we should write $\hat{H}_A(t) \otimes_\alpha \mathbf{I}_\alpha$ instead of $\hat{H}_A(t)$ in order to respect the dimensions, and similarly, add to $\hat{H}_\alpha$ the tensor products with the identity operators of the Hilbert spaces of $A$ and of the baths $\alpha \neq \alpha'$. To alleviate the notation, we omit the identity products throughout the rest of this course.

The system's bare energy changes, $\Delta E_A$, is by definition obtained by measuring the variations of $\hat{H}_A(t)$,

$$\Delta E_A := \mathrm{Tr}[\hat{H}_A(t)\hat{\rho}(t) - \hat{H}_A(0)\hat{\rho}(0)]. \tag{10.30}$$

On the other hand, the energy changes leaving bath $\eta$ between times 0 and $t$, is identified as the heat $Q_\eta$ transmitted by the bath $\eta$ to the system. It is given by measuring $\hat{H}_\eta$,

$$Q_\eta := \mathrm{Tr}[\hat{H}_\eta(\hat{\rho}(0) - \hat{\rho}(t))]. \tag{10.31}$$

Finally, the work exerted by external forces on the total system (system $A$ and baths) is by definition given by the variations of the total Hamiltonian $\hat{H}(t)$,

$$W := \mathrm{Tr}[\hat{H}(t)\hat{\rho}(t) - \hat{H}(0)\hat{\rho}(0)]. \tag{10.32}$$

We highlight that these definitions are compatible with the first law when additionally requiring that the coupling is switched on after the first measurement and switched off before the final measurement: $\hat{V}_\alpha(0) = \hat{V}_\alpha(t) = 0$. Indeed, in this case, the first law is satisfied

$$\Delta E_A = W + \sum_\eta Q_\eta. \tag{10.33}$$

We point out that there exist alternative definitions and methods of measurement of work in quantum thermodynamics; the discussion of these variations go beyond the scope of the course, we refer to [181, 195] for further reading.

**Measuring heat, work and internal energy**   Since $\hat{H}_A(t)$ and all the $\hat{H}_\alpha$ commute, we can measure them simultaneously and define the general characteristic function

$$G(t, \boldsymbol{\lambda}) = \mathrm{Tr}[\hat{\rho}^{\boldsymbol{\lambda}}(t)]; \quad \hat{\rho}^{\boldsymbol{\lambda}}(t) = \hat{U}_{\boldsymbol{\lambda}}(t,0)\hat{\bar{\rho}}(0)\hat{U}_{-\boldsymbol{\lambda}}^{\dagger}(t,0), \tag{10.34}$$

where

$$\hat{U}_{\boldsymbol{\lambda}}(t,0) = e^{i\boldsymbol{\lambda}\cdot\hat{\boldsymbol{H}}(t)/2}\hat{U}(t,0)e^{-i\boldsymbol{\lambda}\cdot\hat{\boldsymbol{H}}(0)/2}, \tag{10.35}$$

where $\hat{\boldsymbol{H}}(t) = (\hat{H}_A(t), \hat{H}_1, ..., \hat{H}_N)$ and $\boldsymbol{\lambda} = (\lambda_A, \lambda_1, ..., \lambda_N)$ respectively denote a vector of system-bath Hamiltonians and a vector of counting fields.

Let's now assume that the initial density matrix is diagonal in a joint eigenbasis of $\hat{H}_A(0), \hat{H}_1, ..., \hat{H}_N$. Such a basis exists since the Hamiltonians of the system and baths commute with each other. Then, we can replace $\hat{\bar{\rho}}(0) = \hat{\rho}(0)$ in (10.34).

To measure the system's bare energy changes, $\Delta E_A$, defined in (10.30), we choose $\lambda_A = \lambda$ and $\lambda_\alpha = 0$ for all $\alpha$, and we obtain the variation of $\hat{H}_A(t)$ between times 0 and $t$,

$$\Delta E_A = \frac{1}{i}\partial_\lambda G(t, \lambda, 0, ..., 0)|_{\lambda=0} = \mathrm{Tr}[\hat{H}_A(t)\hat{\rho}(t) - \hat{H}_A(0)\hat{\rho}(0)]. \tag{10.36}$$

For the heat $Q_\eta$ leaked by the bath $\eta$, defined in (10.31), we choose $\lambda_A = \lambda_\alpha = 0$ for all $\alpha \neq \eta$ and $\lambda_\eta = -\lambda$:

$$Q_\eta = \frac{1}{i}\partial_\lambda G\underbrace{(t, 0, 0, ..., -\lambda, 0, ..., 0)}_{-\lambda \text{ at position } \eta}|_{\lambda=0} = \mathrm{Tr}[\hat{H}_\eta(\hat{\rho}(0) - \hat{\rho}(t))] \tag{10.37}$$

Finally, to measure the work, defined in (10.32), we choose $\lambda_A = \lambda_\alpha = \lambda$,

$$W = \frac{1}{i}\partial_\lambda G(t, \lambda, ..., \lambda)|_{\lambda=0}$$

$$= \frac{1}{i}\partial_\lambda G(t, \lambda, 0, ..., 0)|_{\lambda=0} + \sum_\eta \partial_\lambda G \underbrace{(t, 0, 0, ..., -\lambda, 0, ..., 0)}_{\lambda \text{ at position } \eta}|_{\lambda=0} \qquad (10.38)$$

$$= \Delta E_A - \sum_\eta Q_\eta\,.$$

Again, recall that this approach is here justified when requiring that the coupling is switched on after the first measurement and switched off before the final measurement: $\hat{V}_\alpha(0) = \hat{V}_\alpha(t) = 0$.

### 10.3.4   Entropy

We conclude this section by introducing the entropy. In quantum mechanics, the (von Neumann) entropy of a system described by a density matrix $\hat{\rho}$ is defined as[37]

$$S(\hat{\rho}) = -k_B \text{Tr}[\hat{\rho} \log \hat{\rho}] \qquad (10.39)$$

Using the spectral decomposition $\hat{\rho} = \sum_j p_j |e_j\rangle\langle e_j|$, where $\{|e_j\rangle\}$ is an orthonormal eigenbasis of $\hat{\rho}$ and $\sum_j p_j = 1$, we can re-write

$$S(\hat{\rho}) = -k_B \sum_j p_j \log p_j\,, \qquad (10.40)$$

where we recognize the entropy (also called the Gibbs entropy) introduced in the Chapter 3 in the context of classical systems. $S(\hat{\rho})$ expresses the uncertainty, or lack of knowledge, about the realization of a certain state $|e_j\rangle$ in the mixture.

   Measuring a system increases the entropy. To see this, let us define $\Delta S = S(\hat{\rho}') - S(\hat{\rho})$. Using the projection postulate, we have

$$\Delta S = S\left(\sum_n \hat{\Pi}_n \hat{\rho} \hat{\Pi}_n\right) - S(\hat{\rho})$$

We now introduce the relative entropy: the relative entropy between two density matrices $\hat{\rho}_1, \hat{\rho}_2$ is by definition $D(\hat{\rho}_1||\hat{\rho}_2) = \text{Tr}[\hat{\rho}_1 \log \hat{\rho}_1] - \text{Tr}[\hat{\rho}_1 \log \hat{\rho}_2]$ [180]. It satisfies $D(\hat{\rho}_1||\hat{\rho}_2) \geq 0$. Hence,

$$0 \leq D(\hat{\rho}||\hat{\rho}') = -S(\hat{\rho}) - \text{Tr}[\hat{\rho} \log \hat{\rho}'] \qquad (10.41)$$

Using the fact that $[\hat{\Pi}_n, \hat{\rho}'] = 0$, we obtain

$$-\text{Tr}\left[\hat{\rho} \log \hat{\rho}'\right] = -\text{Tr}\left[\sum_n \hat{\Pi}_n \hat{\rho} \log(\hat{\rho}')\hat{\Pi}_n\right] = -\text{Tr}\left[\sum_n \hat{\Pi}_n \hat{\rho}\hat{\Pi}_n \log(\hat{\rho}')\right] = S(\hat{\rho}') \quad (10.42)$$

and hence $\Delta S \geq 0$.

---

[37]The von Neumann entropy can also be defined without the Boltzmann constant $k_B$.

## 10.4 Quantum fluctuation theorems

An important outcome of the stochastic thermodynamics theory is the discovery of *fluctuation theorems* [37]. Fluctuation theorems provide a symmetry between the probability to observe a certain fluctuation of heat or work in the forward process, and the probability to observe its opposite value in the reversed process. Initially identified for out of equilibrium classical systems, fluctuation theorems have been extended to quantum systems, where they take the form of exact symmetries satisfied by the fluctuations of thermodynamic quantities (e.g., work and heat currents) at the level of the unitary evolution [28,196]. For a review of classical and quantum fluctuation theorems, see e.g. [197].

In what follows, we derive detailed fluctuation theorems, which take the form of exact symmetries of the characteristic function, linking the fluctuating entropy of a given forward process with the one generated in its time reversed counterpart. To do so, we first need to define the reversed process; we will use the approach of [28]. In quantum mechanics, time reversal is defined using the time reversal operator $\Theta$, which satisfies $\Theta^2 = \mathbf{I}$, $\Theta i \Theta = -i$ (see appendix C in [181]). Any observable $\hat{M}$ is either even or odd under $\Theta$, i.e., $\Theta \hat{M} \Theta = \hat{M}$ (even) or $\Theta \hat{M} \Theta = -\hat{M}$ (odd). For instance, the position operator is even, while the momentum operator is odd.

Consider a closed system described by a Hamiltonian $\hat{H}(t)$. The Hamiltonian may contain odd components, such as spins or magnetic fields. We denote all these components by $\hat{\boldsymbol{B}}$, and write explicitly the dependence of in $\hat{\boldsymbol{B}}$ of the Hemiltonian: $\hat{H}(t, \hat{\boldsymbol{B}})$. We then obtain the relation

$$\Theta \hat{H}(\hat{\boldsymbol{B}}, t)\Theta = \hat{H}(-\hat{\boldsymbol{B}}, t). \tag{10.43}$$

Let's now consider a forward process, defined by the initial density matrix $\hat{\rho}(0)$ and the Hamiltonian $\hat{H}(t, \hat{\boldsymbol{B}})$, taking place between the times $t = 0$ and $t = t_f$. As we saw, the postulate 3 implies that the evolution of the density matrix $\hat{\rho}(t)$ follows the Liouville equation (10.12), and is determined by the propagator (10.7). The reversed process is defined in a similar way [28,198]: the propagator of the reversed process, noted $\hat{U}^R(0, t, \hat{\boldsymbol{B}})$, is defined as the operator satisfying

$$i\hbar \partial_t \hat{U}^R(t, 0, -\hat{\boldsymbol{B}}) \quad = \hat{H}(-\hat{\boldsymbol{B}}, t_f - t)\hat{U}^R(t, 0, -\hat{\boldsymbol{B}})$$

$$\hat{U}^R(0, 0, \hat{\boldsymbol{B}}) \qquad = \mathbf{I}. \tag{10.44}$$

It can be shown [198] that the propagators of the forward and backward processes are related by

$$\hat{U}^R(t, 0, -\hat{\boldsymbol{B}}) = \Theta \hat{U}(t_f - t, t_f, \hat{\boldsymbol{B}})\Theta. \tag{10.45}$$

In particular, for $t = t_f$, we have

$$\hat{U}^R(t, 0, -\hat{\boldsymbol{B}}) = \Theta \hat{U}(0, t, \hat{\boldsymbol{B}})\Theta = \Theta \hat{U}^\dagger(t, 0, \hat{\boldsymbol{B}})\Theta. \tag{10.46}$$

We then define the density matrix of the reversed process as

$$\Theta \hat{\rho}^R(t)\Theta := \hat{U}^R(t, 0, -\hat{\boldsymbol{B}})\Theta \hat{\rho}^R(0)\Theta \hat{U}^{R\dagger}(t, 0, -\hat{\boldsymbol{B}}). \tag{10.47}$$

Multiplying (10.47) left and right by $\Theta$ and using (10.46), we obtain

$$\hat{\rho}^R(t) = \hat{U}^\dagger(t, 0, \hat{\boldsymbol{B}})\hat{\rho}^R(0)\hat{U}(t, 0, \hat{\boldsymbol{B}}). \tag{10.48}$$

Following the same path for the tilted dynamics with counting fields, we can identify the characteristic function for the reversed process as [28]

$$G^R(t, \boldsymbol{\lambda}) = \text{Tr}[\hat{\rho}^{R\boldsymbol{\lambda}}(t)] = \text{Tr}\left[\hat{U}^\dagger_{\boldsymbol{\lambda}}(t, 0)\bar{\hat{\rho}}^R_0 \hat{U}_{-\boldsymbol{\lambda}}(t, 0)\right]. \tag{10.49}$$

Now that we have the characteristic function for the reversed process, we may derive quantum fluctuation theorems.

### 10.4.1   Closed systems

We consider first the case of closed systems, i.e., systems exchanging energy but not particles with their environment. In this subsection, the closed system consists of the quantum system described by $\hat{H}_A(t)$ and of the baths $\hat{H}_\alpha$. While the system Hamiltonian $\hat{H}_A(t)$ may be time dependent, we assume that the bath Hamiltonians $\hat{H}_\alpha$ are time independent. We mention that this model allows to study, e.g., quantum refrigerators or quantum heat engines [181].

**Fluctuation theorem**   Let's assume that the initial density matrices of the forward and time reversed processes are given by Gibbs states, that is

$$
\begin{aligned}
\hat{\rho}(0) &= \frac{e^{-\beta_A \hat{H}_A(0)}}{Z_A(0)} \bigotimes_\alpha \frac{e^{-\beta_\alpha \hat{H}_\alpha}}{Z_\alpha} \\[2mm]
\hat{\rho}^R(0) &= \frac{e^{-\beta_A \hat{H}_A(t)}}{Z_A(t)} \bigotimes_\alpha \frac{e^{-\beta_\alpha \hat{H}_\alpha}}{Z_\alpha}
\end{aligned}
\tag{10.50}
$$

where $\beta_A, \beta_\alpha$ are the inverse of the temperatures of the system $A$ and the baths $\alpha$, and with $Z_A(t) = \mathrm{Tr}_A[e^{-\beta_A \hat{H}_A(t)}]$ and $Z_\alpha = \mathrm{Tr}_\alpha[e^{-\beta_\alpha \hat{H}_\alpha}]$. Then, we have the following detailed fluctuation theorem

$$
G^R(t, -\boldsymbol{\lambda} + i\boldsymbol{\beta}) = G(t, \boldsymbol{\lambda}) \frac{Z_A(0)}{Z_A(t)} = G(t, \boldsymbol{\lambda}) e^{\beta_A \Delta F_{eq}},
\tag{10.51}
$$

with inverse temperatures $\boldsymbol{\beta} = (\beta_A, \beta_1, ..., \beta_N)$ and $\Delta F_{eq} = F_{eq}(t) - F_{eq}(0)$ where the equilibrium free energy of the system is $F_{eq}(t) = -\frac{1}{\beta_A} \log Z_A(t)$. To see this, we begin by writing

$$
\begin{aligned}
G(t, \boldsymbol{\lambda}) &= \tfrac{1}{Z(0)} \mathrm{Tr}[\hat{U}(t,0) e^{-i\boldsymbol{\lambda}\cdot\hat{\boldsymbol{H}}(0)} e^{-\boldsymbol{\beta}\cdot\hat{\boldsymbol{H}}(0)} \hat{U}^\dagger(t,0) e^{i\boldsymbol{\lambda}\cdot\hat{\boldsymbol{H}}(t)}], \\[2mm]
G^R(t, \boldsymbol{\lambda}) &= \tfrac{1}{Z(t)} \mathrm{Tr}[\hat{U}^\dagger(t,0) e^{-i\boldsymbol{\lambda}\cdot\hat{\boldsymbol{H}}(t)} e^{-\boldsymbol{\beta}\cdot\hat{\boldsymbol{H}}(t)} \hat{U}(t,0) e^{i\boldsymbol{\lambda}\cdot\hat{\boldsymbol{H}}(0)}],
\end{aligned}
\tag{10.52}
$$

with $Z(t) = Z_A(t) \prod_\alpha Z_\alpha$. Replacing $\boldsymbol{\lambda}$ by $-\boldsymbol{\lambda} + i\boldsymbol{\beta}$ in $G^R(t, \boldsymbol{\lambda})$, we obtain

$$
G^R(t, -\boldsymbol{\lambda} + i\boldsymbol{\beta}) = \frac{1}{Z(t)} \mathrm{Tr}[\hat{U}^\dagger(t,0) e^{i\boldsymbol{\lambda}\cdot\hat{\boldsymbol{H}}(t)} \hat{U}(t,0) e^{-i\boldsymbol{\lambda}\cdot\hat{\boldsymbol{H}}(0)} e^{-\boldsymbol{\beta}\cdot\hat{\boldsymbol{H}}(0)}].
\tag{10.53}
$$

The last factor in the trace is exactly the unnormalized initial state of the forward evolution, given in (10.50). Hence,

$$
G^R(t, -\boldsymbol{\lambda} + i\boldsymbol{\beta}) = \frac{Z_A(0)}{Z_A(t)} \mathrm{Tr}[\hat{U}^\dagger(t,0) e^{i\boldsymbol{\lambda}\cdot\hat{\boldsymbol{H}}(t)} \hat{U}(t,0) e^{-i\boldsymbol{\lambda}\cdot\hat{\boldsymbol{H}}(0)} \hat{\rho}(0)],
\tag{10.54}
$$

and the result follows. Note that the fluctuation theorem (10.51) is valid at all times.

**Crooks and Jarzynski relations**   The fluctuation theorem (10.51) is very general, and allows to obtain special symmetries for the fluctuations of different quantities (heat, work, internal energy). In particular, we can use (10.51) to derive two celebrated results, the Crooks fluctuation theorem and the Jarzynski equality.

The Crooks fluctuation theorem, initially derived in the context of classical statistical mechanics [40] and later extended to quantum systems [199], is a work fluctuation theorem. It states that, given a state initially at thermal equilibrium at temperature $\beta^{-1}$, the ratio between the probability $p(W)$ to observe a work fluctuation $W$ in the forward process and the probability $p^R(-W)$ to observe the reverse fluctuation in the reversed process

is equal to $e^{\beta(W-\Delta F_{eq})}$, where $\Delta F_{eq}$ is the free energy difference between the initial and final equilibrium states. The Jarzynski equality follows immediately from the Crooks fluctuation theorem, although historically it was derived first [42].

To derive these relations from (10.51), we consider the case of a single heat bath $\eta$ at temperature $\beta_\eta^{-1}$ and assume that $\beta_A = \beta_\eta$. As explained previously, to measure the work, we choose the counting fields $\lambda_A = \lambda_\eta =: \lambda$, as in (10.38),

$$W = \frac{1}{i}\partial_\lambda G(t, \boldsymbol{\lambda})|_{\lambda=0} = \text{Tr}[\hat{H}_0(t)\hat{\rho}(t) - \hat{H}_0(0)\hat{\rho}(0)] \tag{10.55}$$

where $\hat{H}_0(t) = \hat{H}_A(t) + \hat{H}_\eta$. Using (10.23), we may write the probability distribution of the work as

$$p(W) = \int_{-\infty}^{+\infty} d\lambda e^{-i\lambda W} G(t, \boldsymbol{\lambda}). \tag{10.56}$$

Using (10.51) and (10.23) again, we obtain

$$\begin{aligned} p(W) &= \int_{-\infty}^{+\infty} d\lambda \int_{-\infty}^{+\infty} dW' e^{-\beta_A \Delta F_{eq}} e^{i(-\lambda + i\beta_A)W'} p^R(W') \\ &= p^R(-W) e^{\beta_A(W - \Delta F_{eq})} \end{aligned} \tag{10.57}$$

which is the Crooks relation. Taking the average, we obtain the Jarzynski equality [42].

$$\langle e^{-\beta_A W} \rangle = \int_{-\infty}^{+\infty} dW e^{-\beta_A W} p(W) = e^{-\beta_A \Delta F_{eq}}. \tag{10.58}$$

**Second law of thermodynamics**   An integral fluctuation theorem and the second law of thermodynamics can be obtained using an identity similar to the fluctuation theorem (10.51). Let's introduce the entropy production $\Sigma$, corresponding to changes in the quantity $\hat{S}(t) + \sum_\alpha \beta_\alpha \hat{H}_\alpha$, where $\hat{S}(t) = -\log \hat{\rho}_A(t)$ is the operator which, once measured, gives the system's entropy, and where $\beta_\alpha$ is the inverse of the temperature of the bath $\alpha$. We point out that this definition of $\Sigma$ corresponds to the entropy production only when the system is coupled to ideal heat baths with fixed temperatures. Further discussion of the notion of entropy production in more general setups go beyond the scope of this course; for a more thorough discussion see [181] or [200].

Assuming this time that the initial density matrices satisfy

$$\begin{aligned} \hat{\rho}(0) &= \hat{\rho}_A(0) \bigotimes_\alpha \frac{e^{-\beta_\alpha \hat{H}_\alpha}}{Z_\alpha} \\ \hat{\rho}^R(0) &= \hat{\rho}_A(t) \bigotimes_\alpha \frac{e^{-\beta_\alpha \hat{H}_\alpha}}{Z_\alpha}, \end{aligned} \tag{10.59}$$

where $\hat{\rho}_A(t)$ is the final state of the system in the forward dynamics, we obtain the following identity

$$G_\Sigma^R(t, -\lambda_\Sigma + i) = G_\Sigma(t, \lambda_\Sigma). \tag{10.60}$$

The identity (10.60) allows to derive an *integral* fluctuation theorem for the entropy production (of the forward dynamics): setting $\lambda_\Sigma = i$, we obtain

$$G_\Sigma(t, i) = \langle e^{-\Sigma} \rangle = 1, \tag{10.61}$$

where the brackets denote an ensemble averaging: $G_\Sigma(\lambda_\Sigma, t) = \int d\Sigma e^{i\lambda_\Sigma \Sigma} P(\Sigma)$. The convexity of the exponential function in (10.61) yields the second law

$$\langle \Sigma \rangle \geq 0 \tag{10.62}$$

**First law of thermodynamics at the fluctuating level** We conclude this section on closed systems with a comment on energy conservation. Let's assume that the total Hamiltonian $\hat{H}$ is time independent. In this case, the average work performed on the system vanishes, as can be seen from (10.55), and the first law writes

$$\Delta E_A - \sum_\eta Q_\eta = 0\,. \tag{10.63}$$

One can then express the requirement for the first law to be valid at the fluctuations level, i.e., for fluctuations in $W$ to vanish, in terms of the symmetry

$$\hat{\rho}^{\boldsymbol{\lambda}+\chi\mathbf{1}}(t) = \hat{\rho}^{\boldsymbol{\lambda}}(t) \tag{10.64}$$

for all times, $\mathbf{1} = (1, 1..., 1)$, $\chi \in \mathbb{R}$. It is clear that (10.64) guaranties the first law on average, as can be seen by setting $\lambda = 0$ and taking the derivative in $\chi$, as in (10.38). The symmetry (10.64) also guaranties that the fluctuations of the system energy correspond exactly to fluctuations of the bath: every fluctuation of the bath counted by the counting field $\chi$ on $\hat{H}_B$ is exactly compensated by the fluctuations of the system, also counted with the counting field $\chi$ on $\hat{H}_A$. This holds true, for instance, when, at all times $t$,

$$\left[\sum_\alpha \hat{V}_\alpha(t), \hat{H}_A + \sum_\alpha \hat{H}_\alpha\right] = 0 \tag{10.65}$$

i.e. when there is a strict energy conservation between the bath and the system.

### 10.4.2 Open systems: quantum master equations

This section is more focused on recent research, specifically [182].

In practice, quantum systems we are interested in are often *open*, i.e., they can exchange energy and particles with the environment. We then focus on the thermodynamics at the level of the quantum system described by $\hat{H}_A(t)$, while the baths $\hat{H}_\alpha$ now constitute the environment; more specifically, we wish to trace out the degrees of freedom of the baths in order to obtain a description of the reduced density matrix $\hat{\rho}_A := \mathrm{Tr}_B[\hat{\rho}]$, where $B$ denotes the Hilbert spaces of all the baths $\alpha$. The evolution of the system's reduced density matrix then does not obey the Liouville equation in general, but can be conveniently described using a quantum master equation [180, 201–203], obtained by tracing out the baths, and performing approximations which we will discuss here. The approximations performed during the derivation of a quantum master equation could break the fluctuation theorem (10.51), leading to a master equation which fails to accurately describe the thermodynamics of the system $A$. A *thermodynamically consistent* quantum master equation should satisfy the fluctuation theorems valid at the unitary level. The purpose of this subsection is to explain under which conditions the thermodynamic consistency is preserved.

**Tilted quantum master equations** In order to trace out the degrees of freedom of the baths, we first decompose the initial state as $\hat{\rho}(0) = \hat{\rho}_A(0) \otimes \sum_\nu \eta_\nu |\nu\rangle\langle\nu|$, where $|\nu\rangle$ are eigenvectors of the baths. The tilted density matrix in (10.34) becomes

$$\hat{\rho}_A^{\boldsymbol{\lambda}}(t) = \sum_{\mu,\nu} \hat{W}_{\mu,\nu}^{\boldsymbol{\lambda}}(t,0)\hat{\rho}_A(0)\hat{W}_{\mu,\nu}^{\boldsymbol{\lambda}\dagger}(t,0) =: \mathcal{M}_{\boldsymbol{\lambda}}(t,0)\hat{\rho}_A(0), \tag{10.66}$$

where $\hat{W}_{\mu,\nu}^{\boldsymbol{\lambda}} = \sqrt{\eta_\nu}\langle\mu|\hat{U}_{\boldsymbol{\lambda}}(t,0)|\nu\rangle$. The operators $\hat{W}_{\mu,\nu}$ are a set of *Kraus operators*. Kraus operators are used to describe quantum operations between quantum states which preserve

the positivity of the density matrix. Such operations are called *quantum maps*; a quantum map $\mathcal{M}$ is a completely positive operators, such that there exists a set of Kraus operators $\hat{W}_j$ satisfying $\mathcal{M}(\hat{\rho}) = \sum_j \hat{W}_j \hat{\rho} \hat{W}_j^\dagger$. The theory of quantum maps is beyond the scope of this course; for more details on this topic, see subsection 3.2 in [180].

We then perform the Markov approximation, also called the *semigroup hypothesis* in the context of quantum maps [180]: $\mathcal{M}_{\boldsymbol{\lambda}}(t, 0) = \mathcal{M}_{\boldsymbol{\lambda}}(t, s)\mathcal{M}_{\boldsymbol{\lambda}}(s, 0)$. Under this assumption, we obtain a time local equation of the form

$$d_t \hat{\rho}_A^{\boldsymbol{\lambda}}(t) = \lim_{\delta \to \delta_0} \frac{1}{\delta}(\mathcal{M}_{\boldsymbol{\lambda}}(t + \delta, t) - \mathbf{I})\hat{\rho}_A^{\boldsymbol{\lambda}}(t) =: \mathcal{L}_{\boldsymbol{\lambda}}(t)\hat{\rho}_A^{\boldsymbol{\lambda}}(t), \tag{10.67}$$

with $\hat{\rho}_A^{\boldsymbol{\lambda}}(0) = \hat{\rho}_A(0)$. Notice that the time increment $\delta$ is not taken to zero, but instead to a finite time $\delta_0$, called *coarse graining time*. The coarse graining time is chosen to be larger than the relaxation time of the bath and smaller than the relaxation time of the system. We will assume further on that the limit in (10.67) exists, and that the resulting equation does not explicitly depend on $\delta_0$. Let's now restrict ourselves to the case where the counting fields are put solely on the baths, $\boldsymbol{\lambda} = (0, \boldsymbol{\lambda_B})$. Then, the generator assumes the general form

$$\mathcal{L}_{0,\boldsymbol{\lambda}_B}(t)\hat{\rho}_A^{0,\boldsymbol{\lambda}_B} = -i[\hat{H}'_{0,\boldsymbol{\lambda_B}}(t), \hat{\rho}_A^{0,\boldsymbol{\lambda}_B}] + \mathcal{D}_{0,\boldsymbol{\lambda}_B}(t)\hat{\rho}_A^{0,\boldsymbol{\lambda}_B}. \tag{10.68}$$

For a detailed derivation, see the supplemental material in [182]. To alleviate the notations, we dropped the $t$ dependence of $\hat{\rho}_A^{0,\boldsymbol{\lambda_B}}(t)$. The term $\hat{H}'_{0,\boldsymbol{\lambda_B}}(t)$ is the sum of the system Hamiltonian $\hat{H}_A$ and of a Lamb shift contribution $\hat{H}_{LS}^{0,\boldsymbol{\lambda_B}}$ – a shift in the system's energies induced by the interaction with the bath – and $\mathcal{D}_{0,\boldsymbol{\lambda_B}}(t)$ describes the dissipation. The dissipator $\mathcal{D}_{0,\boldsymbol{\lambda_B}}(t)$ can generically be written as the sum of an anticommutator and a jump term $\mathcal{J}$,

$$\mathcal{D}_{0,\boldsymbol{\lambda}_B}(t)\hat{\rho}_A^{0,\boldsymbol{\lambda}_B} = \{\hat{G}_{0,\boldsymbol{\lambda}_B}, \hat{\rho}_A^{0,\boldsymbol{\lambda}_B}\} + \mathcal{J}_{0,\boldsymbol{\lambda}_B}(t)\hat{\rho}_A^{0,\boldsymbol{\lambda}_B}. \tag{10.69}$$

We may repeat the same derivation for the time reversed tilted density matrix $\hat{\rho}^{R\boldsymbol{\lambda}}(t)$ given in (10.49), and we obtain

$$\hat{\rho}_A^{R\boldsymbol{\lambda}}(t) = \sum_{\mu,\nu} \hat{W}_{\mu,\nu}^{R\boldsymbol{\lambda}}(t, 0)\hat{\rho}_A^R(0)\hat{W}_{\mu,\nu}^{R\boldsymbol{\lambda}\,\dagger}(t, 0), \tag{10.70}$$

where the Kraus operators are $\hat{W}_{\mu,\nu}^{R\boldsymbol{\lambda}} = \sqrt{\eta_\nu}\langle\mu|\hat{U}_{\boldsymbol{\lambda}}^\dagger(t, 0)|\nu\rangle$.

For the rest of this section, we assume that the system Hamiltonian $\hat{H}_A$ is time-independent.

**Generalized quantum detailed balance condition**    Notice that the Kraus operators of the time reversed dynamics, given in (10.70), obey the symmetry

$$\hat{W}_{\mu,\nu}^{R\boldsymbol{\lambda}} = e^{i\lambda_A \hat{H}_A}(\hat{W}_{\nu,\mu}^{(\lambda_A, \boldsymbol{\lambda}_B + i\boldsymbol{\beta}_B)})^\dagger e^{-i\lambda_A \hat{H}_A}. \tag{10.71}$$

Together with the semigroup hypothesis, the identity (10.71) leads to a sufficient condition for a master equation to satisfy the fluctuation theorems (10.51) and (10.60), which reads

$$\mathcal{L}_{0,-\boldsymbol{\lambda}_B}^\dagger[...] = \mathcal{L}_{0,-\boldsymbol{\lambda}_B + i\boldsymbol{\beta}_B}^R[...], \tag{10.72}$$

where $\boldsymbol{\beta}_B = (\beta_1, ..., \beta_N)$, and where the $\dagger$ on the r.h.s. denotes the *adjoint*: the adjoint $\mathcal{O}^\dagger$ of a superoperator $\mathcal{O}$ as the one fulfilling $\text{Tr}[(\mathcal{O}(X))^\dagger Y] = \text{Tr}[X^\dagger \mathcal{O}^\dagger(Y)]$ for all operators $X, Y$. To see this, we begin by noticing that the property

$$\hat{W}_{\mu\nu}^{\boldsymbol{\lambda}}(t, 0) = e^{i\lambda_A \hat{H}_A/2}\hat{W}_{\mu\nu}^{0,\boldsymbol{\lambda}_B}(t, 0)e^{-i\lambda_A \hat{H}_A/2} \tag{10.73}$$

of the Kraus operators gives, by replacing in (10.66),

$$\hat{\rho}_A^{\boldsymbol{\lambda}}(t) = e^{i\frac{\lambda_A}{2}\hat{H}_A}e^{t\mathcal{L}_{0,\boldsymbol{\lambda}_B}}[e^{-i\frac{\lambda_A}{2}\hat{H}_A}\bar{\hat{\rho}}_A(0)e^{-i\frac{\lambda_A}{2}\hat{H}_A}]e^{i\frac{\lambda_A}{2}\hat{H}_A}. \tag{10.74}$$

Similarly,

$$\hat{\rho}_A^{R\boldsymbol{\lambda}}(t) = e^{i\frac{\lambda_A}{2}\hat{H}_A}e^{t\mathcal{L}_{0,\boldsymbol{\lambda}_B}^R}[e^{-i\frac{\lambda_A}{2}\hat{H}_A}\bar{\hat{\rho}}_A^R(0)e^{-i\frac{\lambda_A}{2}\hat{H}_A}]e^{i\frac{\lambda_A}{2}\hat{H}_A}. \tag{10.75}$$

Let's now assume that the initial density matrices satisfy

$$\begin{aligned} \hat{\rho}_A(0) &= \frac{e^{-\beta_A\hat{H}_A}}{Z_A} \\[6pt] \hat{\rho}_A^R(0) &= \frac{e^{-\beta_A\hat{H}_A}}{Z_A}. \end{aligned} \tag{10.76}$$

Then,

$$\begin{aligned} G(t,\boldsymbol{\lambda}) &= Tr[\hat{\rho}_A^{\boldsymbol{\lambda}}(t)] \\ &= Tr\left[e^{i\lambda_A\hat{H}_A}e^{t\mathcal{L}_{0,\boldsymbol{\lambda}_B}}\left(e^{-i\lambda_A\hat{H}_A}\hat{\rho}_A(0)\right)\right], \end{aligned} \tag{10.77}$$

and, replacing $\boldsymbol{\lambda} \to -\boldsymbol{\lambda} + i\boldsymbol{\beta}$ in (10.75) and using the definition of the adjoint,

$$\begin{aligned} G^R(t,-\boldsymbol{\lambda}+i\boldsymbol{\beta}) &= Tr[\hat{\rho}_A^{R,-\boldsymbol{\lambda}+i\boldsymbol{\beta}}(t)] \\ &= Tr\left[e^{-i\lambda_A\hat{H}_A}e^{-\beta_A\hat{H}_A}e^{t\mathcal{L}_{0,-\boldsymbol{\lambda}_B+i\boldsymbol{\beta}_B}^R}\left(e^{i\lambda_A\hat{H}_A}/Z_A\right)\right] \\ &= Tr\left[\left(e^{t\mathcal{L}_{0,-\boldsymbol{\lambda}_B+i\boldsymbol{\beta}_B}^{R\dagger}}\left(e^{i\lambda_A\hat{H}_A}\hat{\rho}_A(0)\right)\right)^\dagger e^{i\lambda_A\hat{H}_A}\right]. \end{aligned} \tag{10.78}$$

Hence, the symmetry (10.51) is satisfied if

$$\begin{aligned} e^{t\mathcal{L}_{0,\boldsymbol{\lambda}_B}}\left(e^{-i\lambda_A\hat{H}_A}\hat{\rho}_A(0)\right) &= \left(e^{t\mathcal{L}_{0,-\boldsymbol{\lambda}_B+i\boldsymbol{\beta}_B}^{R\dagger}}(e^{i\lambda_A\hat{H}_A}\hat{\rho}_A(0))\right)^\dagger \\ \iff \left(e^{t\mathcal{L}_{0,\boldsymbol{\lambda}_B}}(e^{-i\lambda_A\hat{H}_A}\hat{\rho}_A(0))\right)^\dagger &= e^{t\mathcal{L}_{0,-\boldsymbol{\lambda}_B+i\boldsymbol{\beta}_B}^{R\dagger}}(e^{i\lambda_A\hat{H}_A}\hat{\rho}_A(0)) \\ \iff e^{t\mathcal{L}_{0,-\boldsymbol{\lambda}_B}}\left(e^{i\lambda_A\hat{H}_A}\hat{\rho}_A(0)\right) &= e^{t\mathcal{L}_{0,-\boldsymbol{\lambda}_B+i\boldsymbol{\beta}_B}^{R\dagger}}(e^{i\lambda_A\hat{H}_A}\hat{\rho}_A(0)), \end{aligned} \tag{10.79}$$

where we used the fact that

$$\left(e^{t\mathcal{L}_{0,\boldsymbol{\lambda}_B}}(e^{-i\lambda_A\hat{H}_A}\hat{\rho}_A(0))\right)^\dagger = e^{t\mathcal{L}_{0,-\boldsymbol{\lambda}_B}}(e^{i\lambda_A\hat{H}_A}\hat{\rho}_A(0)), \tag{10.80}$$

as can be seen using the expression (10.66). The condition (10.79) is then satisfied if $\mathcal{L}_{0,-\boldsymbol{\lambda}_B}^\dagger[...] = \mathcal{L}_{0,-\boldsymbol{\lambda}_B+i\boldsymbol{\beta}_B}^R[...]$, which is what we wanted to prove.

The condition (10.72) is called the *generalized quantum detailed balance condition* [182].

The proof of the symmetry (10.60) for the entropy production follows the same steps, but this time the only assumption is that $\hat{\rho}_A(0) = \hat{\rho}_A^R(t)$ and $\hat{\rho}_A^R(0) = \hat{\rho}_A(t)$.

**Strict and average energy conservation for quantum master equations**    We conclude this section with a brief discussion on how the strict energy balance condition (10.64) translates for quantum master equations. Let us note $\mathcal{L}_{\boldsymbol{\lambda}} := \mathcal{L}_{\lambda_A,\boldsymbol{\lambda}_B}$. The equivalent of (10.64) then writes

$$\mathcal{L}_{\boldsymbol{\lambda}}[...] = \mathcal{L}_{\boldsymbol{\lambda}+\chi\mathbf{1}}[...]. \tag{10.81}$$

This relation reveals the connection between invariance properties of the quantum master equation and the symmetries of the generating function [12, 13, 204]. However, we highlight that despite the resemblance between (10.81) and (10.64), the former is in fact less restrictive: indeed, (10.81) is equivalent to imposing (10.64) at every time intervals $\delta_0$, but not at all times.

*Remark:* In practice, many quantum master equations do not meet the strict energy conservation condition (10.81) or the generalized quantum detailed balance condition (10.72). In fact, in order to satisfy both these conditions, it is necessary to perform the secular approximation (we refer to the section 3.3 in [180] for the definition of the secular approximation); see [182] for a detailed proof. However, it is possible to relax the strict energy conservation condition (10.64) and to maintain a thermodynamic consistency on average, provided that the generalized detailed balance condition (10.72) is satisfied. This is for instance the case in the weak coupling limit, as illustrated in [182].

## 10.5   Conclusion and perspectives

In this introduction to quantum thermodynamics course, we covered the basic concepts underlying the theory of quantum statistics and of quantum measurement. We introduced the two-point measurement method, a relatively modern tool still actively used in research in the field of open quantum systems. The core of the physical discussion lies in the detailed fluctuation theorems (10.51) and (10.60). These two theorems are very general, and are valid at all times, not only in the steady state regime. These theorems are formulated using the two point measurement method with counting fields, and take the form of symmetries of the characteristic function. An appeal of this approach is that it can be easily used as a criteria to test the thermodynamic consistency of quantum master equations obtained by tracing out the degrees of freedom of the environment: the counting fields keep track of the energy exchanges occurring at the microscopic level, hence preserving the symmetries at the coarse grained level ensures that the energy exchanges are correctly accounted for by the quantum master equation.

For the discussion on open quantum systems, we have limited ourselves to cases where the Markov approximation applies. However, the framework can be extended beyond this approximation, for instance in repeated interaction setups [205] or in the context of thermalizing scattering maps [206].

Let us also mention that the "work" was introduced here as a projective measurement of the total Hamiltonian. This approach is not always appropriate. For instance, there is an alternative definition of what constitutes a work source, introduced in the context of repeated interactions [205], which is based on a purely thermodynamic criteria: a work source is a system which transmits energy without changing its entropy. Other definitions are reviewed and compared in [195]. These variations go beyond the scope of this course, and in fact, discussions on what should be a definition of work in quantum mechanics are still active.

# 11    Why "(Post)Modern Thermodynamics"?

**Alberto Garilli, Emanuele Penocchio, and Matteo Polettini as eXtemporanea.**
*A moment of collective reflection about fundamental questions in and on thermodynamics.*

## 11.1   Introduction

The last moment of the school (P)MT, also participated by the attendees of the ensuing workshop, was not a frontal lecture but a (semi-serious) structured open discussion. The ground for it was prepared through a survey among participants and by an informal initiative during the lunch break just before the lecture. The discussion was conducted by one of the organizers and ended with a slide presentation undisclosing the rationale behind "Postmodern Thermodynamics" as title of the school and workshop. Finally, after the event we had a second survey among participants to ask their appreciation of the moment. Here we describe the actions, summarize the discussion and the presentation, and draw some conclusions.

The initiative was inspired by similar activities by eXtemporanea, an inclusive collective of students, researchers and activists, operating mainly in Italy, and interested in experimenting new forms of dialogue beyond the dichotomy science vs. society.

## 11.2   Entry survey & breaking the ice

In preparation to the public discussion, in the first two days of the school we launched an online survey, only open to the participants to the school, about the status, perception and role of the laws of thermodynamics. However, by mistake participants were initially directed to the master file rather than to the answer form, and some of them had fun hacking it in interesting ways. Questions were:

- How is thermodynamics linked to the industrial revolution?

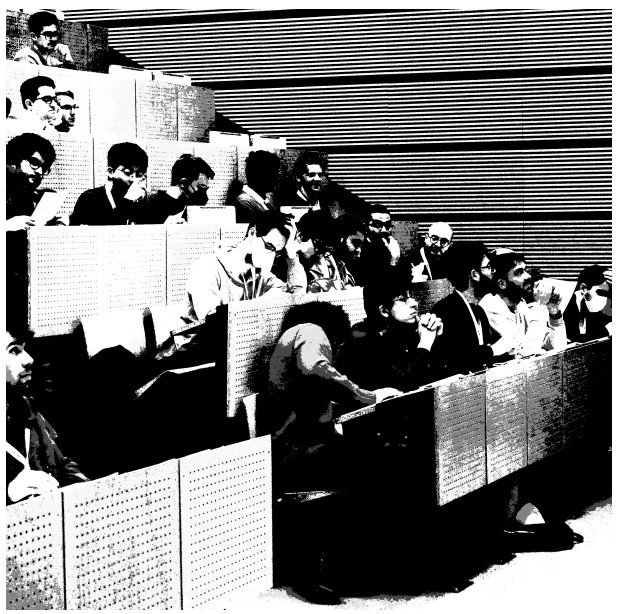

Figure 11.1: A moment at (P)MT.

- The zeroth/first/second/third law is (principles, facts, definitions, etc.)

- Which is the most important past experiment in thermodynamics?

- What would be an important future experiment in thermodynamics?

- How would you define yourself philosophically?

- Thermodynamics is. . .

- Thermodynamics should be. . .

The answers were analyzed overnight. Some initial considerations were:

- Fancy options and answers given by the participants were that with the industrial revolution we gained some followers ("but I wasn't there"), that the second law is a property imposed on our choice of parsimonious scientific description, that the third law is a lie because it does not work for glasses, that thermodynamics is a hammer, or a religion, or a totalizing meta-narrative infused with folklore that a post-modernist would demistify.

- On most issues, in particular on what the laws of thermodynamics are, there was very little consensus, with contradictory answers (ranging from facts to conventions).

- The widespread opinion that thermodynamics was guided by the scientific revolution suggests that the development of science is strongly intertwined with the socio-historical context.

- Most people believe thermodynamics is a framework, with the alternative hacked answer "a hammer" receiving some attention.

- Most people believe thermodynamics should be generalized (along with improved, reformulated, simplified, put in a museum); this is interesting given that, as Einstein put it, classical thermodynamics does not add assumptions with respect to those of science itself, thus it literally applies to any physical system.

- The most popular experiment was Joule's experiment relating energy and heat, which is interesting since it deals with the first law but not with the second. Other answers included experiments to obtain chemical potentials; trying to prove Carnot's limit wrong; boiling water to make pasta.

- While a majority defined itself as pragmatic, few were able to propose future experiments, with all proposed experiments quite vague (measure of calorimetric heat at microscopic scale; an experiment identifying the role of fluctuations of non-thermal origin; measuring the collapse of the wave function using quantum signatures of thermodynamic quantities, ruling out dark matter or prove that you can extract work from it).

Finally, during lunch break three whiteboards were placed at the corners of the hall, one for the first, one for the second, and one for the third law of thermodynamics. People were asked to write their own favourite formulation of the law.

## 11.3   Discussion

What follows is a tentative synthesis of some of the observations that arose during the discussion. The very light-hearted session was structured around the answers of the survey, with the objective to trigger a discussion about how the current (and future) state of thermodynamics is perceived by researchers active in the field. The beginning of the discussion was centered on collocating the development of thermodynamics in a socio-historical framework. Many participants (about 40%) recognized that the early development of the field was driven by practical concerns, such as optimizing energy utilization and solving engineering challenges for application to thermal machines during the industrial revolution. In fact, steam engines were already used for many applications since a few decades when Carnot, considered the father of thermodynamics, published his famous book *Réflexions sur la puissance motrice du feu* about work and heat [207].

**Laws of thermodynamics.**   After breaking the ice with some history, we assessed the participants' perceptions about the laws of thermodynamics. Here opinions were quite heterogeneous, as it can be seen from the variety of answers, and the participants started to actively contribute to the debate. What emerged is that people mostly divide over two main interpretations of the first law: on the one hand as the definition of heat giving a name to the non-useful energy whose mechanical analogue is expressed by friction; on the other hand a conservation law (conservation of energy) of phenomenological origin that is in agreement with human empirical experience. It is important to point out that a conservation law is established with respect to some reference: when we say that energy is conserved, we are considering the energy of an open system and its environment, on the assumption that the two together form a closed or isolated system. This concept can be extended to a universal scale only if the whole universe can be assumed to be closed or isolated. This point turned out to be very important throughout the discussion and appeared many times, especially as regards the second law. One of its formulations is in fact that entropy cannot decrease in an ideal closed universe, a requirement we assume in order to be able to perform mental experiments against a more pragmatic definition based on experience, where we never see heat flowing from cold to hot reservoirs. However in a closed universe the Poincaré recurrence theorem is also at work: this latter theorem was largely addressed during the discussion, and seen by some as a proof that the second law of thermodynamics should be taught as false. However, as counter-argued by some, the recurrence time is far above the age of the universe for quite simple systems, and we cannot even say if the universe we belong is closed or isolated.

Another largely discussed point was about entropy being related to order, seen by many as a relative concept, well expressed by a real-life example that probably everyone experienced during his early years: in a very disordered room you may know where an object you need is, even though anyone else would be struggling to find it. Hence, the room has some order according to you, but it is not the same for your mother who does not have information about the position of the items scattered around. On the other hand, if she suddenly decides to clean up your room, and placing items in a way that is ordered for her, you might not be able to find what you are looking for, since that is her order and not yours. Participants had different perceptions of order and they tried to impose their own view, some arguing that there is no reason for entropy to be subjective.

**Experiments in thermodynamics.**   The experiment that is considered by many participants to be the most important one in thermodynamics is Joule's experiment relating work and heat, that proves the first law of thermodynamics (but it was argued that some might have referred to the Joule effect, which does relate to the second law). Not many

experiments about the second law were proposed, maybe because entropy is quite an abstract concept, not physically measurable at least in a direct way. However, as already pointed out, there are formulations of the second law based on the concept of heat, and any test of efficiency of an engine is a test of the second law. Despite many people defined themselves as pragmatic not many possible future experiments were proposed. Some proposals focused on disproving the limitations imposed by nature itself (such as Carnot's limit) rather than actually proving something or getting new insights for future developments of the theory. This discussion sparked little interest though, probably also because the participants consisted mainly on theoreticians.

**Present and future thermodynamics.** What emerged from the survey is that thermodynamics is mostly seen as a framework that allows to deal with ignorance about systems, and that it has to be generalized. Something everyone agreed on is that it can be made more clean, especially in the way it is taught to students. In fact, it is undeniable that thermodynamics was developed initially as a phenomenological theory and this is reflected in the way it is presented to students approaching it for the first time. This also makes it difficult to lead their intuition and a proposed solution is to make it more systematic. Nowadays it is often introduced as a series of chronologically ordered discoveries and phenomenological laws, but we now have access to more mathematically rigorous ways of introducing the discipline. It was pointed out that historically there have been attempts to provide axiomatic formulations of thermodynamics, as for instance, Carathéodory's work.

## 11.4   Denoument

Finally we undisclosed the rationale behind "Postmodern Thermodynamics" as title of the school and workshop by a frontal presentation, starting off with a poem that one of the participants generated using a chatbot based on large language models:

≪Postmodern thermodynamics,
A concept both strange and new,
A field of study that defies
The rules of what we knew.

It challenges our deepest beliefs
About the nature of the world,
And opens up new possibilities
For scientists to unfurl.

With its theories and its principles,
It changes how we see the past,
And guides us towards a future
That is bright and vast.

So let us celebrate this field
That dares to break the mold,
And embrace the possibilities
Of postmodern thermodynamics bold. ≫

We then discussed three different tastes of postmodernism (that we called the "good", the "bad" and the "ugly"), and introduced the figure of Bruno Latour.

An instance of ugly postmodernism was identified in the so-called "science wars" spurred for example by the so-called Sokal hoax – whereby mathematician Alan Sokal exposed a journal in postmodern studies by publishing the nonsensical paper *Transgressing the boundaries: toward a transformative hermeneutics of quantum gravity* [208] (but see *The conceptual penis as a social construct* by Lindsay and Boyle for a similar more recent endeavour [209]). In a following book with Bricmont [210], Sokal insisted that many philosophers and scholars, mostly of French tradition, abused scientific jargon. More at large, scholars identified as postmodern were accused of cherry-picking their topics, of mystifying their discourse by obscure language (often employing scientific jargon), of fostering anti-scientific beliefs, and of creating vacuous academic careers out of fags.

We introduced bad postmodernism by a quote by Miller, a disciple of Popper and vocal critic of the allegedly naïve defense of Sokal and Bricmont of critical rationalism [211]:

> ≪A more fundamental obstacle to a general return to rationalism is that rationalism itself is usually presented in a way that so unedifyingly infringes its own standards of honesty that the only conversions that it can hope to procure are conversions to irrationalism.≫

We skipped on the complex philosophical arguments by Miller to go to what we perceived as issues in the posture of Sokal and Bricmont, if maintained today. In particular: Can we generalize their criticism to all of the social sciences? Do we, as scientists, own scientific jargon? Do we actually understand postmodernist criticism? Is science different in some essential way? The risk we envisioned is that some of the same tendencies that infected postmodern studies a couple of decades ago may be acting within the sciences today. This cannot be imputed to just a few bad apples but to more systemic pressures.

Finally came the good taste of postmodernism in the form of a brief analysis of the 1979 work of Lyotard *The Postmodern Condition: A Report on Knowledge* [212] (well synthesized in Ref. [213]). According to Lyotard modernity was dominated by two institutions: Science (administering truth), and Society (administering justice). Postmodernism is the crisis of these institutions with respect to their grand narratives, embodied for example in the Humboldtian university, analyzed in greater depth by Bill Readings [214]. In this era institutions come in patches that seek their own legitimation through different strategies: *performativity*, that is, implementing technical criteria believed to be subjective (H-index, impact factor and other quantifiers of excellence/ impact/merit etc.); *consensus* – whose spokesperson at the time was Habermas – that is the idea of re-building society by open participative discussion (a virtuous example in the sciences today may be the IPCC reports on climate change, or the process of deliberation by the gravitational wave community documented by Collins [215]); and *paralogy*, whose champion is Lyotard itself, that forsters the acceptance of disagreement and incommensurableness of positions.

In the second part of the presentation we introduced the figure of Bruno Latour, an influential French intellectual who passed away a few weeks before (P)MT. Latour's work has been a constant topic of conversation during lunch and coffee breaks in the CSSM group that hosted and organized (P)MT, mostly thanks to the group leader's personal passion for his figure. Latour was also among the authors criticized by Sokal and Bricmont. One of his major contributions, that set the stage for all of his future elaborations, was an early ethnography of scientists working at an endocrinology lab [216]. In the '90s he proposed *actor-network theory*: science does not explain society just as society does not explain science, but rather they create networks of mutual definitions and actions. In more recent years his public perception shifted from being an anti-science demon to almost a science cheerleader [217]. In an interview by eXtemporanea [218] he said:

> ≪The situation was entirely different when I started to study forty years ago,

and I was accused of criticizing the scientists because we were describing how they work, and that was seen as a critique. But forty years later the only way to still defend science is to do exactly this.≫

One of his crucial works is *Have we ever been modern?*, which departs from an analysis by Shapin and Schaeffer where the birth of modernity is placed in the creation of the vacuum and thus of the possibility of creating the pure facts of nature (but see [219] for a more nuanced discussion as related to thermodynamics). Latour was also critical of the relevance of the scientific method. In the same interview he said [218]:

≪Science is not actually made of philosophical ideas, it's made of patches of lots of little things, like experimental data, puzzles in your head, thousands of other things which lead you to the capacity to make the things you study convince others. It's a very fascinating system which has nothing to do with rationality. Is there one single technical detail of all your exercise as PhD or writers, which is helped by saying "I'm embodying rationality"? Ridiculous. In the 20th century, different from the 21st, when you were teaching the students, if you were giving addresses, it was good to give the impression that you were incarnating rationality spreading into the world. But now? People will laugh at you!≫

Nevertheless, scientists do resort or at least refer to some sort of "method". If this is not a tool to know reality, then what is it? A set of social practices to legitimate science? A set of scientific practices to legitimate society?

To conclude, we launched a pool tournament following the social dinner, that very night, and we entitled it to another hero of the CSSM group, Noam Chomsky, given that December 8th is his birthday. This is slightly ironic as Chomsky is very much against postmodernism (and possibly also of pool), as in the debate he partakes he finds more useful to refer to "truisms" – pieces of objective, rigorous knowledge that can be achieved by assuming a neutral point of view from the outside, of which he is an embodiment.

## 11.5   Considerations

The automated poem above shows that statistical inference algorithms fed by online material can credibly replicate well-established human formats of online expression. This is one of the reasons why in this school and workshop we included an open, physically and socially interactive, "off-the-track" discussion about the context and foundations of the field itself and about future perspectives as (patches of) a community.

Despite recognizing the novelty and strangeness of the juxtaposition of the two words "postmodern" and "thermodynamics" (very rarely associated online[38]), the algorithm reflects a positivistic and progressive view of science. However, such ideals played almost no role in our discussions. Rather, the lack of consensus about as fundamental facts as "what is a law" or "what does a law do?" in favour of a plurality of positions and adjustments was in line with paralogy, a concept proposed by early analyst of the postmodern condition Lyotard to explain the agency of individuals and groups to create a legitimating metanarrative. Displaying structural disagreement and local convergence may actually increase participation and trust: for example, several students privately or through a following survey praised the moment for giving them the opportunity of seeing for the first time senior scientists engage in foundational discussions and manifest their passions.

---

[38]A few occurrences are interesting. We mention without further discussion Haddad [220], Pynchon [221], Burr [222], and Rosenberg [223] commenting on Gilles Deleuze.

The hacking of the entry survey was a very interesting opportunity for conversation. Amongst the wittiest hacks, it was suggested that thermodynamics is a hammer, which quite possibly comes from the saying "to a person with a hammer everything looks like a nail", thus suggesting that the community may be a bit too self-focused.

The discussion session underlined how the scientific community, and in particular in the framework of thermodynamics, has heterogeneous views and perceptions. Interestingly, the discussion ended up recurrently dealing with speculations about the whole universe.

Opening the discussion programmatically also stimulated several participants to the ensueing workshop to include such kind of considerations in their own talks, e.g. by adding non-scientific citations (e.g. Ivan Illich, Paul Karl Feyerabend), or by framing their contribution in a more overarching perspective, or even by announcing their departure from academia.

Two hours was too short a time to dwell into all of the many interesting discussions that initiated, but nevertheless the effort was appreciated as testified by the results of the poll we launched after the event. This suggests the opportunity and necessity to dedicate structured moments of metadiscussion at community events.

# Acknowledgements

Matteo Polettini thanks Michela Bernini for help with the informal discussion session, Artur Wachtel and Tobias Fischback for helping understanding coding, and Alex Blokhuis for many discussions. We are thankful to Tarun Mascarenhas, Ashwin Gopal, Emanuele Penocchio, Philipp Strasberg, and Raphaël Chétrite for reviewing part of the content.

**Funding information**   The school and workshop (Post)Modern Thermodynamics and the present lecture notes were supported and financed by the University of Luxembourg, by the National Research Fund Luxembourg (project CORE ThermoComp C17/MS/11696700), by the European Research Council, project NanoThermo (ERC-2015-CoG Agreement No. 681456), by the Doctoral Program in Physics and Materials Science. Sara Dal Cengio and Vivien Lecomte acknowledge support by the ANR-18-CE30-0028-01 grant LABS, the EverEvol CNRS MITI project and an IXXI project. Gianmaria Falasco is funded by the European Union (NextGenerationEU) and by the program STARS@UNIPD with project "ThermoComplex". Alexandre Lazarescu was supported by the Belgian Excellence of Science (EOS) initiative through the project 30889451 PRIMA Partners in Research on Integrable Systems and Applications. Ariane Soret acknowledges the funding of Luxembourg National Research Fund ThermoQO C21/MS/15713841.

# A   Proposed unified notation

Thermodynamics is a melting pot of ideas and techniques from many sciences, so no surprise that it is confusing. People put together stochastic processes and differential equations, information theory and quantum mechanics, control theory and martingales. They often use the same word for different concepts and call the same thing by different names.

Ahead of the school we asked lecturers to converge on a unified notation, and we even ventured as far as suggesting our own notation. You can find it below and evaluate the success of our proposal. Even some of us organizers did not follow it...

In the free discussions during the school it emerged that students were more confused about use of words rather than notation, e.g. between "equilibrium" and "detailed balance".

This makes us think that a dictionary between present-day thermodynamics and the other sciences, as well as its past, is still much needed, but that it cannot just be about conventions. We hope the experience of (P)MT, while not solving the issue, will least stimulate discussion about it.

## A.1   Free symbols

Greek letter $\alpha, \beta, \ldots$

Upper-case greek letters $\Gamma, \Delta, \ldots$

Upper-case roman letters $E, F, \ldots$

## A.2   Acronyms

CTMC continuous-time Markov chain

BM Brownian motion

FR fluctuation relation

EP[R]: entropy production [rate]

MAK mass-action kinetics

CRN chemical reaction network

TUR thermodynamic uncertainty relation

KCL Kirchhoff's Current Law

KVL Kirchhoff's Loop Law

LDB Local Detailed Balance

WKB Wentzel–Kramers–Brillouin

## A.3   Relations

= generic identity

:= definition of the left-hand side

=: definition of the right-hand side

$\equiv$ taking values (for random variables and examples)

$\asymp$ asymptotically

$\approx$ approximately

$\sim$ distributed with

$\prec$ some order relation

## A.4   Conventions

sign: positive inward the system

no Einstein convention on index contraction

Boltzmann's constant $k_B \equiv 1$

log natural logarithm

$a_b$ and $a(b)$ dependency; prefer $a_b$ if $b$ is fixed, $a(b)$ if $b$ is variable

operations between vectors: $\boldsymbol{a}! = \prod_i a_i!$ and $\boldsymbol{a}^{\boldsymbol{b}} = \prod_i a_i^{b_i}$

$\hbar$ Planck's constant

## A.5    General

$\mathbb{N}, \mathbb{Z}, \mathbb{R}, \mathbb{R}_+, \mathbb{C}$ natural, integer, real, non-negative reals, complex numbers

$i, j, k \in \mathbb{N}$ generic indices

$\mathscr{A}$ some space

$a, b, c, a', a'' \in \mathscr{A}$ generic variables

$\mathscr{X}$ a discrete state space

$x, y, z, x', x'', \text{etc.} \in \mathscr{X}$ discrete states

$\delta_{a',a} = \mathbf{I}_{a',a}$ Kroenecker delta

$\mathbf{I}_{\mathscr{A}}(a)$ indicator function

$\delta(a - a')$ Dirac delta

$\frac{\mathrm{d}}{\mathrm{d}a}, \frac{d}{da}$ total derivative

$\frac{\partial}{\partial a}$ partial derivative

$đa, đa$ inexact differential

## A.6    Linear algebra

$\mathbf{A}$ a matrix

$\mathbf{A}^\top, \mathbf{A}^\dagger$ matrix transpose, adjoint

$\mathbf{A}_{ij}$ or $\mathbf{A}_{i,j}$ the $i$-th row and $j$-th column matrix entry

$\mathbf{A}_{xy}$ or $\mathbf{A}_{x,y}$ matrix entry relative to states $x, y$

$\mathbf{I}$ identity matrix

$\mathbf{v} = \boldsymbol{v}) = |v\rangle$ a vector

$\mathbf{w}\cdot = \mathbf{w}^\top = \langle w|$ a row-vector (1-form)

$\mathbf{w} \cdot \mathbf{v} = \langle w|v\rangle$ scalar product

$\mathbf{w} \circ \mathbf{v} = |v\rangle \langle w|$ outer product

$\mathbf{1}^\top = \mathbf{1}^\dagger = \mathbf{1}\cdot = (1, 1, \ldots, 1)$ a row-vector with all 1's

$\mathrm{Tr}[\hat{M}]$ trace of an operator $\hat{M}$

$\mathrm{Tr}_X[\hat{M}]$ partial trace of $\hat{M}$ over a subspace $X$

$[\hat{M}, \hat{L}] = \hat{M}\hat{L} - \hat{L}\hat{M}$ commutator of two operators $\hat{M}, \hat{L}$

$\mathcal{H}$ Hilbert space

$\hat{M}$ operators acting on $\mathcal{H}$

## A.7   Oriented graphs

$\mathscr{X}$ vertex set

$\mathscr{E}$ oriented edge set

$e = x \to y \in \mathscr{E}$ an oriented edge

$\mathtt{s}(e) = x, \mathtt{t}(e) = y$ the source and target states of an oriented edge

$\mathbf{D} : \mathscr{E} \to \mathscr{X}$ with $\mathbf{D}_{x,e} = \delta_{x,\mathtt{t}(e)} - \delta_{x,\mathtt{s}(e)}$ incidence matrix

$\mathbf{c} \in \ker \mathbf{D}$ an oriented cycle

$\mathbf{c}^* \in \operatorname{im} \mathbf{D}^\top$ an oriented cocycle

## A.8   Stochastic variables

$\hat{a}$ a random variable taking values $a \in \mathscr{A}$

$p\,(\,\hat{a} \equiv a\,) = p(a)$ probability and probability mass function of a discrete random variable taking a certain value

$q, p', p''$ a different probability

$p\,(\,\hat{a} \in [a, a + da)\,) = f(a)\,da$ probability and probability mass function of a continuous random variable taking values in an interval

$g, f', f''$ other probability mass functions

$\langle \hat{a} \rangle$ average

$\mathcal{K}(\lambda) = \log\langle \exp \lambda \hat{a} \rangle$ cumulant generating function

$\langle \hat{a} \rangle^{(i)} = \frac{\partial^i \mathcal{K}}{\partial \lambda^i}(0)$ $i$-th cumulant

$\operatorname{var} \hat{a} = \langle \hat{a} \rangle^{(2)}$ variance

$\mathcal{S}(\hat{a}) = \mathcal{S}(p) = -\sum_{a \in \mathscr{A}} p(a) \log p(a)$ Shannon entropy

$\mathcal{D}(p\|q)$ relative entropy / Kullback-Leibler divergence

## A.9   Stochastic processes

$\hat{B}(t) = \hat{B}_t$ Brownian motion

$\hat{N}(t) = \hat{N}_t$ Poisson process with unit rate

$n, t$ number of iterations and final time of discrete-time and a continuous-time stochastic process

$\mathbf{a} = \boldsymbol{a} = (a_0, a_1, \ldots, a_n)^\top = \{a_i\}_{i=0}^n$ states visited by a discrete-time stochastic process

$p(\hat{a}_n \equiv a) = p_n(a)$ probability of variable after $n$ transitions

$\mathbf{x} = \boldsymbol{x} = \{x(\tau), \tau \in [0, t)\}$ states visited by a continuous-time stochastic process

$p(a_t \equiv a) = p_t(a)$ probability of variable at time $t$

$\tau_0, \tau_1, \ldots, \tau_n$ the times waited at states $x_0, x_1, \ldots, x_n$

$\bar{\hat{a}}_\tau = \hat{a}_{t-\tau}$ time-reversed process

$\bar{p}$ time-reversed probability

## A.10 Markov chains

$\mathbf{P}$ transition matrix of a discrete-time Markov chain

$p(x|y) = \mathbf{P}_{xy}$ transition probability

$\mathbf{R}$ rate matrix of a continuous-time Markov chain

$r(x|y) = \mathbf{R}_{xy}$ rate of jump $y \to x$, for $x \neq y$

$r(x) = -\mathbf{R}_{xx} = \sum_{y \in \mathcal{X}} r(y|x)$ exit rate out of state $x$

$\boldsymbol{p}_\infty$ unique steady distribution (viz. $\mathbf{P}\boldsymbol{p}_\infty = \boldsymbol{p}_\infty$, $\mathbf{R}\boldsymbol{p}_\infty = 0$)

$\mathcal{I}_\infty(a) = -\lim_{t \to \infty} t^{-1} \log p(\hat{a}_t \equiv at)$ rate function

$\mathcal{K}_\infty(\lambda) = \lim_{t \to \infty} t^{-1} \log \langle \exp \lambda \hat{a}_t \rangle$ scaled-cumulant generating function

## A.11 Thermodynamics

$T, \beta$ temperature, inverse temperature

$\mu$ chemical potential

$\mathcal{E}$ energy

$\mathcal{U}, \mathcal{V}$ generic potentials

$đ\mathcal{Q}$ heat increment

$đ\mathcal{W}$ work increment

$\mathcal{F}, \mathcal{G}$ Helmholtz and Gibbs free energies

$\mathcal{S}^{\mathrm{another}}$ another notion of entropy (different from Shannon's)

$\mathbf{L}$ linear-response matrix

$f_t(x|y) = r(x|y)p_t(y)$ mean edge flux

$j_t(x|y) = f_t(x|y) - f_t(y|x)$ mean edge current

$a(x|y) = \log \frac{r(x|y)}{r(y|x)}$ local-detailed-balance affinity

$\sigma_t = \sum_{x \prec y} j_t(x|y)a(x|y)$ mean environmental EPR

$\hat{F}_t(x|y)$ time-integrated flux

$\hat{J}_t(x|y)$ time-integrated current

$A(\mathbf{c})$ cycle affinity

### A.12  Chemical reaction networks

$X, Y, Z, X', X'',$ etc. $\in \mathscr{X}$ (roman roman letters) chemical species

$\mathbf{X} = (X_1, \ldots, X_s)$ vector of chemical species

$[X] = X \in \mathbb{N}$ population of X

$x, y, x', x'' \in \mathbb{R}_+$ deterministic populations

$\mathscr{C}$ stoichiometric compatibility class

$\pm\rho, \rho \in \{1, \ldots, r\} = \mathscr{R}$ oriented reactions

$k_{\pm\rho}$ reaction rate constant (stochastic)

$\overline{k}_{\pm\rho}$ reaction rate constant (mean-field)

$r_\rho(\boldsymbol{X})$ reaction rate velocity

$\nu$ stoichiometric coefficient

$\mathbf{N}$ stoichiometric matrix

$\mathbf{c}$ right-null vector of $\mathbf{N}$ (chemical cycle)

$\boldsymbol{\ell}\cdot$ left-null vector of $\mathbf{N}$ (conservation law)

$\lambda$ conserved deterministic quantity

$\mathcal{L}$ conserved stochastic quantity

$\delta$ deficiency

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
