# Peer review of "Methods and Conversations in (Post)Modern Thermodynamics"

_SciPost Physics Lecture Notes_

## Round 1 · Referee Report · Anonymous (Referee 1) · 2023-12-30

Strengths

1- Timely subject

2- Broad field of application.

3- Lectures by leading researchers.

Weaknesses

1- See report

Report

This is my report on https://scipost.org/submissions/scipost_202312_00007v1/ Methods and Conversations in (Post)Modern Thermodynamics, by F. Avanzini et al. I recommend publication after the authors consider the points below.

The work may be roughly divided in three parts, namely

i) Foreword, Introduction and Sections 1 to 10.

ii) Section 11.

iii) Appendix A

Part i) is a set of lectures on recent advances in thermodynamics, defined on very broad terms. The subjects range from the time-honored (such as metastability (Lecture 8) and first passage problems (Lecture 9)) to relatively new applications, such as network thermodynamics (Lecture 2). The speakers are in general leaders of their fields and the level of the lectures is excellent. There is some overlap between different lectures (for example, Lecture 2 touches on chemical reaction networks, the subject of Lectures 4 and 5, and on Markovian dynamics, the subject of Lecure 1). However, I believe it is better to allow for this as it makes every lecture more self-contained.

I believe this set of lectures is timely and will be very helpful to the community. I would expect it will have a similar impact than the set of review articles Thermodynamics in the Quantum Regime by F. Binder et al. (Springer, 2018), with Lecture 10 actually serving as a bridge from one to the other.

As a reader I would have preferred for each lecture to have its own set of references, rather than a single listing at the volume’s end.

Part iii) is a bold proposal for a unified nomenclature for all things thermodynamic. I am not aware of similar initiatives on other fields of physics; usually it is some highly successful textbook, such as Landau and Lifshitz, Bjorken and Drell or Misner, Thorne and Wheeler, which sets the notation for its period of dominance. I believe such an agreement would be very valuable, although it will not be generally accepted without some discussion. Personally I would object to EP(R) as entropy production (rate), as I find very difficult to understand EPR as meaning anything other than Einstein, Podolsky and Rosen.

Part ii) is certainly the most controversial. Its declared aim is to summarize an open discussion session and on-line survey which took part at the end of the conference and to draw some conclusions, the overall subject being the rationale behind “Postmodern Thermodynamics" as title of the school and workshop. The text reflects the semi-serious (authors’ words) character of the activity, including a detailed account of how the survey was hacked by the participants.

The discussion itself is a somewhat mixed account on society at large’s impact on the development of Physics, the character of physical law, the relationship between theory and experiment, the future of thermodynamics and the dialog (or lack thereof) between Physics and the Humanities, as exemplified by the infamous Sokal hoax. Along the discussion the usual suspects are rounded up, with some discussion of Bruno Latour’s views on science and Jean-François Lyotard views on post-modernism, but also mentions to other authors such as Feyerabend, Habermas, Shapin and Schaeffer and a somewhat contrived mention to Chomsky.

Personally I would commend the authors for raising the topic and I enjoyed the informal style. However, as the text is going to achieve some kind of permanence as a published work, I would ask the authors to consider the following points.

My main misgiving is that I do not really see how the discussion here relates to the preceding lectures, which other than the novelty of the subjects were delivered in the conventional fashion. From this point of view, I think this lecture in its present form fails short of its stated goal of justifying the (Post) in the title.

I believe this faillure relates in part to the fact that the text does not fully explain what being modern or postmodern means – I would accept that part of being postmodern is not to need such explanations. We are told that

"According to Lyotard modernity was dominated by two institutions: Science (administering truth), and Society (administering justice). Postmodernism is the crisis of these institutions with respect to their grand narratives, embodied for example in the Humboldtian university, analyzed in greater depth by Bill Readings [214]. In this era institutions come in patches that seek their own legitimation through different strategies: performativity, that is, implementing technical criteria believed to be subjective (H-index, impact factor and other quantifiers of excellence/ impact/merit etc.); consensus ( whose spokesperson at the time was Habermas ) that is the idea of re-building society by open participative discussion (a virtuous example in the sciences today may be the IPCC reports on climate change, or the process of deliberation by the gravitational wave community documented by Collins [215]); and paralogy, whose champion is Lyotard itself, that forsters the acceptance of disagreement and incommensurableness of positions. "

(by the way, didn’t the authors mean that the technical criteria were believed to be "objective"?) and then further on

"...the lack of consensus about as fundamental facts as “what is a law” or “what does a law do?” in favour of a plurality of positions and adjustments was in line with paralogy, a concept proposed by early analyst of the postmodern condition Lyotard to explain the agency of individuals and groups to create a legitimating metanarrative."

I haven’t been able to find examples of such acceptance of disagreement and incommensurableness of positions in any of the ten preceding lectures; I would be glad if it turns out I simply missed them, and then I would appreciate they are pointed out explicitly. Otherwise, I must suggest considering taking the (Post) out of the title.

I would take the opportunity to ask the authors to comment on a feature of the text I found surprising. On the subject of the overall impact of society at large on the development of Physics all the given examples concern the technological or economic, such as

- How is thermodynamics linked to the industrial revolution?

- The widespread opinion that thermodynamics was guided by the scientific revolution suggests that the development of science is strongly intertwined with the sociohistorical context.

- Many participants (about 40%) recognized that the early development of the field was driven by practical concerns, such as optimizing energy utilization and solving engineering challenges for application to thermal machines during the industrial revolution.

I find it surprising (and hard to believe this is the authors’ position) that the text suggests the only impact of society at large on Physics comes from technological demands, the fact that the founding fathers of thermodynamics (particularly, say, from mid XIX to mid XX centuries) were all white males living in or near imperial centers being irrelevant. Even with respect to Physics and the Industrial Revolution it could be argued that the connection is much deeper than just the technological driver. In the words of Evelyn Fox Keller,

"...it is a fantasy that any human product could be free of human values. And science is a human product." [1]

To conclude, I recommend publication after the authors consider my optional suggestions.

[1] Evelyn Fox Keller in “Evelyn Fox Keller: The Gendered Language of Science”, interview by Bill Moyers, May 6th, 1990. Online https://billmoyers.com/content/evelyn-fox-keller/ Consulted on 12/30/2023.

Requested changes

See report.

  • validity: top
  • significance: top
  • originality: high
  • clarity: top
  • formatting: good
  • grammar: -

Author:  Pedro Harunari  on 2024-01-22  [id 4269]

(in reply to Report 1 on 2023-12-30)

We thank the referee for the very insightful and overarching view on the lecture notes. We modified the lecture notes in several parts marked in red (especially the Introduction and Lecture 11).

Part i) is a set of lectures on recent advances in thermodynamics […] I believe this set of lectures is timely and will be very helpful to the community. I would expect it will have a similar impact than the set of review articles Thermodynamics in the Quantum Regime by F. Binder et al. (Springer, 2018), with Lecture 10 actually serving as a bridge from one to the other.

Thanks for the appreciation. Let us point out that the fact that there are overlaps is intentional and has been the output of intensive preparatory work ahead of the school.

We included a mention of F. Binder et al. (Springer, 2018) in Lecture 10.

As a reader I would have preferred for each lecture to have its own set of references, rather than a single listing at the volume’s end.

We solved this problem.

Part iii) is a bold proposal for a unified nomenclature for all things thermodynamic […]

We were undecided about publishing the unified notation, that was proposed in preparation of the school and was followed only in part by the lecturers. Informally, many participants agreed on the usefulness of this conversation. However, this might not be the right place and mode to make a proposal. Therefore we removed this Appendix and moved its opening paragraph to the Introduction.

Part ii) is certainly the most controversial. […] with […] a somewhat contrived mention to Chomsky.

We agree the comment on Chomsky is contrived and only kept “(This is slightly ironic as Chomsky on several occasions expressed contempt for postmodernist analysis.)”

Personally I would commend the authors for raising the topic […] I think this lecture in its present form fails short of its stated goal of justifying the (Post) in the title.

The “(Post)” in the title is also the title of the school we ran in December 2022, which was very successful also because it was informal in many other respects (especially from an organizational point of view). We did not take it too seriously, and we hope the reader will not either. For us it would be a bit difficult to disentangle the school and the lecture notes, so for the moment we keep the title as is, but we are open to future suggestions from other referees.

[…] the text does not fully explain what being modern or postmodern means – I would accept that part of being postmodern is not to need such explanations.

About the birth of modernity: we added a short explanation of (what we understand of) We have never been modern by Latour, which was inspirational. We also added a quote from the suggested article by Fox Keller that resonates with it.

We are told that […] (by the way, didn’t the authors mean that the technical criteria were believed to be “objective”?) and then further on

Of course!

“…the lack of consensus […] to create a legitimating metanarrative.” I haven’t been able to find examples of such acceptance of disagreement and incommensurableness of positions in any of the then preceding lectures; I would be glad if it turns out I simply missed them, and then I would appreciate they are pointed out explicitly. Otherwise, I must suggest considering taking the (Post) out of the title.

This is a very good point, thanks for raising it. The lectures themselves are rather technical and orthodox, following the Markov process approach to thermodynamics, which of course entails a lot of debatable assumptions.

This final “lecture” was the only moment where we could taste some outspoken disagreement, and our impression was it could go on for a full day or more, and that it could get quite heated… Unfortunately in the transcript we could not give a full account.

So indeed the title may be misleading if one expects a systematic critical analysis of the foundations of the field and of its relations to other theoretical approaches to thermodynamics.

However some hints of disagreement and incommensurableness can be read between the lines of the main lectures, by an expert eye. It is not up to us editors to point them out. Just a couple of examples that regard directly one of the editors’ contributions: - The discussion surrounding Fig. 5.8 and the conclusions of Chapter 5 suggest that one of the most acclaimed results in the field, the fluctuation relation can only be tested in regimes where it is explained by the previous best theory (the linear regime hypothesis). This makes it pretty post-modern… - About the proposed unified notation, we noticed that people converged on some mathematical and fundamental physics conventions, but could not converge on the thermodynamic ones – which is odd given the school and workshop were dedicated to thermodynamics!

I would take the opportunity to ask the authors to comment on a feature of the text I found surprising. On the subject of the overall impact of society at large on the development of Physics all the given examples concern the technological or economic, such as - How is thermodynamics linked to the industrial revolution? - The widespread opinion that thermodynamics was guided by the scientific revolution suggests that the development of science is strongly intertwined with the sociohistorical context. - Many participants (about 40%) recognized that the early development of the field was driven by practical concerns, such as optimizing energy utilization and solving engineering challenges for application to thermal machines during the industrial revolution. I find it surprising (and hard to believe this is the authors’ position) that the text suggests the only impact of society at large on Physics comes from technological demands, the fact that the founding fathers of thermodynamics (particularly, say, from mid XIX to mid XX centuries) were all white males living in or near imperial centers being irrelevant. Even with respect to Physics and the Industrial Revolution it could be argued that the connection is much deeper than just the technological driver. In the words of Evelyn Fox Keller, “…it is a fantasy that any human product could be free of human values. And science is a human product.” [1]

Yes. This sociological consideration is very narrow and does not capture the full picture. Other factors such as gender/power inequality, political or religious beliefs etc. should be taken into account. In fact, if we follow Latour to the end (who criticized sociological reductionism - and postmodernism! - just as much scientific reductionism) it would be even more interesting to eliminate social explanations of science and use the objects of science themselves (e.g. the second law) to create meaningful nexuses that “reassemble the social”, as per the title of an important book by him.

The first question on the industrial revolution was meant to be an icebreaker towards an audience that might not have had any previous exposure to meta-narratives of science. We used a discrepancy with history as is told in high-school textbooks. Of course this does not exhaust the discussion.

Thanks for pointing out the interview to Fox Keller. We added a quote from it and a brief discussion.

(addendum: the interviewer writes “my friend at the University of Texas, a physicist, Steven Weinberg, talks about the universe as being one of overwhelming hostility”: isn’t this just a reframing of homo homini lupus and of Hobbes’s assumptions about the “state of nature”?).

---

## Editorial Decision

resubmitted